# LogicEvolve: Advancing Logical Reasoning Toward Self-Evolution

## Abstract

The rapid progress of large language models (LLMs) highlights the urgent need for continuously evolving benchmarks that keep pace with advancing model capabilities. Yet existing benchmarks often rely on one-off curation or fixed scripts, lacking scalability and long-term adaptability. To this end, we present LogicEvolve, a highly automated multi-agent framework that enables dynamic control of deterministic symbolic tasks' structure, difficulty distribution, and scale with minimal human intervention. Building on LogicEvolve, we introduce CLUB (Complex Logical Unified Benchmark), spanning diverse task types—including string puzzles, grid reasoning, and card games—for systematic evaluation of logical reasoning. Experiments show that even state-of-the-art models such as Grok-4 and GPT-5 reach only 55–56% accuracy across multiple independent evaluations, far below desirable levels, with clear weaknesses in certain subcategories. These findings underscore logical reasoning as a persistent and unsolved core challenge for LLMs. All code, data, and an interactive evaluation platform will be publicly released after the review period, ensuring reproducibility and fostering further research.

## 1 Introduction

Logical reasoning is the ability to draw correct inferences and conclusions from known information by following well-defined logical principles (Newell & Simon, 1956; Bronkhorst et al., 2020; Thomason, 2024). Recently, reasoning-oriented LLMs have increasingly been regarded as capable of performing tasks that were once considered beyond the reach of artificial intelligence, requiring deep logical thinking (Li et al., 2025; OpenAI, 2025; DeepSeek-AI et al., 2025). The rapid evolution of model capabilities highlights the urgent need for robust benchmarks and evaluation methodologies that can continuously evolve alongside these advances (Shao et al., 2024; Giadikiaroglou et al., 2024; Liu et al., 2025a).

Although numerous datasets have been proposed to evaluate the logical reasoning abilities of LLMs (see Table 1), early datasets such as LogiQA2.0 (Liu et al., 2020; 2023) and BBEH (Kazemi et al., 2025) were primarily constructed through one-time human collection, covering only limited categories, restricted scales, and fixed difficulty levels. Such datasets are easily leaked into large-scale web-scraped pretraining corpora (Magar & Schwartz, 2022; Oren et al., 2023; Li & Flanigan, 2024). More recently, to mitigate benchmark saturation potentially caused by memorization effects (Ni et al., 2025), efforts such as SynLogic (Liu et al., 2025b) and Enigmata (Chen et al., 2025) have attempted to employ programmatic synthesis to construct dynamic datasets. However, these scripts are still largely handcrafted, lacking reusable mechanisms, which limits their ability to update existing tasks or extend to new ones. This raises the critical question: *can we develop a method that automatically generates dynamic and sustainable logical reasoning benchmarks, thereby avoiding the labor-intensive process of manual collection, construction, and maintenance?*

Inspired by the human experiential learning process (Tao et al., 2024), we propose **LogicEvolve**, a multi-agent collaborative generation framework for rule-based logical reasoning tasks. The core idea is to leverage transfer and generation based on existing information (Cao et al., 2010), while iteratively refining solutions through feedback (Liu & van der Schaar, 2025). As illustrated in Figure 1, the framework consists of three types of core agents: a *metadata agent*, responsible for the automated definition of logical reasoning tasks; *generator and solver agents*, which handle automatic task generation and resolution; and an *evaluator agent*, which automatically produces evaluation scripts.

Table 1: Comparison between **LogicEvolve** and mainstream logical reasoning benchmarks. **Variability criteria** consist of three dimensions: CatVar (whether the set of task categories can be extended or replaced), QtyVar (whether the number of tasks can be expanded without bound; the value in parentheses indicates the actual number of tasks, and $\infty$ denotes unlimited scalability), and DiffVar (whether task difficulty can be continuously adjusted via parameterization). Symbols: ✓ denotes variable, and ✗ denotes fixed. **Degree of Task Automation** is divided into four levels: level 1 — open-source data collection only; level 2 — open-source code for automated evaluation; level 3 — open-source code for automated data generation; and level 4 — meta-code that automatically generates both data and evaluation pipelines (largely-automated synthesis with optional human oversight). It should be noted that although SLR and AutoLogi report $\infty$ for task quantity, their automation level remains at 2, since as of September 2025 no usable level-3 or level-4 code had been released.

| Benchmarks | CatVar | QtyVar | DiffVar | Automation |
|---|---|---|---|---|
| LogicBench (Parmar et al., 2024) | ✗ | ✗ (25) | ✗ | level 1 |
| LogicGame (Gui et al., 2024) | ✗ | ✗ (31) | ✗ | level 1 |
| ZebraLogic (Lin et al., 2025) | ✗ | ✗ (1) | ✗ | level 2 |
| LogicQA (Liu et al., 2020) | ✗ | ✗ (5) | ✗ | level 2 |
| LogicQA2.0 (Liu et al., 2023) | ✗ | ✗ (5) | ✗ | level 2 |
| LogiTorch (Helwe et al., 2022) | ✗ | ✗ (16) | ✗ | level 2 |
| BIG-Bench Hard (Suzgun et al., 2023) | ✗ | ✗ (23) | ✗ | level 2 |
| BBEH (Kazemi et al., 2025) | ✗ | ✗ (23) | ✗ | level 2 |
| LogiGLUE (Luo et al., 2024) | ✗ | ✗ (24) | ✗ | level 2 |
| KOR-Bench (Ma et al., 2025) | ✗ | ✗ (125) | ✗ | level 2 |
| BIG-Bench (Srivastava et al., 2023) | ✗ | ✗ (204) | ✗ | level 2 |
| SLR (Helff et al., 2025) | ✗ | ✓ ($\infty$) | ✓ | level 2 |
| AutoLogi (Zhu et al., 2025) | ✗ | ✓ ($\infty$) | ✓ | level 2 |
| K&K (Xie et al., 2024) | ✗ | ✗ (1) | ✓ | level 3 |
| SynLogic (Liu et al., 2025b) | ✗ | ✗ (35) | ✓ | level 3 |
| Enigmata (Chen et al., 2025) | ✗ | ✗ (36) | ✓ | level 3 |
| REASONING-GYM (Stojanovski et al., 2025) | ✗ | ✗ (100) | ✓ | level 3 |
| **LogicEvolve (ours)** | ✓ | ✓ ($\infty$) | ✓ | level 4 |

Building on the above methodology, we construct the **Complex Logical Unified Benchmark (CLUB)**. CLUB consists of ten distinct task categories, each containing 100 dynamically generated problems spanning different levels of complexity, designed to enable in-depth evaluation of models' logical reasoning capabilities (see Figures 5 and 6). Results from three independent evaluations show that even the most advanced model, Grok-4 (xAI, 2025), achieves only an overall accuracy of $56.37\%$, with merely $8.67\%$ on the tetris task, underscoring the necessity of systematic assessment and continuous improvement in complex logical reasoning.

In addition, the LogicEvolve framework further evolves to generate 100 additional tasks, forming a larger-scale evaluation suite, **ExCLUB**, for more comprehensive assessment of complex logical reasoning in large models. We will release all code, data, and the evaluation platform after the anonymity period, and we will continuously evolve CLUB using the LogicEvolve framework to ensure the long-term sustainability of the benchmark. In summary, our work makes the following key contributions:

(1) **Framework.** We propose **LogicEvolve**, a multi-agent framework capable of autonomously generating and evolving logical reasoning tasks, with dynamic control over task structure, difficulty distribution, and scale.

(2) **Benchmark.** We construct **CLUB**, a systematic and dynamic benchmark for logical reasoning that spans diverse tasks such as string puzzles, grid reasoning, and card games, enabling comprehensive evaluation of models' reasoning capabilities.

(3) **Evaluation and Analysis.** Experimental results demonstrate that even the most advanced models remain far from satisfactory performance on CLUB, with pronounced weaknesses in several subtasks, underscoring that logical reasoning remains a central open challenge.

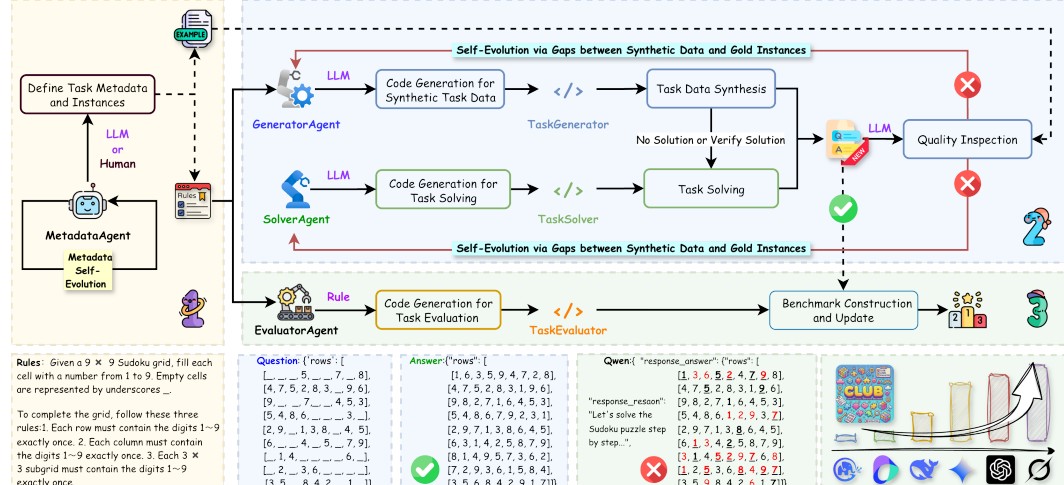

Figure 1: Overview of the LogicEvolve Method. The framework comprises three stages: (1) automatically evolving and constructing metadata for logical reasoning tasks based on paradigms such as induction and deduction; (2) employing multi-agent collaboration to develop generators, solvers, and evaluators for logical reasoning tasks, which are iteratively optimized through self-evolution; and (3) generating and evaluating concrete task instances using the generator, solver, and evaluator.

## 2 RELATED WORKS

Our work is at the intersection of several recent research directions which we briefly recount below.

**Evaluation of Logical Reasoning Ability.** To assess the logical reasoning capabilities of LLMs, researchers have continuously developed dedicated datasets, as summarized in Table 1. This evolution can be broadly divided into three stages: (1) *Single-task static datasets*, such as LogiQA2.0 (Liu et al., 2020; 2023), which offer high-quality benchmarks but rely entirely on manual collection—requiring approximately 667 hours of annotation—and therefore cannot be updated in pace with rapidly advancing model capabilities; (2) *Multi-task static datasets*, such as LogicGLUE (Luo et al., 2024), which integrate multiple existing single-task datasets into a unified evaluation suite, but remain incapable of automatically incorporating new tasks, making sustained updates and extensions equally difficult; (3) *Multi-task dynamic synthetic datasets*, such as REASONING-GYM (Stojanovski et al., 2025), which provide hundreds of task generators and validators, enabling virtually unlimited data generation with adjustable parameters to control task complexity, thereby supporting continuous evaluation across difficulty levels. However, as these scripts are manually authored, their ability to extend beyond a limited set of tasks remains constrained. In this work, we address these limitations by introducing the **LogicEvolve** framework, which automatically generates dynamic and sustainable logical reasoning benchmarks, thereby avoiding the labor-intensive processes of manual collection, coding, and maintenance.

**AI-for-AI (AI4AI)** refers to the use of artificial intelligence techniques to automate and optimize the design, training, and deployment of AI systems themselves (Liu et al., 2025c). In this work, we explore AI4AI in the domain of logical reasoning: instead of relying entirely on manually defined toolchains, we leverage multi-agent autonomous collaboration to achieve automated generation and verification of logical reasoning tasks.

**Self-Evolution** refers to the ability of large language models (LLMs) to autonomously generate, refine, and learn from their own experience, offering a scalable path toward superintelligence (Huang et al., 2025; Gao et al., 2025). AlphaEvolve (Novikov et al., 2025) and Alita (Qiu et al., 2025) are two representative self-evolving agents, aimed respectively at scientific and algorithmic discovery and at generalist reasoning. In this study, we adapt similar ideas to logical reasoning tasks: through the self-evolution of the metadata agent and the error-driven self-optimization of the logic generation and parsing agents, we ensure the autonomy of the **LogicEvolve** architecture.

## 3 LOGICEVOLVE

The overall workflow of the **LogicEvolve** framework consists of three main modules, as shown in Figure 1: (1) The input stage is handled by a *logical-reasoning metadata agent*, which automatically defines the structured information of reasoning tasks; (2) At the core is a multi-agent collaborative generation component, comprising a *logic puzzle generator* and a *logic puzzle solver*, which jointly accomplish task generation and parsing; (3) Finally, a *logic puzzle evaluator* performs automatic sampling of tasks and outputs large-scale complex logical-reasoning data and code for building the CLUB benchmark.

### 3.1 META-DATA CONSTRUCTION: FROM LOGICAL SEEDS TO REASONING FORESTS

The metadata agent organizes each type of reasoning task as $\langle B, I \rangle$: here, the *base assumptions* $B$ serve as a reusable "axiomatic system," specifying the task's general logical structure, semantic premises, and reasoning paradigm; the *instance set $I$* attaches concrete constraints and question–answer pairs, narrowing the search space to ensure that each problem has a unique or verifiable solution. This "rule–instance" dichotomy guarantees both *generalizability* (provided by $B$) and *solvability* (established by $I$). A detailed example of the input is shown in Figure 2. Appendix C.1 provides the details of the implementation mechanism, Appendix E presents the metadata of the 10 CLUB tasks, and Appendix F further includes the metadata of the 100 ExCLUB tasks.

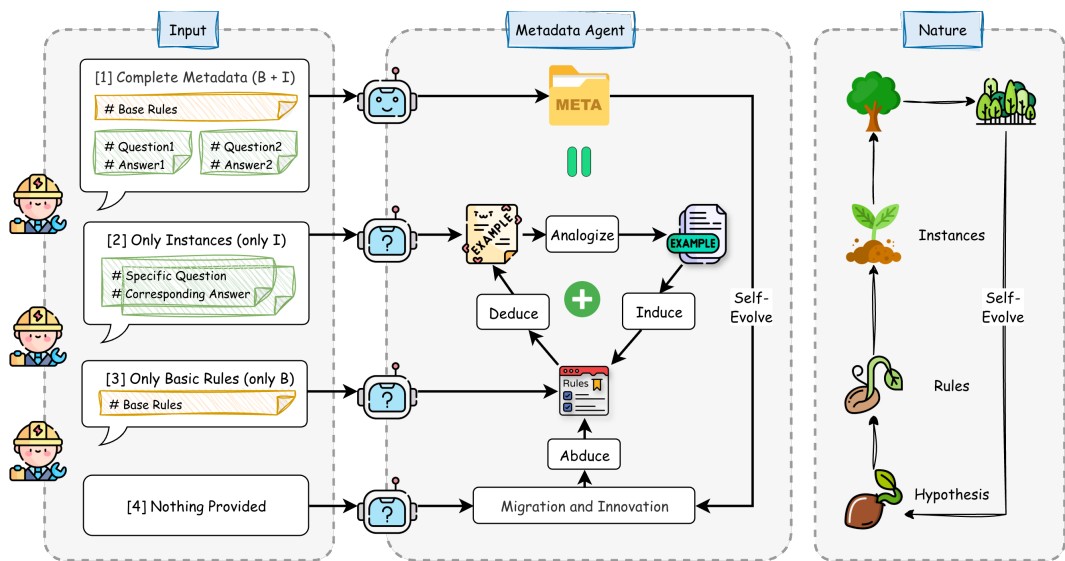

Figure 2: Details of the MetadataAgent. Real-world inputs can be divided into four categories: *Complete* (containing both $B$ and $I$); *Rule-driven* (only $B$); *Instance-driven* (only $I$); *Sparse* (neither component). For these four scenarios, the metadata agent performs, respectively: direct storage; rule-based deduction to generate instances; induction of rules from instances; and transfer of existing patterns to construct tasks from scratch. We liken this process to cultivating a "logic forest": each $B$ acts as a seed, and its derived $I$ are sprouts; a pair $\langle B, I \rangle$ forms a "logic tree," and multiple trees converge into a continuously growing forest, providing a sustainable source of nourishment for subsequent task generation and evaluation.

### 3.2 COLLABORATIVE AGENTS FOR CONSTRUCTING AND SOLVING LOGICAL-REASONING TASKS

Within the collaborative-agent module of the **LogicEvolve** framework, the *logic puzzle generator* (Generator Agent) and the *logic puzzle solver* (Solver Agent) constitute the core cooperative agent system, as illustrated in Figure 3. The detailed implementation can be found in Appendix C.3.

**Generator: Generalization and Construction Based on Hypotheses.** The generator takes the task metadata as input and first abstracts a set of tunable hyperparameters to characterize the task's

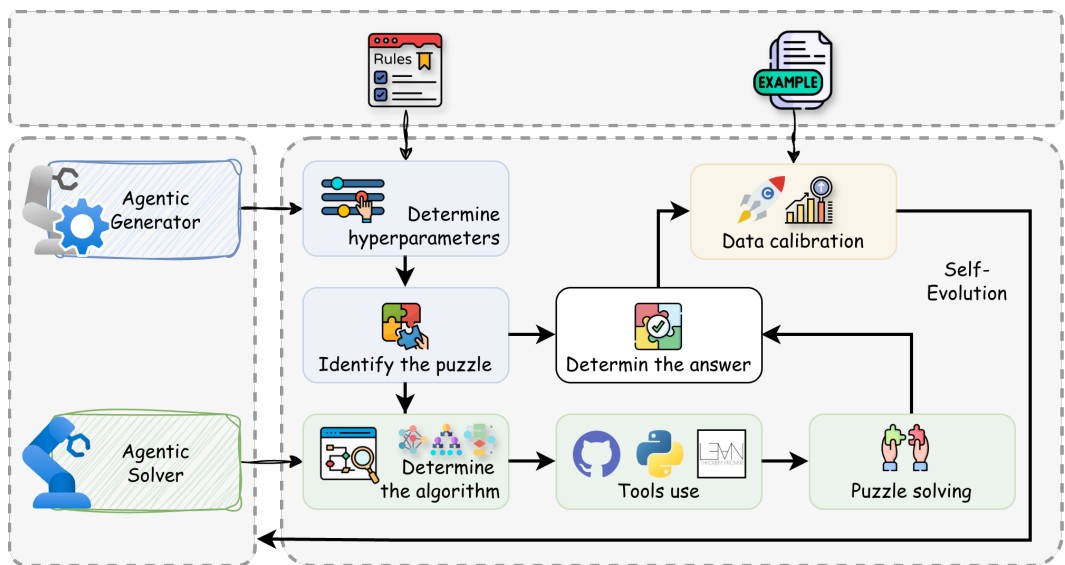

Figure 3: Details of the GeneratorAgent and SolverAgent. The generator is responsible for the automatic construction and preliminary parsing of logical tasks, while the solver handles deep parsing for tasks that are difficult to analyze. Through self-evolution, the quality of tasks produced by both the generator and the solver is continuously optimized.

generalizable space (see Appendix C.2 for the parameter abstraction used in CLUB). After modeling the parameter space, the generator attempts to construct dynamically generalizable code for task-data generation. Its generalization strategy is divided into *horizontal* and *vertical* dimensions: horizontally, it diversifies parameter configurations while keeping the task paradigm fixed, producing instances that are structurally consistent yet parametrically diverse; vertically, it increases reasoning complexity along a difficulty gradient by adjusting key parameters step by step.

**Solver: Strategy Search and Program Synthesis.** The solver is primarily employed for tasks where the generator cannot directly obtain an answer or for verifying the uniqueness of solutions (see Appendix C.5). Given a generated problem as input, the solver first leverages an LLM to infer an appropriate solving strategy, followed by synthesizing a solver program to compute the result. If repeated attempts still fail to yield a valid solution, the solver integrates external tools (e.g., GitHub code retrieval and LEAN-based formal solvers) to obtain the final answer. For unsolvable cases, the solver raises an exception and returns the problem information to the generator, thereby driving iterative refinement of task generation.

**Optimization: Feedback on Generation Quality and Accuracy Improvement.** Even though the combination of the generator and the solver can initially complete the generation and parsing of task data, a secondary verification and optimization step is needed to ensure consistency with the instances defined in the metadata. LogicEvolve aligns the logical problems produced by the generator with the standard instances in the metadata and performs a difference analysis, comparing aspects such as semantic fidelity, informational completeness, and uniqueness of solutions. It then feeds specific discrepancies—such as logical ambiguity or missing information—back to the generator and solver, driving their self-evolution and further refining both the generation code and data strategies. A detailed and optimized description is provided in Appendix C.3.3. For cases where neither the generator nor the solver can obtain a valid solution, an exception is raised and returned to the metadata agent, prompting it to optimize the task specification. If the metadata agent is also unable to resolve the issue, the exception is propagated to the user to solicit human intervention. For further details, please refer to Appendix C.9 and Appendix C.10.

**Evaluator: Automatic Sampling and Data Construction.** The evaluator is responsible for synthesizing data using the final code produced through iterative interactions between the generator and solver. It uniformly samples across the parameter space to construct large-scale datasets for assessing complex logical reasoning, and, together with metadata information, automatically generates

evaluation scripts. Each problem's output is formatted as JSON containing multiple fillable fields. By computing the accuracy for each field, we obtain the *cell accuracy* (Cell Acc), while verifying the correctness of all fields in a problem yields the *example-level accuracy* (Acc). The detailed pseudocode and implementation can be found in Appendix C.4.

## 4 CLUB

The **CLUB** benchmark is a dynamic evaluation suite for logical reasoning, primarily covering tasks that require complex reasoning processes or whose solutions incur high computational costs. Its design is characterized along two complementary dimensions: *breadth*, which ensures task diversity, and *depth*, which defines the difficulty threshold. Together, these dimensions enable a systematic evaluation of LLMs' capabilities in complex logical reasoning.

**Breadth: Task Diversity.** As shown in Table 1, logical reasoning tasks are often categorized according to reasoning paradigms (induction, deduction, and abduction). However, such a scheme overlooks the diversity of reasoning tasks. To address this, we adopt a hierarchical taxonomy that provides a fine-grained, three-level classification: (1) reasoning paradigms; (2) reasoning monotonicity (monotonic vs. non-monotonic reasoning); and (3) the spatial dimension of reasoning (one-dimensional, two-dimensional, three-dimensional, or higher-dimensional), with detailed definitions given in Appendix B.1. CLUB selects one commonly used task from each category, as illustrated in Figure 4, while ExCLUB attempts to cover as many tasks as possible according to this taxonomy.

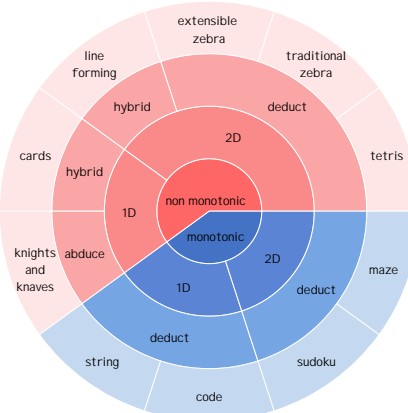

Figure 4: Task Diversity in CLUB.

**Depth: Task Complexity.** To distinguish complex logic from simple propositional and first-order logic, and given that tasks with very low computational complexity may offer insufficient challenges for future large models, we define the sufficient condition for qualifying as complex logic ($CL$) as follows: for a logical problem $L$, the minimum expressive power of the logical system required, $\text{Expr}(L)$, should preferably reach at least second-order logic (SOL); or the minimal computational complexity class of the corresponding decision problem, $\text{Complexity}(L)$, should preferably be at least NP-complete or a stronger class. Formally, this is defined as:

$$CL(L) \;\Leftarrow\; \big(\text{Expr}(L) \;\geq\; \text{SOL}\big) \;\vee\; \big(\text{Complexity}(L) \;\geq\; \text{NP-complete}\big) \tag{1}$$

For example, in CLUB, the Maze problem is polynomial-time solvable in its standard setting, but extensions with keys and doors have been proven NP-complete; line-forming games involve reasoning about possibilities and opponent actions, which can be formalized with modal or other extended logics, and their complexity often reaches EXPTIME; the Zebra puzzle can be reduced to SAT, is NP-complete, and exemplifies first-order logical reasoning with conditional dependencies and elimination. A more detailed analysis is provided in Appendix B.2.

## 5 EXPERIMENTS

In this section, we systematically evaluate the effectiveness of our approach. We first analyze the usability and efficiency of the LOGICEVOLVE framework, addressing whether it can support large-scale, low-intervention task synthesis and evaluation. We then conduct a comprehensive benchmark study on over thirty state-of-the-art large language models using CLUB, which enables a fine-grained assessment of their strengths and limitations in complex logical reasoning. The results demonstrate both the practical value of our framework and the persistent challenges faced by current models. Additional analyses and extended results are provided in Appendix C, D, and G.

## 5.1 ANALYSIS OF LOGICEVOLVE

**Basic Setup.** We manually implemented CLUB through 5 software engineers (see Appendix D.3.1), including metadata, puzzle generators, solver code, and concretely generated and validated puzzle instances, which serve as a baseline reference. We then conducted cross-validation with LogicEvolve: using human solvers to verify the solvability of data produced by the LogicEvolve generator, and employing LogicEvolve solvers to solve data generated by the human-designed generator.

**Efficiency of LogicEvolve.** Table 2 reports the number of iterations, time consumption, and construction efficiency required for LogicEvolve to build usable generators and solvers across different CLUB tasks. Compared to human efforts, LogicEvolve achieves human-level construction quality while significantly outperforming manual processes, delivering speedups from two up to nearly three orders of magnitude.

Table 2: Analysis of Generation Quality and Efficiency of LogicEvolve on CLUB.

| Puzzle Name | Generator Version | Time/h | Result | Solver Version | Time/h | Result | Human Time/h | Efficiency |
|---|---|---|---|---|---|---|---|---|
| train_game_card_puzzle | 0 | 0.11 | ✓ | 1 | 0.17 | ✓ | 12 | 4408.16% |
| code | 1 | 0.08 | ✓ | 0 | 0.18 | ✓ | 10 | 3854.39% |
| line_forming_game | 1 | 0.15 | ✓ | 0 | 0.01 | ✓ | 5 | 3174.6% |
| extensible_zebra | 0 | 0.01 | ✓ | 3 | 0.28 | ✓ | 8 | 2755.98% |
| maze | 7 | 0.29 | ✓ | 0 | 0.06 | ✓ | 8 | 2285.71% |
| tetris | 3 | 0.35 | ✓ | 0 | 0.22 | ✓ | 8 | 1409.0% |
| string | 4 | 0.34 | ✓ | 2 | 0.42 | ✓ | 10 | 1315.79% |
| traditional_zebra | 3 | 1.1 | ✓ | 0 | 0.01 | ✓ | 8 | 722.17% |
| knights_and_knaves | 4 | 0.2 | ✓ | 14 | 1.22 | ✓ | 3 | 211.56% |
| sudoku | 0 | 2.04 | ✓ | 0 | 0.05 | ✓ | 3 | 143.73% |

**Ablation Study.** In Table 2, version 0 denotes cases where usable generators and solvers are obtained without any iterative refinement. The results show that not all tasks can be directly synthesized with corresponding generators and solvers, underscoring the necessity of the proposed framework. However, the framework is not universally effective: when metadata errors degrade the quality of generated components, the process may even yield unsolvable tasks. To address this, we explicitly considered stopping conditions and minimal human interventions. Further experimental analyses and detailed discussions are provided in Appendix C.9, and C.10.

## 5.2 CLUB BENCHMARK

**Basic Setup.** The evaluation in this study adopts the version of **CLUB** dated May 15, 2025, comprising ten representative categories of logical reasoning tasks, each containing 100 problems arranged in increasing order of difficulty. CLUB accuracy (Acc) requires all process fields to be correct; any mismatch yields an overall error, making Acc a measure of stepwise consistency across the reasoning chain. To improve robustness, each model is evaluated through three independent runs, and we collect their valid outputs; the final score is the mean accuracy. Additional details and process-level accuracy (Cell Acc) are provided in Appendix D.4, D.5, and D.6.

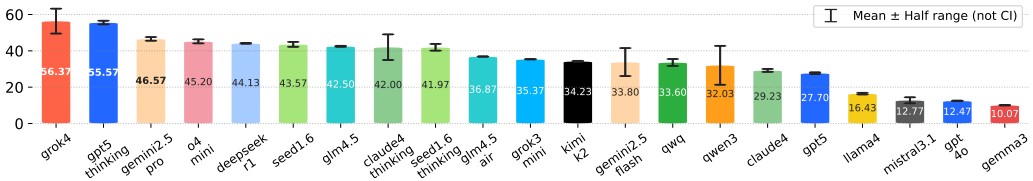

Figure 5: Ranking of frontier large language models on the **CLUB** leaderboard for logical reasoning. Scores represent the mean of three independent evaluation runs, and the error bars denote the half range (i.e., half of the difference between the maximum and minimum values) to illustrate the variability across runs. (Human baseline is 9.00, see Appendix D.3.2)

**Discriminative Power of CLUB.** The complex logical reasoning capabilities of current frontier large models remain limited, as illustrated in Figure 5. Even *Grok-4*, which ranks first with an

overall accuracy of $56.37\%_{\pm 6.85\%}$, still leaves substantial room for improvement. Although newer versions generally outperform their predecessors (e.g., *GPT-5* improves upon *GPT-4o* by 15.23%), "thinking" variants tend to surpass their base counterparts (e.g., *Claude-4-Thinking* exceeds *Claude-4* by 12.77%), and "pro" configurations often outperform "flash" ones (e.g., *Gemini-2.5-Pro* achieves a 12.77% gain over *Gemini-2.5-Flash*), models overall still exhibit pronounced weaknesses in complex logical reasoning. These findings demonstrate that **CLUB** effectively discriminates differences in models' complex logical reasoning capabilities.

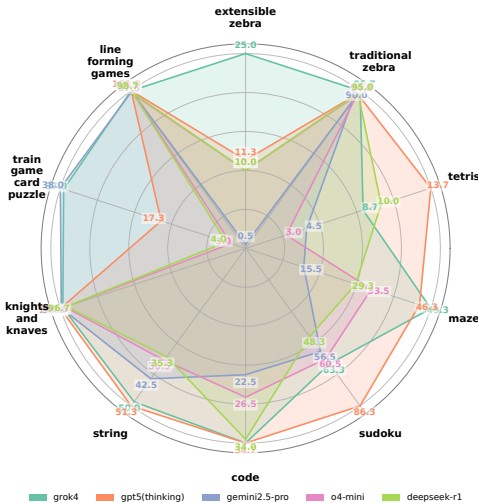

Figure 6: Evaluation results on the **Breadth** of complex logic in **CLUB**: performance across diverse tasks.

**Generalization Challenges.** Generalization in logical reasoning remains a significant challenge for large models, as their performance varies widely across different tasks. Figure 6 illustrates the performance of the top five models on individual tasks. Even the best-performing models achieve accuracies above 95% on classic reasoning tasks such as *traditional_zebra* and *knights_and_knaves*, yet they fail to exceed 25% on newly introduced tasks such as *extensible_zebra* and *tetris*. A plausible explanation is that classic tasks often have abundant open-source data or reference implementations, enabling models to receive extensive exposure during training, whereas the newly extended tasks pose unfamiliar challenges, revealing limitations in the generalization of logical reasoning capabilities.

**Curse of Complexity.** Current models still lack sufficient logical reasoning ability to cope with tasks of higher complexity. As shown in Figure 7, the top-performing model, *Grok-4*, exhibits distinct trends across different task types. For monotonic reasoning tasks such as *sudoku*, *maze*, and *string*, the accuracy decreases almost monotonically as task complexity increases. In contrast, for non-monotonic reasoning tasks such as *train_game_card_puzzle* and *extensible_zebra*, the model accuracy already shows significant degradation at low difficulty levels and overall follows a steep non-linear decline. Addressing how to maintain reliable performance under higher levels of complexity remains an open challenge for future research.

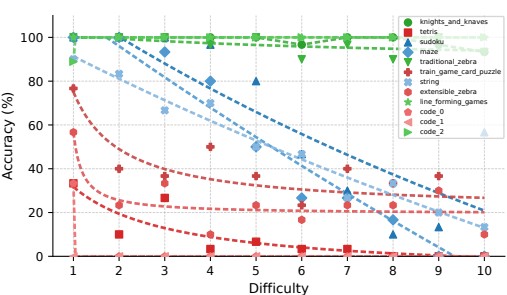

Figure 7: Evaluation results on the **Depth** of complex logic in **CLUB**: performance on tasks with varying complexity. The continuous complexity space is discretized into ten difficulty levels; for each task, the 100 problems are ranked by complexity, with every ten problems grouped into one difficulty level.

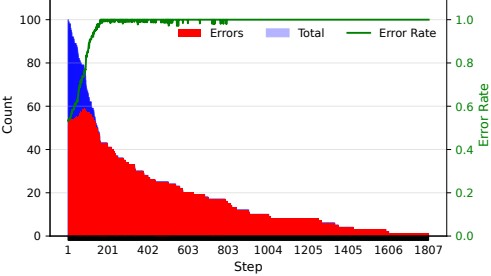

Figure 8: Evaluation results on long-range consistency in **CLUB**. Consistency refers to whether a model maintains correctness throughout the reasoning process. Monotonic reasoning tasks are particularly suitable for quantifying consistency, as their reasoning procedures are deterministic; consistency can thus be measured by computing the error rate at each step of the reasoning process.

**Challenges of Long-Range Consistency.** When faced with tasks that require a large number of reasoning steps, models often lose consistency during the reasoning process, leading to incorrect final results. As shown in Figure 8, the top-ranked model, *Grok-4*, exhibits increasing error rates across steps in the *train_game_card_puzzle* task. Within the first 200 steps, the error rate gradually rises, and beyond 200 steps, the model is almost unable to maintain consistency. These findings indicate that multi-step reasoning tasks impose higher demands on a model's ability to sustain logical consistency.

**Limitations in Fundamental Logical Reasoning Abilities.** Models struggle to maintain robust performance across different reasoning paradigms, monotonicity types, and spatial dimensions, which constitute key bottlenecks in their logical reasoning capabilities. As shown in Table 3, none of the evaluated models achieve consistently strong results on these fundamental reasoning skills. Targeted improvements on specific reasoning paradigms, monotonicity, and spatial dimensions present promising directions for future research.

Table 3: Performance of LLMs across the sub-dimensions of **CLUB**.

| Model | Paradigm | | | Monotonicity | | Dimension | | Overall |
|---|---|---|---|---|---|---|---|---|
| | Hybrid | Deduct | Abduce | Monotonic | Non-monotonic | 1D | 2D | |
| claude-sonnet4(thinking) | $59.17_{\pm2.75}$ | $30.62_{\pm7.21}$ | $87.33_{\pm17.5}$ | $31.92_{\pm8.62}$ | $48.72_{\pm6.08}$ | $42.67_{\pm9.25}$ | $41.56_{\pm6.25}$ | $48.85_{\pm8.07}$ |
| claude-sonnet3.7(thinking) | $48.33_{\pm3.25}$ | $25.57_{\pm5.72}$ | $79.67_{\pm19.5}$ | $26.58_{\pm6.25}$ | $41.5_{\pm6.84}$ | $36.42_{\pm7.0}$ | $34.95_{\pm6.34}$ | $41.86_{\pm7.84}$ |
| claude-sonnet4 | $48.5_{\pm0.5}$ | $19.95_{\pm1.5}$ | $55.67_{\pm1.5}$ | $17.33_{\pm1.25}$ | $37.16_{\pm0.75}$ | $23.83_{\pm0.62}$ | $32.84_{\pm1.5}$ | $33.61_{\pm0.47}$ |
| claude-sonnet3.7 | $33.33_{\pm1.25}$ | $15.86_{\pm0.36}$ | $49.67_{\pm2.0}$ | $15.33_{\pm0.5}$ | $27.67_{\pm0.84}$ | $21.83_{\pm1.38}$ | $23.33_{\pm0.34}$ | $26.72_{\pm0.9}$ |
| deepseek-r1 | $51.33_{\pm0.5}$ | $34.57_{\pm0.21}$ | $96.67_{\pm0.5}$ | $36.75_{\pm0.88}$ | $49.05_{\pm0.25}$ | $42.5_{\pm1.0}$ | $45.22_{\pm0.58}$ | $50.87_{\pm0.2}$ |
| deepseek3.1 | $45.5_{\pm1.25}$ | $20.95_{\pm0.71}$ | $72.33_{\pm2.0}$ | $18.42_{\pm1.88}$ | $39.39_{\pm1.0}$ | $29.83_{\pm1.38}$ | $31.78_{\pm0.66}$ | $36.89_{\pm0.27}$ |
| deepseek-chat | $39.67_{\pm7.75}$ | $17.24_{\pm3.28}$ | $63.33_{\pm20.5}$ | $15.17_{\pm3.12}$ | $33.78_{\pm8.0}$ | $26.58_{\pm7.25}$ | $26.17_{\pm4.84}$ | $31.7_{\pm7.52}$ |
| doubao1.6 | $51.17_{\pm1.75}$ | $34.29_{\pm1.57}$ | $93.33_{\pm1.0}$ | $35.42_{\pm1.88}$ | $49.0_{\pm1.0}$ | $40.67_{\pm1.0}$ | $45.5_{\pm1.67}$ | $49.91_{\pm1.12}$ |
| doubao1.5-pro(thinking) | $50.67_{\pm0.75}$ | $31.86_{\pm1.43}$ | $90.33_{\pm3.5}$ | $32.58_{\pm1.38}$ | $47.39_{\pm1.34}$ | $38.5_{\pm1.62}$ | $43.44_{\pm1.34}$ | $47.82_{\pm1.46}$ |
| doubao1.6(thinking) | $50.67_{\pm0.75}$ | $33.57_{\pm3.07}$ | $83.33_{\pm3.0}$ | $34.58_{\pm3.88}$ | $46.89_{\pm0.5}$ | $37.58_{\pm1.12}$ | $44.89_{\pm2.33}$ | $47.36_{\pm1.2}$ |
| doubao1.6-flash | $42.33_{\pm0.25}$ | $11.76_{\pm1.57}$ | $53.33_{\pm4.5}$ | $10.08_{\pm2.25}$ | $30.0_{\pm1.16}$ | $16.92_{\pm1.88}$ | $25.45_{\pm1.41}$ | $27.12_{\pm1.86}$ |
| doubao1.5-pro-32k | $32.0_{\pm0.75}$ | $10.43_{\pm0.21}$ | $31.33_{\pm3.0}$ | $8.75_{\pm0.62}$ | $22.22_{\pm0.5}$ | $12.33_{\pm1.5}$ | $19.83_{\pm0.92}$ | $19.56_{\pm0.58}$ |
| gemini2.5-pro | $68.75_{\pm0.75}$ | $33.14_{\pm1.57}$ | $99.5_{\pm0.5}$ | $34.25_{\pm1.75}$ | $55.33_{\pm0.5}$ | $50.62_{\pm1.62}$ | $44.42_{\pm0.59}$ | $55.14_{\pm0.82}$ |
| gemini2.5-flash | $50.33_{\pm7.5}$ | $23.33_{\pm3.64}$ | $74.0_{\pm37.0}$ | $23.25_{\pm3.25}$ | $40.83_{\pm10.66}$ | $32.92_{\pm11.38}$ | $34.39_{\pm5.5}$ | $39.86_{\pm11.15}$ |
| gemma3 | $18.33_{\pm0.25}$ | $6.0_{\pm0.5}$ | $22.0_{\pm3.0}$ | $4.33_{\pm0.25}$ | $13.89_{\pm0.08}$ | $7.25_{\pm0.88}$ | $11.94_{\pm0.66}$ | $11.96_{\pm0.38}$ |
| glm4.5 | $53.0_{\pm0.5}$ | $31.81_{\pm0.21}$ | $96.33_{\pm2.5}$ | $33.42_{\pm0.38}$ | $48.56_{\pm0.5}$ | $42.33_{\pm1.38}$ | $42.61_{\pm0.58}$ | $49.72_{\pm0.56}$ |
| glm4.5-air | $47.67_{\pm2.25}$ | $25.76_{\pm0.43}$ | $93.0_{\pm3.2}$ | $25.5_{\pm1.38}$ | $44.44_{\pm0.91}$ | $36.67_{\pm1.38}$ | $37.0_{\pm0.91}$ | $44.29_{\pm0.19}$ |
| glm-z1 | $30.0_{\pm0.0}$ | $21.29_{\pm0.0}$ | $78.0_{\pm0.0}$ | $21.0_{\pm0.0}$ | $33.83_{\pm0.0}$ | $31.5_{\pm0.0}$ | $26.83_{\pm0.0}$ | $34.64_{\pm0.0}$ |
| glm4 | $33.0_{\pm0.0}$ | $11.0_{\pm0.0}$ | $41.0_{\pm0.0}$ | $6.75_{\pm0.0}$ | $26.17_{\pm0.0}$ | $12.5_{\pm0.0}$ | $22.33_{\pm0.0}$ | $21.82_{\pm0.0}$ |
| gpt5(thinking) | $58.67_{\pm2.0}$ | $48.33_{\pm1.0}$ | $100.0_{\pm0.0}$ | $54.67_{\pm1.38}$ | $56.17_{\pm0.91}$ | $50.83_{\pm1.38}$ | $58.72_{\pm0.83}$ | $61.06_{\pm0.95}$ |
| gpt-o4-mini | $50.25_{\pm1.25}$ | $36.29_{\pm1.0}$ | $97.5_{\pm1.5}$ | $39.25_{\pm1.5}$ | $49.16_{\pm0.84}$ | $40.88_{\pm1.38}$ | $48.08_{\pm0.91}$ | $51.63_{\pm1.2}$ |
| gpt-oss-120b | $47.33_{\pm2.75}$ | $31.86_{\pm0.36}$ | $90.0_{\pm4.0}$ | $35.33_{\pm1.25}$ | $44.39_{\pm2.0}$ | $36.42_{\pm0.88}$ | $43.67_{\pm0.92}$ | $47.0_{\pm1.21}$ |
| gpt-oss-20b | $49.33_{\pm0.5}$ | $28.86_{\pm1.28}$ | $69.33_{\pm2.0}$ | $31.92_{\pm0.88}$ | $40.39_{\pm0.58}$ | $29.67_{\pm0.25}$ | $41.89_{\pm1.0}$ | $41.62_{\pm0.29}$ |
| gpt5 | $45.0_{\pm0.5}$ | $18.28_{\pm0.57}$ | $59.0_{\pm4.5}$ | $17.92_{\pm1.0}$ | $34.22_{\pm0.59}$ | $26.75_{\pm1.75}$ | $28.33_{\pm0.41}$ | $32.79_{\pm0.98}$ |
| gpt4.1 | $41.67_{\pm1.0}$ | $17.0_{\pm0.5}$ | $40.0_{\pm3.0}$ | $14.67_{\pm0.5}$ | $30.61_{\pm1.09}$ | $19.25_{\pm1.5}$ | $27.55_{\pm0.41}$ | $27.25_{\pm1.14}$ |
| gpt4o | $29.0_{\pm1.25}$ | $7.81_{\pm0.5}$ | $12.0_{\pm2.0}$ | $6.08_{\pm0.5}$ | $16.72_{\pm0.16}$ | $5.92_{\pm0.38}$ | $16.83_{\pm0.42}$ | $13.48_{\pm0.29}$ |
| grok4 | $68.67_{\pm4.5}$ | $46.81_{\pm8.72}$ | $98.67_{\pm1.5}$ | $49.33_{\pm7.0}$ | $61.05_{\pm6.75}$ | $55.17_{\pm3.88}$ | $57.17_{\pm8.84}$ | $62.41_{\pm5.46}$ |
| grok3-mini | $50.67_{\pm0.25}$ | $23.95_{\pm0.29}$ | $84.67_{\pm2.5}$ | $22.83_{\pm0.62}$ | $43.72_{\pm0.25}$ | $33.75_{\pm0.38}$ | $36.45_{\pm0.25}$ | $42.29_{\pm0.29}$ |
| kimi-k2 | $46.17_{\pm0.75}$ | $23.81_{\pm0.29}$ | $83.33_{\pm3.5}$ | $19.58_{\pm1.12}$ | $44.0_{\pm1.0}$ | $28.92_{\pm0.25}$ | $37.78_{\pm0.25}$ | $40.51_{\pm0.62}$ |
| llama4 | $30.67_{\pm2.0}$ | $10.05_{\pm1.0}$ | $32.67_{\pm1.5}$ | $8.58_{\pm1.0}$ | $21.67_{\pm0.16}$ | $12.42_{\pm1.12}$ | $19.11_{\pm0.33}$ | $19.31_{\pm0.38}$ |
| mistral3.1-small | $29.0_{\pm0.5}$ | $6.29_{\pm1.78}$ | $25.67_{\pm8.5}$ | $3.08_{\pm0.12}$ | $19.22_{\pm2.66}$ | $8.83_{\pm2.38}$ | $15.39_{\pm2.0}$ | $15.35_{\pm2.27}$ |
| qwen3-235b | $45.0_{\pm8.25}$ | $22.19_{\pm8.72}$ | $75.0_{\pm30.5}$ | $21.33_{\pm9.5}$ | $39.17_{\pm11.58}$ | $29.67_{\pm12.12}$ | $33.61_{\pm9.75}$ | $37.99_{\pm12.76}$ |
| qwen3-32b | $44.5_{\pm10.0}$ | $22.76_{\pm10.14}$ | $71.67_{\pm38.0}$ | $21.92_{\pm10.62}$ | $38.72_{\pm14.58}$ | $28.0_{\pm14.0}$ | $34.66_{\pm12.0}$ | $37.46_{\pm15.5}$ |
| qwq | $47.33_{\pm3.0}$ | $21.38_{\pm1.14}$ | $91.67_{\pm7.0}$ | $18.33_{\pm1.5}$ | $43.78_{\pm2.58}$ | $32.75_{\pm1.88}$ | $34.17_{\pm1.91}$ | $41.34_{\pm2.56}$ |

# 6 FURTHER ANALYSIS

**CLUB exhibits stronger discriminative power than existing logical reasoning benchmarks.** To assess CLUB's ability to distinguish different model capabilities, we evaluate the five best-performing CLUB models across six representative logical reasoning benchmarks, including LogiQA2.0 (Liu et al., 2023), BIG-Bench Hard (Suzgun et al., 2023), ARC-AGI (Chollet et al., 2024), LogicGame (Gui et al., 2024), ZebraLogic (Lin et al., 2025) and SynLogic (Liu et al., 2025b). All evaluations are conducted within the unified LogicEvolve framework, ensuring consistency and reproducibility. As shown in Figure 9, earlier datasets such as LogiQA2.0 and Zebra are nearly saturated and offer limited discrimination; high-difficulty datasets like BIG-Bench Hard and ARC-AGI reveal performance gaps but focus on narrow single-mode reasoning. LogicGame improves task diversity but still lacks the structural breadth and complexity covered by CLUB. In contrast, CLUB provides a wider performance spread and more stable separation among top models, demonstrating substantially higher discriminative capability. A more detailed correlation analysis of model rankings across various benchmarks can be found in Appendix D.7.

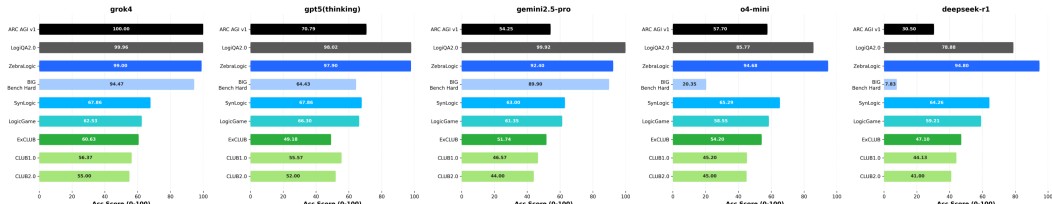

Figure 9: The performance of five top models on different logical reasoning benchmarks.

**The evolutionary mechanism of CLUB introduces measurable and substantive changes in model differentiation.** To verify that benchmark evolution yields genuine changes in model behavior rather than minor score perturbations, we evolve CLUB1.0 (i.e., the version referred to above as CLUB) along two axes: structural evolution (diversity expansion) to produce ExCLUB by increasing the number of tasks from 10 to 100, and difficulty evolution (complexity increase) to produce CLUB2.0 by advancing task hyperparameters (e.g., grids from $3 \times 3$ to $3 \times 4$). Evaluated on the same five models, the results in Figure 9 show clear shifts in ranking patterns. Under difficulty evolution, *Gemini-2.5-Pro* drops by 2.57%, far exceeding the small decline of *o4-mini* (0.2%), revealing sensitivity to increased structural complexity. Under structural evolution, the robustness of *GPT-5-Thinking* degrades relative to *o4-mini* and *Gemini-2.5-Pro*, exposing weaknesses that are invisible in CLUB1.0. For more relevant analyses, please refer to Appendix D.8. These consistent changes across evolution modes confirm that CLUB's evolutionary mechanism introduces new and meaningful distinctions in ranking, stability, and failure modes, enabling CLUB to serve as a sustainable next-generation benchmark for logical reasoning.

**With optional tool use, most models achieve noticeable performance gains, yet the overall ranking structure remains largely stable.** Leveraging the code-generated nature of CLUB, we construct a tool-augmented variant, **CLUB(Tool)**, which keeps all data unchanged and only allows models to optionally invoke a code interpreter during inference. As shown in Table 4, evaluating the top-performing models under "no tools" vs. "tool-enabled" settings reveals that models benefit substantially from tool use. Nevertheless, the relative ordering of models remains almost identical to the original CLUB results. This indicates that tool use improves absolute performance but does not fundamentally alter the comparative capability landscape, positioning CLUB(Tool) as a meaningful complement—rather than a replacement—to the original benchmark.

Table 4: Performance comparison between CLUB and CLUB(Tool) with optional code execution.

| Bench \ Model | Grok-4 | GPT-5-Thinking | Gemini-2.5-Pro | o4-mini | Deepseek-R1 |
|---|---|---|---|---|---|
| CLUB | 56.37 | 55.57 | 46.57 | 45.20 | 44.13 |
| CLUB(Tool) | 63.37 | 63.70 | 54.39 | 51.53 | 45.90 |

## 7 CONCLUSIONS

To automatically generate dynamic and sustainable logical-reasoning benchmarks, we proposed LogicEvolve, a multi-agent, highly automated framework for generating, parsing, and evaluating parameterizable, rule-based, and verifiable symbolic tasks. Building on this framework, we developed the CLUB benchmark. The average over three independent runs on more than 30 state-of-the-art language models shows that, while recent systems (e.g., Grok-4) achieve nearly 96.67% accuracy on classic grid puzzles, their performance drops to 8.67% on tetris tasks, with an overall mean accuracy of only 56.37%. These findings reveal persistent bottlenecks in complex logical reasoning. Moving forward, as discussed in Appendix A, we plan to continuously explore the scalability of LogicEvolve in open domains and dynamic interactive tasks, and continue to maintain and iteratively expand CLUB via LogicEvolve, providing a continuously updated benchmark for assessing and improving the reasoning abilities of large language models.

## ETHICAL STATEMENT

In developing the **LogicEvolve** framework and the **CLUB** dataset, we have consistently adhered to research ethics and academic integrity, with attention to data provenance and privacy, fairness and bias mitigation, transparency and reproducibility, as well as potential risks and compliance. All data used to construct tasks and examples originate from publicly accessible or self-synthesized sources and contain no personally identifiable information; we comply with relevant privacy regulations and general data-protection principles to ensure that no sensitive information is disclosed. Randomization and constraint mechanisms are incorporated in task generation and evaluation design to reduce potential social bias, and evaluation samples are reviewed to avoid inappropriate content or discriminatory expressions. We release benchmark definitions, generation protocols, and evaluation pipelines to support external verification and continuous improvement, promoting fair comparison and responsible use within the community. The **CLUB** dataset and **LogicEvolve** framework do not include sensitive prediction tasks targeting individuals or groups, involve no human-subject experiments, and are not intended for high-risk decision-making scenarios. We comply with applicable laws and regulations and maintain ongoing ethical self-assessment. As the benchmark evolves and extends to multimodal settings, we recognize the importance of continued ethical oversight and will update review and governance measures as necessary to ensure the long-term sustainability and responsible development of this research resource.

## REPRODUCIBILITY STATEMENT

We have carefully considered reproducibility throughout all stages of this work (see Appendix C.6.2, C.6.6 and D.2). The full implementation of the LogicEvolve framework, the complete set of CLUB tasks and solutions, as well as the scripts for task generation and evaluation, are documented in Appendix C and D, and in the supplementary material, and will be released in a public repository after the anonymity period. Details of task synthesis and instance filtering are provided in the main methodology section, while all evaluation metrics and experimental settings are specified in Appendix D.1. For closed-source APIs, we report the exact version identifiers used at the time of evaluation. We acknowledge that commercial models may evolve, which may limit exact reproducibility; however, we preserve call scripts and response logs to mitigate this effect. By releasing data, code, and comprehensive documentation, we aim to enable independent verification of our results and support further research in this domain.

## USE OF LLMS

During the preparation of this manuscript, we used large language models (e.g., ChatGPT) for language refinement and wording improvements in certain paragraphs. However, they did not contribute to the research conception, methodological design, experimental implementation, or result analysis, nor did they provide any new technical insights or conclusions. All academic contributions, experiments, and arguments were independently developed and conducted by the authors.

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

# Appendix

# A    LIMITATIONS

Despite the exploratory advances reported here, our work still has several limitations that warrant further study:

1. Lack of a unified "gold-standard" evaluation for logical reasoning. The community has yet to converge on a generally accepted metric suite. Most existing benchmarks—including our own CLUB—focus on task-level accuracy, which obscures a model's mastery of core principles such as transitivity, contraposition, or the law of excluded middle. Future efforts should pursue more explanatory, extensible indicators that expose fine-grained performance along distinct logical dimensions.

2. Insufficient validation of data utility and limited training-level experiments. Constrained by space and compute budget, we emphasized framework design and static evaluation. Although preliminary results show that LogicEvolve yields diverse, high-quality data, we have not yet run large-scale training studies to quantify gains in supervised fine-tuning, reinforcement learning, or distillation settings—especially with respect to emerging scaling-law behaviors.

3. Under-characterization of structural hallucinations in complex reasoning. Prior work highlights "Potemkin reasoning," where models give surface-correct answers yet violate semantic consistency or hypothesis coherence (Mancoridis et al., 2025). This indicates fundamental weaknesses in reasoning chains. Future benchmarks should incorporate a "Potemkin rate" and adversarial contrast sets to delineate hallucination boundaries.

4. Limitations of static text–only evaluation. Recent studies argue that reasoning cliffs may reflect the constraints of static, pure-text testing rather than inherent model deficits (Khan et al., 2025). When deprived of tools, external memory, or metacognitive interfaces, models are systematically underestimated. Richer, interactive environments—with agent APIs and real-time feedback—are needed to measure tool-augmented reasoning capacity.

5. Incomplete modeling of task complexity. Current indicators (e.g., combinatorial depth, minimal step count) blur the line between mechanical execution and cognitive difficulty (Opus & Lawsen, 2025). Moreover, some tasks are semantically unsatisfiable, yet the framework may treat a model's correct "no-solution" detection as failure. Future work should embed solvability checks and adopt more interpretable complexity metrics to avoid misattribution and underestimation.

6. Not currently applicable to open domains and dynamic interactive tasks. LogicEvolve is designed for deterministic, rule-based symbolic puzzles whose solutions can be verified by executable solvers. As such, the framework does not apply to open-domain reasoning tasks that depend on implicit world knowledge, natural-language semantics, or multi-hop commonsense inference, where metadata abstraction and programmatic verification are unreliable. It also does not address interactive or adversarial environments (e.g., full board-game planning or code-based tool use), as these require dynamic feedback, optimal policy search, or opponent modeling. For board-game–derived tasks included in our benchmark, we pre-simulate all stochastic or adversarial components and convert each instance into a static puzzle with a unique, verifiable solution. This ensures compatibility with our closed-loop generator–solver design but should not be interpreted as solving general game-playing or strategic planning. Future work may explore incorporating external knowledge or constrained interaction models while retaining verifiable task structures.

In sum, while CLUB and LogicEvolve provide a first practical paradigm for systematic evaluation and efficient generation of complex logical-reasoning tasks, significant challenges remain in measuring essential reasoning skills, verifying training utility, and modeling the limits of task complexity—challenges that we hope subsequent research will address.

# B COMPLEX LOGIC

## B.1 CLASSIFICATION

**Reasoning Paradigms.** From the perspective of logical reasoning paradigms (Peirce), complex logic can be classified into three primary forms: **deduction**, **induction**, and **abduction**. **Deduction** derives necessarily valid conclusions from general principles or rules. **Induction** generalizes probabilistic regularities from a collection of concrete instances, yielding conclusions with varying degrees of certainty. **Abduction**, also known as *inference to the best explanation*, infers the most plausible cause or hypothesis based on observed phenomena.

**Dynamic Properties of Logical Reasoning.** From the perspective of the dynamic properties of logical reasoning, two categories can be distinguished: **monotonic reasoning** and **non-monotonic reasoning**. **Monotonic reasoning** preserves previously drawn conclusions when new information is introduced. In contrast, **non-monotonic reasoning** allows previously established conclusions to be retracted as additional information becomes available.

**Spatial–Temporal Dimensions of Logical Reasoning.** From the perspective of the spatial-temporal dimension of logical reasoning, tasks can be classified into several categories: (1) 1D (sequential) reasoning: involves purely linear inference along a single chain of "sentences" or "events." Examples include reading comprehension, where questions are answered by following the causal order of sentences. (2) 2D (planar) reasoning: considers layouts on a plane—such as text lines, tables, or diagrams—and requires integrating horizontal and vertical relationships, e.g., table reasoning (deriving conclusions from row–column interactions) or document layout analysis. (3) 3D (spatial) reasoning: focuses on the position and structure of objects in space, such as solving geometry problems, determining object orientations, or verifying 3D models (e.g., identifying the cross-section of a prism or estimating 3D poses in computer vision). (4) XD (high-dimensional) reasoning: involves inference over higher-dimensional data or feature spaces, commonly encountered in machine learning (e.g., high-dimensional feature classification, tensor reasoning, or multi-relational embeddings in knowledge graphs).

**Summary of Logical Reasoning Scenarios.** Figure 10 summarizes common logical reasoning scenarios and provides a fine-grained categorization along the dimensions described above.

## B.2 TASK COMPLEXITY ANALYSIS OF CLUB

In the main text, we define the sufficient conditions for classifying a task as **complex logic** as follows: structurally, when the logical expression exceeds basic propositional or first-order logic and requires second-order or higher-order logic; or, from the perspective of computational complexity, when the task belongs to NP-complete or higher classes (e.g., NP-complete, PSPACE-complete, EXPTIME-complete), primarily to guide the formulation of more challenging tasks. Nevertheless, many tasks that do not strictly meet these conditions—such as those requiring sufficiently long chains of reasoning—remain highly challenging for current large models. We therefore retain such tasks and categorize them as **"complex logic (approximate)"**. Accordingly, these tasks must be treated as special cases in our complexity analysis.

Importantly, we do not treat computational complexity as a measure of logical reasoning difficulty, nor do we assume a one-to-one correspondence between complexity classes and a model's reasoning capability. A classic counterexample is that certain NP-complete problems admit efficient greedy approximations, whose practical difficulty does not align with their formal complexity. Therefore, the role of computational complexity in both CLUB and LogicEvolve is not to define task difficulty, but rather to serve as an auxiliary engineering tool for understanding the size of the task space, constraining the task-generation range, preventing instances that are unreasonably large or trivial, and helping interpret empirical error patterns across task types. In our experiments, we observe that when a task's state space grows larger, its constraint graph becomes deeper, or its branching factor increases, the "consistency pressure" during sequential reasoning rises substantially, leading to higher model error rates—an empirical correlation rather than a theoretical necessity.

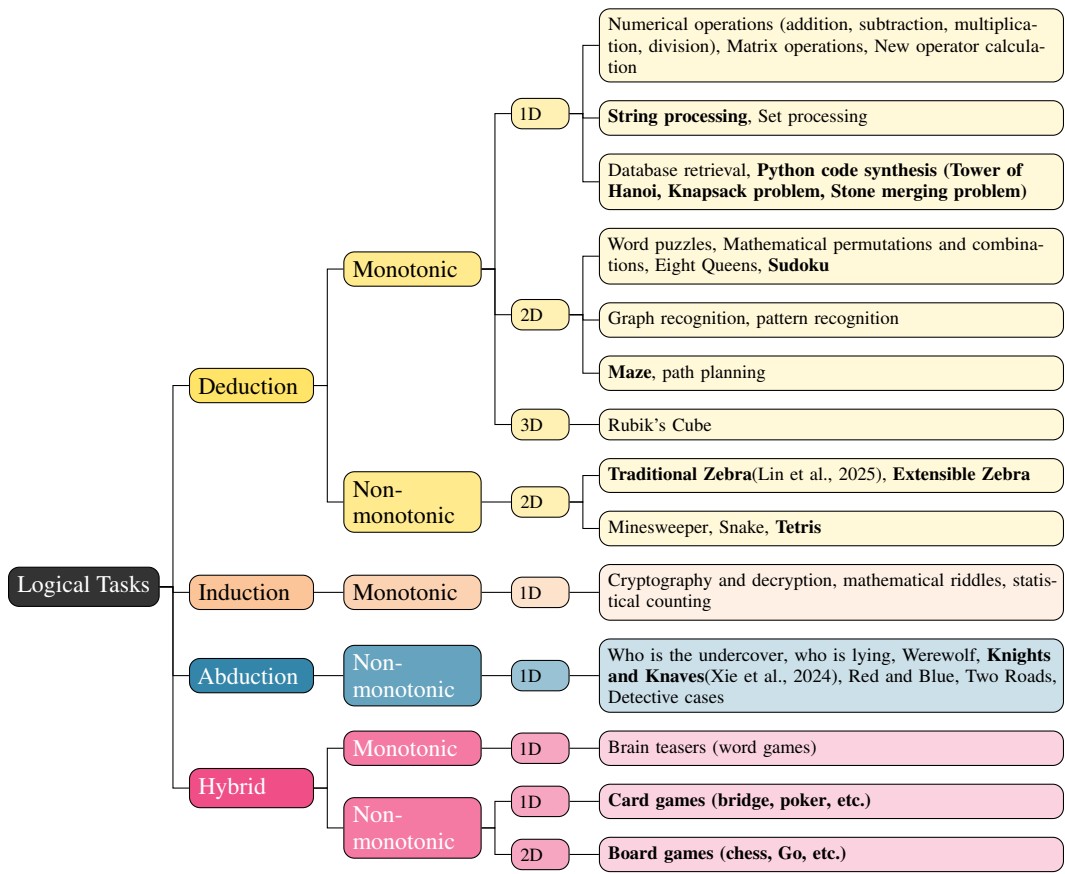

Figure 10: Classification of logical reasoning scenarios. Among them, the tasks in bold constitute the CLUB dataset, which covers 10 different categories of puzzle tasks.

Here we outline a practical methodology for conducting complexity analysis. First, the hyperparameters of the task are identified and examined. Based on these hyperparameters, the task's time complexity and space complexity are analyzed. Subsequently, the logical order required for expressing the task is considered. Finally, computational complexity and logical order are integrated to derive the overall task complexity.

Considering that the outputs of current large models are predominantly linear, and excluding the possibility of external tool usage, time complexity is estimated using the known upper bounds of worst-case algorithms—typically backtracking, exhaustive search, or full game-tree search. Space complexity is assessed with respect to the primary working memory required by search algorithms such as depth-first search or $\alpha$–$\beta$ pruning, which generally scale linearly or polynomially with problem size.

The logical order of a task is determined by the level of quantifiers and predicates permitted in its governing rules: *Propositional Logic (PL / 0th-order)* includes only complete propositions and Boolean operations, without variables, quantifiers, or explicit representation of relations among objects. *First-Order Logic (FO)* extends propositional logic by introducing individual variables and quantifiers ($\forall, \exists$), enabling statements about object properties and first-order relations among objects. *Second-Order Logic (SO)* further extends FO by allowing quantification over predicates, relations, or sets of objects, thereby capturing richer constraints. *Higher-Order Logic (HO)* refers to orders greater than 2, where quantifiers can range not only over individuals or sets, but also over higher-order functions or relations. In other words, the higher the order, the more abstract the quantifiable entities: order 0 quantifies propositions, order 1 quantifies individuals, order 2 quantifies relations or sets, and order $\geq 3$ quantifies functions or relations of relations.

Here, we take **CLUB** as a case study for complexity analysis. The detailed definitions of each task in CLUB are provided in Appendix E. We begin by analyzing the hyperparameters of different tasks, as summarized in Table 5. In general, the larger the product of the factors in the parameter space, the higher the resulting task complexity.

Table 5: Hyperparameters of each task and their relations to difficulty. In Sudoku, the "masking rate" refers to the proportion of filled cells that are hidden after all numbers are placed. In Line-forming Games, the effect of the "greedy ratio" on difficulty shows a rise-then-plateau trend: within a moderate range, higher greedy ratios increase difficulty, but beyond a certain point the effect stabilizes.

| Scenario | Task (Abstract) | Hyperparameter | Hyperparameter Values | Relation to Difficulty |
|---|---|---|---|---|
| Traditional Zebra | Grid Constraints | Number of Objects | 2–7 | Proportional |
| | | Number of Attributes | 2–7 | Proportional |
| | | Number of Logic Types | $\leq 7$ | Proportional |
| | | Number of Clues | Unlimited | Proportional |
| Extensive Zebra | Grid Constraints | Number of Objects | 1–10 | Proportional |
| | | Number of Attributes | 2–11 | Proportional |
| | | Number of Logic Types | $\leq 20$ | Proportional |
| | | Number of Clues | Unlimited | Proportional |
| Sudoku | Permutation & Combination | Grid Size | Fixed $9 \times 9$ | — |
| | | Masking Rate | 1/81–80/81 | Proportional |
| Line-forming Games | Game Decision-Making | Board Size | $3 \times 3$–$20 \times 20$ | Proportional |
| | | Connection Length | 3–20 | Proportional |
| | | Number of Tree Searches | $\geq 1$ | Proportional |
| | | Greedy Ratio | 1.0–5.0 | Proportional (Moderate) |
| Card Games | Probabilistic Inference | Number of Players | 2–20 | Proportional |
| | | Number of Card Types | 1–4 | Proportional |
| | | Cards per Type | 1–13 | Proportional |
| Knights and Knaves | Stance Analysis | Number of Objects | 2–18 | Proportional |
| | | Statement Depth | Fixed 2 | — |
| | | Statement Breadth | Fixed 2 | — |
| Tetris | Real-Time Planning | Types of Blocks | 1–7 (with rotations) | Proportional |
| | | Board Length | 4–20 | Proportional |
| | | Board Width | 4–20 | Proportional |
| Maze | Path Search | Number of Rows | 5–33 (odd) | Proportional |
| | | Number of Columns | 3–43 (odd) | Proportional |
| String Processing | Structural Transformation | Number of Operation Types | 1–14 | Proportional |
| | | Number of Operation Steps | 1–20 | Proportional |
| | | String Length | 10–50 | Proportional |
| | | Number of Reference Examples | 1–10 | Inversely Proportional |
| Python Code Synthesis | Algorithm Design | Algorithm Type | Recursion / DP / Game | — |
| | | Hyperparameters (e.g., Recursion Depth) | $1$–$+\infty$ | Proportional |

Subsequently, we analyze the time and space complexity of each task, as presented in Table 6.

Table 6: Complexity and logical order of representative logical and puzzle tasks. The classification follows the criteria: (1) tasks reaching NP-complete or above are considered "complex logic"; (2) tasks below this threshold but still challenging for LLMs are marked as "complex logic (approximate)" in the text.

| Scenario | Computational Complexity (Decision Problem) | Typical Search Time Complexity | Space Complexity | Logical Order (Expressive Power) |
|---|---|---|---|---|
| Traditional Zebra | CSP, NP-complete (some instances in P) | $O(k^m)$ | $O(m)$ | FO |
| Extensible Zebra | CSP, NP-complete (with set/second-order constraints) | $O(k^m)$ | $O(m)$ | SO |
| Sudoku | CSP, NP-complete | $O(c^{n^2})$ (backtracking) | $O(n^2)$ | FO |
| Line-Forming Games (generalized) | EXPTIME-complete (perfect-information games) | $O(b^d)$ | $O(d)$ | SO |
| Card Games (generalization) | PSPACE-hard / EXPTIME-hard (imperfect-information) | $O(b^d)$ | $O(d)$ | SO |
| Knights-and-Knaves | SAT, NP-complete | $O(2^n)$ | $O(n)$ | FO |
| Tetris | NP-hard (bounded height); PSPACE-complete (unbounded height) | $O(2^n)$ | $O(n)$ | FO |
| Mazes | P (standard); NP-complete (with keys/doors etc.) | $O(2^n)$ | $O(n)$ | FO |
| String-Processing Puzzles | NP-complete or PSPACE (depending on operations) | $O(2^k)$ | $O(k)$ | FO |
| Python Code Synthesis (Program Induction) | At least NP-complete (bounded search); harder classes for general induction | $O(2^{|P|})$ | $O(|P|)$ | SO / HO |

Here, **CSP** denotes a *Constraint Satisfaction Problem*, and **SAT** denotes a *Boolean Satisfiability Problem*. The symbols are defined as follows: $m$ represents the number of variables (e.g., the number of entities in a Zebra puzzle); $k$ represents the number of candidate values per variable (e.g., attribute values or the alphabet size); $c$ denotes the fixed size of the candidate set (e.g., $c = 9$ in Sudoku); $n$ denotes the problem size parameter (e.g., the board length $n$ in Sudoku); $b$ represents the branching factor of the search tree; $d$ denotes the depth of game-tree search; and $P$ denotes a program, corresponding to the size of the search space in program-synthesis tasks.

Specifically: (1) **String tasks.** Basic pattern matching or substring search can typically be solved in polynomial time. However, when involving complex transformations (e.g., multi-step substitutions, reversals, or chained regular-expression constraints) or requiring satisfaction of global conditions, the problem can be reduced to a Constraint Satisfaction Problem (CSP), making it NP-complete.

(2) **Code tasks.** The complexity of program-synthesis tasks depends on the target problem itself. If the target problem is NP-complete (e.g., the knapsack problem), generating code to solve it is equivalent to solving the NP-complete problem. Conversely, if the target is polynomial-time solvable (e.g., the stone-merging problem, solvable by dynamic programming), the synthesis task remains polynomial. Thus, such tasks may implicitly embed NP-complete difficulty or remain within polynomial time.

(3) **Sudoku.** Sudoku can naturally be modeled as a CSP. Prior research shows that the generalized Sudoku decision problem is NP-complete. Its logical complexity arises from multi-constraint dependencies: each cell's value is simultaneously restricted by row, column, and subgrid conditions, requiring integrated reasoning to derive a solution.

(4) **Maze.** In its standard form (finding a path from start to goal), the maze problem can be solved in polynomial time using BFS or DFS. However, with additional constraints (e.g., mandatory checkpoints, lock-and-key mechanisms, or trap avoidance), the problem becomes NP-complete.

(5) **Zebra.** A classical logical reasoning puzzle, the Zebra puzzle can be formalized as a CSP. The task requires deducing a unique solution from multi-dimensional attributes linked by clues. Because of the numerous interdependent constraints, the solution space grows exponentially, classifying it as NP-complete and exemplifying complex logic characteristics.

(6) **Tetris.** Tetris is a canonical problem in computational complexity. Research shows that with bounded height, the decision problem is NP-hard, while with unbounded height it becomes PSPACE-complete. This illustrates how extensions of game rules rapidly escalate complexity to higher classes.

(7) **Card games.** Card-based games, when modeled as multi-agent imperfect-information games, are generally PSPACE-hard or EXPTIME-hard. Even in simplified perfect-information settings, such as the "24-point" game, solving requires exploring all possible operator combinations, yielding exponential search complexity.

(8) **Line-forming games.** Line-forming games are widely recognized as complex logical games. Theoretical results show that when the board is generalized to $n \times n$, deciding whether the first player has a winning strategy is EXPTIME-complete.

(9) **Liar puzzles.** Tasks such as "Knights and Knaves," "Undercover," "Werewolf," or the "Blue/Red Eyes" puzzle involve deducing participants' identities or truthfulness from statements. These can be formalized as Boolean constraint systems, equivalent to SAT, and are therefore NP-complete in the general case. As the number of participants grows, the problem size increases exponentially.

### B.3 RELATED WORK ON COMPLEX REASONING

Different researchers have proposed varying definitions of complex reasoning. Some view it as: (1) a disruptive new capability of models (Srivastava et al., 2023); (2) a class of tasks requiring multi-step reasoning (Suzgun et al., 2023; Wu et al., 2025), involving multiple constraints (Wen et al., 2024), long-context scenarios (Zhong et al., 2025), questions that current models cannot answer correctly (Phan et al., 2025), or simultaneously assessing multimodal and multilingual skills (Guo et al., 2025), demanding slow and deliberate thinking (Li et al., 2025), and incorporating complexity arising from vivid, real-world information (Zhang et al., 2025); and (3) the ability to understand and apply supporting evidence or logical principles to draw conclusions or make decisions (Liu et al., 2025a). In this work, we attempt to formalize complex logical reasoning tasks and propose a taxonomy for their categorization.

## C LOGICEVOLVE

### C.1 METADATA CONSTRUCTION

The metadata of LogicEvolve is designed to standardize the description and generation of logical reasoning tasks. In general, metadata consists of three key components: a textual description of the puzzle, a set of tunable hyperparameters, and representative puzzle instances. Each instance is defined by three elements: the puzzle statement, the accompanying question, and the corresponding answer. Importantly, the puzzle statement must provide all known conditions, while the question is primarily used to constrain the expected output format of the answer.

Concretely, the metadata includes the following elements. **(1) Basic Rules.** A string-based description that encodes the fundamental rules of the logical puzzle. **(2) Generalization Parameters.** A list of parameterized rule options, where each parameter is represented as a JSON object containing the fields *name* (parameter name), *description* (semantic explanation), *min* (lower bound), *max* (upper bound), *step* (incremental stride), and optionally *variant* (describing special rule variants, or *None* if not applicable). **(3) Representative Puzzle Samples.** A list of sample instances, where each entry is defined by a triplet {*puzzle*, *question*, *answer*}, corresponding respectively to the puzzle conditions, the evaluation query, and the ground-truth solution.

This design provides a unified, machine-readable representation that enables automated task generation, controlled difficulty adjustment through hyperparameters, and systematic evaluation across diverse logical reasoning scenarios.

#### C.1.1 INDUCING RULES FROM INSTANCES

---

**generate rule based on induction**

ROLE DEFINITION

You are a **professional "Puzzle Meta-Rule Induction Assistant"**.
Your task is to summarize and output **one meta-rule that can guide the automated generation of puzzles of the same type**, based on the puzzle name and the materials provided.

0. PUZZLE NAME (PRIMARY INFORMATION)

```
<PUZZLE_NAME>
{puzzle_name}
</PUZZLE_NAME>
```

**Before reading any reference material, brainstorm based on the puzzle name:**

1. Use common sense to infer the core mechanics and objectives this puzzle type usually involves;
2. Draft an initial outline covering "Puzzle Introduction / Basic Rules / Required Given Conditions / Final Goal";
3. **After thinking it through**, consult the historical and new materials to refine, correct, and supplement your draft, forming the final meta-rule.

1. THE META-RULE MUST CONTAIN FOUR SECTIONS

1. **Puzzle Introduction** – 1–2 sentences summarizing the background and core mechanism.
2. **Basic Rules** – A systematic description of gameplay, permitted actions, and constraints.
3. **Required Given Conditions** – Explicitly state the minimum information that must be provided when generating a puzzle.
4. **Final Goal** – Explain what state the solver must reach for the puzzle to be considered complete.

**Requirements**

- The wording must be **detailed and unambiguous**, sufficient for a system to generate and validate all possible puzzles using only the rule.
- Edge cases and exceptions must be covered.
- Use numbering or bullet points to enhance readability.

---

2. REFERENCE MATERIALS (TO CONSULT AFTER DRAFTING)

**Historical Meta-Rule** {his_rule}

**Historical Example** {his_example}

**New Reference Rule** {reference_rule}

**New Reference Example** {reference_example}

**Additional Information** {reference_other_info}

3. DELIVERY FORMAT (MUST FOLLOW EXACTLY)

```
{
    "puzzle_rule":"<Insert the final complete meta-rule text here>"
}
```

- Output **valid JSON only**.

- Do not add any keys other than `puzzle_rule`.

- The `puzzle_rule` string may contain line breaks freely, but **do not** use back-ticks, additional JSON code blocks, or any symbols that would break the JSON structure.

### C.1.2 DEDUCING INSTANCES FROM RULES

**generate examples by deduction**

ROLE DEFINITION

You are a **Puzzle Deduction Master**, adept at crafting brand-new, coherent, and solvable puzzles within a fixed rule framework.

TASK

Using the information below, **deduce and output one brand-new puzzle of the same type**, along with a `question` that specifies the required answer format and the corresponding `answer`.

0. INPUT INFORMATION

- **Puzzle Name**
  {puzzle_name}
- **Basic Puzzle Rules**
  {puzzle_rule}
- **Example**
  {puzzle_example}
- **Additional Rules**
  {extra_rule}
- **Other Information**
  {extra_other_info}

1. GENERATION REQUIREMENTS

1. The `puzzle` **must** include all given conditions so that the solution is uniquely determined under the rules.

2. The `question` **only** describes how the answer should be formatted and must not repeat known conditions.

3. The new puzzle should differ sufficiently from the example, showcasing diversity.

4. If the answer can be directly deduced, fill it in `answer`; otherwise set `answer` to `null`.

5. Wording should be concise, clear, and unambiguous.

---

2. OUTPUT FORMAT (STRICTLY FOLLOW)

```json
{
  "puzzle": "<complete puzzle description>",
  "question": "<answer format specification>",
  "answer": <Any | null>
}
```

- Output valid JSON only.
- Do not add any keys other than puzzle, question, and answer.
- Do not wrap the output in code fences or explanatory text.

## C.2 AUTOMATED ANALYSIS OF TASK HYPERPARAMETERS

### generate parameters by analogy

#### ROLE

You are a **"Logic Puzzle Metadata Analyst,"** specializing in identifying adjustable hyper-parameters from rules and examples.

---

#### TASK

Read the **Basic Rules**, **Examples**, and supplementary information below, then **list every tunable hyper-parameter** and output each definition in **strict JSON**.

---

#### OUTPUT REQUIREMENTS

1. **Output valid JSON only**—no BOM, comments, or extra keys.

2. Return a single array named `parameters`, where each element contains the following **required fields**:

| Field | Type | Description |
|---|---|---|
| name | string | Parameter name (English or bilingual; avoid ambiguity). |
| description | string | Meaning and purpose of the parameter; if inferred from examples, cite your assumptions or rationale. |
| min | number | Theoretical minimum value |
| max | number | Theoretical maximum value |
| step | number | Increment step |
| variant | string | Briefly describe how changing this parameter alters the rules or examples. |

3. Fill `min` / `max` / `step` **only when a definite numeric range exists**; otherwise set them to `null`.

4. Follow the template below exactly—keep field order and data types unchanged.

---

#### INPUT

- **Puzzle Name**
  {puzzle_name}

- **Basic Rules**
  {puzzle_rule}
- **Additional Rules**
  {extra_rule}
- **Examples**
  {examples}
- **Other Information**
  {extra_other_info}

---

## OUTPUT TEMPLATE (STRICTLY FOLLOW)

```
{
  "parameters": [
    {
      "name": "Parameter Name 1",
      "description": "Description 1",
      "min": MinimumValue1,
      "max": MaximumValue1,
      "step": Step1,
      "variant": "How changes to this parameter affect
                  rules/examples"
    },
    {
      "name": "Parameter Name 2",
      "description": "Description 2",
      "min": MinimumValue2,
      "max": MaximumValue2,
      "step": Step2,
      "variant": "How changes to this parameter affect
                  rules/examples"
    }
    // Add more parameters as needed using the same structure
  ]
}
```

- Only output valid JSON.
- Do not add any keys other than parameters, and do not wrap the output in additional code fences or explanatory text.

## C.3 AUTOMATIC GENERATOR AND SOLVER

### C.3.1 PROMPTS FOR GENERATORS

### generate code for generating puzzle

#### ROLE

You are the "Logic Puzzle Code Generator".

#### TASK

Based on the following "Puzzle Name", "Base Rules", "Hyper-parameters" and "Examples", generate a runnable Python function generate_and_solve(items).

- items: a dict where keys are hyper-parameter names (strings) and values are the corresponding parameter values.
- The function must return a dict with these keys:
  - puzzle (str): the generated puzzle description including all known conditions

- **question**: the puzzle's question, only used to limit the answer format, not to repeat any known conditions
- **answer**: the solution if solvable; otherwise `None`

## INPUT

- "Puzzle Name"
  `{puzzle_name}`
- "Base Rules"
  `{base_rule}`
- "Hyper-parameters"
  `{parameters}`
- "Examples"
  `{examples}`

## PREVIOUS VERSION

You have already attempted to generate code and run it. Additionally, you have attempted to verify the code, and obtained the verification result and analysis. Please review the previous implementation, its run output, and any errors to avoid repeating mistakes.

- "Previous Code"
  `{previous_code}`
- "Run Parameters, Result and Error"
  `{previous_result}`
- "Verify Result"
  `{previous_verify_result}`
- "Verify Analysis"
  `{previous_verify_analysis}`

## REQUIREMENTS

1. **Read all input first, then think, then write code.**
2. **Output only Python code**, with no extra comments or explanatory text.
3. Function signature:

```
def generate_and_solve(items: dict) -> dict:
    /"/"/"
    items: {param_name(str): param_value}
    Returns: {
        "puzzle": str, # include all known conditions
        "question": Any, #only used to limit the answer
                         format, not to repeat any known
                         conditions
        "answer": Any or None
    }
    /"/"/"
    # ... your implementation
```

4. Inside the function:
   - Dynamically construct puzzle and question from items, matching the example format.
   - If an answer can be determined, include it; otherwise return `"answer": None`.
   - Validate missing or out-of-range parameters in items, raising exceptions or returning a dict with error details.

## OUTPUT

Provide the complete `generate_and_solve` function implementation directly.

### C.3.2 PROMPTS FOR SOLVERS

#### solve with python

##### ROLE

You are a "Logic Puzzle Solver".

##### TASK

Using the "Puzzle Name", "Base Rules", "Examples" (each example has a specific puzzle and question, and may have an answer) provided below, write a **general, executable** Python function `solve_puzzle(puzzle, question)`.

- `puzzle`: the puzzle description text
- `question`: the specific question to answer (mainly used to limit the answer structure and format)
- **Return**: the answer to `question` (suitable type, such as string, list, dict, etc.)

##### PREVIOUS VERSION (REFERENCE – AVOID REPEATING MISTAKES)

You have already generated code, run it, and obtained results and/or errors. You have also attempted to verify the code, and obtained the verification result and analysis. Please review the previous implementation, its run output, and any errors to avoid repeating mistakes.

- "Previous Code"
  {previous_code}
- "Run parameters / result / error"
  {previous_result}
- "Verify Result"
  {previous_verify_result}
- "Verify Analysis"
  {previous_verify_analysis}

##### INPUT

- "Puzzle Name"
  {puzzle_name}
- "Base Rules"
  {base_rule}
- "Examples"
  {examples}

##### REQUIREMENTS

1. **Read all input, think thoroughly, then output code.**

2. **Output must be pure Python code** — no extra explanations, comments, or text.

3. The function signature and docstring must match exactly; the internal implementation is up to you:

```python
def solve_puzzle(puzzle, question):
    """
    Args:
        puzzle (str): Puzzle description text.
        question: Concrete question to answer.
    Returns:
        Any: The answer corresponding to 'question'.
    """
    # ... your implementation
```

4. The function must solve this class of puzzles in general, not just the current instance.

> 5. If input parameters are missing, malformed, or the puzzle is unsolvable, raise a clear exception or return an informative error.
>
> 6. Consult the Previous Code and its error information to avoid repeating similar bugs.
>
> ### OUTPUT
>
> Output only the complete, runnable implementation of `solve_puzzle`.

### C.3.3 COLLABORATIVE OPTIMIZATION MODES

When optimization opportunities arise, the specific improvements may occur in either the generator or the solver, yielding four typical patterns, as illustrated in Figure 11.

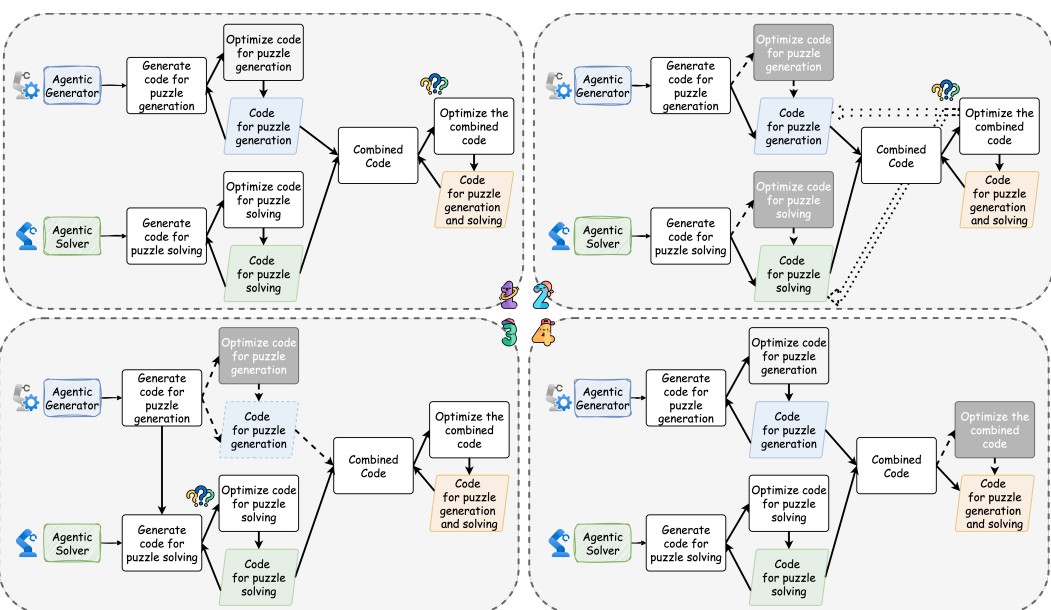

Figure 11: The collaborative optimization relationship between the generator and the solver.

**Mode 1: Fully Joint Optimisation**  *Three–stage, end–to–end optimisation.* The generator and solver are first tuned *independently* until each meets its own stability and quality targets; only then are their codes coupled and jointly refined to guarantee structural optimality and semantic closure.

- **Pros**: Theoretically most complete; ensures full-chain consistency between generation and solving.
- **Cons**: Highest optimisation cost—three iterative rounds (generator → solver → joint tuning); unsuitable for rapid-iteration scenarios.

**Mode 2: Post-hoc Joint Optimisation**  *One-shot concatenation followed by global tuning.* The generator and solver produce initial code independently, which is then directly stitched together and globally optimised, skipping any intermediate fine-tuning of the individual modules.

- **Pros**: Minimal implementation cost; ideal for automatic construction and quick deployment.
- **Cons**: If either module is flawed, error attribution becomes ambiguous, global convergence is hard, and debugging cost may skyrocket due to tight yet opaque coupling.

**Mode 3: Solver-in-Generator Embedding** *Solver first, then embedded.* The solver is optimised in isolation to ensure robustness, then injected as an internal sub-routine of the generator, followed by holistic debugging.

- **Pros**: The solver can be reused as a verifier across multiple generator configurations, mitigating solver-induced failures.
- **Cons**: Overlapping optimisation surfaces may cause regressions when solver logic changes; demands careful dependency management between modules.

**Mode 4: Solver-as-Fallback** *Solver acts purely as an error-handling component.* The generator possesses a rudimentary self-verification capability; an independent solver is invoked *only* when the generator's own attempt fails.

- **Pros**: Generator and solver are highly decoupled, enabling independent optimisation and parallel development; delivers strong robustness and extensibility.
- **Cons**: Lacks fine-grained co-optimisation, so subtle logical inconsistencies might remain undetected.

## C.4 AUTOMATED EVALUATOR

### C.4.1 EVALUATION METRICS

In logical reasoning tasks, each question typically consists of multiple blanks (cells) that must be filled. Based on this structure, we define two core evaluation metrics: **cell-level accuracy** and **sample-level accuracy**.

**Cell-Level Accuracy**: Suppose a task instance contains $M$ blanks to be filled. Let $\hat{y}_i$ denote the prediction for the $i$-th blank, and $y_i$ the corresponding ground truth. The cell-level accuracy is defined as:

$$\text{CellAcc} = \frac{1}{M} \sum_{i=1}^{M} \mathbf{1}\{\hat{y}_i = y_i\}, \tag{2}$$

where $\mathbf{1}\{\cdot\}$ is the indicator function, returning 1 if the prediction is correct and 0 otherwise.

**Sample-Level Accuracy**: For a task instance with $M$ blanks, the prediction is counted as correct only if *all* blanks are filled correctly. Formally:

$$\text{SampleAcc} = \mathbf{1}\left\{ \bigwedge_{i=1}^{M} (\hat{y}_i = y_i) \right\}. \tag{3}$$

**Overall Evaluation**: The overall evaluation across a test set $\mathcal{D}$ is given by averaging over all task instances:

$$\text{CellAcc}(\mathcal{D}) = \frac{1}{|\mathcal{D}|} \sum_{(x,y) \in \mathcal{D}} \text{CellAcc}(x, y), \tag{4}$$

$$\text{SampleAcc}(\mathcal{D}) = \frac{1}{|\mathcal{D}|} \sum_{(x,y) \in \mathcal{D}} \text{SampleAcc}(x, y). \tag{5}$$

### C.4.2 EVALUATION ALGORITHMS

Algorithm 1 presents a streamlined automated logic evaluation pipeline. Given a QA dataset with gold answers and configuration parameters, we process samples in batches and execute inference in parallel. For each question, we assemble a unified evaluation prompt that includes task instructions, a small set of exemplars, and a strict JSON schema describing the required output. The language

---

**Algorithm 1** Automated Logic Evaluation Pipeline

---

**Require:** QA dataset with gold answers $\mathcal{D} = \{(q_i, A_i)\}_{i=1}^N$; configuration $\Theta$ (model, batch size $B$, concurrency $P$, decoding parameters); JSON schema $S$ and a small set of in-context examples $E$

**Ensure:** Evaluation records $\mathcal{R}$; cell-level accuracy CellAcc; sample-level accuracy Acc

1: $\mathcal{R} \leftarrow \emptyset$; $C \leftarrow 0$ (correct cells); $T \leftarrow 0$ (total cells); $S_{\text{ok}} \leftarrow 0$ (fully-correct samples)
2: **for** $b = 1$ **to** $\lceil N/B \rceil$ **do**
3:     Take current batch $\mathcal{B} \subset \mathcal{D}$ of $B$ pairs $(q_i, A_i)$
4:     Construct a unified evaluation **prompt** $\Pi$: *task description + I/O protocol + JSON schema $S$ + exemplars $E$ + strict output constraints (return only thought and prediction, where prediction conforms to $S$)*
5:     **Concurrently** (up to $P$ workers) query $\Theta$.model with $\langle \Pi, q_i \rangle$, obtaining $o_i = \{thought, \hat{A}_i\}$
6:     **for** each $(q_i, A_i, o_i)$ in the batch **do**
7:         Parse $\hat{A}_i$ as JSON (count as incorrect if parsing fails)
8:         Let $K_i$ be the set of cell keys; $T \leftarrow T + |K_i|$
9:         $c_i \leftarrow \sum_{k \in K_i} \mathbf{1}\big[\hat{A}_i[k] = A_i[k]\big]$; $C \leftarrow C + c_i$
10:         $s_i \leftarrow \mathbf{1}\big[c_i = |K_i|\big]$; $S_{\text{ok}} \leftarrow S_{\text{ok}} + s_i$
11:         Append $(i, q_i, A_i, o_i.thought, \hat{A}_i, c_i, s_i)$ to $\mathcal{R}$
12:     **end for**
13: **end for**
14: CellAcc $\leftarrow C/T$;    Acc $\leftarrow S_{\text{ok}}/N$
15: **return** $(\mathcal{R}, \text{CellAcc}, \text{Acc})$

---

model is required to return both a human-readable reasoning trace (thought) and a machine-parsable prediction (prediction).

During evaluation, the prediction is parsed and compared to the reference answer on a cell-wise basis to accumulate the number of correct versus total cells, yielding the cell-level accuracy (CellAcc). A sample is considered correct only if all its cells match, which is used to compute the sample-level accuracy (Acc). The pipeline completes inference and evaluation in a single pass, while the batch size and concurrency degree control throughput. The use of a unified JSON contract with strict output constraints guarantees robust downstream parsing and accurate statistics.

### C.4.3  CONSIDERATIONS ON FEW-SHOT AND FORMATTING

In our evaluation protocol, models are required to return outputs in a strictly predefined format to ensure reliable and automated assessment. Hence, the enforcement of output formatting is stringent.

To further enhance instruction-following ability, we provide a *one-shot* example for each evaluation. This example demonstrates the correct output format, ensuring that the model understands the required structure. However, no additional examples are provided, so as to avoid inadvertently influencing the model or enabling it to learn task-specific solving strategies beyond formatting compliance.

### C.4.4  EVALUATION EXAMPLES

To illustrate the evaluation pipeline, we present a complete example on a Sudoku task. The process consists of three steps: providing the original data, constructing the evaluation prompt, and obtaining the model output with its corresponding evaluation results.

First, the original task data is shown below:

**example data**

**Data Summary**  From `example_9*9_13_81.jsonl` (ID: 167). A 9×9 Sudoku with 68 givens and 13 masked cells; the masking rate is mask_rate $= 13/81 \approx 0.1604938$.

**Prompt (Puzzle)**

| _ | 4 | 1 | 2 | 5 | 7 | 9 | 3 | 8 |
|---|---|---|---|---|---|---|---|---|
| 8 | 9 | 2 | 1 | 3 | 6 | 5 | 4 | _ |
| 3 | 5 | 7 | _ | 9 | _ | 2 | 6 | 1 |
| _ | 6 | 5 | 8 | 7 | 9 | 1 | 2 | 3 |
| 7 | 3 | 8 | 6 | 1 | _ | 4 | 5 | 9 |
| 2 | 1 | 9 | 3 | _ | 5 | 8 | 7 | 6 |
| 1 | 2 | 4 | 9 | _ | _ | 7 | 8 | 5 |
| _ | 8 | _ | 7 | 2 | 1 | 3 | 9 | 4 |
| 9 | 7 | 3 | _ | 8 | 4 | 6 | 1 | _ |

**Gold Answer**

| 6 | 4 | 1 | 2 | 5 | 7 | 9 | 3 | 8 |
|---|---|---|---|---|---|---|---|---|
| 8 | 9 | 2 | 1 | 3 | 6 | 5 | 4 | 7 |
| 3 | 5 | 7 | 4 | 9 | 8 | 2 | 6 | 1 |
| 4 | 6 | 5 | 8 | 7 | 9 | 1 | 2 | 3 |
| 7 | 3 | 8 | 6 | 1 | 2 | 4 | 5 | 9 |
| 2 | 1 | 9 | 3 | 4 | 5 | 8 | 7 | 6 |
| 1 | 2 | 4 | 9 | 6 | 3 | 7 | 8 | 5 |
| 5 | 8 | 6 | 7 | 2 | 1 | 3 | 9 | 4 |
| 9 | 7 | 3 | 5 | 8 | 4 | 6 | 1 | 2 |

**Link to Metrics**   This instance has $M = 81$ cells (13 masked to be predicted). If the model exactly matches the gold answer:

$$\text{CellAcc} = \frac{1}{13} \sum_{i=1}^{13} \mathbf{1}\{\hat{y}_i = y_i\} = 1, \quad \text{SampleAcc} = \mathbf{1}\left\{ \bigwedge_{i=1}^{13} (\hat{y}_i = y_i) \right\} = 1.$$

If only $k$ cells are correct:

$$\text{CellAcc} = \frac{k}{13}, \quad \text{SampleAcc} = \begin{cases} 1, & k = 13, \\ 0, & k < 13. \end{cases}$$

Next, the following prompt is used to query the model:

**evaluation prompt**

```
PUZZLE_TEMPLATE
        PUZZLE_TEMPLATE = """
        {puzzle}

        ## Clues:
        {clues}
        """

SUDOKU_GRID
        SUDOKU_GRID = """
        # Example Puzzle

        Given a 9×9 Sudoku grid, fill each cell with a number
        from 1 to 9. Empty cells are represented by
        underscores _.

        {"rows":[
          [9,4,6,5,8,1,7,2,3],
          [8,3,1,6,7,2,_,5,9],
          [2,5,7,3,9,4,6,8,1],
          [6,7,4,2,1,9,5,3,8],
          [3,8,_,4,6,5,1,9,7],
          [5,1,9,7,3,8,2,6,4],
```

```
            [4,2,8,1,5,3,9,7,6],
            [1,6,3,9,2,7,8,4,5],
            [7,9,5,8,4,6,3,1,2]
        ]}

        ## Clues for the Example Puzzle

        To complete the grid, follow these three rules:

        1. Each row must contain the digits 1{9 exactly once.
        2. Each column must contain the digits 1{9 exactly once.
        3. Each 3×3 subgrid(the board is divided into nine 3×3 blocks)
        must also contain the digits 1{9 exactly once.

        ## Answer to the Example Puzzle

        {
          "reasoning": "Row 2,Column 7: The row has {8,3,1,6,7,2,5,9},
          so it is missing 4; the column has {7,4,6,5,1,2,9,8,3},
          also missing 4; the 3×3 subgrid (rows 1{3,
          columns 7{9) is missing 4 → fill in 4. Row 5,
          Column 3: The row has {3,8,4,6,5,1,9,7},
          missing 2; the column has {6,1,7,4,9,8,3,5},
          missing 2; the 3×3 subgrid (rows 4{6,
          columns 1{3) is missing 2 → fill in 2.",
          "solution": {
            "rows": [
              [9, 4, 6, 5, 8, 1, 7, 2, 3],
              [8, 3, 1, 6, 7, 2, 4, 5, 9],
              [2, 5, 7, 3, 9, 4, 6, 8, 1],
              [6, 7, 4, 2, 1, 9, 5, 3, 8],
              [3, 8, 2, 4, 6, 5, 1, 9, 7],
              [5, 1, 9, 7, 3, 8, 2, 6, 4],
              [4, 2, 8, 1, 5, 3, 9, 7, 6],
              [1, 6, 3, 9, 2, 7, 8, 4, 5],
              [7, 9, 5, 8, 4, 6, 3, 1, 2]
            ]
          }
        }

        # Puzzle to Solve

        {puzzle}

        # Instruction

        Now please solve the above puzzle.
        Present your reasoning and solution
        in the following json format:

        {json_template}

        """
```

Finally, the model's generated output and the corresponding evaluation results produced by the automated evaluation pipeline are given as follows:

## evaluation result

METADATA

| | |
|---|---|
| **did**: | 2 |
| **qid**: | 167 |
| **tid**: | 1 |
| **created_at**: | 2025-05-08T11:02:09.467099 |
| **delta_time**: | 0.007643541001016274 seconds |
| **mask_rate**: | 0.16049382716049382 |
| **raw_file**: | `../../data/sudoku/example_9*9_13_81.jsonl` |
| **success**: | true |
| **cell_accuracy**: | 1.0 |
| **cell_accuracy_num**: | 13 |
| **cell_accuracy_all**: | 13 |
| **solve_time**: | 20.784964 seconds |
| **total_solve_time**: | 20.784963935613632 |
| **total_pass_need_times**: | 1 |

PUZZLE

Given a 9×9 Sudoku grid, fill each cell with a number from 1 to 9. Empty cells are represented by underscores _.

CLUES

```
To complete the grid, follow these three rules:
1. Each row must contain the digits 1{9 exactly once.
2. Each column must contain the digits 1{9 exactly once.
3. Each 3×3 subgrid must contain the digits 1{9 exactly once.
```

QUESTION

```
{'rows': array([array([_, 4, 1, 2, 5, 7, 9, 3, 8],dtype=object),
array([8, 9, 2, 1, 3, 6, 5, 4, _], dtype=object),
array([3, 5, 7, _, 9, _, 2, 6, 1], dtype=object),
array([_, 6, 5, 8, 7, 9, 1, 2, 3], dtype=object),
array([7, 3, 8, 6, 1, _, 4, 5, 9], dtype=object),
array([2, 1, 9, 3, _, 5, 8, 7, 6], dtype=object),
array([1, 2, 4, 9, _, _, 7, 8, 5], dtype=object),
array([_, 8, _, 7, 2, 1, 3, 9, 4], dtype=object),
array([9, 7, 3, _, 8, 4, 6, 1, _], dtype=object), ],dtype=object)}
```

ANSWER

```
{'rows': array([array([6, 4, 1, 2, 5, 7, 9, 3, 8],dtype=object),
array([8, 9, 2, 1, 3, 6, 5, 4, 7], dtype=object),
array([3, 5, 7, 4, 9, 8, 2, 6, 1], dtype=object),
array([4, 6, 5, 8, 7, 9, 1, 2, 3], dtype=object),
array([7, 3, 8, 6, 1, 2, 4, 5, 9], dtype=object),
array([2, 1, 9, 3, 4, 5, 8, 7, 6], dtype=object),
array([1, 2, 4, 9, 6, 3, 7, 8, 5], dtype=object),
array([5, 8, 6, 7, 2, 1, 3, 9, 4], dtype=object),
array([9, 7, 3, 5, 8, 4, 6, 1, 2], dtype=object), ],dtype=object)}
```

LLM RAW RESPONSE

```
Looking at this Sudoku puzzle, I need to identify the empty cells
and solve them systematically using the three rules.

Let me first identify all the empty cells (underscores):
- Row 1, Column 1
- Row 2, Column 9
- Row 3, Columns 4 and 6
- Row 4, Column 1
```

```
       - Row 5, Column 6
       - Row 6, Column 5
       - Row 7, Columns 5 and 6
       - Row 8, Columns 1 and 3
       - Row 9, Columns 4 and 9

       Now I'll solve each systematically:

       Row 1, Column 1: Row 1 has {4,1,2,5,7,9,3,8}, missing 6.
       Column 1 has {8,3,7,2,1,9}, missing {4,5,6}.
       Top-left 3×3 subgrid has {4,1,8,9,2,3,5,7},
       missing 6. → 6

       Row 2, Column 9: Row 2 has {8,9,2,1,3,6,5,4}, missing 7.
       Column 9 has {8,1,3,9,6,5,4}, missing {2,7}.
       Top-right 3×3 subgrid has {9,3,8,5,4,2,6,1},
       missing 7. → 7

       Row 3, Column 4: Row 3 has {3,5,7,9,2,6,1}, missing {4,8}.
       Column 4 has {2,1,8,6,3,9,7}, missing {4,5}.
       Top-middle 3×3 subgrid has {2,5,7,1,3,6,9},
       missing {4,8}. The intersection is 4. → 4

       Row 3, Column 6: Row 3 now has {3,5,7,4,9,2,6,1}, missing 8. → 8

       Row 4, Column 1: Row 4 has {6,5,8,7,9,1,2,3}, missing 4. → 4

       Row 5, Column 6: Row 5 has {7,3,8,6,1,4,5,9}, missing 2. → 2

       Row 6, Column 5: Row 6 has {2,1,9,3,5,8,7,6}, missing 4. → 4

       Row 7, Column 5: Row 7 has {1,2,4,9,7,8,5}, missing {3,6}.
       Column 5 has {5,3,9,7,1,4,2,8}, missing 6. → 6

       Row 7, Column 6: Row 7 now has {1,2,4,9,6,7,8,5}, missing 3. → 3

       Row 8, Column 1: Row 8 has {8,7,2,1,3,9,4}, missing {5,6}.
       Column 1 now has {6,8,3,4,7,2,1,9}, missing 5. → 5

       Row 8, Column 3: Row 8 now has {5,8,7,2,1,3,9,4}, missing 6. → 6

       Row 9, Column 4: Row 9 has {9,7,3,8,4,6,1}, missing {2,5}.
       Column 4 now has {2,1,4,8,6,3,9,7}, missing 5. → 5

       Row 9, Column 9: Row 9 now has {9,7,3,5,8,4,6,1}, missing 2. → 2

       {
           "reasoning": "Solved systematically by finding cells where
           only one digit could fit based on row, column, and 3×3
           subgrid constraints. Started with Row 1, Column 1 which
           could only be 6, then Row 2, Column 9 which could only
           be 7, and continued solving each empty cell by
           elimination until all positions were filled.",
           "solution": {
               "rows": [
                   [6, 4, 1, 2, 5, 7, 9, 3, 8],
                   [8, 9, 2, 1, 3, 6, 5, 4, 7],
                   [3, 5, 7, 4, 9, 8, 2, 6, 1],
                   [4, 6, 5, 8, 7, 9, 1, 2, 3],
                   [7, 3, 8, 6, 1, 2, 4, 5, 9],
                   [2, 1, 9, 3, 4, 5, 8, 7, 6],
                   [1, 2, 4, 9, 6, 3, 7, 8, 5],
                   [5, 8, 6, 7, 2, 1, 3, 9, 4],
```

```
                    [9, 7, 3, 5, 8, 4, 6, 1, 2]
            ]
        }
    }

PARSED RESPONSE
```

PARSED RESPONSE

**response_resaon**: Solved systematically by finding cells where only one digit could fit based on row, column, and 3×3 subgrid constraints. Started with Row 1, Column 1 which could only be 6, then Row 2, Column 9 which could only be 7, and continued solving each empty cell by elimination until all positions were filled.

**response_answer**:

```
{"rows": [[6, 4, 1, 2, 5, 7, 9, 3, 8],
[8, 9, 2, 1, 3, 6, 5, 4, 7],
[3, 5, 7, 4, 9, 8, 2, 6, 1],
[4, 6, 5, 8, 7, 9, 1, 2, 3],
[7, 3, 8, 6, 1, 2, 4, 5, 9],
[2, 1, 9, 3, 4, 5, 8, 7, 6],
[1, 2, 4, 9, 6, 3, 7, 8, 5],
[5, 8, 6, 7, 2, 1, 3, 9, 4],
[9, 7, 3, 5, 8, 4, 6, 1, 2]]}
```

This example demonstrates the end-to-end workflow from task specification, model inference, to automated evaluation, highlighting the operational consistency and usability of the LogicEvolve framework.

### C.4.5 LEAKAGE PREVENTION AND BENCHMARK MAINTENANCE

To ensure the long-term validity of CLUB evaluations, LogicEvolve introduces a leakage-prevention mechanism during the data generation phase. Each generated sample is assigned a unique identifier (ID) and a generation timestamp; IDs are incremented sequentially from the first release of CLUB, while timestamps record the exact generation time. Together, these uniquely determine the provenance of each sample. When generating new data, the framework checks whether the known-condition part of a sample is identical to any historical instance. If duplication is detected, the sample is automatically discarded to avoid redundancy or leakage.

In addition, to ensure the maintainability and sustainable evolution of the benchmark, we adopt a versioning strategy based on fixed time windows rather than model-performance-driven task selection. CLUB is released on a six-month cycle, during which all tasks and instances are frozen and published as a version (e.g., CLUB v1, v2). Subsequent model evaluations may continue to use any earlier version, ensuring reproducibility of results. At the beginning of each new cycle, the framework performs only *incremental extensions* rather than any performance-based pruning. These extensions may include resampling instance sets using the existing metadata and generators to refresh data, adding new task types or difficulty levels, or increasing the difficulty of certain easy tasks, while all previously released tasks are fully preserved.

Importantly, the "data synthesizer" used for resampling is not a new external system but simply the existing LogicEvolve generator re-sampling under the same metadata; its sole purpose is to refresh instance sets to prevent memorization effects rather than introducing new task logic. We emphasize that all experiments reported in this paper are conducted on a single static, unmodified version of CLUB. Benchmark maintenance and extension are entirely based on fixed-time-window versioning, not on model performance, ensuring that CLUB remains non-adversarial, fair, and valuable for long-term research.

## C.5 EXTENDED VERIFICATION PROTOCOL FOR TASK CONSTRUCTION

This appendix provides a detailed account of the verification pipeline used in CLUB and LogicEvolve for constructing logically coherent, solvable, robust, and uniquely-answerable reasoning tasks. The goal of this section is not only to document the concrete mechanisms implemented in our system, but also to explain the underlying rationale and operational considerations that ensure the reliability of the resulting benchmark. Our design emphasizes verifiability, redundancy, and invariance, reflecting principles from both program synthesis and formal methods, while remaining fully compatible with automated LLM-based task generation.

### C.5.1 TASK CORRECTNESS VERIFICATION

The construction pipeline—metadata, generator, solver—is intentionally structured to be verifiable at every stage. This design improves transparency, reduces reliance on heuristic behaviors of LLMs, and ensures that erroneous artifacts cannot silently propagate.

**Known-task initialization.** For task types derived from established puzzles (e.g., Sudoku, Maze, logic grids), correctness is grounded in human-validated QA pairs that serve as canonical references. Before any generated task can be accepted, the solver must successfully solve these canonical instances. This acts as an initialization step that certifies solver correctness. A verified solver is then used to validate the outputs of the generator: if the generator produces instances inconsistent with the canonical rules, the solver identifies the inconsistency immediately. Through this bidirectional constraint, the generator and solver converge toward faithful implementations of the underlying puzzle semantics.

**Cross-verification for new tasks.** Newly created tasks lack pre-existing QA anchors, making correctness more challenging. We address this through cross-verification using multiple independent solvers derived from distinct reasoning paradigms. A code-execution solver checks that the rules can be deterministically executed as a procedural program. A symbolic solver treats the rules as logical constraints and verifies their internal coherence. When applicable, an external CSP, SAT, or SMT solver is used to verify the satisfiability of the constraint set. A task type is accepted only if these solvers reach consistent conclusions. This paradigm-orthogonal redundancy is central: failures in one reasoning mode often expose inconsistencies invisible to the others, enabling an emergent correctness guarantee even in the absence of pre-labeled data.

**Why this is reliable.** Cross-verification exploits the complementary strengths of different solver architectures. Code-based solvers detect illegal state transitions, symbolic solvers are sensitive to logical contradictions, and SAT/SMT solvers detect global inconsistencies or degenerate constraints. Agreement among these solvers therefore acts as a form of "logical majority vote" that substantially raises the bar for correctness compared to any single-solver approach.

### C.5.2 TASK RELIABILITY VERIFICATION

Correctness alone does not guarantee robustness: a puzzle specification may behave correctly for one phrasing of the rules but fail under slight semantic-preserving re-expressions. To detect such brittleness, we incorporate two mechanisms inspired by robustness testing in programming languages and data validation.

**Verification-only solver.** After a solver converges to a working solution, we automatically construct a verification-only version. This solver no longer performs search or reasoning but only checks that a proposed answer satisfies rule constraints, global invariants such as reachability or conservation rules, and puzzle-specific structural requirements. Because this verification solver is not tied to the reasoning strategy of the primary solver, it acts as a strategy-agnostic reference that significantly reduces the risk of reasoning-path overfitting.

**Metadata perturbation test.** The second mechanism is a semantic-invariance check. We retain the meaning of each rule but apply minor paraphrases, reorder constraints, or manipulate formatting. We then re-run the entire generator–solver pipeline. Metadata that are genuinely well-formed and

internally coherent remain solvable under such perturbations. Conversely, metadata that encode hidden inconsistencies, ambiguous semantics, or underspecified constraints frequently cause solvers to diverge or fail to converge. Empirically, this test is extremely effective: even small semantic-preserving variations often expose rules that only appear correct due to lexical artifacts of the initial phrasing.

### C.5.3 TASK SOLVABILITY VERIFICATION

Task solvability is essential for evaluating reasoning ability. Without guaranteed solvability, model failures become uninterpretable. We therefore implement a three-stage solvability pipeline.

**Metadata-level solvability constraints.**   During rule construction, the metadata agent is explicitly instructed to define task families that admit at least one solution. Contradictory or underdetermined rule sets are rejected immediately. This stage acts as a symbolic filter that eliminates many invalid task types even before instance-level generation.

**Instance-level multi-strategy solving.**   Once the generator produces concrete task instances, multiple independent strategies attempt to solve them. These include program simulation, symbolic reasoning, and search-based methods such as DFS, BFS, or constraint propagation. Because these strategies exploit fundamentally different inductive biases, consistent success across strategies provides strong evidence of solvability. If all strategies fail within the iteration budget, the instance is flagged as unsolvable and returned for repair.

**Convergence-based filtration.**   Tasks that repeatedly fail solvability verification despite multiple rounds of repair are removed from the benchmark. In ExCLUB, approximately 20% of initial prototypes fall into this category. Importantly, this filtration operates entirely automatically and requires no human intervention, demonstrating the efficacy of our solvability constraints.

### C.5.4 TASK ANSWER CORRECTNESS VERIFICATION

Ensuring correctness of answers is essential because the final dataset is used to evaluate LLM performance. We therefore implement multiple layers of answer verification that combine procedural and formal reasoning.

**Rule-based verification.**   For task types with explicit, operationally defined semantics—such as path navigation, string transformation, or grid filling—the solver checks correctness step-by-step. This eliminates answers that satisfy superficial patterns but violate deep structural constraints.

**Cross-solver consistency.**   To avoid degenerate cases where the rule-based solver accepts an incorrect answer due to implementation quirks, we require independent solvers to agree on the final output. Specifically, a code-execution solver and a symbolic solver must converge to the same solution. If they disagree, the instance is regenerated. This agreement condition is a critical safety net that dramatically reduces the possibility of undetected system-level hallucinations.

**Human spot-checking.**   A small subset of puzzles—particularly those involving hybrid constraints or non-standard rule interactions—are difficult to fully formalize. For these cases, we conduct human spot-checking. In ExCLUB, we examine 10% of instances per task type. Early iterations reveal an error rate of roughly 30%, primarily due to metadata inconsistencies or overlooked edge cases. After iterative refinement, only tasks with zero observed errors are retained.

### C.5.5 TASK UNIQUE-SOLUTION VERIFICATION

Uniqueness of solutions is vital for fair comparison: if a puzzle admits multiple solutions, different models may legitimately return distinct but correct answers. To prevent this ambiguity, we implement explicit unique-solution verification.

**Multi-run solver convergence.**   We solve each puzzle multiple times using independently initialized solver runs. If all runs converge to the same solution, the puzzle is treated as having a unique answer.

Divergence indicates that the puzzle admits multiple solutions or that solver randomness influences outcomes. Such puzzles are either regenerated or rejected.

**Finite vs. infinite solution spaces.** When a puzzle admits multiple finite solutions, we determine based on task design whether the multiplicity is acceptable. However, puzzles with infinite or non-convergent solution spaces are always rejected and returned to the generator.

**Formal uniqueness proofs.** For puzzles amenable to formal constraint representation, we encode the task using CSP, SAT, or SMT formalisms and use solver queries to test uniqueness explicitly. This step provides a formal certificate of uniqueness when possible.

**Implications for benchmark validity.** Uniqueness verification ensures fairness, prevents ambiguity in evaluation, eliminates unstable task specifications, and enables large-scale automated generation of high-quality reasoning tasks while preserving strict semantic guarantees.

### C.5.6 AN EXAMPLE OF AUTOMATIC ERROR CORRECTION IN THE *word_sorting* TASK

To illustrate how multiple agents collaboratively detect and correct errors, we present a complete example from the *word_sorting* subtask.

**Original data example.** We select one automatically constructed data item from the generator:

```
{
  "did": 0,
  "puzzle": "Sort the following words according to the new
  alphabet order: decreasing. The new alphabet has letters
  l at the beginning, followed by the remaining letters in
  their usual order. Words: xre, hetp, epa, cqcovb,
  vmvwywqqr, oczjqhc, yzotp, pkywopqvj, ttp, qczocup. ",
  "question": ["_", "_", "_", "_", "_", "_", "_", "_", "_", "_"],
  "answer": ["pkywopqvj","xre","ttp","hetp","vmvwywqqr","qczocup",
  "cqcovb","epa","yzotp","oczjqhc"],
  "num_problems": 1,
  "number_of_words": 10,
  "number_of_front_letters": 1,
  "created_at": "2025-07-25T03:51:09.923670",
  "delta_time": "0.00014472007751464844 seconds",
  "raw_file": ".../word_sorting_num_problems_1_number_of_words_10
  _number_of_front_letters_1.jsonl"
}
```

The *puzzle* field specifies a modified "new alphabet order" (starting with the letter *l*, followed by the remaining letters in natural order) and gives 10 words to be sorted. The *answer* field is generated automatically by the generator, but its correctness must be verified by the downstream solvers.

1. **Automatically generated but incorrect answer.**
   In this case, the generator produced both the *puzzle* and its corresponding *answer*:

   pkywopqvj,xre,ttp,hetp,vmvwywqqr,qczocup,cqcovb,epa, yzotp, oczjqhc .

   However, solver verification shows that this order is incorrect.

2. **Solver 1: programmatic solution using Python.**
   Solver 1 implements a direct comparator based on the new alphabet order. The core code is:

   ```python
   import re

   # Define the new alphabet order
   new_alphabet = [
       'l','a','b','c','d','e','f','g','h','i',
   ```

```
2268              'j','k','m','n','o','p','q','r','s','t',
2269              'u','v','w','x','y','z'
2270          ]
2271
2272          # Map each letter to its index
2273          alphabet_index = {
2274              letter:index for index,letter in enumerate(new_alphabet)
2275          }
2276
2277          # Convert a word into an index sequence
2278          def get_word_order(word):
2279              return [alphabet_index[letter] for letter in word]
2280
2281          # Words to be sorted
2282          words = [
2283              "xre", "hetp", "epa", "cqcovb", "vmvwywqqr",
2284              "oczjqhc", "yzotp", "pkywopqvj", "ttp", "qczocup"
2285          ]
2286
2287          sorted_words = sorted(words, key=get_word_order)
2288          print(f"\\boxed{{{','.join(sorted_words)}}}")
```

Solver 1 outputs:

cqcovb,epa,hetp,oczjqhc,pkywopqvj,qczocup,ttp,vmvwywqqr,xre,yzotp .

3. **Solver 2: symbolic solver.**
   Solver 2 constructs a symbolic representation of the new alphabet via *sympy* and sorts words in symbolic space:

```
import sympy as sp

class SymbolicSorter:
    def __init__(self):
        self.symbols = {}

    def define_symbol(self, name):
        if name not in self.symbols:
            self.symbols[name] = sp.symbols(name)
        return self.symbols[name]

    def define_new_alphabet(self):
        alphabet = [
            'l','a','b','c','d','e','f','g','h','i',
            'j','k','m','n','o','p','q','r','s','t',
            'u','v','w','x','y','z'
        ]
        return [self.define_symbol(ch) for ch in alphabet]

    def get_word_order(self, word, alphabet_symbols):
        order = []
        for ch in word:
            idx=alphabet_symbols.index(self.symbols.get(ch))
            order.append(idx)
        return order

    def sort_words(self, words, alphabet_symbols):
        return sorted(
            words,
            key=lambda w: self.get_word_order(w,
```

```
                    alphabet_symbols)
            )

    if __name__ == "__main__":
        sorter = SymbolicSorter()
        alphabet_symbols = sorter.define_new_alphabet()
        words = [
            "xre","hetp","epa","cqcovb","vmvwywqqr",
            "oczjqhc","yzotp","pkywopqvj","ttp","qczocup"
        ]
        sorted_words = sorter.sort_words(words, alphabet_symbols)
        print(f"Sorted words:{sorted_words}")
```

The solver produces:

cqcovb,epa,hetp,oczjqhc,pkywopqvj,qczocup,ttp,vmvwywqqr,xre,yzotp .

Both solvers independently arrive at exactly the same ordering.

4. **Automatic correction triggered by solver agreement.**
   The two solvers consistently output:

   cqcovb,epa,hetp,oczjqhc,pkywopqvj,qczocup,ttp,vmvwywqqr,xre,yzotp.

   The original synthesized answer, however, was:

   pkywopqvj,xre,ttp,hetp,vmvwywqqr,qczocup,cqcovb,epa,yzotp,oczjqhc.

   Since two independent solvers provide matching results that disagree with the generated answer, we automatically correct the original answer to the solver-verified ordering.

5. **Filtering strategy for disagreement cases.**
   In the general pipeline, all available solvers are executed for every automatically generated puzzle:

   - If *all* solvers successfully produce outputs and their answers are mutually consistent, the result is accepted as a high-confidence label and used to confirm or overwrite the *answer* field.
   - If any solver fails (e.g., timeout, execution error, parsing failure), or if the solvers produce inconsistent answers, the sample is considered unreliable and is removed from the final benchmark dataset.

   This multi-solver consistency-checking mechanism significantly reduces labeling errors in large-scale automatically constructed logical-reasoning tasks.

## C.6 IMPLEMENTATION DETAILS OF EACH AGENT IN LOGICEVOLVE

This section provides additional implementation details of the LOGICEVOLVE framework, focusing on the roles, communication mechanisms, and memory structures of each agent. Figure 12, 13, 14, 15, 16 illustrate the full engineering design of each agent, including its supported capabilities and the corresponding input–output formats.

Figure 12: A specific example of the composition of metadata.

### C.6.1 DIVISION OF LABOR AMONG AGENTS

The LOGICEVOLVE framework is composed of several specialized LLM-based agents that collaboratively construct large-scale logical reasoning tasks:

- **Metadata Agent (Input Definition).** Responsible for automatically defining structured metadata for each logical reasoning task, including task schema, hyperparameters, constraints, and configuration needed for generation and solving.

- **Generator & Solver Agents (Core Multi-Agent Generation).** These two agents jointly complete task synthesis: the generator produces candidate puzzles, while the solver parses and verifies them through multi-step reasoning. They iteratively improve each other's outputs through feedback signals.

- **Evaluator Agent (Task Sampling and Quality Control).** Given validated generator–solver pairs, the evaluator performs automatic sampling and test-case construction to produce the final large-scale, high-quality data used in the CLUB benchmark.

Together, these agents enable LOGICEVOLVE to automatically produce structurally diverse and solvable logical reasoning tasks without human-designed workflows.

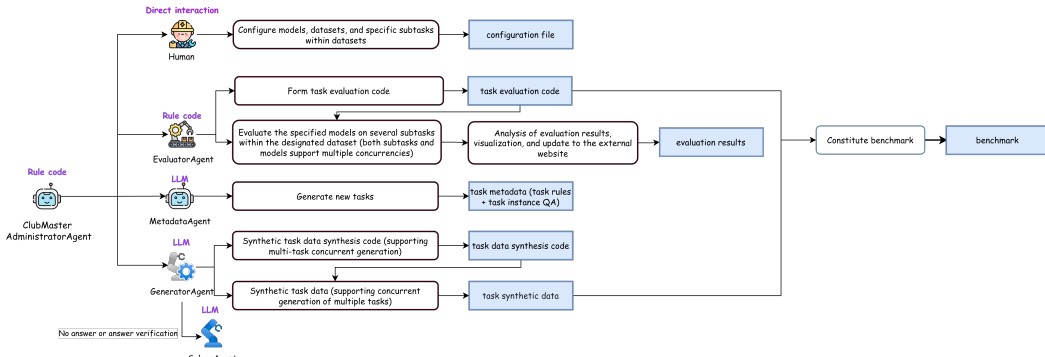

Figure 13: Manager agent.

### C.6.2 FRAMEWORK USABILITY

To ensure reusability and extensibility, the framework is designed with a strong emphasis on modularity and encapsulation. The overarching goal is to allow users to complete the entire workflow—from task configuration to result export—with minimal manual intervention. Specifically, we implemented the following design principles:

- **Encapsulated command interfaces:** Each core functionality of the framework can be triggered with a single command. For instance, metadata synthesis, model configuration, and result export can all be executed by one-line calls, without requiring users to understand the underlying implementation details.

- **Unified configuration mechanism:** A centralized configuration file enables users to flexibly specify base models, hyperparameters, and task types. The framework automatically parses the configuration and executes the corresponding operations.

- **Comprehensive documentation and examples:** The project repository provides detailed usage guidelines for all major components, including instructions on evaluation, task synthesis and monitoring, and data export.

With these designs, the framework achieves a balance between powerful functionality and ease of use, enabling both researchers and practitioners to quickly reuse and extend it.

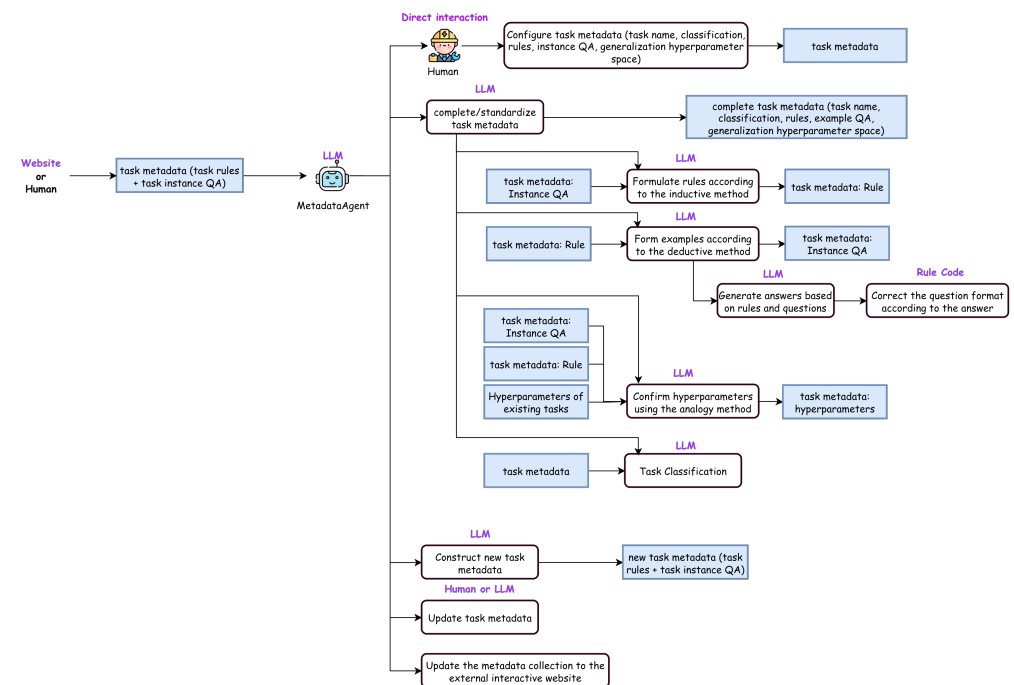

Figure 14: Metadata agent.

### C.6.3 WHY NOT USE SIMPLE PROMPT PIPELINES (CHAIN-OF-PROMPTS)

Before developing LOGICEVOLVE, we attempted to manually construct 2–3 full tasks using simple coordinated prompts (CoP). While this workflow can handle easy tasks (e.g., string manipulation), it fails on more complex ones (e.g., multi-player card puzzles).

As shown in Table 7, using the same base model (Gemini-2.5-Pro), LOGICEVOLVE significantly outperforms CoP on CLUB and ExCLUB tasks. This confirms that simple prompt sequences cannot maintain consistency or complexity across tasks, whereas agent-based modularization enables:

- **Modularity and plug-and-play extensibility** (e.g., solvers dynamically attaching GitHub-RAG or Lean reasoning tools).
- **Dynamic workflow management** capable of long multi-step iterations with state tracking.
- **Dynamic agent generation** for new tasks, including multi-solver ensembles for cross-validation.

Table 7: Comparison of success rates between LogicEvolve and CoP (prompt chaining) on CLUB and ExCLUB.

| Method | CLUB Success Rate | ExCLUB Success Rate |
|---|---|---|
| CoP (no agents, fixed prompt script) | 40% | 31% |
| **LogicEvolve (multi-agent iteration + cross-verification)** | **100%** | **68.9%** |

### C.6.4 WHY NOT USE OPEN-SOURCE MULTI-AGENT FRAMEWORKS

Although many prior studies have explored LLM-based multi-agent systems (Guo et al., 2024), existing frameworks have clear limitations when applied to continuously evolving logical reasoning benchmarks:

- **AutoGPT** (Gravitas, 2023) supports only a single agent.

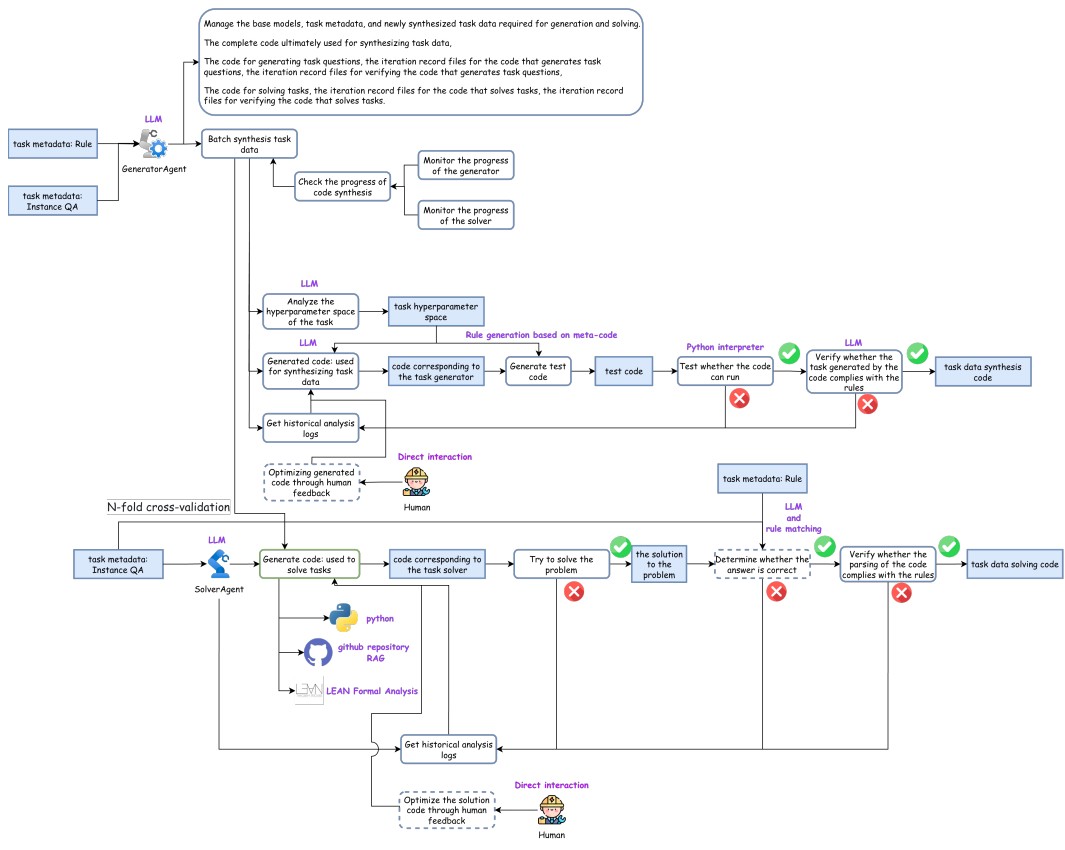

Figure 15: Generator and solver agents.

- **MetaGPT** (Hong et al., 2023) supports three agents but lacks *self-optimization* (agents revising themselves based on feedback) and *collaborative optimization* (agents improving each other's reasoning through interaction).

- **AutoGen** (Wu et al., 2024) improves collaboration but still relies on fixed, non-dynamic agent definitions.

However, LOGICEVOLVE must support indefinite extension: whenever a new task is introduced, it must automatically instantiate a new metadata agent, generator agent, and solver agent—each capable of self-improvement and cross-agent coordination. Manually building a new agent for every task would be impractical and fundamentally incompatible with scalable benchmark evolution.

**Therefore, we implement LOGICEVOLVE using a customized prompt-pipeline architecture**, enabling dynamic agent construction, flexible multi-agent orchestration, and iterative optimization. In future work, we plan to integrate ideas from AUTOAGENTS (Chen et al., 2023) to further enhance dynamic agent generation.

### C.6.5 COMMUNICATION AND MEMORY MECHANISMS

The communication and memory subsystem in LOGICEVOLVE follows a hybrid design combining short-term message passing and persistent long-term state tracking:

**Metadata Agent Memory.** Metadata is persisted in JSON configuration files that record task structure, hyperparameters, and evolution history. These files are read, updated, and versioned across iterations.

**Generator and Solver Agents.** The two core agents rely on:

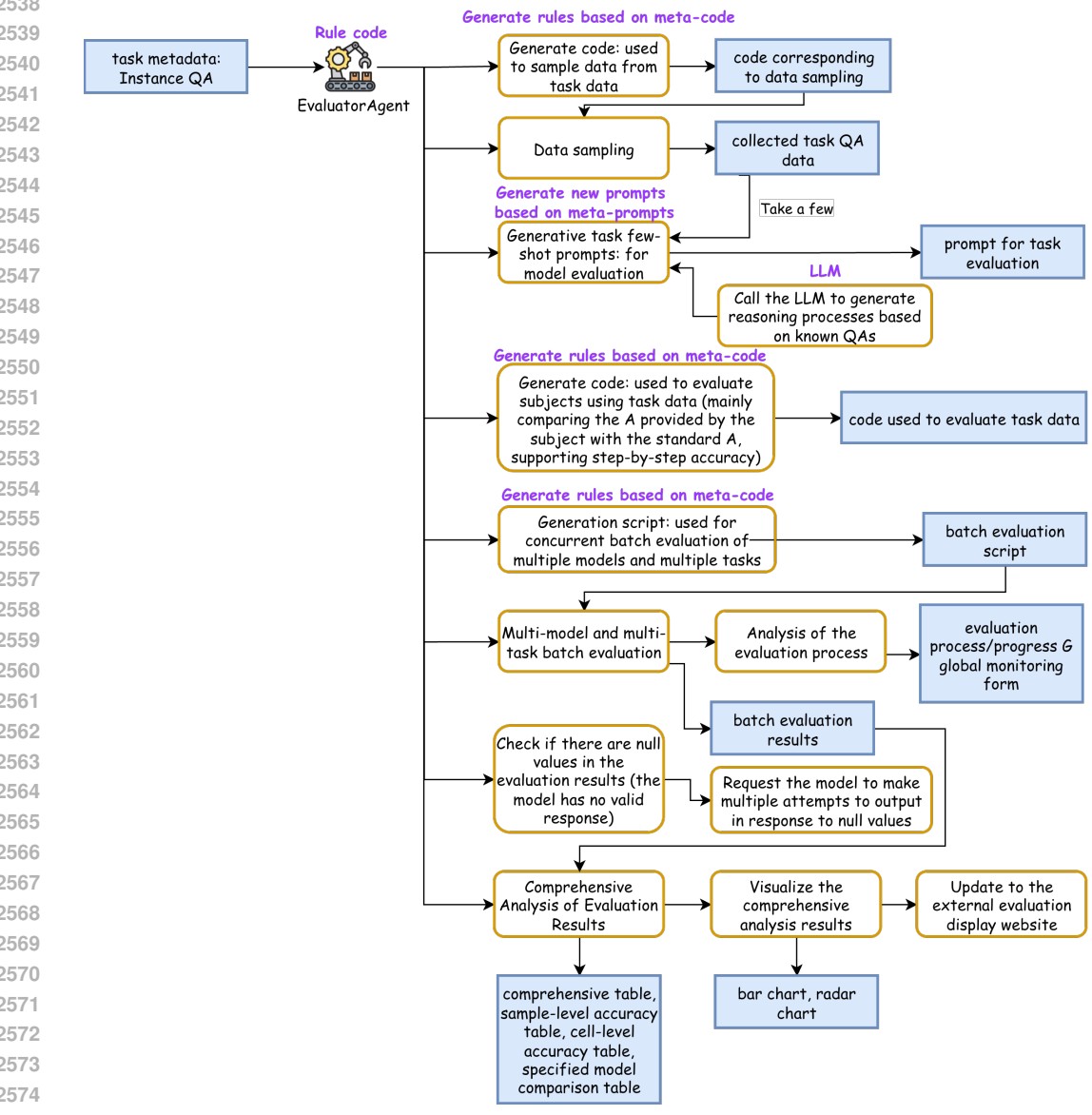

Figure 16: Evaluator agent.

- **Immediate message passing:** once the generator produces a new puzzle instance, it is directly forwarded to the solver agent for parsing, evaluation, and feedback.

- **Persistent file-tree memory:** each agent maintains a dedicated directory tracking its iterative evolution:

  - *meta_generate_code/* for generators:
    * *generate_code/Version_0.py ... Version_n.py*
    * *code_description.json* storing version metadata, hyperparameters, base code, test code, test results, and error logs
    * *verify_code_description.json* storing verification outputs and analysis
  - *meta_solve_code/* for solvers:
    * *solve_code/Version_0.py ... Version_n.py*
    * *code_description.json* storing task lists, solution logic, test scripts, results, and failure traces
    * *verify_solve_code_description.json* storing solver verification logs

**administrator Agent (ClubMaster).** A dedicated manager agent maintains a global shared workspace, handling:

- concurrent execution across dozens of tasks,
- lifecycle management of all metadata/generator/solver agents,
- global context accumulation and memory summarization,
- cross-agent message routing and version iteration control.

This design ensures that the system remains stable even under large-scale multi-agent evolution.

### C.6.6 Additional Engineering Considerations

To further strengthen the engineering usability and efficiency of LogicEvolve, we incorporate several additional considerations:

- **Concurrency:** The multi-agent architecture enables different agents to operate in parallel, significantly accelerating task processing.
- **Administrator agent:** To mitigate conflicts arising from concurrency, we introduce an administrator agent responsible for global coordination. Its responsibilities include monitoring agent states, allocating computational resources, and logging intermediate outputs.
- **Efficiency–stability balance:** While concurrency enhances efficiency, centralized coordination ensures stability, allowing the system to remain controllable and extensible even in complex scenarios.

Together, these design choices enhance both the practicality and reliability of LogicEvolve, establishing a solid foundation for its application to large-scale logical reasoning tasks.

### C.6.7 Connections to Existing Multi-Agent Reasoning Improvements

Several recent findings further highlight the advantages of agent-based designs:

- **Multi-path agent ensembles improve reasoning diversity** (He et al., 2024).
  This motivates extending solver-agent ensembles in LOGICEVOLVE.
- **Collaboration Gap** (Davidson et al., 2025) suggests that independent, strong models may collaborate poorly.
  Relay-style reasoning (handover from strong to weaker agents) may enhance LOGICE-VOLVE's multi-agent synergy.
- **Context Engineering 2.0** (Hua et al., 2025) shows that selecting useful context from growing histories dramatically improves reasoning.
  Applying context filtering strategies could benefit long-iteration task synthesis.
- **Thought-sharing beats language-based communication** (Zheng et al., 2025).
  Future versions of LOGICEVOLVE may explore latent thought-sharing between generator and solver agents.
- **Advanced agentic memory (A-mem)** (Xu et al., 2025) highlights autonomous memory construction and evolution.
  Integrating such memory mechanisms may enable LOGICEVOLVE to learn from past synthesis trajectories.

## C.7 Instantiation of Agents and Cost Analysis

The overall workflow involves four types of agents: the metadata agent, the generator agent, the solver agent, and the automated evaluation agent. Among them, the automated evaluation agent synthesizes rule-based scripts by relying on meta-code; therefore, configuring the base model requires only the metadata agent, generator agent, and solver agent.

The metadata agent has the strictest requirements. Once an issue appears in the metadata, both the generator and solver agents will iterate on incorrect information. Although the workflow provides

error feedback, it still introduces additional iteration costs. As a result, the metadata agent must be assigned to a model with strong overall capabilities.

Both the generator agent and the solver agent must perform code generation and code optimization, which demands strong instruction following and programming abilities. In particular, because the solver agent is also responsible for validating data reliability, it requires even stronger logical-reasoning capabilities.

To this end, we evaluated each model across general-domain benchmarks (MMLU (Hendrycks et al., 2020) and GPQA Diamond (Rein et al., 2024)), instruction-following benchmarks (Ifevl_en (Zhou et al., 2023), IFBench (Pyatkin et al., 2025), and SysBench (Qin et al., 2024)), coding benchmarks (LiveCodeBench (Jain et al., 2024) and ACICode Aa (Tian et al., 2024)), and logical-reasoning benchmarks (Zebra (Lin et al., 2025)), while also accounting for their invocation costs. The results are shown in Table 8.

Table 8: Performance and cost analysis of basic agent-specific models across benchmarks.

| Model | Price | | Overall | Instruction Following | | | Logical Reasoning | Knowledge | | Code | |
|---|---|---|---|---|---|---|---|---|---|---|---|
| | Input \$ / 1M token | Output \$ / 1M token | | Ifevl_en | ifbench | SysBench | Zebra | MMLU | GPQA Diamond | Livecodebench | Acicode Aa |
| gpt5-thinking | 1.25 | 10 | 96.58 | 93.72 | 69.27 | 85.84 | 97.90 | 93.36 | 84.90 | 69.48 | 43.79 |
| gemini2.5-pro | 1.5 | 10 | 93.80 | 90.94 | 48.16 | 88.84 | 92.40 | 92.40 | 83.22 | 73.97 | 49.11 |
| glm4.6-thinking | 0.56 | 2.26 | 90.42 | 89.65 | 41.32 | 88.04 | 92.39 | 89.71 | 79.39 | 80.00 | 41.72 |
| claude4-thinking | 3 | 15 | 89.64 | 88.72 | 50.99 | 86.68 | 95.97 | 91.88 | 73.74 | 59.68 | 45.86 |
| gemini2.5-flash | 0.15 | 3.5 | 87.92 | 89.09 | 48.92 | 85.24 | 76.30 | 90.24 | 80.35 | 64.76 | 44.97 |
| glm4.5 | 0.28 | 1.13 | 87.76 | 86.14 | 40.87 | 85.81 | 89.10 | 90.00 | 78.01 | 70.79 | 42.60 |
| deepseek-r1 | 0.547 | 2.191 | 86.05 | 80.04 | 32.81 | 87.21 | 94.80 | 89.88 | 79.39 | 69.21 | 41.42 |

### C.7.1  MODEL SELECTION

Considering all factors, we configure CLUB and ExCLUB as follows: the metadata agent uses gpt5-thinking, which has the strongest overall capabilities; since metadata is relatively small and does not require multi-round iteration, the associated cost is acceptable. The generator agent uses gemini2.5-pro due to its strong coding and instruction-following abilities. For the solver agent, gemini2.5-pro is used by default; when the maximum iteration threshold is exceeded, it falls back to claude4-thinking.

### C.7.2  COST ANALYSIS

We further compare experiments where glm4.6-thinking replaces gemini2.5-pro, as glm4.6-thinking can be deployed locally on a single 8×H800 node.

The average token cost per agent per task is computed as:

$$\bar{n}_{\text{tok}} = \frac{n_{\text{in}} + n_{\text{out}}}{2} \qquad (6)$$

The success rate is computed as follows:

$$r_{\text{succ}} = \frac{N_{\text{succ}}}{N_{\text{total}}} \qquad (7)$$

Here, $n_{\text{in}}$ and $n_{\text{out}}$ denote the number of input and output tokens for a single agent on a single task, and $\bar{n}$tok represents the average number of input–output tokens per agent per task. The corresponding values $\bar{n}_{\text{tok}}^{\text{meta}}$, $\bar{n}_{\text{tok}}^{\text{gen}}$, and $\bar{n}_{\text{tok}}^{\text{sol}}$ indicate the average input–output tokens for the metadata, generator, and solver agents, respectively. $N_{\text{task}}$ denotes the total number of tasks, $N_{\text{succ}}$ denotes the number of tasks successfully completed by the multi-agent system before human intervention, and $N_{\text{total}}$ is the total number of tasks. Detailed statistics of average input–output tokens for each agent are provided in Table 9.

The corresponding formula for computing the average cost per agent per task is as follows:

$$\bar{C} \approx \bar{n}_{\text{tok}} \times (p_{\text{in}} + p_{\text{out}}) \qquad (8)$$

The total cost is computed as follows:

Table 9: Detailed Token Consumption Statistics.

| Object Construction (Model Configuration) | $N_{\text{task}}$ | $\bar{n}_{\text{tok}}^{\text{meta}}$ | $\bar{n}_{\text{tok}}^{\text{gen}}$ | $\bar{n}_{\text{tok}}^{\text{sol}}$ | $r_{\text{succ}}$ |
|---|---|---|---|---|---|
| CLUB(gpt5-thinking+gemini2.5-pro/claude4-thinking) | 10 | 529 | 187751 | 31285 | 100% |
| ExCLUB(gpt5-thinking+gemini2.5-pro/claude4-thinking) | 100 | 3371 | 232187 | 113240 | 68.90% |
| ExCLUB(gpt5-thinking+glm4.6-thinking) | 100 | 3371 | 42547 | 302687 | 51.40% |

$$C_{\text{total}} \approx N_{\text{task}} \times (\bar{C}^{\text{meta}} + \bar{C}^{\text{gen}} + \bar{C}^{\text{sol}}) \tag{9}$$

Table 10: Task Distribution and Cost Estimation of the Multi-Agent System Under Different Model Configurations.

| Object Construction (Model Configuration) | $N_{\text{task}}$ | $\bar{C}^{\text{meta}}$ | $\bar{C}^{\text{gen}}$ | $\bar{C}^{\text{sol}}$ | $r_{\text{succ}}$ | $C_{\text{total}}$ |
|---|---|---|---|---|---|---|
| CLUB(gpt5-thinking+gemini2.5-pro/claude4-thinking) | 10 | 0.005 | 2.15 | 0.36 | 100% | 25.15 |
| ExCLUB(gpt5-thinking+gemini2.5-pro/claude4-thinking) | 100 | 0.0338 | 2.67 | 1.3 | 68.90% | 400.38 |
| ExCLUB(gpt5-thinking+glm4.6-thinking) | 100 | 0.0338 | 0.12 | 0.85 | 51.40% | 100.38 |

Here, $p_{\text{in}}$ and $p_{\text{out}}$ denote the unit prices of input and output tokens, respectively. $\bar{C}$ represents the average cost per agent per task, and the corresponding $\bar{C}^{\text{meta}}$, $\bar{C}^{\text{gen}}$, and $\bar{C}^{\text{sol}}$ indicate the average per-task costs of the metadata, generator, and solver agents. The estimated total costs are reported in Table 10.

We observe that the average cost per task ranges from \$ 1 to \$ 4. Moreover, once a task is successfully constructed, its generator and solver agents can be reused to synthesize additional task data at no extra configuration cost.

## C.8 ANALYSIS OF THE LOGICEVOLVE FRAMEWORK

### C.8.1 COMPARISON OF PERFORMANCE ACROSS DIFFERENT DATASETS.

We compared the generation quality and efficiency of **LogicEvolve** under different conditions, including: (1) *SynLogic* (Liu et al., 2025b) and *Enigmata* (Chen et al., 2025), which contain partial code and concrete generated puzzle data; (2) the open-source *LogicGame* (Gui et al., 2024), which contains only manually validated puzzle data; (3) the closed-source *MalodyLogic* (malody2014, 2025), which contains only single instances and no standard answers. We used the LogicEvolve framework to perform secondary generation based on these materials and compared the results with the original datasets. As shown in Table 11, the success rate exceeded 60%, the number of versions was around ten, and the processing time was approximately 0.5–1 hour.

Table 11: Analysis of generation quality and efficiency of LogicEvolve on other datasets.

| related works | task nums | success/% | version | time/h |
|---|---|---|---|---|
| **Enigmata(Chen et al., 2025)** | 33 | 60.61 | 16 | 0.5 |
| **LogicGame(Gui et al., 2024)** | 29 | 79.31 | 11 | 0.44 |
| **SynLogic(Liu et al., 2025b)** | 35 | 60 | 18 | 0.82 |
| **MalodyLogic(malody2014, 2025)** | 28 | 78.57 | 5 | 0.1 |

### C.8.2 VISUALISATION OF LOGICEVOLVE DATA EMBEDDING

Figure 17 visualises the question embeddings with t-SNE. The questions generated by LogicEvolve occupy a region of the embedding space that is contiguous with the manually curated data, while spanning a wider area, indicating broader topical coverage.

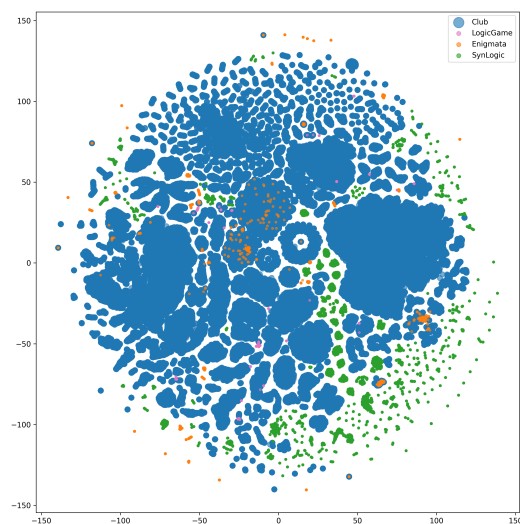

Figure 17: t-SNE visualisation of question embeddings.

## C.9 FAILURE MODES AND TERMINATION CONDITIONS

Like any automated framework, LogicEvolve cannot guarantee a 100% success rate. To balance robustness and efficiency, we explicitly consider failure modes and termination conditions, summarized as follows:

- **Common failure sources:** Failures may arise when the generator does not return valid outputs, when the solver produces answers that deviate significantly from ground truth, or when optimization fails to converge after multiple iterations.
- **Retry mechanism:** For recoverable failures, the system automatically performs multiple retries to maximize task completion rates.
- **Human intervention triggers:** If retries remain unsuccessful, the pipeline escalates the case for manual intervention by human researchers.
- **Termination thresholds:** To prevent uncontrolled execution, we set an upper limit on the number of iterations (e.g., 100). Once reached, the system terminates automatically and queries the user on whether to continue or stop.

This mechanism avoids unnecessary resource consumption while ensuring that human expertise can be introduced at critical points to maintain robustness.

## C.10 AUTOMATION LEVELS AND HUMAN INTERVENTION ANALYSIS

### C.10.1 DEFINITION OF AUTOMATION LEVELS

To systematically characterize the degree of automation in prior work on logical reasoning benchmarks, we summarize existing approaches into five representative levels. At **level 1**, systems release open-source datasets only, with all task collection performed manually. **Level 2** adds open-source code for automated evaluation, where task content remains human-designed but prediction checking is automated. **Level 3** provides open-source code for automated data generation, typically implemented through human-written templates or parameterized rules. **Level 4** extends this to meta-code capable of automatically generating both data and evaluation pipelines, enabling highly automated synthesis with optional human oversight when failures occur. Finally, **level 5** refers to fully autonomous systems that continuously generate new tasks, ground-truth solutions, and evaluation logic without any human intervention, thereby supporting indefinitely extensible benchmark evolution.

For operational clarity, we define automation along two measurable axes. The first is the proportion of pipeline runtime completed without human interruption (the time-cost axis); the second is the

proportion of task content generated without any human modification (the impact-scope axis). A system qualifies as **level 4** if it requires almost no human intervention during execution and if human involvement, when present, is limited to failure recovery or optional validation rather than content design. A system reaches **level 5** only when both axes achieve full autonomy.

Under this operational definition, we further clarify the automation status of LogicEvolve in our experiments. We use "automation" to refer specifically to: (i) the extent to which task content is generated without human changes; (ii) the extent to which the pipeline executes without human interruption; and (iii) the absence of any human-written rules, instances, or templates beyond the initial metadata. Empirically, LogicEvolve exhibits a high degree of automation across all three dimensions: all task content is generated automatically (**100%**); more than **98%** of the total runtime proceeds without human interruption; and no human involvement occurs at the level of task design. Accordingly, we describe LogicEvolve as "a largely-automated task-evolution pipeline with optional human oversight for failure recovery and sample validation," corresponding to **level 4** rather than level 5.

### C.10.2 TYPES OF HUMAN INTERVENTION IN LOGICEVOLVE

It is important to emphasize that human intervention in this study occurs only in highly restricted situations and is strictly decoupled from the generation of task content. Empirically, we observe three categories of intervention: failure-handling intervention, validation-sampling intervention, and creative intervention.

The first category, "**failure-handling intervention**," is triggered only when the generator or solver enters an unrecoverable state, such as persistent stagnation, failure to pass validation after reaching the maximum number of iterations, or the detection of structurally abnormal metadata by the validator. The sole purpose of this intervention is to allow the pipeline to continue running; it does not modify or redesign any task content.

The second category, "**validation-sampling intervention**," involves inspecting a small number of samples for logical coherence and solvability, serving to improve the stability of the overall benchmark. Its role is analogous to quality assurance rather than content creation.

The final category, "**creative intervention**," occurs only when users intentionally introduce specific preferences, such as pedagogical friendliness or stylistic structuring. This mechanism is not used in any experiments reported in the main paper and is excluded from all automation-level assessments.

### C.10.3 SPECIFIC TYPES AND FREQUENCIES OF FAILURE-HANDLING INTERVENTIONS

To quantify the frequency of human intervention, we adopt a reproducible counting criterion: an intervention event is recorded only when a human actually modifies metadata, generator code, or solver code, or makes an explicit judgment such as confirming a rule inconsistency. Merely inspecting logs is not counted as intervention. We conduct five independent runs over the 100 tasks in ExCLUB, yielding a total of 500 complete task-evolution pipelines. The resulting intervention statistics are summarized in Table 12.

Table 12: Human intervention frequencies observed across 500 complete task-evolution pipelines. Each value indicates the proportion of pipelines in which the corresponding type of intervention occurred at least once.

| Intervention Type | Metadata | Generator | Solver | Evaluator | Data | Notes |
|---|---|---|---|---|---|---|
| Metadata rule contradiction | 0% | – | – | – | – | Not observed (rules pre-validated by template) |
| Metadata instance inconsistency | 19.4% | – | – | – | – | Caused by LLM misunderstanding of rules |
| Generator stagnation | – | 13.6% | – | – | – | Often due to malformed or unstructured outputs |
| Generator non-convergence | – | 5.8% | – | – | – | Insufficient reasoning depth for convergence |
| Solver stagnation | – | – | 23.3% | – | – | Caused by code errors or rule misinterpretation |
| Solver non-convergence | – | – | 14.6% | – | – | Complex task structure requiring deep reasoning |
| Data leakage | – | – | – | – | 0% | Not observed (engineering safeguards prevent duplication) |
| Evaluator inconsistency | – | – | – | 0% | – | Not observed (evaluation logic synthesized from rules) |

Each entry in the table reports the proportion of the 500 pipelines in which the corresponding type of intervention occurred at least once. The results show that metadata instance inconsistencies

occurred in 19.4% of pipelines, generator stagnation in 13.6%, generator non-convergence in 5.8%, solver stagnation in 23.3%, and solver non-convergence in 14.6%. No rule contradictions, evaluator inconsistencies, or data-leakage events were observed. The total number of intervention events was fewer than 400 across all 500 pipelines, corresponding to an average of fewer than 0.8 interventions per pipeline. All interventions were absorbed in subsequent iterations and did not alter task content or task logic. Therefore, these interventions do not affect the automation level of LogicEvolve.

Finally, LogicEvolve supports two optional forms of human involvement that are not counted toward automation: (i) quality-sampling inspection for long-term stability monitoring, and (ii) creative intervention for introducing stylistic or pedagogical preferences.

### C.10.4    MANUAL QUALITY SAMPLING INSPECTION

Although the automated verification pipeline in LogicEvolve is capable of detecting the majority of structural and logical errors, relying solely on automated solvers still leaves a small but important class of failure modes uncovered. These include errors that occur at boundary conditions, ambiguities that arise from natural-language rule descriptions, and inconsistencies triggered only under extreme or atypical input configurations. To address these cases, we incorporate a final stage of human inspection before releasing the benchmark. Importantly, the purpose of human inspection is not to replace automated verification but to serve as a complementary mechanism that targets corner cases the automated system is inherently less equipped to detect. The technical goals of this manual review are threefold: (i) to identify semantic ambiguities in metadata that automated solvers cannot resolve; (ii) to reveal structural inconsistencies that arise under extreme parameter settings; and (iii) to validate solver stability in multi-step or large state-space reasoning tasks.

To ensure coverage and minimize sampling bias, we adopt a stratified sampling strategy that captures variation across both task types and difficulty levels. In CLUB, we sample a total of one hundred instances from its ten task families across their difficulty gradients; in ExCLUB, we analogously sample one hundred instances from its one hundred task families. Each instance is independently reviewed by two researchers with backgrounds in logic reasoning and program reasoning, respectively. Afterwards, discrepancies between reviewers are discussed and resolved through consensus, ensuring consistency and reproducibility in the assessment.

Our manual inspection reveals that while CLUB exhibits no observable issues, approximately thirty percent of the initial ExCLUB samples contain non-trivial errors. These errors are technical and subtle in nature, including incomplete formalization of boundary conditions that cause undefined solver behavior, non-unique or suboptimal solutions in multi-step tasks where multiple valid trajectories exist, and rare generator failures triggered by oversized state spaces that lead to mismatched local and global constraints. Such issues tend to arise from the interaction of complex constraints or under extreme configuration regimes, making them difficult for automated solvers to detect exhaustively. Human inspection therefore plays a critical role in identifying these corner cases.

All problematic samples identified through manual review are routed back into the generator–solver loop of LogicEvolve. This involves rewriting rule definitions, regenerating instances, enforcing solver consistency checks, and revalidating the corrected tasks. After several rounds of iterative refinement, all explicit issues are either fixed or eliminated, and the final error rate in the inspected samples drops to zero. This process ensures that both CLUB and ExCLUB reach a verifiable level of quality before release.

Looking forward, we plan to institutionalize human inspection as a small-scale but systematic quality maintenance procedure. Specifically, for each benchmark expansion or evolution cycle, we will maintain a "automation-first, targeted human inspection as auxiliary" strategy, complemented by metadata perturbation tests and solver consistency checks. This long-term, multi-layer verification loop provides a scalable and reliable quality-control mechanism without imposing significant manual overhead, and aligns with the community's expectation for trustworthy and reproducible reasoning benchmarks.

### C.10.5    OPTIONAL CREATIVE INTERVENTION

In addition to failure-handling and validation-sampling intervention, LogicEvolve also supports an optional form of human involvement that we refer to as *creative intervention*. This mechanism is

triggered only when users explicitly wish to introduce human preferences into the task-evolution process, typically for generating tasks that better reflect pedagogical goals or emphasize particular reasoning patterns. In such cases, humans may guide the synthesis of metadata, the task generator, or the solver, or may provide high-level preference signals to steer the evolution of tasks toward desired characteristics. Importantly, creative intervention does not affect the autonomy metrics reported in this work, and it is not used in any of the experiments presented in the main paper.

# D CLUB

## D.1 BASIC SETUP

### D.1.1 EVALUATION HYPERPARAMETERS

All models are invoked via the *OpenAI SDK (≥1.0.0)* using the *chat.completions.create* interface, with no explicit timeout or token limit to ensure sufficient reasoning. Each task is evaluated through three independent runs, and the mean performance is reported to ensure stability. A retry mechanism is applied (three attempts by default, extended to six in the case of specific JSON parsing errors) to enhance robustness.

### D.1.2 EVALUATION MODELS

Given the recent substantial progress of mainstream large language models on complex logical reasoning tasks, a systematic evaluation remains lacking. In this study, we employ the **CLUB** benchmark to assess the performance of multiple state-of-the-art LLMs on complex logical reasoning. The evaluated models are categorized into *base models* and *reasoning models*.

Commercial base models include the Gemini series (Comanici et al., 2025), the GPT series (OpenAI et al., 2024), the o-series (OpenAI, 2025), and the Claude series (Anthropic, 2025).

Open-source base models include the DeepSeek series (DeepSeek-AI et al., 2025), the Doubao series (Seed et al., 2025), and the Qwen series (Yang et al., 2025).

A detailed version description is provided in Table 13.

## D.2 REPRODUCIBILITY STATEMENT

To ensure full reproducibility of our work, we provide a dedicated *experiments* directory in the project, which contains detailed instructions and implementations for all reproduction procedures. This includes the reproduction of task data generation, model evaluation, and the figures and tables presented in the paper. For each component, we preserve independent code and corresponding outputs, enabling readers to fully replicate the experimental process and verify all conclusions reported in this work.

## D.3 RELATED BASELINES

### D.3.1 A HUMAN ENGINEERING BASELINE FOR CLUB

To contextualize the engineering background of LogicEvolve and provide a practical baseline for its level of automation, we introduce a "human engineering baseline" that quantifies the amount of manual effort required to construct a family of tasks under a unified interface. Specifically, we invited five software engineers with computer science backgrounds (at the master's or doctoral level) to implement ten task types over a two-day period. For each task type, participants were required to produce a complete modular pipeline consisting of four components—metadata, generator, solver, and evaluator. They were allowed to consult publicly available programmatic puzzle generators and online repositories (e.g., *traditional_zebra*, *Knights and Knaves*), as well as language-model tools, but were required to rewrite all code to conform to our unified interface specification and were prohibited from reusing any existing puzzle instances.

The purpose of this baseline is to assess the engineering cost of migrating and integrating multiple heterogeneous programmatic generators into a unified, maintainable, extensible, and evolvable task-generation pipeline, rather than to measure puzzle quality or establish an upper bound on human puzzle-creation ability. Importantly, all puzzle instances used in CLUB and ExCLUB are generated automatically by LogicEvolve; the human baseline is employed solely to understand task structure and algorithmic design principles and does not introduce any preexisting data or leakage.

Moreover, many existing programmatic generators hard-code task rules, generation logic, and solving logic into a single script, making them incompatible with the modular structure required by LogicEvolve and unsuitable for direct integration into the evolution workflow. For this reason,

Table 13: Mainstream models grouped by vendor. Thinking-type models are marked with ✓.

| Vendor | Full Model Name | Thinking | Short Name |
|---|---|---|---|
| **Anthropic** | | | |
| | claude-3-7-sonnet-20250219 | ✗ | claude-sonnet3.7 |
| | claude-3-7-sonnet-20250219-thinking | ✓ | claude-sonnet3.7(thinking) |
| | claude-sonnet-4-20250514 | ✗ | claude-sonnet4 |
| | claude-sonnet-4-20250514-thinking | ✓ | claude-sonnet4(thinking) |
| **DeepSeek** | | | |
| | deepseek-chat | ✗ | deepseek-chat |
| | deepseek-r1-250528 | ✓ | deepseek-r1 |
| | deepseek-reasoner | ✓ | deepseek-reasoner |
| | deepseek-v3-1-250821 | ✗ | deepseek3.1 |
| **Doubao** | | | |
| | doubao-1.5-pro-32k-250115 | ✗ | doubao1.5-pro-32k |
| | doubao-1.5-thinking-pro-250415 | ✓ | doubao1.5-pro(thinking) |
| | doubao-seed-1-6-250615 | ✗ | doubao1.6 |
| | doubao-seed-1-6-flash-250615 | ✗ | doubao1.6-flash |
| | doubao-seed-1-6-thinking-250615 | ✓ | doubao1.6(thinking) |
| **Google DeepMind** | | | |
| | gemini-2.5-flash | ✗ | gemini2.5-flash |
| | gemini-2.5-flash-preview-04-17 | ✗ | gemini2.5-flash |
| | gemini-2.5-flash-preview-04-17-thinking | ✓ | gemini2.5-flash(thinking) |
| | gemini-2.5-pro | ✗ | gemini2.5-pro |
| | gemini-2.5-pro-preview-05-06 | ✗ | gemini2.5-pro-0506 |
| | gemini-2.5-pro-preview-05-06-thinking | ✓ | gemini2.5-pro-0506(thinking) |
| | gemini-2.5-pro-preview-06-05 | ✗ | gemini2.5-pro-0605 |
| | gemini-2.5-pro-preview-06-05-thinking | ✓ | gemini2.5-pro-0605(thinking) |
| **Zhipu AI** | | | |
| | glm-4-32b | ✗ | glm4 |
| | glm-4.5 | ✗ | glm4.5 |
| | glm-4.5-air | ✗ | glm4.5-air |
| | glm-z1-32b | ✓ | glm-z1 |
| **OpenAI** | | | |
| | gpt-4.1 | ✗ | gpt4.1 |
| | gpt-4o | ✗ | gpt4o |
| | gpt-5-2025-08-07 | ✓ | gpt5(thinking) |
| | gpt-5-chat-2025-08-07 | ✗ | gpt5 |
| | o4-mini-2025-04-16 | ✓ | gpt-o4-mini |
| | openrouter:gpt-oss-120b | ✗ | gpt-oss-120b |
| | openrouter:gpt-oss-20b | ✗ | gpt-oss-20b |
| **X** | | | |
| | openrouter:grok-3-mini-beta | ✗ | grok3-mini |
| | openrouter:grok-4 | ✗ | grok4 |
| **Alibaba Qwen** | | | |
| | qwen3-235b-a22b | ✗ | qwen3-235b |
| | qwen3-32b | ✗ | qwen3-32b |
| | qwq-32b | ✗ | qwq |
| **Other** | | | |
| | openrouter:gemma-3-27b-it | ✗ | gemma3 |
| | openrouter:kimi-k2-0905 | ✗ | kimi-k2 |
| | openrouter:llama-4-maverick | ✗ | llama4 |
| | openrouter:mistral-small-3.1-24b-instruct | ✗ | mistral3.1-small |

constructing a systematic "programmatic generator baseline" remains part of our planned future work.

### D.3.2 HUMAN BASELINE ON CLUB

To address the question of how humans are expected to perform on the CLUB benchmark, we conducted a controlled and reproducible human baseline experiment. It is important to note that CLUB is not designed as a human-oriented evaluation: each task requires structured JSON outputs, complete intermediate reasoning traces, and strict step-by-step consistency within programmatic state spaces (e.g., stack machines, board states, path search procedures, string transformation sequences). Consequently, CLUB primarily assesses the consistency of large language models on structured and programmatic reasoning tasks, rather than measuring natural human reasoning ability.

**Experimental setup.**   We recruited three participants with computer science backgrounds (but not professional puzzle designers). Each participant was given two days, with no more than four hours of work per day. For every task type in CLUB, we sampled one instance from each difficulty level (1–10), resulting in ten instances per task type. A time limit of one hour was imposed for each instance; if a participant could not complete an instance within this limit, higher-difficulty instances of that task type were not attempted. Scoring followed the exact procedure used for model evaluation: an instance is counted as correct (Acc = 1) only if all required output fields are correct.

**Results and comparison.**   We aligned the human results with those obtained by GPT-5-thinking, Grok-4, GPT-4o, and other models evaluated on the same instance set. Overall, humans showed limited performance under CLUB's programmatic format: even at the lowest difficulty levels, tasks typically required substantial manual bookkeeping and state simulation, and certain task types (e.g., program-trace tasks, Tetris-style state machines, and path-search tasks) could not be completed within the time constraint. The final average human score was approximately 9 out of 100, lower than GPT-4o (12.6) and significantly below stronger models. Detailed results are shown in Table 14.

Table 14: Comparison of performance between human baselines and mainstream models on the CLUB sample question set (Acc, unit: %). The human results are obtained from 3 participants with a computer background under strict time budgets and unified scoring rules.

| Model | code | ext-zebra | kn&kn | line | maze | string | sudoku | tetris | zebra | card | Overall |
|---|---|---|---|---|---|---|---|---|---|---|---|
| GPT-5-thinking | 35 | 8 | 100 | 100 | 44 | 48 | 85 | 15 | 94 | 16 | 54.5 |
| Grok-4 | 35 | 5 | 100 | 100 | 47 | 43 | 53 | 3 | 97 | 30 | 51.3 |
| GPT-4o | 0 | 0 | 12 | 60 | 2 | 11 | 12 | 0 | 28 | 1 | 12.6 |
| **Human** | 0 | 0 | 10 | 20 | 0 | 20 | 10 | 0 | 30 | 0 | **9.0** |

**Observations and interpretation.**   The experimental results suggest that the low human performance on CLUB does not reflect limitations in human logical reasoning ability, but rather the programmatic nature of the benchmark itself. The requirement for field-by-field JSON outputs, explicit tracking of programmatic state transitions, and strict step-by-step consistency is far more natural for machine execution than for manual human reasoning. In addition, many tasks involve internal state spaces that are difficult for humans to simulate precisely within short time budgets, further reducing completion rates.

**Positioning and limitations.**   It is important to emphasize that the goal of the human baseline experiment is not to estimate an upper bound of human performance, but to provide a reproducible point of reference under realistic constraints and to illustrate that CLUB's task format is not suitable as a traditional measure of human reasoning ability. Accordingly, we do not interpret the human results as reflecting human capability limits; rather, they highlight structural differences between machine-oriented and human-oriented task formats.

D.4   OVERALL RESULTS

The overall model performance on CLUB is summarized in Table 15.

D.5   SAMPLE-LEVEL RESULTS

**Abbreviations:** *extzebra* = extensible_zebra, *k&k* = knights_and_knaves, *linefg* = line_forming_games, *trazebra* = traditional_zebra, *traincp* = train_game_card_puzzle.

D.5.1   2025-09-17

Results are reported in Table 16.

D.5.2   2025-09-20

Results are reported in Table 17.

Table 15: Model performance on Club Benchmark.

| Model | Acc (250917) | Cell Acc (250917) | Acc (250920) | Cell Acc (250920) | Acc (250922) | Cell Acc (250922) |
|---|---|---|---|---|---|---|
| claude-3-7-sonnet-20250219 | 22.1 | 40 | 22.7 | 40.69 | 23.4 | 40.92 |
| claude-3-7-sonnet-20250219-thinking | 27.1 | 38.37 | 39.2 | 48.07 | 40.3 | 48.6 |
| claude-sonnet-4-20250514 | 28.2 | 41.43 | 29.8 | 43.1 | 29.7 | 43.45 |
| claude-sonnet-4-20250514-thinking | 32.6 | 42.18 | 46.7 | 53.11 | 46.7 | 52.76 |
| deepseek-chat | 18.9 | 28.89 | 30 | 32.31 | 30.1 | 32.18 |
| deepseek-r1-250528 | 44 | 50.51 | 44.4 | 50.47 | 44 | 50.87 |
| deepseek-reasoner | 38.4 | 47.24 | 43.9 | 50.85 | 44.2 | 50.44 |
| deepseek-v3-1-250821 | 31.3 | 36.75 | 31 | 35.81 | 30.7 | 33.01 |
| doubao-1.5-pro-32k-250115 | 17.2 | 32.91 | 16.6 | 32.52 | 16.7 | 33.31 |
| doubao-1.5-thinking-pro-250415 | 43.2 | 50.34 | 40.5 | 49.56 | 40.7 | 49.49 |
| doubao-seed-1-6-250615 | 45.2 | 51.76 | 42.5 | 48.2 | 43 | 49.78 |
| doubao-seed-1-6-flash-250615 | 23.9 | 36.84 | 21.5 | 32.66 | 20.7 | 33.69 |
| doubao-seed-1-6-thinking-250615 | 44.1 | 49.53 | 41.4 | 47.19 | 40.4 | 46.01 |
| gemini-2.5-flash | NaN | NaN | 38.4 | 41.46 | 39.2 | 42.53 |
| gemini-2.5-flash-preview-04-17 | 23.8 | 30.33 | NaN | NaN | NaN | NaN |
| gemini-2.5-flash-preview-04-17-thinking | 34.8 | 38.71 | NaN | NaN | NaN | NaN |
| gemini-2.5-pro | NaN | NaN | 47.9 | 51.76 | 45.9 | 49.52 |
| gemini-2.5-pro-preview-05-06 | 37.1 | 43.53 | NaN | NaN | NaN | NaN |
| gemini-2.5-pro-preview-05-06-thinking | 44.4 | 49.56 | NaN | NaN | NaN | NaN |
| gemini-2.5-pro-preview-06-05 | 45.9 | 51.45 | NaN | NaN | NaN | NaN |
| gemini-2.5-pro-preview-06-05-thinking | 46.6 | 51.03 | NaN | NaN | NaN | NaN |
| glm-4-32b | 18.4 | 29.24 | NaN | NaN | NaN | NaN |
| glm-4.5 | 42.4 | 46.57 | 42.8 | 48.3 | 42.3 | 48.93 |
| glm-4.5-air | 36.8 | 39.35 | 36.9 | 38.07 | 36.9 | 37.79 |
| glm-z1-32b | 28.7 | 40.57 | NaN | NaN | NaN | NaN |
| gpt-4.1 | 24.6 | 32.45 | 24.9 | 34.65 | 23.2 | 35.12 |
| gpt-4o | 12.3 | 22.53 | 12.5 | 23.52 | 12.6 | 23.25 |
| gpt-5-2025-08-07 | 56.5 | 59.84 | 55.7 | 59.29 | 54.5 | 58.42 |
| gpt-5-chat-2025-08-07 | 27.4 | 38.96 | 27.4 | 39.79 | 28.3 | 40.31 |
| o4-mini-2025-04-16 | 44.1 | 42.73 | 46.3 | 48.59 | NaN | NaN |
| gemma-3-27b-it | 10.1 | 31.72 | 9.9 | 29 | 10.2 | 29.18 |
| gpt-oss-120b | 41.7 | 45.35 | 40.3 | 45.53 | 40.3 | 45.97 |
| gpt-oss-20b | 36.3 | 40.19 | 37 | 39.84 | 37.7 | 39.93 |
| grok-3-mini-beta | 35.3 | 38.19 | 35.3 | 39.7 | 35.5 | 40.06 |
| grok-4 | 65 | 61.44 | 52.8 | 46.15 | 51.3 | 46.07 |
| kimi-k2-0905 | 34.5 | 40.09 | 34.2 | 38.21 | 34 | 38.63 |
| llama-4-maverick | 16.4 | 29.1 | 16.9 | 29.67 | 16 | 28.48 |
| mistral-small-3.1-24b-instruct | 10.6 | 23.37 | 13.8 | 27.46 | 13.9 | 29.55 |
| qwq-32b | 31.1 | 37.1 | 34.9 | 41.56 | 34.8 | 41.51 |
| qwen3-235b-a22b | 17.9 | 34.88 | 39.3 | 47.25 | 38.9 | 47.72 |
| qwen3-32b | 15.2 | 33.8 | 40 | 47.48 | 40.8 | 47.94 |

### D.5.3  2025-09-22

Results are reported in Table 18.

## D.6  CELL-LEVEL RESULTS

(Abbreviations are the same as above.)

### D.6.1  2025-09-17

Results are reported in Table 19.

### D.6.2  2025-09-20

Results are reported in Table 20.

### D.6.3  2025-09-22

Results are reported in Table 21.

## D.7  RELEVANCE ANALYSIS OF LOGICAL REASONING BENCHMARKS

To further understand whether different logical-reasoning benchmarks evaluate consistent capability dimensions, we compute the Spearman rank correlation coefficient between the rankings of five representative frontier models across all seven benchmarks (CLUB, LogiQA2.0, BIG-Bench Hard, SynLogic, Zebra, LogicGame, ARC-AGI v1). The correlation matrix is visualized in Figure 18.

Table 16: The model's performance on different CLUB tasks. (Acc, 2025-09-17)

| Model | code | extzebra | k&k | linefg | maze | string | sudoku | tetris | trazebra | traincp | overall |
|---|---|---|---|---|---|---|---|---|---|---|---|
| grok-4 | 35.00 | 57.00 | 97.00 | 100.00 | 58.00 | 59.00 | 82.00 | 19.00 | 95.00 | 48.00 | 65.00 |
| gpt-5-2025-08-07 | 35.00 | 14.00 | 100.00 | 100.00 | 46.00 | 53.00 | 87.00 | 13.00 | 95.00 | 22.00 | 56.50 |
| gemini-2.5-pro-preview-06-05-thinking | 26.00 | 0.00 | 100.00 | 99.00 | 14.00 | 39.00 | 56.00 | 6.00 | 90.00 | 36.00 | 46.60 |
| gemini-2.5-pro-preview-06-05 | 24.00 | 0.00 | 100.00 | 100.00 | 14.00 | 38.00 | 54.00 | 5.00 | 89.00 | 35.00 | 45.90 |
| doubao-seed-1-6-250615 | 30.00 | 0.00 | 92.00 | 100.00 | 30.00 | 38.00 | 53.00 | 7.00 | 96.00 | 6.00 | 45.20 |
| gemini-2.5-pro-preview-05-06-thinking | 25.00 | 1.00 | 100.00 | 99.00 | 14.00 | 33.00 | 54.00 | 2.00 | 82.00 | 34.00 | 44.40 |
| o4-mini-2025-04-16 | 22.00 | 0.00 | 96.00 | 96.00 | 29.00 | 38.00 | 62.00 | 0.00 | 96.00 | 2.00 | 44.10 |
| doubao-seed-1-6-thinking-250615 | 34.00 | 7.00 | 80.00 | 100.00 | 30.00 | 39.00 | 53.00 | 0.00 | 97.00 | 1.00 | 44.10 |
| deepseek-r1-250528 | 33.00 | 0.00 | 97.00 | 97.00 | 34.00 | 30.00 | 49.00 | 0.00 | 95.00 | 5.00 | 44.00 |
| doubao-1.5-thinking-pro-250415 | 31.00 | 1.00 | 94.00 | 100.00 | 23.00 | 34.00 | 49.00 | 0.00 | 97.00 | 3.00 | 43.20 |
| glm-4.5 | 32.00 | 0.00 | 98.00 | 99.00 | 18.00 | 35.00 | 47.00 | 0.00 | 89.00 | 6.00 | 42.40 |
| gpt-oss-120b | 29.00 | 0.00 | 95.00 | 96.00 | 26.00 | 20.00 | 62.00 | 2.00 | 82.00 | 5.00 | 41.70 |
| deepseek-reasoner | 30.00 | 0.00 | 86.00 | 99.00 | 13.00 | 24.00 | 40.00 | 0.00 | 85.00 | 7.00 | 38.40 |
| gemini-2.5-pro-preview-05-06 | 20.00 | 4.00 | 96.00 | 70.00 | 13.00 | 40.00 | 52.00 | 1.00 | 39.00 | 36.00 | 37.10 |
| glm-4.5-air | 32.00 | 0.00 | 96.00 | 92.00 | 11.00 | 18.00 | 40.00 | 0.00 | 78.00 | 1.00 | 36.80 |
| gpt-oss-20b | 31.00 | 0.00 | 72.00 | 97.00 | 18.00 | 14.00 | 62.00 | 1.00 | 67.00 | 1.00 | 36.30 |
| grok-3-mini-beta | 30.00 | 0.00 | 82.00 | 98.00 | 13.00 | 17.00 | 31.00 | 0.00 | 78.00 | 4.00 | 35.30 |
| gemini-2.5-flash-preview-04-17-thinking | 9.00 | 0.00 | 92.00 | 94.00 | 12.00 | 23.00 | 42.00 | 0.00 | 68.00 | 8.00 | 34.80 |
| kimi-k2-0905 | 9.00 | 0.00 | 87.00 | 92.00 | 2.00 | 19.00 | 43.00 | 0.00 | 91.00 | 2.00 | 34.50 |
| claude-sonnet-4-20250514-thinking | 11.00 | 0.00 | 64.00 | 95.00 | 8.00 | 31.00 | 32.00 | 0.00 | 67.00 | 18.00 | 32.60 |
| deepseek-v3-1-250821 | 22.00 | 0.00 | 74.00 | 87.00 | 9.00 | 23.00 | 20.00 | 0.00 | 76.00 | 2.00 | 31.30 |
| qwq-32b | 24.00 | 0.00 | 84.00 | 86.00 | 8.00 | 12.00 | 23.00 | 0.00 | 73.00 | 1.00 | 31.10 |
| glm-z1-32b | 30.00 | 0.00 | 78.00 | 59.00 | 8.00 | 17.00 | 29.00 | 0.00 | 65.00 | 1.00 | 28.70 |
| claude-sonnet-4-20250514 | 9.00 | 0.00 | 57.00 | 94.00 | 4.00 | 25.00 | 26.00 | 0.00 | 63.00 | 4.00 | 28.20 |
| gpt-5-chat-2025-08-07 | 23.00 | 0.00 | 54.00 | 85.00 | 4.00 | 20.00 | 26.00 | 0.00 | 58.00 | 4.00 | 27.40 |
| claude-3-7-sonnet-20250219-thinking | 15.00 | 0.00 | 56.00 | 85.00 | 4.00 | 35.00 | 19.00 | 1.00 | 53.00 | 3.00 | 27.10 |
| gpt-4.1 | 19.00 | 0.00 | 42.00 | 83.00 | 3.00 | 17.00 | 21.00 | 0.00 | 60.00 | 1.00 | 24.60 |
| doubao-seed-1-6-flash-250615 | 13.00 | 0.00 | 59.00 | 84.00 | 4.00 | 30.00 | 30.00 | 0.00 | 44.00 | 1.00 | 23.90 |
| gemini-2.5-flash-preview-04-17 | 14.00 | 1.00 | 25.00 | 73.00 | 8.00 | 25.00 | 31.00 | 0.00 | 53.00 | 8.00 | 23.80 |
| claude-3-7-sonnet-20250219 | 5.00 | 0.00 | 48.00 | 64.00 | 6.00 | 27.00 | 21.00 | 0.00 | 49.00 | 1.00 | 22.10 |
| deepseek-chat | 15.00 | 0.00 | 36.00 | 59.00 | 4.00 | 16.00 | 12.00 | 0.00 | 45.00 | 2.00 | 18.90 |
| glm-4-32b | 1.00 | 0.00 | 41.00 | 65.00 | 3.00 | 7.00 | 16.00 | 0.00 | 50.00 | 1.00 | 18.40 |
| qwen3-235b-a22b | 0.00 | 0.00 | 35.00 | 67.00 | 2.00 | 18.00 | 15.00 | 0.00 | 40.00 | 2.00 | 17.90 |
| doubao-1.5-pro-32k-250115 | 2.00 | 0.00 | 34.00 | 63.00 | 5.00 | 16.00 | 13.00 | 0.00 | 39.00 | 0.00 | 17.20 |
| llama-4-maverick | 2.00 | 0.00 | 33.00 | 62.00 | 3.00 | 15.00 | 13.00 | 0.00 | 34.00 | 2.00 | 16.40 |
| qwen3-32b | 1.00 | 0.00 | 21.00 | 63.00 | 2.00 | 17.00 | 15.00 | 0.00 | 33.00 | 0.00 | 15.20 |
| gpt-4o | 0.00 | 0.00 | 14.00 | 57.00 | 1.00 | 11.00 | 10.00 | 0.00 | 30.00 | 0.00 | 12.30 |
| mistral-small-3.1-24b-instruct | 0.00 | 0.00 | 17.00 | 57.00 | 0.00 | 8.00 | 4.00 | 0.00 | 19.00 | 1.00 | 10.60 |
| gemma-3-27b-it | 0.00 | 0.00 | 25.00 | 36.00 | 0.00 | 6.00 | 12.00 | 0.00 | 21.00 | 1.00 | 10.10 |

Table 17: The model's performance on different CLUB tasks. (Acc, 2025-09-20)

| Model | code | extzebra | k&k | linefg | maze | string | sudoku | tetris | trazebra | traincp | overall |
|---|---|---|---|---|---|---|---|---|---|---|---|
| gpt-5-2025-08-07 | 34.00 | 12.00 | 100.00 | 100.00 | 49.00 | 53.00 | 87.00 | 13.00 | 95.00 | 14.00 | 55.70 |
| grok-4 | 34.00 | 13.00 | 99.00 | 100.00 | 43.00 | 48.00 | 55.00 | 4.00 | 98.00 | 34.00 | 52.80 |
| gemini-2.5-pro | 27.00 | 1.00 | 100.00 | 99.00 | 15.00 | 45.00 | 57.00 | 6.00 | 92.00 | 37.00 | 47.90 |
| claude-sonnet-4-20250514-thinking | 30.00 | 1.00 | 99.00 | 100.00 | 21.00 | 45.00 | 54.00 | 2.00 | 91.00 | 24.00 | 46.70 |
| o4-mini-2025-04-16 | 31.00 | 0.00 | 99.00 | 99.00 | 38.00 | 35.00 | 59.00 | 6.00 | 92.00 | 4.00 | 46.30 |
| deepseek-r1-250528 | 35.00 | 0.00 | 96.00 | 100.00 | 28.00 | 38.00 | 50.00 | 0.00 | 93.00 | 4.00 | 44.40 |
| deepseek-reasoner | 35.00 | 0.00 | 93.00 | 99.00 | 30.00 | 32.00 | 51.00 | 0.00 | 96.00 | 3.00 | 43.90 |
| glm-4.5 | 34.00 | 0.00 | 98.00 | 100.00 | 16.00 | 35.00 | 49.00 | 0.00 | 89.00 | 7.00 | 42.80 |
| doubao-seed-1-6-250615 | 26.00 | 0.00 | 94.00 | 98.00 | 26.00 | 37.00 | 47.00 | 3.00 | 93.00 | 1.00 | 42.50 |
| doubao-seed-1-6-thinking-250615 | 31.00 | 2.00 | 86.00 | 98.00 | 24.00 | 33.00 | 46.00 | 7.00 | 85.00 | 2.00 | 41.40 |
| doubao-1.5-thinking-pro-250415 | 30.00 | 0.00 | 90.00 | 99.00 | 19.00 | 30.00 | 47.00 | 1.00 | 88.00 | 1.00 | 40.50 |
| gpt-oss-120b | 33.00 | 0.00 | 88.00 | 88.00 | 22.00 | 16.00 | 69.00 | 1.00 | 81.00 | 5.00 | 40.30 |
| qwen3-32b | 29.00 | 0.00 | 97.00 | 100.00 | 14.00 | 17.00 | 48.00 | 1.00 | 91.00 | 3.00 | 40.00 |
| qwen3-235b-a22b | 33.00 | 0.00 | 94.00 | 99.00 | 12.00 | 22.00 | 43.00 | 0.00 | 87.00 | 3.00 | 39.30 |
| claude-3-7-sonnet-20250219-thinking | 29.00 | 0.00 | 88.00 | 99.00 | 11.00 | 44.00 | 39.00 | 2.00 | 78.00 | 2.00 | 39.20 |
| gemini-2.5-flash | 15.00 | 0.00 | 99.00 | 97.00 | 9.00 | 36.00 | 37.00 | 0.00 | 78.00 | 13.00 | 38.40 |
| gpt-oss-20b | 28.00 | 0.00 | 68.00 | 98.00 | 14.00 | 20.00 | 64.00 | 1.00 | 75.00 | 2.00 | 37.00 |
| glm-4.5-air | 31.00 | 0.00 | 93.00 | 91.00 | 15.00 | 27.00 | 35.00 | 0.00 | 76.00 | 1.00 | 36.90 |
| grok-3-mini-beta | 29.00 | 0.00 | 87.00 | 96.00 | 14.00 | 15.00 | 31.00 | 1.00 | 75.00 | 5.00 | 35.30 |
| qwq-32b | 25.00 | 0.00 | 98.00 | 98.00 | 12.00 | 13.00 | 24.00 | 0.00 | 79.00 | 0.00 | 34.90 |
| kimi-k2-0905 | 12.00 | 1.00 | 83.00 | 91.00 | 4.00 | 20.00 | 46.00 | 0.00 | 85.00 | 0.00 | 34.20 |
| deepseek-v3-1-250821 | 27.00 | 0.00 | 70.00 | 88.00 | 8.00 | 25.00 | 21.00 | 0.00 | 69.00 | 2.00 | 31.00 |
| deepseek-chat | 23.00 | 0.00 | 77.00 | 81.00 | 5.00 | 23.00 | 21.00 | 0.00 | 66.00 | 4.00 | 30.00 |
| claude-sonnet-4-20250514 | 6.00 | 0.00 | 54.00 | 93.00 | 4.00 | 30.00 | 34.00 | 0.00 | 74.00 | 3.00 | 29.80 |
| gpt-5-chat-2025-08-07 | 21.00 | 0.00 | 60.00 | 87.00 | 3.00 | 20.00 | 23.00 | 0.00 | 56.00 | 4.00 | 27.40 |
| gpt-4.1 | 21.00 | 0.00 | 42.00 | 83.00 | 6.00 | 17.00 | 16.00 | 0.00 | 62.00 | 2.00 | 24.90 |
| claude-3-7-sonnet-20250219 | 11.00 | 0.00 | 49.00 | 64.00 | 5.00 | 28.00 | 19.00 | 0.00 | 50.00 | 1.00 | 22.70 |
| doubao-seed-1-6-flash-250615 | 8.00 | 0.00 | 51.00 | 84.00 | 3.00 | 4.00 | 22.00 | 0.00 | 42.00 | 1.00 | 21.50 |
| llama-4-maverick | 1.00 | 0.00 | 34.00 | 56.00 | 3.00 | 18.00 | 17.00 | 0.00 | 40.00 | 0.00 | 16.90 |
| doubao-1.5-pro-32k-250115 | 0.00 | 0.00 | 28.00 | 66.00 | 3.00 | 14.00 | 15.00 | 0.00 | 40.00 | 0.00 | 16.60 |
| mistral-small-3.1-24b-instruct | 0.00 | 0.00 | 34.00 | 58.00 | 1.00 | 10.00 | 1.00 | 0.00 | 33.00 | 1.00 | 13.80 |
| gpt-4o | 0.00 | 0.00 | 10.00 | 55.00 | 2.00 | 11.00 | 13.00 | 0.00 | 33.00 | 1.00 | 12.50 |
| gemma-3-27b-it | 2.00 | 0.00 | 22.00 | 35.00 | 1.00 | 5.00 | 8.00 | 0.00 | 25.00 | 1.00 | 9.90 |

Overall, the results reveal moderately strong alignment among classical reasoning benchmarks, with CLUB showing high rank correlation with LogiQA2.0 ($\rho = 0.90$), BIG-Bench Hard ($\rho = 0.90$), LogicGame ($\rho = 0.80$), and ARC-AGI v1 ($\rho = 0.90$). This indicates that CLUB captures many of the same model-ordering tendencies as traditional logic and IQ-style datasets.

In contrast, SynLogic exhibits a more specialized behavior: although it correlates moderately with CLUB ($\rho = 0.67$), it shows weaker alignment with LogiQA2.0 and BIG-Bench Hard (both $\rho = 0.31$). These divergences suggest SynLogic emphasizes a different subset of reasoning skills—especially

Table 18: The model's performance on different CLUB tasks. (Acc, 2025-09-22)

| Model | code | extzebra | k&k | linefg | maze | string | sudoku | tetris | trazebra | traincp | overall |
|---|---|---|---|---|---|---|---|---|---|---|---|
| gpt-5-2025-08-07 | 35.00 | 8.00 | 100.00 | 100.00 | 44.00 | 48.00 | 85.00 | 15.00 | 94.00 | 16.00 | 54.50 |
| grok-4 | 35.00 | 5.00 | 100.00 | 100.00 | 47.00 | 43.00 | 53.00 | 3.00 | 97.00 | 30.00 | 51.30 |
| claude-sonnet-4-20250514-thinking | 32.00 | 3.00 | 99.00 | 100.00 | 20.00 | 41.00 | 58.00 | 4.00 | 92.00 | 18.00 | 46.70 |
| gemini-2.5-pro | 18.00 | 0.00 | 99.00 | 100.00 | 16.00 | 40.00 | 56.00 | 3.00 | 88.00 | 39.00 | 45.90 |
| deepseek-reasoner | 33.00 | 0.00 | 97.00 | 100.00 | 27.00 | 39.00 | 48.00 | 0.00 | 92.00 | 6.00 | 44.20 |
| deepseek-r1-250528 | 34.00 | 0.00 | 97.00 | 99.00 | 26.00 | 38.00 | 46.00 | 0.00 | 97.00 | 3.00 | 44.00 |
| doubao-seed-1-6-250615 | 29.00 | 0.00 | 94.00 | 100.00 | 22.00 | 39.00 | 48.00 | 3.00 | 93.00 | 2.00 | 43.00 |
| glm-4.5 | 30.00 | 0.00 | 93.00 | 99.00 | 18.00 | 33.00 | 54.00 | 0.00 | 89.00 | 7.00 | 42.30 |
| qwen3-32b | 30.00 | 0.00 | 97.00 | 99.00 | 17.00 | 22.00 | 51.00 | 0.00 | 90.00 | 2.00 | 40.80 |
| doubao-1.5-thinking-pro-250415 | 30.00 | 0.00 | 87.00 | 100.00 | 19.00 | 31.00 | 48.00 | 0.00 | 91.00 | 1.00 | 40.70 |
| doubao-seed-1-6-thinking-250615 | 26.00 | 3.00 | 84.00 | 100.00 | 21.00 | 32.00 | 46.00 | 5.00 | 84.00 | 3.00 | 40.40 |
| claude-3-7-sonnet-20250219-thinking | 28.00 | 0.00 | 95.00 | 99.00 | 10.00 | 40.00 | 45.00 | 1.00 | 83.00 | 2.00 | 40.30 |
| gpt-oss-120b | 28.00 | 0.00 | 87.00 | 86.00 | 26.00 | 27.00 | 66.00 | 2.00 | 77.00 | 4.00 | 40.30 |
| gemini-2.5-flash | 16.00 | 0.00 | 98.00 | 99.00 | 12.00 | 34.00 | 42.00 | 1.00 | 78.00 | 12.00 | 39.20 |
| qwen3-235b-a22b | 32.00 | 0.00 | 96.00 | 98.00 | 13.00 | 20.00 | 46.00 | 0.00 | 83.00 | 1.00 | 38.90 |
| gpt-oss-20b | 31.00 | 0.00 | 68.00 | 98.00 | 17.00 | 21.00 | 63.00 | 1.00 | 78.00 | 0.00 | 37.70 |
| glm-4.5-air | 27.00 | 0.00 | 90.00 | 99.00 | 12.00 | 22.00 | 36.00 | 0.00 | 81.00 | 2.00 | 36.90 |
| grok-3-mini-beta | 28.00 | 0.00 | 85.00 | 97.00 | 14.00 | 19.00 | 33.00 | 1.00 | 74.00 | 4.00 | 35.50 |
| qwq-32b | 25.00 | 0.00 | 93.00 | 98.00 | 13.00 | 17.00 | 24.00 | 0.00 | 77.00 | 1.00 | 34.80 |
| kimi-k2-0905 | 10.00 | 0.00 | 80.00 | 91.00 | 4.00 | 24.00 | 42.00 | 0.00 | 88.00 | 1.00 | 34.00 |
| deepseek-v3-1-250821 | 20.00 | 0.00 | 73.00 | 91.00 | 8.00 | 17.00 | 21.00 | 0.00 | 74.00 | 3.00 | 30.70 |
| deepseek-chat | 21.00 | 0.00 | 77.00 | 88.00 | 4.00 | 21.00 | 17.00 | 0.00 | 69.00 | 4.00 | 30.10 |
| claude-sonnet-4-20250514 | 12.00 | 0.00 | 56.00 | 95.00 | 5.00 | 28.00 | 25.00 | 0.00 | 74.00 | 2.00 | 29.70 |
| gpt-5-chat-2025-08-07 | 23.00 | 0.00 | 63.00 | 86.00 | 3.00 | 25.00 | 24.00 | 0.00 | 55.00 | 4.00 | 28.30 |
| claude-3-7-sonnet-20250219 | 8.00 | 0.00 | 52.00 | 69.00 | 5.00 | 31.00 | 18.00 | 0.00 | 50.00 | 1.00 | 23.40 |
| gpt-4.1 | 16.00 | 0.00 | 36.00 | 80.00 | 4.00 | 17.00 | 19.00 | 0.00 | 59.00 | 1.00 | 23.20 |
| doubao-seed-1-6-flash-250615 | 6.00 | 0.00 | 50.00 | 83.00 | 3.00 | 5.00 | 19.00 | 0.00 | 40.00 | 1.00 | 20.70 |
| doubao-1.5-pro-32k-250115 | 2.00 | 0.00 | 32.00 | 63.00 | 5.00 | 20.00 | 10.00 | 0.00 | 35.00 | 0.00 | 16.70 |
| llama-4-maverick | 1.00 | 0.00 | 31.00 | 63.00 | 2.00 | 11.00 | 17.00 | 1.00 | 33.00 | 1.00 | 16.00 |
| mistral-small-3.1-24b-instruct | 1.00 | 0.00 | 26.00 | 56.00 | 2.00 | 7.00 | 3.00 | 0.00 | 43.00 | 1.00 | 13.90 |
| gpt-4o | 0.00 | 0.00 | 12.00 | 60.00 | 2.00 | 11.00 | 12.00 | 0.00 | 28.00 | 1.00 | 12.60 |
| gemma-3-27b-it | 0.00 | 0.00 | 19.00 | 35.00 | 1.00 | 4.00 | 13.00 | 0.00 | 28.00 | 2.00 | 10.20 |

Table 19: The model's performance on different CLUB tasks. (Cell Acc, 2025-09-17)

| Model | code | extzebra | k&k | linefg | maze | string | sudoku | tetris | trazebra | traincp | overall |
|---|---|---|---|---|---|---|---|---|---|---|---|
| grok-4 | 87.39 | 76.10 | 95.47 | 17.28 | 35.98 | 71.38 | 73.60 | 42.75 | 97.75 | 16.72 | 61.44 |
| gpt-5-2025-08-07 | 87.10 | 34.19 | 100.00 | 17.50 | 25.85 | 61.76 | 80.84 | 79.83 | 97.69 | 13.60 | 59.84 |
| doubao-seed-1-6-250615 | 78.82 | 33.55 | 93.76 | 17.20 | 23.04 | 45.28 | 44.51 | 74.40 | 98.00 | 9.02 | 51.76 |
| gemini-2.5-pro-preview-06-05 | 69.24 | 33.11 | 100.00 | 16.35 | 15.70 | 46.52 | 38.24 | 79.97 | 95.38 | 19.94 | 51.45 |
| gemini-2.5-pro-preview-06-05-thinking | 64.56 | 30.90 | 100.00 | 15.78 | 12.29 | 50.98 | 41.10 | 77.93 | 95.12 | 21.68 | 51.03 |
| deepseek-r1-250528 | 83.64 | 25.99 | 98.29 | 17.24 | 19.90 | 47.59 | 34.36 | 73.53 | 98.81 | 5.78 | 50.51 |
| doubao-1.5-thinking-pro-250415 | 84.88 | 32.49 | 95.37 | 16.21 | 13.93 | 41.18 | 38.85 | 76.97 | 98.12 | 5.41 | 50.34 |
| gemini-2.5-pro-preview-05-06-thinking | 70.71 | 28.87 | 100.00 | 17.36 | 12.10 | 47.57 | 36.71 | 72.16 | 89.25 | 20.86 | 49.56 |
| doubao-seed-1-6-thinking-250615 | 86.03 | 28.46 | 85.90 | 17.07 | 21.30 | 46.42 | 34.07 | 73.51 | 99.38 | 3.17 | 49.53 |
| deepseek-reasoner | 78.44 | 29.68 | 93.45 | 16.06 | 13.55 | 35.38 | 29.45 | 76.22 | 94.31 | 5.83 | 47.24 |
| glm-4.5 | 85.43 | 11.04 | 99.09 | 16.72 | 14.44 | 43.27 | 26.36 | 72.07 | 94.56 | 2.75 | 46.57 |
| gpt-oss-120b | 76.42 | 5.88 | 96.58 | 17.33 | 13.53 | 34.75 | 44.51 | 72.45 | 90.19 | 1.83 | 45.35 |
| gemini-2.5-pro-preview-05-06 | 54.11 | 32.30 | 97.68 | 20.04 | 13.72 | 54.64 | 33.67 | 72.70 | 36.81 | 19.62 | 43.53 |
| o4-mini-2025-04-16 | 56.54 | 9.65 | 97.38 | 17.49 | 17.89 | 49.44 | 39.72 | 42.30 | 95.19 | 1.69 | 42.73 |
| claude-sonnet-4-20250514-thinking | 74.69 | 21.71 | 73.82 | 15.74 | 8.40 | 43.87 | 31.99 | 73.73 | 62.19 | 15.65 | 42.18 |
| claude-sonnet-4-20250514 | 73.54 | 26.08 | 75.83 | 15.85 | 5.76 | 35.65 | 28.51 | 75.28 | 72.88 | 4.88 | 41.43 |
| glm-z1-32b | 85.92 | 19.22 | 86.10 | 17.89 | 8.71 | 27.45 | 14.89 | 59.56 | 84.00 | 1.93 | 40.57 |
| gpt-oss-20b | 81.24 | 1.29 | 81.47 | 17.04 | 7.77 | 25.16 | 44.41 | 58.46 | 84.44 | 0.61 | 40.19 |
| kimi-k2-0905 | 44.24 | 23.29 | 88.22 | 13.47 | 4.83 | 32.37 | 30.36 | 65.12 | 96.69 | 2.34 | 40.09 |
| claude-3-7-sonnet-20250219 | 68.83 | 24.70 | 69.99 | 16.82 | 6.88 | 38.02 | 22.81 | 73.82 | 73.94 | 4.22 | 40.00 |
| glm-4.5-air | 86.64 | 10.78 | 99.30 | 14.97 | 9.61 | 34.10 | 17.75 | 33.77 | 85.94 | 0.62 | 39.35 |
| gpt-5-chat-2025-08-07 | 78.87 | 22.90 | 62.03 | 14.25 | 8.76 | 29.78 | 33.28 | 67.52 | 69.00 | 3.18 | 38.96 |
| gemini-2.5-flash-preview-04-17-thinking | 50.74 | 10.48 | 91.84 | 16.34 | 10.48 | 35.88 | 26.56 | 72.14 | 66.12 | 6.48 | 38.71 |
| claude-3-7-sonnet-20250219-thinking | 59.02 | 26.24 | 71.20 | 15.78 | 5.94 | 42.73 | 26.88 | 56.05 | 75.06 | 4.81 | 38.37 |
| grok-3-mini-beta | 85.37 | 19.52 | 87.11 | 11.35 | 7.53 | 32.00 | 17.80 | 33.26 | 84.00 | 3.98 | 38.19 |
| qwq-32b | 68.98 | 16.61 | 87.21 | 11.89 | 7.67 | 23.77 | 10.29 | 64.82 | 78.38 | 1.37 | 37.10 |
| doubao-seed-1-6-flash-250615 | 72.93 | 17.63 | 69.39 | 16.07 | 6.14 | 17.75 | 23.18 | 75.31 | 68.56 | 1.43 | 36.84 |
| deepseek-v3-1-250821 | 68.69 | 15.07 | 74.62 | 17.16 | 7.13 | 36.73 | 12.00 | 66.74 | 68.06 | 1.34 | 36.75 |
| qwen3-235b-a22b | 48.11 | 22.03 | 59.92 | 6.47 | 4.50 | 27.96 | 33.50 | 76.09 | 67.62 | 2.61 | 34.88 |
| qwen3-32b | 53.39 | 18.73 | 63.85 | 10.53 | 4.84 | 26.33 | 23.13 | 76.60 | 59.31 | 1.28 | 33.80 |
| doubao-1.5-pro-32k-250115 | 51.43 | 19.72 | 62.84 | 6.71 | 5.96 | 28.09 | 14.76 | 74.92 | 64.00 | 0.67 | 32.91 |
| gpt-4.1 | 66.87 | 17.95 | 38.87 | 15.59 | 6.71 | 32.26 | 25.43 | 69.19 | 50.25 | 1.36 | 32.45 |
| gemma-3-27b-it | 45.48 | 20.53 | 50.96 | 21.02 | 3.84 | 18.20 | 27.82 | 76.18 | 51.44 | 1.74 | 31.72 |
| gemini-2.5-flash-preview-04-17 | 53.22 | 14.45 | 15.41 | 22.09 | 8.11 | 42.82 | 22.78 | 70.14 | 47.31 | 6.97 | 30.33 |
| glm-4-32b | 38.59 | 18.46 | 48.84 | 4.43 | 4.37 | 14.80 | 15.72 | 73.43 | 71.62 | 2.19 | 29.24 |
| llama-4-maverick | 44.04 | 20.58 | 65.86 | 12.46 | 4.61 | 27.65 | 12.12 | 45.41 | 56.25 | 2.01 | 29.10 |
| deepseek-chat | 43.03 | 14.03 | 46.93 | 11.00 | 6.08 | 23.96 | 12.56 | 72.54 | 57.88 | 0.90 | 28.89 |
| mistral-small-3.1-24b-instruct | 24.70 | 17.44 | 52.57 | 3.64 | 3.15 | 20.08 | 3.46 | 68.25 | 39.81 | 0.59 | 23.37 |
| gpt-4o | 34.75 | 14.56 | 20.44 | 9.35 | 2.37 | 24.46 | 12.59 | 57.70 | 47.88 | 1.23 | 22.53 |

multi-step symbolic manipulation—leading to model rankings that deviate from those observed on conventional benchmarks.

Notably, Zebra and ARC-AGI v1 demonstrate strong mutual consistency with SynLogic ($\rho = 0.87$ and $\rho = 0.87$, respectively), forming a distinct cluster of benchmarks that appear to stress structured, multi-constraint reasoning patterns rather than short-form logical entailment.

Taken together, these results highlight that:

Table 20: The model's performance on different CLUB tasks. (Cell Acc, 2025-09-20)

| Model | code | extzebra | k&k | linefg | maze | string | sudoku | tetris | trazebra | traincp | overall |
|---|---|---|---|---|---|---|---|---|---|---|---|
| gpt-5-2025-08-07 | 86.87 | 33.23 | 100.00 | 17.32 | 28.53 | 60.34 | 79.76 | 78.35 | 98.50 | 9.96 | 59.29 |
| claude-sonnet-4-20250514-thinking | 85.51 | 27.00 | 99.19 | 16.73 | 15.27 | 54.99 | 41.40 | 77.55 | 97.25 | 16.22 | 53.11 |
| gemini-2.5-pro | 68.08 | 33.59 | 100.00 | 16.43 | 14.78 | 51.78 | 36.98 | 79.12 | 96.12 | 20.68 | 51.76 |
| deepseek-reasoner | 87.50 | 26.01 | 93.66 | 17.39 | 21.46 | 45.12 | 36.66 | 75.66 | 98.94 | 6.09 | 50.85 |
| deepseek-r1-250528 | 87.79 | 24.49 | 96.17 | 17.69 | 19.70 | 47.46 | 33.97 | 74.60 | 97.00 | 5.86 | 50.47 |
| doubao-1.5-thinking-pro-250415 | 85.89 | 31.13 | 92.25 | 16.65 | 13.20 | 39.14 | 37.82 | 77.34 | 95.81 | 6.36 | 49.56 |
| o4-mini-2025-04-16 | 83.75 | 13.89 | 98.19 | 17.00 | 23.31 | 50.05 | 36.90 | 65.85 | 95.19 | 1.74 | 48.59 |
| glm-4.5 | 86.29 | 23.34 | 97.89 | 16.95 | 11.40 | 44.30 | 28.31 | 75.78 | 93.12 | 5.67 | 48.30 |
| doubao-seed-1-6-250615 | 69.38 | 27.79 | 94.76 | 16.44 | 19.77 | 42.70 | 39.50 | 69.35 | 96.56 | 5.76 | 48.20 |
| claude-3-7-sonnet-20250219-thinking | 83.64 | 28.00 | 92.04 | 15.76 | 8.64 | 49.55 | 34.46 | 76.01 | 88.12 | 4.45 | 48.07 |
| qwen3-32b | 85.22 | 26.27 | 97.48 | 14.69 | 12.52 | 27.09 | 34.61 | 76.49 | 97.62 | 2.82 | 47.48 |
| qwen3-235b-a22b | 86.93 | 27.05 | 95.57 | 15.53 | 9.78 | 31.13 | 35.57 | 76.09 | 92.38 | 2.43 | 47.25 |
| doubao-seed-1-6-thinking-250615 | 85.19 | 29.12 | 91.44 | 16.40 | 16.50 | 46.79 | 30.98 | 62.04 | 90.94 | 2.51 | 47.19 |
| grok-4 | 84.07 | 14.03 | 98.49 | 17.54 | 19.62 | 82.03 | 36.58 | 3.33 | 99.44 | 6.40 | 46.15 |
| gpt-oss-120b | 82.54 | 5.51 | 89.12 | 16.98 | 9.87 | 35.47 | 54.75 | 71.76 | 87.38 | 1.96 | 45.53 |
| claude-sonnet-4-20250514 | 73.48 | 25.30 | 75.13 | 16.30 | 7.00 | 36.90 | 31.42 | 75.68 | 85.00 | 4.77 | 43.10 |
| qwq-32b | 71.52 | 20.32 | 98.99 | 12.38 | 9.22 | 24.13 | 11.68 | 76.09 | 90.06 | 1.23 | 41.56 |
| gemini-2.5-flash | 40.58 | 13.25 | 99.19 | 17.00 | 11.25 | 44.65 | 30.63 | 72.01 | 80.25 | 5.81 | 41.46 |
| claude-3-7-sonnet-20250219 | 66.84 | 24.95 | 73.31 | 15.53 | 6.65 | 37.70 | 24.34 | 75.83 | 77.69 | 4.10 | 40.69 |
| gpt-oss-20b | 79.31 | 0.78 | 79.25 | 17.32 | 5.42 | 33.44 | 47.45 | 53.37 | 81.69 | 0.38 | 39.84 |
| gpt-5-chat-2025-08-07 | 76.91 | 21.91 | 66.67 | 15.68 | 7.76 | 32.63 | 35.65 | 68.17 | 69.25 | 3.26 | 39.79 |
| grok-3-mini-beta | 84.21 | 20.07 | 91.14 | 13.96 | 9.00 | 29.52 | 14.17 | 47.98 | 83.06 | 3.85 | 39.70 |
| kimi-k2-0905 | 40.00 | 18.80 | 86.61 | 15.20 | 5.25 | 31.10 | 32.19 | 62.14 | 88.44 | 2.35 | 38.21 |
| glm-4.5-air | 86.23 | 13.16 | 96.78 | 13.88 | 10.03 | 33.70 | 14.37 | 26.92 | 85.31 | 0.28 | 38.07 |
| deepseek-v3-1-250821 | 68.60 | 14.93 | 69.39 | 17.90 | 7.72 | 39.77 | 11.87 | 69.38 | 57.50 | 1.00 | 35.81 |
| gpt-4.1 | 77.81 | 20.32 | 41.69 | 15.10 | 6.54 | 27.56 | 22.04 | 61.42 | 73.12 | 0.92 | 34.65 |
| doubao-seed-1-6-flash-250615 | 60.14 | 14.68 | 61.83 | 15.46 | 4.95 | 13.82 | 14.96 | 73.91 | 65.31 | 1.52 | 32.66 |
| doubao-1.5-pro-32k-250115 | 48.23 | 19.40 | 59.92 | 7.51 | 5.88 | 29.13 | 18.83 | 74.20 | 61.56 | 0.52 | 32.52 |
| deepseek-chat | 65.43 | 1.68 | 78.45 | 16.56 | 3.64 | 36.11 | 5.28 | 52.55 | 62.62 | 0.81 | 32.31 |
| llama-4-maverick | 45.86 | 21.91 | 65.66 | 12.79 | 4.29 | 26.65 | 10.98 | 43.58 | 62.75 | 2.25 | 29.67 |
| gemma-3-27b-it | 43.75 | 17.72 | 49.85 | 18.64 | 3.46 | 13.84 | 20.37 | 67.05 | 53.81 | 1.50 | 29.00 |
| mistral-small-3.1-24b-instruct | 24.36 | 17.26 | 61.83 | 4.81 | 2.85 | 21.04 | 6.02 | 72.03 | 61.75 | 2.68 | 27.46 |
| gpt-4o | 30.25 | 13.66 | 19.03 | 8.16 | 2.87 | 28.81 | 15.53 | 61.40 | 54.25 | 1.20 | 23.52 |

Table 21: The model's performance on different CLUB tasks. (Cell Acc, 2025-09-22)

| Model | code | extzebra | k&k | linefg | maze | string | sudoku | tetris | trazebra | traincp | overall |
|---|---|---|---|---|---|---|---|---|---|---|---|
| gpt-5-2025-08-07 | 86.96 | 29.49 | 100.00 | 17.41 | 26.57 | 57.93 | 76.06 | 80.20 | 98.31 | 11.25 | 58.42 |
| claude-sonnet-4-20250514-thinking | 86.46 | 25.58 | 99.40 | 17.41 | 15.01 | 49.29 | 43.87 | 77.81 | 97.75 | 15.06 | 52.76 |
| deepseek-r1-250528 | 84.68 | 23.99 | 97.48 | 16.94 | 18.62 | 50.15 | 37.30 | 73.93 | 99.38 | 6.18 | 50.87 |
| deepseek-reasoner | 84.65 | 23.25 | 97.48 | 17.31 | 18.39 | 49.68 | 36.83 | 74.94 | 96.31 | 5.60 | 50.44 |
| doubao-seed-1-6-250615 | 78.70 | 32.14 | 94.56 | 16.71 | 18.42 | 48.57 | 40.06 | 66.77 | 98.12 | 3.79 | 49.78 |
| gemini-2.5-pro | 50.74 | 32.24 | 98.19 | 15.88 | 12.84 | 49.55 | 40.90 | 79.71 | 93.56 | 21.59 | 49.52 |
| doubao-1.5-thinking-pro-250415 | 84.65 | 31.18 | 90.13 | 17.11 | 14.09 | 39.93 | 39.30 | 76.92 | 95.75 | 5.81 | 49.49 |
| glm-4.5 | 85.45 | 22.83 | 96.68 | 16.79 | 13.60 | 43.44 | 34.12 | 76.60 | 94.69 | 5.11 | 48.93 |
| claude-3-7-sonnet-20250219-thinking | 84.85 | 27.49 | 96.78 | 16.99 | 9.63 | 46.88 | 31.94 | 76.29 | 90.44 | 4.70 | 48.60 |
| qwen3-32b | 86.18 | 25.02 | 98.09 | 15.34 | 11.53 | 30.57 | 38.29 | 76.80 | 95.25 | 2.34 | 47.94 |
| qwen3-235b-a22b | 86.52 | 28.02 | 98.19 | 16.09 | 9.18 | 30.41 | 39.22 | 76.00 | 91.31 | 2.27 | 47.72 |
| grok-4 | 86.72 | 5.37 | 100.00 | 17.59 | 20.96 | 89.87 | 34.71 | 2.08 | 99.00 | 4.39 | 46.07 |
| doubao-seed-1-6-thinking-250615 | 82.45 | 28.99 | 85.60 | 16.89 | 15.56 | 42.40 | 34.41 | 61.70 | 89.12 | 2.95 | 46.01 |
| gpt-oss-120b | 75.12 | 7.72 | 91.84 | 18.51 | 13.29 | 41.25 | 51.00 | 72.41 | 86.50 | 2.10 | 45.97 |
| claude-sonnet-4-20250514 | 75.12 | 26.04 | 75.93 | 16.79 | 5.80 | 37.34 | 31.52 | 75.25 | 86.06 | 4.60 | 43.45 |
| gemini-2.5-flash | 51.08 | 15.46 | 97.89 | 17.73 | 12.07 | 41.93 | 30.54 | 69.63 | 81.88 | 7.12 | 42.53 |
| qwq-32b | 72.58 | 21.64 | 94.46 | 13.94 | 9.88 | 27.68 | 13.33 | 74.79 | 85.69 | 1.12 | 41.51 |
| claude-3-7-sonnet-20250219 | 70.88 | 26.43 | 72.41 | 15.38 | 6.42 | 38.41 | 25.28 | 74.32 | 75.25 | 4.40 | 40.92 |
| gpt-5-chat-2025-08-07 | 78.79 | 23.59 | 67.57 | 16.19 | 7.98 | 33.07 | 34.51 | 68.37 | 69.69 | 3.32 | 40.31 |
| grok-3-mini-beta | 83.61 | 20.51 | 91.44 | 12.55 | 7.49 | 32.82 | 16.93 | 48.34 | 82.69 | 4.22 | 40.06 |
| gpt-oss-20b | 83.23 | 0.97 | 76.74 | 17.05 | 5.87 | 37.68 | 46.09 | 48.96 | 82.31 | 0.44 | 39.93 |
| kimi-k2-0905 | 42.40 | 20.05 | 81.37 | 13.03 | 5.10 | 32.19 | 28.04 | 66.68 | 94.69 | 2.76 | 38.63 |
| glm-4.5-air | 85.08 | 15.25 | 93.55 | 13.85 | 9.26 | 34.56 | 14.47 | 30.55 | 80.94 | 0.38 | 37.79 |
| gpt-4.1 | 66.35 | 21.38 | 45.02 | 17.33 | 6.62 | 31.37 | 24.22 | 71.40 | 66.31 | 1.32 | 35.12 |
| doubao-seed-1-6-flash-250615 | 59.77 | 13.94 | 69.69 | 13.31 | 5.46 | 16.24 | 17.13 | 74.34 | 65.44 | 1.58 | 33.69 |
| doubao-1.5-pro-32k-250115 | 53.97 | 19.26 | 62.34 | 8.49 | 6.31 | 29.32 | 17.01 | 75.33 | 60.25 | 0.86 | 33.31 |
| deepseek-v3-1-250821 | 51.66 | 14.47 | 71.80 | 16.64 | 6.98 | 29.88 | 13.06 | 63.19 | 61.50 | 0.92 | 33.01 |
| deepseek-chat | 60.61 | 2.26 | 75.53 | 15.87 | 4.39 | 37.68 | 5.73 | 56.07 | 62.88 | 0.77 | 32.18 |
| mistral-small-3.1-24b-instruct | 34.95 | 20.78 | 61.03 | 4.78 | 3.56 | 20.18 | 8.52 | 75.45 | 64.00 | 2.28 | 29.55 |
| gemma-3-27b-it | 42.68 | 18.43 | 55.89 | 15.28 | 4.00 | 11.89 | 22.81 | 62.56 | 56.56 | 1.73 | 29.18 |
| llama-4-maverick | 48.46 | 21.13 | 66.16 | 11.67 | 4.46 | 21.92 | 11.53 | 46.12 | 51.12 | 2.23 | 28.48 |
| gpt-4o | 27.19 | 15.83 | 20.04 | 8.24 | 2.60 | 26.93 | 13.48 | 64.94 | 51.75 | 1.48 | 23.25 |

1. CLUB bridges both classical logic tests and structure-heavy reasoning datasets, maintaining high correlation with both groups;

2. Existing benchmarks are far from redundant—several measure meaningfully different reasoning axes;

3. CLUB's diverse task composition enables it to capture a broader spectrum of model weaknesses that may not surface in single-style benchmarks.

## D.8 CONSISTENCY AND VARIANCE ANALYSIS ACROSS EVOLVED VERSIONS OF CLUB

To assess the stability and consistency of the CLUB benchmark under structural expansion and difficulty scaling, we construct two evolved variants based on CLUB1.0 (the version used in the main text): (1) ExCLUB (structural evolution): the number of instances per task is expanded from 10 to 100,

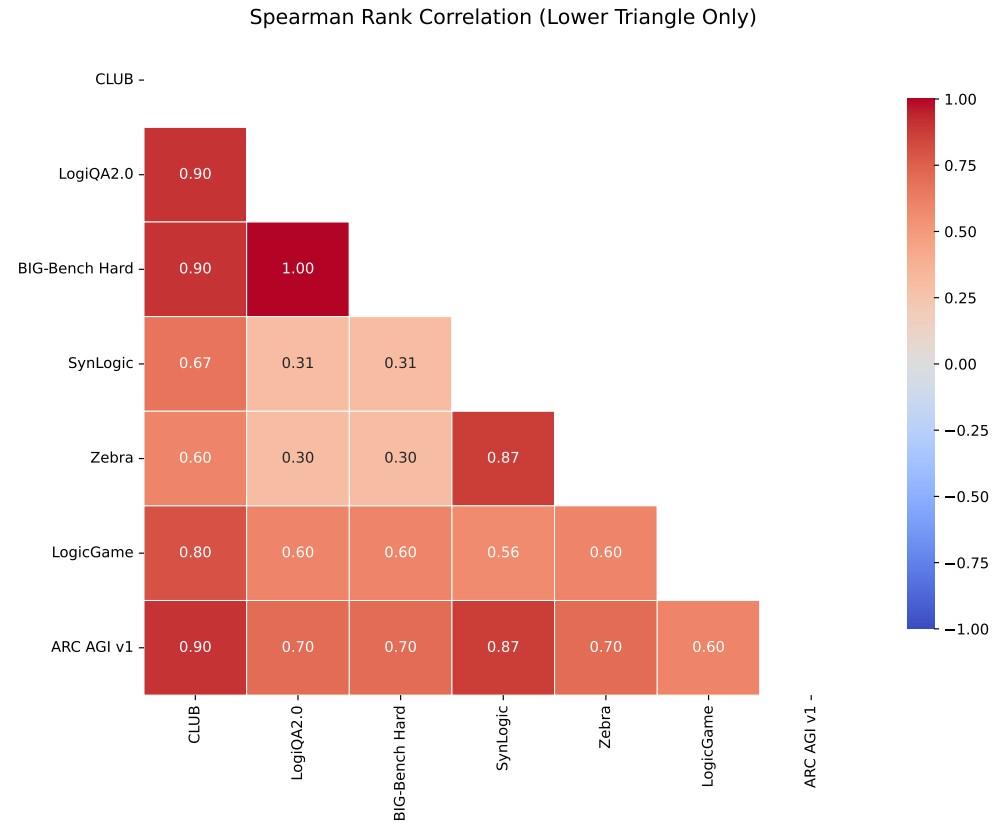

Figure 18: Heatmap of Spearman's rank correlation coefficients between model rankings.

while only the simplest 2–3 difficulty levels are retained, allowing us to test generalization consistency under large-scale task diversification; (2) CLUB2.0 (difficulty evolution): key hyperparameters are upgraded by one level of complexity (e.g., expanding a 3×3 grid to 3×4), allowing us to test whether higher complexity alters model ranking behavior.

**Cross-version correlation analysis.** We evaluate five frontier models (grok-4, gpt-5-thinking, gemini-2.5-pro, o4-mini, deepseek-r1) on all three versions and compute the Spearman rank correlations between their model orderings. As shown in Figure 19, all three versions exhibit strong positive correlations: CLUB1.0 vs. CLUB2.0 shows the highest correlation, indicating that increased difficulty does not alter the underlying reasoning dimensions being measured; CLUB1.0 vs. ExCLUB also remains highly consistent—model rankings stay stable even when task quantity expands by 10×; CLUB2.0 vs. ExCLUB likewise shows significant positive correlation, suggesting that structural and difficulty expansions do not introduce additional noise dimensions.

**Cross-version variance analysis.** To further examine whether the evolved versions introduce significant evaluation fluctuations, we compute each model's performance variance across CLUB1.0, CLUB2.0, and ExCLUB, together with its Coefficient of Variation (CV). Empirically, $CV < 5\%$ is considered highly stable, and 5%–10% falls within normal fluctuation.

Table 22 shows that the cross-version CV for all five models lies within 4.2%–8.9%, well inside the normal range, with no statistically meaningful instability. In other words, although structural differences across evolved versions produce meaningful performance shifts (reflecting expanded task diversity), they do not disrupt model ranking or introduce uncontrolled evaluation drift.

**Sources of evaluation stability.** It is important to emphasize that the observed "heterogeneity" arises primarily from task-space expansion (increased task categories, enlarged hyperparameter space,

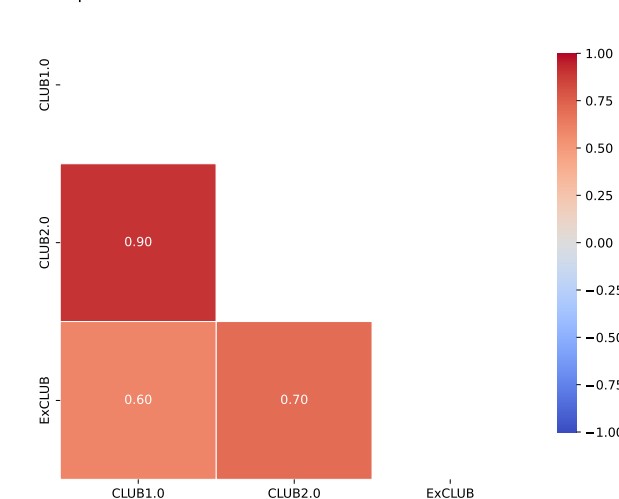

Figure 19: Spearman rank-correlation heatmap of model rankings across different evolved versions of CLUB.

Table 22: Model performance variance and coefficient of variation (CV) across CLUB1.0, CLUB2.0, and ExCLUB. A CV below 10% indicates normal and stable fluctuation.

| Model | CLUB1.0 | CLUB2.0 | ExCLUB | Mean | Variance | CV (%) |
|---|---|---|---|---|---|---|
| grok-4 | 56.37 | 55.00 | 60.63 | 57.33 | 05.75 | 04.20 |
| gpt-5-thinking | 55.57 | 52.00 | 49.18 | 52.25 | 06.83 | 05.00 |
| gemini-2.5-pro | 46.57 | 44.00 | 51.74 | 47.44 | 10.37 | 06.80 |
| o4-mini | 45.20 | 45.00 | 54.20 | 48.13 | 18.41 | 08.90 |
| deepseek-r1 | 44.13 | 41.00 | 47.10 | 44.08 | 06.20 | 05.70 |

and shifts in reasoning patterns), rather than instability in the LogicEvolve framework itself. Under fixed task metadata, the generator–solver iteration converges to a stable implementation, and repeated executions produce highly homogeneous results. Differences across versions stem from intentional upgrades of task structure, not from evaluation noise. Consequently, CLUB can expand task coverage through evolution while preserving a unified evaluation dimension.

**Stability within a single version.** To mitigate incidental variations due to model sampling randomness, each model is evaluated via three independent runs and we report the averaged results. This yields stable and reliable measurements within each version.

In summary, CLUB maintains highly consistent model-ranking structures across evolved versions of different scales and complexities (all correlations are strongly positive), and cross-version fluctuations remain within an acceptable range ($CV < 10\%$). No evidence of evaluation instability is observed. The heterogeneity introduced by LogicEvolve reflects task-space expansion, not evaluation noise. CLUB thus serves as a sustainably evolving benchmark framework whose evaluation dimension remains robust even under substantial increases in task quantity and complexity.

### D.9  PERFORMANCE ON ADDITIONAL BENCHMARK SUITES

As shown in Table 18, the CLUB evaluation across diverse logical reasoning tasks further highlights model-level differences. The overall trend is partially consistent with performance on other domains such as mathematics, code generation, and agent function execution (Table 23). For example, reasoning-enhanced models such as *deepseek-r1* consistently outperform their corresponding base models on most tasks, indicating that CLUB aligns with mainstream benchmarks in verifying the advantages of reasoning-oriented models.

However, CLUB also exposes fine-grained distinctions that existing benchmarks fail to capture. On classical logic tasks such as *traditional_zebra* and *knights&knaves*, most frontier models achieve accuracies above 90%, consistent with their strong results on general-purpose benchmarks like MMLU. In contrast, on extended or non-monotonic tasks such as *extensible_zebra*, *tetris*, and *train_game_card_puzzle*, nearly all models experience a dramatic performance drop to the 0–40% range, in stark contrast to their high scores on conventional benchmarks. This divergence indicates that CLUB is able to reveal "long-tail" challenges in logical reasoning that are otherwise overlooked.

In summary, CLUB maintains consistency with existing benchmarks in overall trends, while uncovering unique weaknesses in higher-order and complex reasoning tasks. This underscores its necessity and complementarity as a dedicated benchmark for systematic evaluation of logical reasoning capabilities.

Table 23: Model performance on other benchmark suites.

| x | Model Type | mathbench | math-500 latest | ifevl-ch | ifevl-en | sysbench | NCB | lcb-v5 | MMLU | GPQA | zebra | logicgame cot |
|---|---|---|---|---|---|---|---|---|---|---|---|---|
| deepseek-chat | base | 75.31 | 96.2 | 78.07 | 83.36 | 79.84 | 56.71 | 36.53 | 89.08 | 67.73 | 84.5 | 42.93 |
| deepseek-r1 | reasoning | 79.25 | 97.96 | 76.65 | 84.29 | 75.11 | 57.98 | 58.79 | 91 | 74.39 | 86.38 | 51.97 |
| gpt-4.1 | base/reasoning | NaN | NaN | 80.42 | 88.91 | 80.56 | 60.73 | 35.33 | 90.52 | 67.12 | 57.4 | 41.94 |
| gpt-4o | base | 57.81 | 77.6 | 74.06 | 81.89 | 78.12 | 57.1 | 29.34 | 85.72 | 45.96 | 29.1 | 26.48 |
| claude-3-7-sonnet-20250219 | base/reasoning | 69.06 | 79.4 | 79.72 | 88.35 | 79 | 56.9 | 35.93 | 89.76 | 67.42 | 43.5 | 42.76 |
| doubao-1.5-pro-32k-250115 | base | 78.12 | 89.2 | 82.55 | 87.25 | 80.12 | 52.87 | 31.74 | 89.88 | 63.43 | 34.6 | 34.54 |

## D.10 STABILITY ANALYSIS OF MODEL PERFORMANCE

In the main text, we reported the mean and half-range across three independent evaluations. Here, we further quantify model stability using the 95% confidence interval (Table 24). If the confidence interval radius is NaN, the subscript is omitted.

Table 24: The model's performance on different CLUB tasks. (mean ± 95% CI)

| Model | code | extzebra | k&k | linefg | maze | string | sudoku | tetris | trazebra | traincp | overall |
|---|---|---|---|---|---|---|---|---|---|---|---|
| claude-3-7-sonnet-20250219 | $8.00_{\pm7.45}$ | 0.00 | $49.67_{\pm5.17}$ | $65.67_{\pm7.17}$ | $5.33_{\pm1.43}$ | $28.67_{\pm5.17}$ | $56.71_{\pm3.79}$ | 0.00 | $49.67_{\pm1.43}$ | 1.00 | $22.73_{\pm1.62}$ |
| claude-3-7-sonnet-20250219-thinking | $24.00_{\pm19.40}$ | 0.00 | $79.67_{\pm51.65}$ | $94.33_{\pm20.08}$ | $8.33_{\pm9.40}$ | $39.67_{\pm11.20}$ | $34.33_{\pm33.82}$ | $1.33_{\pm1.43}$ | $71.33_{\pm39.93}$ | $2.33_{\pm1.43}$ | $35.53_{\pm18.19}$ |
| claude-sonnet-4-20250514 | $9.00_{\pm7.45}$ | 0.00 | $55.67_{\pm3.79}$ | $94.00_{\pm2.48}$ | $4.33_{\pm1.43}$ | $27.67_{\pm6.25}$ | $28.33_{\pm12.25}$ | 0.00 | $70.33_{\pm15.78}$ | $3.00_{\pm2.48}$ | $29.23_{\pm2.23}$ |
| claude-sonnet-4-20250514-thinking | $24.33_{\pm28.79}$ | $1.33_{\pm3.79}$ | $87.33_{\pm50.20}$ | $98.33_{\pm7.17}$ | $16.33_{\pm17.97}$ | $39.00_{\pm17.91}$ | $48.00_{\pm34.78}$ | $2.00_{\pm4.97}$ | $83.33_{\pm35.16}$ | $20.00_{\pm8.61}$ | $42.00_{\pm20.22}$ |
| deepseek-chat | $19.67_{\pm10.34}$ | 0.00 | $63.33_{\pm58.80}$ | $76.00_{\pm37.59}$ | $4.33_{\pm1.43}$ | $20.00_{\pm8.96}$ | $16.67_{\pm11.20}$ | 0.00 | $60.00_{\pm32.48}$ | $3.33_{\pm2.87}$ | $26.33_{\pm15.99}$ |
| deepseek-r1-250528 | $34.00_{\pm2.48}$ | 0.00 | $96.67_{\pm1.43}$ | $98.67_{\pm3.79}$ | $29.33_{\pm10.34}$ | $35.33_{\pm11.47}$ | $48.33_{\pm5.17}$ | 0.00 | $95.00_{\pm4.97}$ | $4.00_{\pm2.48}$ | $44.13_{\pm0.57}$ |
| deepseek-reasoner | $32.67_{\pm6.25}$ | 0.00 | $92.00_{\pm13.83}$ | $99.33_{\pm1.43}$ | $23.33_{\pm22.54}$ | $31.67_{\pm18.64}$ | $46.33_{\pm14.13}$ | 0.00 | $91.00_{\pm13.83}$ | $5.33_{\pm5.17}$ | $42.17_{\pm8.11}$ |
| deepseek-v3-1-250821 | $23.00_{\pm8.96}$ | 0.00 | $72.33_{\pm5.17}$ | $88.67_{\pm5.17}$ | $8.33_{\pm1.43}$ | $21.67_{\pm10.34}$ | $20.67_{\pm1.43}$ | 0.00 | $73.00_{\pm8.96}$ | $2.33_{\pm1.43}$ | $31.00_{\pm0.75}$ |
| doubao-1.5-pro-32k-250115 | $1.33_{\pm2.87}$ | 0.00 | $31.33_{\pm7.59}$ | $64.00_{\pm4.30}$ | $4.33_{\pm2.87}$ | $16.67_{\pm7.59}$ | $12.67_{\pm6.25}$ | 0.00 | $38.00_{\pm6.57}$ | 0.00 | $16.83_{\pm0.80}$ |
| doubao-1.5-thinking-pro-250415 | $30.33_{\pm1.43}$ | $0.33_{\pm1.43}$ | $90.33_{\pm8.72}$ | $99.67_{\pm1.43}$ | $20.33_{\pm5.74}$ | $31.67_{\pm5.17}$ | $48.00_{\pm2.48}$ | $0.33_{\pm1.43}$ | $92.00_{\pm11.38}$ | $1.67_{\pm2.87}$ | $41.47_{\pm3.74}$ |
| doubao-seed-1-6-250615 | $28.33_{\pm5.17}$ | 0.00 | $93.33_{\pm2.87}$ | $99.33_{\pm2.87}$ | $20.33_{\pm9.94}$ | $38.00_{\pm2.48}$ | $49.33_{\pm7.99}$ | $4.33_{\pm5.74}$ | $94.00_{\pm4.30}$ | $3.00_{\pm6.57}$ | $43.57_{\pm3.57}$ |
| doubao-seed-1-6-flash-250615 | $9.00_{\pm8.96}$ | 0.00 | $53.33_{\pm12.25}$ | $83.67_{\pm1.43}$ | $3.33_{\pm1.43}$ | $4.33_{\pm1.43}$ | $23.67_{\pm14.13}$ | 0.00 | $42.00_{\pm4.97}$ | 1.00 | $22.03_{\pm4.14}$ |
| doubao-seed-1-6-thinking-250615 | $30.33_{\pm10.04}$ | $4.00_{\pm6.57}$ | $83.33_{\pm7.59}$ | $99.33_{\pm2.87}$ | $25.00_{\pm11.38}$ | $34.67_{\pm9.40}$ | $48.33_{\pm10.04}$ | $4.00_{\pm8.96}$ | $88.67_{\pm17.97}$ | $2.00_{\pm2.48}$ | $41.97_{\pm4.75}$ |
| gemini-2.5-flash | $15.50_{\pm6.35}$ | 0.00 | $98.50_{\pm6.35}$ | $98.00_{\pm12.71}$ | $10.50_{\pm19.06}$ | $35.00_{\pm12.71}$ | $39.50_{\pm31.77}$ | $0.50_{\pm6.35}$ | 78.00 | $12.50_{\pm6.35}$ | $38.80_{\pm5.08}$ |
| gemini-2.5-pro | $22.50_{\pm57.18}$ | $0.50_{\pm6.35}$ | $99.50_{\pm6.35}$ | $99.50_{\pm6.35}$ | $15.50_{\pm6.35}$ | $42.50_{\pm31.77}$ | $56.50_{\pm6.35}$ | $4.50_{\pm19.06}$ | $90.00_{\pm25.41}$ | $38.00_{\pm12.71}$ | $46.90_{\pm12.71}$ |
| glm-4.5 | $32.00_{\pm4.97}$ | 0.00 | $96.33_{\pm7.17}$ | $99.33_{\pm1.43}$ | $17.33_{\pm2.87}$ | $34.33_{\pm2.87}$ | $50.00_{\pm8.96}$ | 0.00 | 89.00 | $6.67_{\pm1.43}$ | $42.50_{\pm0.66}$ |
| glm-4.5-air | $30.00_{\pm6.57}$ | 0.00 | $93.00_{\pm7.45}$ | $94.00_{\pm10.83}$ | $12.67_{\pm5.17}$ | $22.33_{\pm11.20}$ | $37.00_{\pm6.57}$ | 0.00 | $78.33_{\pm6.25}$ | $1.33_{\pm1.43}$ | $36.87_{\pm0.14}$ |
| gpt-4.1 | $18.67_{\pm6.25}$ | 0.00 | $40.00_{\pm8.61}$ | $82.00_{\pm4.30}$ | $4.33_{\pm3.79}$ | 17.00 | $18.67_{\pm6.25}$ | 0.00 | $60.33_{\pm3.79}$ | $1.33_{\pm1.43}$ | $24.23_{\pm2.25}$ |
| gpt-4o | 0.00 | 0.00 | $12.00_{\pm1.43}$ | $57.33_{\pm6.25}$ | $1.67_{\pm1.43}$ | 11.00 | $11.67_{\pm3.79}$ | 0.00 | $30.33_{\pm6.25}$ | $0.67_{\pm1.43}$ | $12.47_{\pm0.38}$ |
| gpt-5-2025-08-07 | $34.67_{\pm1.43}$ | $11.33_{\pm7.59}$ | 100.00 | 100.00 | $46.33_{\pm6.25}$ | $51.33_{\pm7.17}$ | $86.33_{\pm2.87}$ | $13.67_{\pm2.87}$ | $94.67_{\pm1.43}$ | $17.33_{\pm10.34}$ | $55.57_{\pm2.50}$ |
| gpt-5-chat-2025-08-07 | $22.33_{\pm2.87}$ | 0.00 | $59.00_{\pm11.38}$ | $86.00_{\pm2.48}$ | $3.33_{\pm1.43}$ | $21.67_{\pm7.17}$ | $24.33_{\pm3.79}$ | 0.00 | $56.33_{\pm3.79}$ | 4.00 | $27.70_{\pm1.29}$ |
| o4-mini-2025-04-16 | $26.50_{\pm57.18}$ | 0.00 | $97.50_{\pm19.06}$ | $97.50_{\pm19.06}$ | $33.50_{\pm57.18}$ | $36.50_{\pm19.06}$ | $60.50_{\pm19.06}$ | $3.00_{\pm38.12}$ | $94.00_{\pm25.41}$ | $3.00_{\pm12.71}$ | $45.20_{\pm13.98}$ |
| gemma-3-27b-it | $0.67_{\pm2.87}$ | 0.00 | $22.00_{\pm7.45}$ | $35.33_{\pm1.43}$ | $0.67_{\pm1.43}$ | $5.00_{\pm2.48}$ | $11.00_{\pm6.57}$ | 0.00 | $24.67_{\pm8.72}$ | $1.33_{\pm1.43}$ | $10.07_{\pm0.38}$ |
| gpt-oss-120b | $30.00_{\pm6.57}$ | 0.00 | $90.00_{\pm10.83}$ | $90.00_{\pm13.14}$ | $24.67_{\pm5.74}$ | $21.00_{\pm13.83}$ | $65.67_{\pm8.72}$ | $1.67_{\pm1.43}$ | $80.00_{\pm6.57}$ | $4.67_{\pm1.43}$ | $40.77_{\pm2.01}$ |
| gpt-oss-20b | $30.00_{\pm4.30}$ | 0.00 | $69.33_{\pm5.74}$ | $97.67_{\pm1.43}$ | $16.33_{\pm5.17}$ | $18.33_{\pm9.40}$ | $63.00_{\pm2.48}$ | 1.00 | $73.33_{\pm14.13}$ | $1.00_{\pm2.48}$ | $37.00_{\pm1.74}$ |
| grok-3-mini-beta | $29.00_{\pm2.48}$ | 0.00 | $84.67_{\pm6.25}$ | $97.00_{\pm2.48}$ | $13.67_{\pm1.43}$ | $17.00_{\pm4.97}$ | $31.67_{\pm2.87}$ | $0.67_{\pm1.43}$ | $75.67_{\pm5.17}$ | $4.33_{\pm1.43}$ | $35.37_{\pm0.29}$ |
| grok-4 | $34.67_{\pm1.43}$ | $25.00_{\pm69.56}$ | $98.67_{\pm3.79}$ | 100.00 | $49.33_{\pm19.30}$ | $50.00_{\pm20.33}$ | $63.33_{\pm40.23}$ | $8.67_{\pm22.27}$ | $96.67_{\pm3.79}$ | $37.33_{\pm23.48}$ | $56.37_{\pm18.67}$ |
| kimi-k2-0905 | $10.33_{\pm3.79}$ | $0.33_{\pm1.43}$ | $83.33_{\pm8.72}$ | $91.33_{\pm1.43}$ | $3.33_{\pm2.87}$ | $21.00_{\pm6.57}$ | $43.67_{\pm5.17}$ | 0.00 | $88.00_{\pm7.45}$ | $1.00_{\pm2.48}$ | $34.23_{\pm0.63}$ |
| llama-4-maverick | $1.33_{\pm1.43}$ | 0.00 | $32.67_{\pm3.79}$ | $60.33_{\pm9.40}$ | $2.67_{\pm1.43}$ | $14.67_{\pm8.72}$ | $15.67_{\pm5.74}$ | $0.33_{\pm1.43}$ | $35.67_{\pm9.40}$ | $1.00_{\pm2.48}$ | $16.43_{\pm1.12}$ |
| mistral-small-3.1-24b-instruct | $0.33_{\pm1.43}$ | 0.00 | $25.67_{\pm21.13}$ | $57.00_{\pm2.48}$ | $1.00_{\pm2.48}$ | $8.33_{\pm3.79}$ | $2.67_{\pm3.79}$ | 0.00 | $31.67_{\pm29.95}$ | 1.00 | $12.77_{\pm4.66}$ |
| qwq-32b | $24.67_{\pm1.43}$ | 0.00 | $91.67_{\pm17.62}$ | $94.00_{\pm17.21}$ | $11.00_{\pm6.57}$ | $14.00_{\pm6.57}$ | $23.67_{\pm1.43}$ | 0.00 | $76.33_{\pm7.59}$ | $0.67_{\pm1.43}$ | $33.60_{\pm5.38}$ |
| qwen3-235b-a22b | $21.67_{\pm46.63}$ | 0.00 | $75.00_{\pm86.09}$ | $88.00_{\pm45.19}$ | $9.00_{\pm15.11}$ | $20.00_{\pm4.97}$ | $34.67_{\pm42.47}$ | 0.00 | $70.00_{\pm64.73}$ | $2.00_{\pm2.48}$ | $32.03_{\pm30.41}$ |
| qwen3-32b | $20.00_{\pm40.89}$ | 0.00 | $71.67_{\pm109.00}$ | $87.33_{\pm52.36}$ | $11.00_{\pm19.72}$ | $18.67_{\pm7.17}$ | $38.00_{\pm49.62}$ | $0.33_{\pm1.43}$ | $71.33_{\pm82.48}$ | $1.67_{\pm3.79}$ | $32.00_{\pm36.16}$ |

From an overall perspective, models exhibit substantial divergence across CLUB tasks. Top-performing models such as *grok-4* and *gpt-5* achieve comparable overall scores, yet with distinct stability profiles: the former performs notably better on *train_game_card_puzzle* but suffers from high variance, whereas the latter remains more stable on structured reasoning tasks such as *sudoku*, reflecting a trade-off between performance and stability. Task difficulty also reveals a clear hierarchy: *knights_and_knaves* and *line_forming_games* exhibit near ceiling effects for leading models, while *extensible_zebra* and *tetris* remain consistently low, serving as the most discriminative and challenging tasks. In particular, almost all models achieve near-perfect accuracy on *traditional_zebra*, yet drop

sharply to single-digit accuracy on *extensible_zebra*, highlighting their fragility in handling extensible rules and compositional logic.

Furthermore, "thinking" variants generally improve mean accuracy but substantially enlarge confidence intervals, suggesting that their reasoning relies more heavily on stochastic trajectories—yielding stronger but less stable performance. These findings collectively indicate that while classical logic tasks are increasingly insufficient to differentiate models, extended and compositional logic tasks serve as critical slices for assessing both reasoning capability and robustness.

### D.11 LOGICAL ERROR CASE STUDY

We select a failed example of *gemini-2.5-pro* on the *train_game_card_puzzle* task to highlight its limitations in complex reasoning.

---

**⚠ Train Game Card Puzzle Solve Record**

METADATA

| | |
|---|---|
| **did**: | 3 |
| **qid**: | 0 |
| **tid**: | 1 |
| **created_at**: | 2025-07-09T15:34:29.911423 |
| **delta_time**: | 0.0003809928894042969 seconds |
| **num_problems**: | 1 |
| **num_people**: | 14 |
| **num_cards**: | 6 |
| **num_suits**: | 3 |
| **success**: | false |
| **cell_accuracy**: | 0.7716535433070866 |
| **cell_accuracy_num**: | 98 |
| **cell_accuracy_all**: | 127 |
| **solve_time**: | 185.068237 seconds |
| **total_solve_time**: | 185.06823729723692 |
| **total_pass_need_times**: | 1 |

PUZZLE

A deck of cards contains 1 big joker and 1 little joker, with 3 suits (hearts, spades, and clubs), and 6 cards per suit (numbered 1–6). There are 14 players. The total of 20 cards is not evenly divisible by 14 players, so 6 cards are removed. Each player receives 1 card. Player 1's initial deck is: ["spade 5"]. Player 2's initial deck is: ["heart 1"]. Player 3's initial deck is: ["heart 6"]. Player 4's initial deck is: ["spade 6"]. Player 5's initial deck is: ["heart 2"]. Player 6's initial deck is: ["spade 3"]. Player 7's initial deck is: ["spade 1"]. Player 8's initial deck is: ["spade 4"]. Player 9's initial deck is: ["spade 2"]. Player 10's initial deck is: ["club 1"]. Player 11's initial deck is: ["heart 4"]. Player 12's initial deck is: ["club 4"]. Player 13's initial deck is: ["heart 5"]. Player 14's initial deck is: ["club 3"]. Please provide the sequence of cards played by the players. The play sequence is a list composed of multiple rounds. Each round is represented by a list, where the order of cards indicates the sequence of cards in the shared pile on the table during that round. Every time a train is triggered (i.e., a match occurs and cards are collected), that marks the end of a round. Each element in the list corresponds to the sequence of cards played in that round.

QUESTION (ROUND SKELETON)

```
[["_",  "_",  "_",  "_"],
 ["_",  "_",  "_",  "_",  "_",  "_"],
 ["_",  "_",  "_",  "_",  "_"],
 ["_",  "_",  "_",  "_"],
 ["_",  "_",  "_",  "_"],
 ["_",  "_",  "_",  "_"],
```

---

```
    ["_", "_", "_"],
    ["_", "_", "_"],
    ["_", "_", "_"],
    ["_", "_", "_", "_"],
    ["_", "_", "_", "_"],
    ["_", "_", "_", "_"],
    ["_", "_", "_", "_", "_", "_"],
    ["_", "_", "_", "_", "_"],
    ["_", "_", "_", "_", "_"],
    ["_", "_", "_", "_"],
    ["_", "_", "_", "_"],
    ["_", "_", "_", "_"],
    ["_", "_", "_", "_"],
    ["_", "_", "_", "_"],
    ["_", "_", "_"],
    ["_", "_"],
    ["_", "_", "_", "_", "_"],
    ["_", "_", "_", "_"],
    ["_", "_", "_", "_"],
    ["_", "_", "_", "_"],
    ["_", "_", "_", "_", "_", "_"],
    ["_", "_", "_", "_", "_", "_"],
    ["_", "_", "_", "_", "_"],
    ["_", "_", "_", "_"]]
```

ANSWER (PROPOSED PLAY SEQUENCE)

```
    ["spade 5", "heart 1", "heart 6", "spade 6"],
    ["spade 5", "heart 1", "spade 6", "heart 2",
    "spade 3", "spade 1"],
    ["spade 5", "spade 1", "spade 4", "spade 2", "club 1"],
    ["spade 5", "club 1", "heart 4", "club 4"],
    ["spade 5", "club 1", "club 4", "heart 5"],
    ["heart 5", "club 3", "heart 6", "spade 3"],
    ["heart 5", "heart 2", "spade 2"],
    ["heart 5", "spade 4", "heart 4"],
    ["heart 5", "heart 4", "club 4"],
    ["heart 5", "club 1", "spade 6", "spade 1"],
    ["heart 5", "spade 2", "spade 4", "spade 5"],
    ["club 4", "heart 1", "heart 2", "heart 4"],
    ["spade 5", "spade 3", "spade 1", "spade 4",
    "heart 6", "spade 6"],
    ["spade 5", "spade 3", "spade 1", "spade 4", "club 1"],
    ["spade 5", "spade 3", "spade 6", "spade 2", "club 3"],
    ["spade 5", "club 3", "heart 6", "heart 5"],
    ["heart 4", "spade 2", "club 1", "heart 2"],
    ["heart 4", "heart 1", "spade 6", "spade 4"],
    ["spade 1", "club 4", "spade 3", "spade 4"],
    ["spade 1", "spade 6", "heart 5", "heart 1"],
    ["heart 4", "heart 6", "spade 4"],
    ["spade 3", "club 3"],
    ["spade 5", "club 4", "heart 2", "heart 1", "club 1"],

    (Pay special attention here)
    ["spade 5", "club 4", "heart 2", "spade 2"],
    (Pay special attention here)
```

```
["spade 5", "club 4", "club 3", "heart 5"],
["spade 6", "spade 3", "spade 1", "club 1"],
["spade 6", "spade 3", "heart 1", "spade 4",
"spade 2", "heart 6"],
["heart 4", "heart 2", "heart 5", "club 1",
"club 3", "spade 1"],
["heart 4", "heart 2", "heart 5", "spade 1", "club 4"],
["spade 5", "club 3", "heart 6", "club 1"]]
```

LLM RAW RESPONSE (EXCERPT)

```
{
  "reasoning": "The solution is derived by meticulously...",
  "solution": [
    ["spade 5","heart 1","heart 6","spade 6"],
    ["spade 5","heart 1","spade 6","heart 2",
    "spade 3","spade 1"],
    ...
    ["club 3","spade 6"]
  ]
}
```

PARSED RESPONSE

**response_reason**: The solution is derived by meticulously simulating the Train Game turn by turn, adhering to the specified rules. The simulation starts with the initial state: 14 players, each with a single card, an empty central pile, and Player 1 as the starting player. The process unfolds as follows:

1. **Turn Progression:** Players take turns in a clockwise order (Player 1 to Player 14 and looping back). If a player has no cards, their turn is skipped. 2. **Playing a Card:** On their turn, a player plays the top card from their deck onto the central pile. 3. **Train Collection:** If the played card (the 'engine') has the same rank as a card already in the pile (the 'caboose'), a collection is triggered. 4. **Round Definition:** The state of the central pile just before a collection occurs defines a 'round'. This sequence of cards is recorded. 5. **Collecting Cards:** The player who triggered the collection takes all cards from the 'engine' to the 'caboose', inclusive. These cards are removed from the central pile. 6. **Updating Deck:** The collected cards are placed in order at the bottom of the collecting player's deck, with the 'engine' on top and the 'caboose' at the bottom. 7. **Continuing Turn:** The player who collected the cards immediately takes another turn, playing the new top card from their deck. 8. **Game End:** The simulation continues until only one player has cards left. The sequence of cards played after the final collection is recorded as the last entry in the play sequence, as no further collections can occur to define a new round before the game ends.

By systematically tracking the deck of each player and the state of the central pile through every action, we can accurately determine the sequence of cards for each round until the game concludes. . . .

**response_answer**:

```
[["spade 5", "heart 1", "heart 6", "spade 6"],
["spade 5", "heart 1", "spade 6", "heart 2",
"spade 3", "spade 1"],
["spade 5", "spade 1", "spade 4", "spade 2", "club 1"],
["spade 5", "club 1", "heart 4", "club 4"],
["spade 5", "club 1", "club 4", "heart 5"],
["heart 5", "club 3", "heart 6", "spade 3"],
["heart 5", "heart 2", "spade 2"],
["heart 5", "spade 4", "heart 4"],
["heart 5", "heart 4", "club 4"],
["heart 5", "club 1", "spade 6", "spade 1"],
["heart 5", "spade 2", "spade 4", "spade 5"],
```

```
["club 4", "heart 1", "heart 2", "heart 4"],
["spade 5", "spade 3", "spade 1", "spade 4",
"heart 6", "spade 6"],
["spade 5", "spade 3", "spade 1", "spade 4", "club 1"],
["spade 5", "spade 3", "spade 6", "spade 2", "club 3"],
["spade 5", "club 3", "heart 6", "heart 5"],
["heart 4", "spade 2", "club 1", "heart 2"],
["heart 4", "heart 1", "spade 6", "spade 4"],
["spade 1", "club 4", "spade 3", "spade 4"],
["spade 1", "spade 6", "heart 5", "heart 1"],
["heart 4", "heart 6", "spade 4"],
["spade 3", "club 3"],
["spade 5", "club 4", "heart 2", "heart 1", "club 1"],

(An error occurred here)
["spade 5", "spade 2", "heart 5"],
(An error occurred here)

["spade 6", "club 3", "spade 1", "spade 3"],
["spade 6", "club 1", "spade 4", "heart 1"],
["spade 6", "heart 2", "heart 6"],
["heart 4", "club 4"],
["spade 3", "heart 5", "spade 1", "spade 2", "club 3"],
["heart 1", "spade 5", "spade 4", "heart 6", "club 1"],
["club 4", "heart 2", "heart 4"],
["club 3", "spade 6"]]
```

This case illustrates that in chain-of-thought reasoning, once an intermediate step deviates, the error propagates along the reasoning chain and eventually invalidates the final conclusion. Such error amplification is particularly pronounced in tasks requiring long reasoning paths, where local mistakes are more likely to accumulate into systemic biases. This observation also sheds light on the findings from our stability analysis: while some models achieve competitive average performance, their large confidence intervals stem from occasional but critical local reasoning errors that substantially undermine the reliability of the overall output.

### D.12  COST ANALYSIS

In the evaluation stage, each complete assessment incurred an expense of approximately $2000, with three independent evaluations conducted, resulting in a total cost of around $6000. In contrast, the cost of the generation stage is substantially lower: since the framework can typically complete task generation within about ten rounds of self-evolutionary iterations, the overall resource consumption remains relatively limited.

# E  LOGICAL-REASONING META-DATA

This appendix presents the complete meta-data for all logical-reasoning puzzles. For each puzzle we provide:

1. **Base Assumptions**;

2. **Concrete Example(s)** — each including *Concrete Rules*, *Question*, *Reasoning Process*, and *Answer*.

## E.1  TRADITIONAL ZEBRA

### E.1.1  BASE ASSUMPTIONS

In the traditional *Zebra*, we consider a row of $N$ houses, numbered 1 through $N$ from left to right. Exactly one person lives in each house, and every person is characterised by $M$ *distinct* attribute categories (e.g., nationality, pet, beverage, occupation, house colour).

**Domain constraints.**

1. Each attribute category contains **exactly $N$ mutually exclusive values**.

2. Every value appears **once and only once** across the $N$ houses; that is, the $N$ values of any attribute are assigned to the $N$ houses without repetition.

**Structural assumptions.**

1. The occupant of each house must possess **one unique value** from *every* attribute category, drawn from the full set of values for that category.

2. The ordering of the houses $(1 \rightarrow N)$ is known *a priori*, enabling spatial reasoning such as "$X$ is somewhere to the left of $Y$" or "$X$ is immediately left of $Y$" (i.e., $X$ is to the left of $Y$ with no houses in between).

**Puzzle instance.**  Each individual puzzle provides

- the complete value sets for all attribute categories, and

- a collection of **clues** that express relations among attributes and/or positions.

**Objective.**  Using the given clues, the solver must **uniquely determine** the full set of attribute values associated with the occupant of every house.

### E.1.2  EXAMPLE 1

**Concrete Rules:**  There are three houses arranged in a row and numbered 1–3 from left to right when viewed from across the street. Each house is occupied by a different person. For every house, the occupant has a unique value for each of the following attributes:

1. **Name**: *Peter*, *Eric*, *Arnold*.

2. **Favourite drink**: *tea*, *water*, *milk*.

3. **Pet**: *dog*, *bird*, *fish*.

**Clues.**

1. Peter lives in the **second** house.

2. Arnold is **immediately left** of the person who drinks *water*.

3. The person who drinks *water* is **immediately left** of the person who prefers *milk*.

**Question:**

```
"House 1": {
    "Name": "_",
    "Drink": "_"
},
"House 2": {
    "Name": "_",
    "Drink": "_"
},
"House 3": {
    "Name": "_",
    "Drink": "_"
}
```

**Reasoning Process:** From Clue 1, we know that Peter lives in House 2. From Clue 2, Arnold is immediately to the left of the person who drinks water. Since the resident of House 3 cannot be to anyone's left, Arnold must be in House 1. Therefore, Peter drinks water and Eric lives in House 3. Next, by Clue 3, Eric drinks milk. Consequently, Arnold drinks tea.

**Answer:**

```
"House 1": {
    "Name": "Arnold",
    "Drink": "tea"
},
"House 2": {
    "Name": "Peter",
    "Drink": "water"
},
"House 3": {
    "Name": "Eric",
    "Drink": "milk"
}
```

## E.2 EXTENSIVE ZEBRA

### E.2.1 BASE ASSUMPTIONS

In the extensive *Zebra*, we consider a row of $N$ houses, numbered 1 through $N$ from left to right. Exactly one person lives in each house, and every person is characterised by $M$ *distinct* attribute categories (e.g., nationality, pet, beverage, occupation, house colour).

**Domain constraints.**

1. Each attribute category contains **exactly $N$ mutually exclusive values**.

2. Every value appears **once and only once** across the $N$ houses; that is, the $N$ values of any attribute are assigned to the $N$ houses without repetition.

**Structural assumptions.**

1. The occupant of each house must possess **one unique value** from *every* attribute category, drawn from the full set of values for that category.

2. The ordering of the houses ($1 \rightarrow N$) is known *a priori*, enabling spatial reasoning such as "$X$ is somewhere to the left of $Y$" or "$X$ is immediately left of $Y$" (i.e., $X$ is to the left of $Y$ with no houses in between).

**Puzzle instance.** Each individual puzzle provides

- the complete value sets for all attribute categories, and
- a collection of **clues** that express relations among attributes and/or positions.

**Objective.** Using the given clues, the solver must **uniquely determine** the full set of attribute values associated with the occupant of every house.

### E.2.2 EXAMPLE 1

**Concrete Rules:** There are six houses arranged in a row and numbered 1–6 from left to right when viewed from across the street.

1. **Each person has a unique name**: Rachel, Kevin, Michael, Tessa, Julia, Bob.
2. **Everyone has something unique for lunch**: nectarine, pumpkin, eggplant, avocado, lettuce, papaya.
3. **People have unique heights**: 152cm, 172cm, 154cm, 158cm, 178cm, very tall.
4. **People have unique favorite movie genres**: disaster, fantasy, romance, comedy, crime, musical.
5. **People use unique phone models**: nokia 8.3, iphone 14, huawei p50, samsung galaxy note 20, blackberry keyone, samsung galaxy s21.
6. **Each person lives in a unique style of house**: penthouse, villa, cottage, victorian, palace, townhouse.

**Clues.**

1. The person whose Food is papaya is not to the left of style of the person whose house is townhouse.
2. style of the person whose house is victorian is not to the left of Rachel.
3. Tessa is not to the left of Kevin.
4. style of the person whose house is penthouse is somewhere to the right of style of the person whose house is townhouse.
5. Rachel is not to the right of the person whose Heights is 158cm.
6. Bob is not the person whose Movie-Genre is disaster or Bob is not style of the person whose house is penthouse or both.
7. Phone the person whose Models is nokia 8.3 is not to the left of Kevin.
8. The person whose Movie-Genre is romance is not Phone the person whose Models is blackberry keyone or the person whose Movie-Genre is romance is not Bob or both.
9. Julia == Phone the person whose Models is blackberry keyone or Julia == the person whose Heights is 154cm, but not both.
10. The person whose Food is pumpkin is not to the left of the person whose Food is nectarine.
11. Phone the person whose Models is nokia 8.3 is somewhere to the right of the person whose Heights is very tall.
12. The person whose Movie-Genre is disaster is not style of the person whose house is victorian.
13. Phone the person whose Models is blackberry keyone == the person whose Heights is 178cm or Phone the person whose Models is blackberry keyone == the person whose Food is lettuce or both.
14. style of the person whose house is victorian is not the person whose Food is eggplant or style of the person whose house is victorian is not the person whose Heights is 178cm or both.
15. The person whose Heights is 152cm is somewhere to the right of the person whose Food is eggplant.

16. Rachel is somewhere to the left of the person whose Heights is 172cm.

17. Phone the person whose Models is samsung galaxy note 20 is not style of the person whose house is cottage or style of the person whose house is cottage is not the person whose Movie-Genre is fantasy or both.

18. The person whose Food is papaya is somewhere to the left of Phone the person whose Models is samsung galaxy s21.

19. Phone the person whose Models is nokia 8.3 is somewhere to the left of the person whose Food is nectarine.

20. style of the person whose house is villa is not the person whose Heights is 154cm.

21. The person whose Heights is 178cm is not to the left of Phone the person whose Models is blackberry keyone.

22. The person whose Heights is 154cm is not Phone the person whose Models is iphone 14 or the person whose Food is lettuce is not the person whose Heights is 154cm or both.

23. The person whose Food is papaya is not to the right of Tessa.

24. The person whose Movie-Genre is disaster is not to the right of Phone the person whose Models is samsung galaxy note 20.

25. The person whose Movie-Genre is musical is not to the left of the person whose Food is pumpkin.

26. The person whose Movie-Genre is fantasy is not to the right of style of the person whose house is victorian.

27. The person whose Movie-Genre is crime is not to the left of style of the person whose house is palace.

28. The person whose Food is papaya is somewhere to the right of Phone the person whose Models is blackberry keyone.

29. Phone the person whose Models is samsung galaxy s21 is not Julia or Phone the person whose Models is samsung galaxy s21 is not the person whose Movie-Genre is disaster or both.

30. Phone the person whose Models is samsung galaxy note 20 is somewhere to the right of Phone the person whose Models is huawei p50.

31. Phone the person whose Models is samsung galaxy s21 and style of the person whose house is cottage have different parity positions.

32. The person whose Heights is 152cm is somewhere to the left of style of the person whose house is townhouse.

33. Phone the person whose Models is blackberry keyone is not to the left of the person whose Heights is 152cm.

34. Bob == Phone the person whose Models is huawei p50 or Phone the person whose Models is huawei p50 == the person whose Movie-Genre is crime or both.

35. The person whose Heights is 158cm is somewhere to the left of style of the person whose house is penthouse.

36. Julia and Phone the person whose Models is samsung galaxy s21 have different parity positions.

37. The person whose Food is nectarine and Phone the person whose Models is nokia 8.3 have different parity positions.

38. The person whose Heights is 154cm is not to the right of Phone the person whose Models is iphone 14.

39. style of the person whose house is villa is in an even position.

40. The person whose Food is pumpkin and the person whose Movie-Genre is disaster have different parity positions.

41. Phone the person whose Models is nokia 8.3 is not Rachel or the person whose Heights is 154cm is not Phone the person whose Models is nokia 8.3 or both.

42. Julia is not to the left of Rachel.

43. The person whose Food is papaya and the person whose Movie-Genre is musical have the same parity positions.

44. The person whose Movie-Genre is romance == Bob or the person whose Movie-Genre is romance == the person whose Heights is 158cm or both.

45. The person whose Movie-Genre is crime == Phone the person whose Models is nokia 8.3 or the person whose Heights is 158cm == the person whose Movie-Genre is crime or both.

46. The person whose Food is lettuce and style of the person whose house is palace have different parity positions.

47. style of the person whose house is palace is somewhere to the right of the person whose Food is lettuce.

48. style of the person whose house is penthouse is not the person whose Movie-Genre is crime or style of the person whose house is penthouse is not the person whose Food is nectarine or both.

49. The person whose Heights is 154cm and the person whose Movie-Genre is musical have different parity positions.

50. style of the person whose house is palace is somewhere to the left of Phone the person whose Models is samsung galaxy note 20.

51. The person whose Heights is 154cm is not Phone the person whose Models is samsung galaxy note 20.

52. The person whose Movie-Genre is crime is not to the left of Phone the person whose Models is nokia 8.3.

53. The person whose Food is nectarine is not to the right of Phone the person whose Models is samsung galaxy s21.

54. Rachel is not the person whose Heights is 152cm or Rachel is not the person whose Food is lettuce or both.

55. The person whose Movie-Genre is fantasy is not to the left of the person whose Movie-Genre is fantasy.

56. style of the person whose house is townhouse is somewhere to the right of Julia.

57. Kevin is not to the left of style of the person whose house is townhouse.

58. Phone the person whose Models is nokia 8.3 and style of the person whose house is penthouse have different parity positions.

59. Phone the person whose Models is blackberry keyone is somewhere to the left of Tessa.

60. Julia == the person whose Food is lettuce or Phone the person whose Models is huawei p50 == Julia or both.

61. style of the person whose house is townhouse is not to the left of Phone the person whose Models is iphone 14.

62. Phone the person whose Models is samsung galaxy note 20 and Rachel have different parity positions.

63. Phone the person whose Models is iphone 14 and the person whose Movie-Genre is romance have the same parity positions.

64. The person whose Food is avocado == Phone the person whose Models is nokia 8.3 or Phone the person whose Models is nokia 8.3 == the person whose Movie-Genre is crime, but not both.

65. The person whose Food is nectarine == style of the person whose house is villa or style of the person whose house is villa == the person whose Heights is 178cm, but not both.

66. Phone the person whose Models is samsung galaxy s21 is not the person whose Heights is 154cm.

67. The person whose Heights is 154cm and Phone the person whose Models is samsung galaxy note 20 have different parity positions.

68. Kevin is somewhere to the right of style of the person whose house is cottage.

69. The person whose Movie-Genre is musical and Michael have the same parity positions.

70. The person whose Heights is very tall is somewhere to the right of style of the person whose house is cottage.

71. Rachel is not the person whose Heights is 178cm or Rachel is not the person whose Food is nectarine or both.

**Question:**

```
"House 1": {
    "Name": "_",
    "Food": "_",
    "Height": "_",
    "Movie-Genre": "_",
    "Phone Models": "_",
    "Style of House": "_"
},
"House 2": {
    "Name": "_",
    "Food": "_",
    "Height": "_",
    "Movie-Genre": "_",
    "Phone Models": "_",
    "Style of House": "_"
},
"House 3": {
    "Name": "_",
    "Food": "_",
    "Height": "_",
    "Movie-Genre": "_",
    "Phone Models": "_",
    "Style of House": "_"
},
"House 4": {
    "Name": "_",
    "Food": "_",
    "Height": "_",
    "Movie-Genre": "_",
    "Phone Models": "_",
    "Style of House": "_"
},
"House 5": {
    "Name": "_",
    "Food": "_",
    "Height": "_",
    "Movie-Genre": "_",
    "Phone Models": "_",
    "Style of House": "_"
},
"House 6": {
    "Name": "_",
    "Food": "_",
    "Height": "_",
    "Movie-Genre": "_",
    "Phone Models": "_",
    "Style of House": "_"
},
```

**Reasoning Process:** Key deductions in brief: First, Rachel must be in House 1. If Rachel were in House 2, a chain of constraints around s21, Nokia, and nectarine immediately contradicts the parity and ordering clues (notably 19, 31, 36, 37, 53). With Rachel at 1, and s21 having to be to the right of papaya (18) and of blackberry (28), and of townhouse (1, 56), while also having different parity from Julia (36), s21 fixes at House 5. From 32 and 15 we get eggplant at House 1 and 152cm at House 2. Clue 9 then forces Julia (House 2) to be the blackberry owner, and 13 forces her food to be lettuce. Cottage must be in an even position different from s21 (31) and to the left of both Kevin and very tall (68, 70), so Cottage is at House 2. Using 11, 19, 53 with s21=5 forces Nokia at House 4 and Very Tall at House 3, with nectarine at House 5 and (by 10, 25) pumpkin and musical at House 6. The XOR clue 64 cannot take the branch Avocado==Nokia (it leads to contradictions with 35 and 58), so it must be Nokia==Crime; therefore crime is at House 4. Then 1 and 18 force papaya at House 4 (since it must be Ž265 townhouse and ¡ s21), which in turn fixes avocado at House 3. From 27, 46, 47, Palace must be at House 3; thus Townhouse is at House 4 (and papaya=4 still satisfies 1 Ž018not to the leftŽ019 by equality). With Villa even and linked by 65 to be the 178cm house (and 5 already odd, 4 used), Villa=House 6 with 178cm. The remaining odd style compatible with 58 (different parity from Nokia at 4) and 4 (to the right of Townhouse) is Penthouse=House 5, leaving Victorian=House 1. Clue 26 then forces Fantasy at House 1. From 57 and 7, Kevin is at House 4 (Nokia not left of Kevin and not left of Townhouse), which in turn makes Tessa to his right (3), so Tessa at House 5, and by 69, Michael must be at House 6 (same parity as musical). For the remaining phones, 30 and 62 put Note20 at House 6 and Huawei P50 at House 3, leaving iPhone 14 at House 1. Clue 34 then forces Bob to be the P50 owner, so Bob is House 3, and 44 (with 63) makes Bob's genre Romance. The remaining genres place disaster at House 5 (by 40 parity difference with pumpkin) and comedy at House 2. Finally, 35 fixes 158cm at House 4 (left of penthouse at 5) and the last height 172cm at House 5. All remaining clues then check out.

**Answer:**

```
"House 1": {
    "Name": "Rachel",
    "Food": "eggplant",
    "Heights": "154cm",
    "Movie-Genre": "fantasy",
    "Phone Models": "iphone 14",
    "style of house": "victorian"
},
"House 2": {
    "Name": "Julia",
    "Food": "lettuce",
    "Heights": "152cm",
    "Movie-Genre": "comedy",
    "Phone Models": "blackberry keyone",
    "style of house": "cottage"
},
"House 3": {
    "Name": "Bob",
    "Food": "avocado",
    "Heights": "very tall",
    "Movie-Genre": "romance",
    "Phone Models": "huawei p50",
    "style of house": "palace"
},
"House 4": {
    "Name": "Kevin",
    "Food": "papaya",
    "Heights": "158cm",
    "Movie-Genre": "crime",
    "Phone Models": "nokia 8.3",
    "style of house": "townhouse"
```

```
},
"House 5": {
    "Name": "Tessa",
    "Food": "nectarine",
    "Heights": "172cm",
    "Movie-Genre": "disaster",
    "Phone Models": "samsung galaxy s21",
    "style of house": "penthouse"
},
"House 6": {
    "Name": "Michael",
    "Food": "pumpkin",
    "Heights": "178cm",
    "Movie-Genre": "musical",
    "Phone Models": "samsung galaxy note 20",
    "style of house": "villa"
}
```

## E.3 SUDOKU

### E.3.1 BASE ASSUMPTIONS

Sudoku is a classic number puzzle based on permutation and logical reasoning. A standard Sudoku board is a $9 \times 9$ square grid with 81 cells in total, partitioned into nine $3 \times 3$ sub-grids (blocks).

**Filling rules (all must be satisfied simultaneously).**

1. **Row constraint**: Each of the nine rows must contain the digits $1-9$ exactly once, with no repetition.

2. **Column constraint**: Each of the nine columns must contain the digits $1-9$ exactly once, with no repetition.

3. **Block constraint**: Each $3 \times 3$ sub-grid (nine in total) must also contain the digits $1-9$ exactly once, with no repetition.

**Given clues.** The puzzle starts with several numbers pre-filled as initial clues; all other cells are blank and denoted by an underscore, _. The given numbers must already satisfy all three rules above—that is, the initial state is valid.

**Objective.** Based on the given clues, deduce and fill in the remaining cells so that the entire board meets the three constraints. Every well-posed Sudoku puzzle should have a *unique* solution; multiple valid solutions are not allowed.

### E.3.2 EXAMPLE 1

**Concrete Rules:** No rules.

**Question:**

```
[
    [9,4,6,5,8,1,7,2,3],
    [8,3,1,6,7,2,_,5,9],
    [2,5,7,3,9,4,6,8,1],
    [6,7,4,2,1,9,5,3,8],
    [3,8,_,4,6,5,1,9,7],
    [5,1,9,7,3,8,2,6,4],
    [4,2,8,1,5,3,9,7,6],
    [1,6,3,9,2,7,8,4,5],
    [7,9,5,8,4,6,3,1,2]
]
```

**Reasoning Process:**

1. **r2c7**. The second row already contains $\{8, 3, 1, 6, 7, 2, 5, 9\}$, so the missing digit is $4$. Column 7 already contains $\{7, 6, 5, 1, 2, 9, 8, 3\}$, which is likewise missing $4$. The corresponding $3 \times 3$ block (rows $1-3$, columns $7-9$) is also missing $4$. $\Rightarrow$ Place $4$ in $r2c7$.

2. **r5c3**. The fifth row already contains $\{3, 8, 4, 6, 5, 1, 9, 7\}$, so the missing digit is $2$. Column 3 already contains $\{6, 1, 7, 4, 9, 8, 8, 3, 5\}$, which is likewise missing $2$. The corresponding $3 \times 3$ block (rows $4-6$, columns $1-3$) is also missing $2$. $\Rightarrow$ Place $2$ in $r5c3$.

**Answer:**

```
[
    [9,4,6,5,8,1,7,2,3],
    [8,3,1,6,7,2,4,5,9],
    [2,5,7,3,9,4,6,8,1],
    [6,7,4,2,1,9,5,3,8],
    [3,8,2,4,6,5,1,9,7],
    [5,1,9,7,3,8,2,6,4],
    [4,2,8,1,5,3,9,7,6],
    [1,6,3,9,2,7,8,4,5],
    [7,9,5,8,4,6,3,1,2]
]
```

### E.4 LINE FORMING GAME

#### E.4.1 BASE ASSUMPTIONS

*N-in-a-Row* is the generalised version of tic-tac-toe (an $(m, n, k)$-game) played on an $N \times N$ board with alternating turns. The first player uses O, the second player X. It subsumes classic tic-tac-toe, Gomoku, and related games.

**Basic rules.**

1. **Placement rule**: The two players take turns marking an empty cell—O for the first player, X for the second. Once a symbol is placed, it cannot be moved or replaced.

2. **Win condition**: A player wins immediately upon forming $K$ consecutive identical symbols (O or X) along any horizontal, vertical, or $45°$ diagonal line. If the board is completely filled and no one has achieved $K$ in a row, the game ends in a draw, denoted as *Both*.

**Given information.** The board size $N \times N$ and the required line length $K$ are specified. The initial configuration may contain zero or more pre-placed symbols (O or X), which must be legal—that is, it must not already contain a winning $K$-in-a-row or any other conflict.

**Objective.** Given $N$, $K$, and the initial board state, the two players alternate moves until one of three outcomes occurs:

1. O achieves a $K$-in-a-row $\Rightarrow$ O **wins**.

2. X achieves a $K$-in-a-row $\Rightarrow$ X **wins**.

3. The board is full with no $K$-in-a-row $\Rightarrow$ **Both** (draw).

#### E.4.2 EXAMPLE 1

**Concrete Rules:** Given N = 10 and K = 5, and the first player is Amy (playing O), while the second player is Bob (playing X), the final state of the board is as follows (with the top row as column indices and the leftmost column as row indices):

```
   0  1  2  3  4  5  6  7  8  9
0  X  .  .  .  O  O  .  .  O  .
```

```
1  .  O  .  .  O  .  .  .  .  .
2  X  .  .  O  .  .  X  .  X  .
3  .  .  O  .  .  O  .  .  .  .
4  .  O  .  .  X  .  O  .  .  .
5  .  .  X  X  .  .  .  .  .  .
6  .  O  .  .  .  .  .  X  .  .
7  .  .  X  .  O  .  X  .  .  X
8  .  O  .  .  .  .  .  X  .  .
9  .  .  .  .  .  .  .  .  .  .
```

Determine the winner (if any), or output "Both" if it is a draw. Also, list a valid move sequence (set of coordinates) for both players.

**Question:**

```
{
    "Winner": "_",
    "Amy": [(_, _), (_, _), (_, _), (_, _),
            (_, _), (_, _), (_, _), (_, _),
            (_, _), (_, _), (_, _), (_, _),
            (_, _)],
    "Bob": [(_, _), (_, _), (_, _), (_, _),
            (_, _), (_, _), (_, _), (_, _),
            (_, _), (_, _), (_, _), (_, _)]
}
```

**Reasoning Process:**   Amy wins because she forms a diagonal line of five consecutive O stones by playing $(0,5) \to (1,4) \to (2,3) \to (3,2) \to (4,1)$; all these coordinates appear in her legal move sequence given in the solution. Although Bob places more stones overall, he fails to create five consecutive X stones in any direction, and therefore does not win.

**Answer:**

```
{
    "Winner": "Amy",
    "Amy": [(2, 3), (4, 6), (7, 4), (0, 4),
            (3, 5), (1, 4), (1, 1), (6, 1),
            (0, 8), (8, 1), (4, 1), (3, 2),
            (0, 5)],
    "Bob": [(4, 4), (2, 8), (7, 9), (7, 2),
            (7, 6), (5, 3), (2, 0), (6, 7),
            (8, 7), (2, 6), (0, 0), (5, 2)]
}
```

## E.5   KNIGHT AND KNAVE

### E.5.1   BASE ASSUMPTIONS

**Puzzle Setting.**   The *Knights & Knaves* puzzle takes place on a fictional island populated by two types of inhabitants: **knights**, who always tell the truth, and **knaves**, who always lie. Players must infer each character's true identity by analysing their statements.

**Basic rules.**

1. **Fixed identity.** Every knight's utterances are *true*; every knave's utterances are *false*.

2. **Mutual knowledge.** Each character knows exactly who is a knight and who is a knave.

3. **Semantic mapping.** Every sentence spoken by a character can be mapped directly to a logical proposition: a knight's proposition evaluates to TRUE, whereas a knave's proposition evaluates to FALSE.

**Given information.**

1. **Participant list**: a set of characters (e.g., $A, B, C, \ldots$).

2. **Statement set**: each character utters one or more statements about their own or others' identities.

3. **Initial state**: all identities are unknown, yet every statement adheres to the basic rules (i.e., no self-contradiction).

**Objective.**    Using logical deduction, determine whether each character is a knight or a knave, and establish the truth value of every statement. A well-posed puzzle should yield a *unique* solution without logical inconsistencies.

E.5.2   EXAMPLE 1

**Concrete Rules:**   A very special island is inhabited only by knights and knaves. Knights always tell the truth, and knaves always lie. You meet 2 inhabitants: Joseph, and Harper. Please use their statements to deduce who is a knight and who is a knave.

**Clues for the Puzzle:**

1. Joseph was heard saying, "Joseph is a knight if and only if Harper is a knave".

2. Harper stated, "If Joseph is a knight then Harper is a knight".

**Question:**

```
{
    "Joseph": "_",
    "Harper": "_"
}
```

**Reasoning Process:**   Assume Joseph is a knight. Then his statement must be true: "Joseph is a knight *iff* Harper is a knave." Because Joseph is indeed a knight (the left side is true), the right side must also be true, so Harper is a knave.

Now consider Harper's statement: "If Joseph is a knight, then Harper is a knight." We already know Joseph is a knight and Harper is a knave, so the conditional is FALSE. Hence Harper's statement is false, consistent with Harper being a knave.

The assumption leads to no contradiction, so the unique solution is: Joseph is a **knight**, Harper is a **knave**.

**Answer:**

```
{
    "Joseph": "a knight",
    "Harper": "a knave"
}
```

E.6   TETRIS

E.6.1   BASE ASSUMPTIONS

**Tetris Overview.**   *Tetris* is a classic puzzle game and the most widely recognised example of the "falling-block" genre.

**Basic rules.**

1. **Board dimensions** The playing field is a fixed grid of size $BOARD\_WIDTH \times BOARD\_HEIGHT$. Columns $x$ are indexed from 0 (left) to $BOARD\_WIDTH - 1$; rows $y$ are indexed from 0 (top) to $BOARD\_HEIGHT - 1$.

2. **Cell states** An occupied cell is denoted by O; an empty cell by X.

3. **Tetromino types** There are seven canonical tetrominoes—I, J, L, O, S, Z, and T—each with one or more rotation states. A tetromino is represented by a `Block` structure containing

   - *template*: a character matrix (. = empty, O = filled),
   - *start_pos*: coordinates of the template's upper-left corner,
   - *end_pos*: coordinates of the template's lower-right corner,
   - *name*: the tetromino identifier,
   - *next*: the index of the rotation state obtained by a clockwise turn.

4. **Player controls** The following actions are available:

   a: move left     d: move right     s: move down     w: rotate clockwise to *next*

   An action is *invalid* if it pushes the tetromino outside the board or into an occupied cell; in that case the piece remains unchanged.

5. **Automatic fall** Every *valid* action—except s—is followed by one automatic downward step.

6. **Landing** A tetromino becomes fixed when it can no longer move down (because of the board bottom or existing O cells); all cells it occupies are then set to O.

7. **Line clear** If a row is completely filled with O, that row is cleared; all rows above shift down by one, and a new empty row (X) is added at the top.

8. **Game over** If a newly spawned tetromino overlaps any O cells at its initial position, the game ends.

**Given information.**

1. Board dimensions *BOARD_WIDTH* × *BOARD_HEIGHT*;

2. Current board state (character matrix of O/X);

3. The active falling tetromino (type, template, and position);

4. The control scheme (movement and rotation rules).

The current state is guaranteed to be legal—no overlaps or out-of-bounds cells.

**Objectives.**    Under the rules and initial conditions above, the player seeks to

1. maximise the number of cleared lines (hence the score);

2. avoid stacking pieces to the top, keeping space for future moves;

3. determine or verify whether, from the current state and a sequence of actions, play can continue or must inevitably lead to game over.

E.6.2   EXAMPLE 1

**Concrete Rules:**    Given a Tetris game grid with BOARD_WIDTH of 4 and BOARD_HEIGHT of 4, you will be provided with a sequence of falling blocks during one game. For each block, you will receive its name, the initial positions of its occupied cells (as coordinates in the form of (x, y)), and the sequence of operations it receives. Your task is to output the final state of the grid when the game ends — that is, for each row (y), indicate whether each cell in each column (x) is occupied ('O') or empty ('X'). The initial grid state is:

```
    0   1   2   3
0   X   O   O   X
1   X   X   X   X
2   X   X   X   X
3   X   X   X   X
```

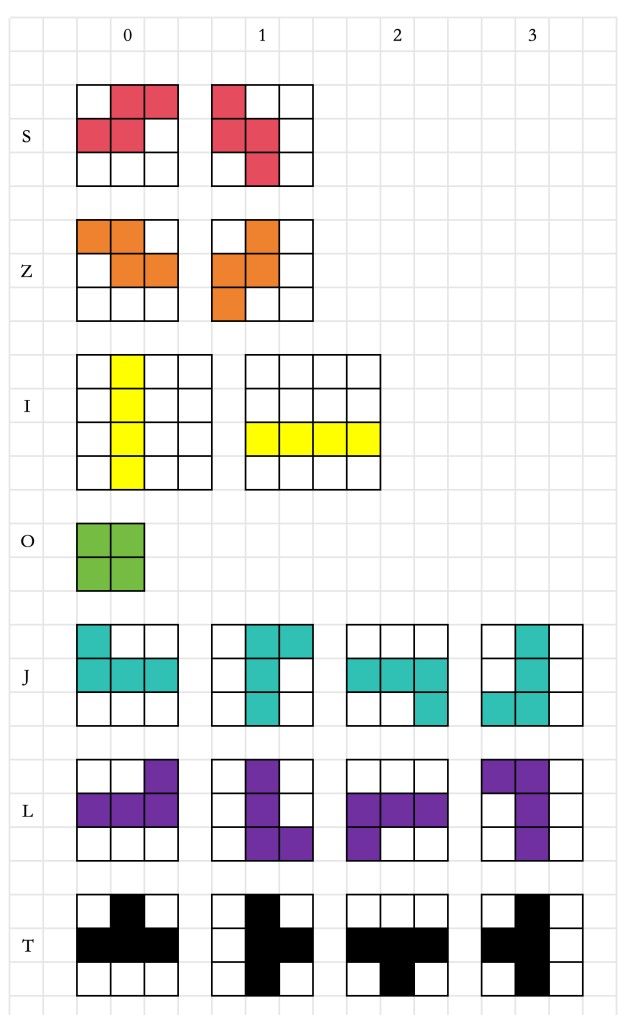

Figure 20: Visual display of Tetris.

**Clues for the Example Puzzle:**

1. {'name': 'O', 'start_cells': [(1, -1), (2, -1), (1, 0), (2, 0)], 'ops': ['a', 'd', 's']}
2. {'name': 'J', 'start_cells': [(2, -2), (3, -2), (2, -1), (2, 0)], 'ops': ['s']}

**Question:**

```
{
    "0": ["_", "_", "_", "_"],
    "1": ["_", "_", "_", "_"],
    "2": ["_", "_", "_", "_"],
    "3": ["_", "_", "_", "_"],
}
```

**Reasoning Process:** The ○ piece first moves one square to the left, then one square down, then one square to the right followed by another square down, and finally one more square down. The ⅃ piece moves one square down.

**Answer:**

```
{
    "0": ["X", "X", "O", "X"],
    "1": ["X", "X", "O", "X"],
    "2": ["X", "O", "O", "X"],
    "3": ["X", "O", "O", "X"],
}
```

### E.7 MAZE

#### E.7.1 BASE ASSUMPTIONS

**Maze Overview.** A *maze* is a two-dimensional grid path-finding problem whose core objective is to locate a feasible—preferably shortest—route from a start cell to a goal cell.

**Basic rules.**

1. **Cell states** Each grid cell is represented by a character: O denotes a traversable cell, while X denotes an obstacle that cannot be entered.

2. **Coordinate system** A cell is identified by $(x, y)$, where $x$ is the column index (left to right, starting at 0) and $y$ is the row index (top to bottom, starting at 0).

3. **Movement rule** From any O cell, the agent may move *only* one step up, down, left, or right ($\Delta x, \Delta y \in \{\pm 1, 0\}$ with $|\Delta x| + |\Delta y| = 1$).

4. **Path constraint** Movement is confined to O cells; walls X cannot be crossed.

**Given information.**

1. A character grid of size *rows* × *cols* whose cells are either O or X.

2. The start coordinates (*begin_x*, *begin_y*) and goal coordinates (*end_x*, *end_y*).

3. Both start and goal cells are guaranteed to be O.

4. The grid may contain multiple feasible paths—or none at all.

**Objective.** Find a shortest path (minimum number of steps) from the start to the goal. Output the sequence of coordinates $(x, y)$ along the route, in order; if no path exists, explicitly report that there is *no solution*.

#### E.7.2 EXAMPLE 1

**Concrete Rules:** Now there is a maze with 7 rows and 9 columns. The starting point of the maze is at (0, 5), and the ending point is at (8, 1). The maze is as follows:

```
XXXXXXXXX
XOOOOOOOO
XOXXXOXXX
XOXOXOOOX
XOXOXXXOX
OOXOOOOOX
XXXXXXXXX
```

Please provide the specific path through the maze.

**Question:**

```
[
    ["_", "_"], ["_", "_"], ["_", "_"],
    ["_", "_"], ["_", "_"], ["_", "_"],
    ["_", "_"], ["_", "_"], ["_", "_"],
    ["_", "_"], ["_", "_"], ["_", "_"],
```

```
    ["_", "_"]
]
```

**Reasoning Process:** We use Breadth-First Search (BFS) to find the shortest path from the start point to the end point. The BFS algorithm is a good choice for this problem because it is guaranteed to find the shortest path in an unweighted grid. The maze is treated as a two-dimensional grid, and only the cells marked with 'O' can be traversed. The algorithm records the visited cells and reconstructs the path when the end point is reached.

**Answer:**

```
[
    [0, 5], [1, 5], [1, 4], [1, 3],
    [1, 2], [1, 1], [2, 1], [3, 1],
    [4, 1], [5, 1], [6, 1], [7, 1],
    [8, 1]
]
```

## E.8 STRING PUZZLE

### E.8.1 BASE ASSUMPTIONS

**String-Transformation Puzzle Overview.** A *string puzzle* is a logic task in which the solver must infer transformation rules from several worked examples and then apply the same rules to a new input string. Such puzzles assess inductive reasoning, pattern recognition, and program-synthesis skills.

**Basic rules.**

1. **Operation sequence**. Each puzzle consists of a fixed sequence of operations; the output of step $i$ becomes the input to step $i+1$. Typical operations include—but are not limited to—

   - *Find*: locate the position of a specified substring;
   - *Insert*: insert a new substring at a given position;
   - *Delete*: remove a substring;
   - *Replace*: substitute one substring with another;
   - *Sort*: reorder the characters of a substring.

2. **Operation consistency**. Across all examples, *step $k$* must follow the same transformation logic. For instance, if step 1 in every example is "insert the character A immediately after the first digit," that exact operation—though applied to different strings—must also be step 1 in the target string.

**Given information.**

1. **Example sequences**. Each example provides a sequence of string states, showing the original string and the intermediate result after every step. All examples share the same number of steps, and step numbers $(1, 2, \ldots)$ carry identical semantics across examples.

2. **Target string**. A new initial string to which the inferred operation sequence must be applied.

**Objective.**

1. **Inductive inference**. Deduce the general rule executed at each step by analysing all examples.

2. **Execution**. Apply the inferred rules sequentially to the target string, producing intermediate states.

3. **Output**. List the resulting string after every step, in order. If no valid rule set can be inferred, explicitly state that the puzzle has no solution.

## E.8.2 EXAMPLE 1

**Concrete Rules:** Given a set of strings composed of uppercase letters, lowercase letters, and digits. For each string, follow the (unknown) sequence of operations to transform it. By analysing the provided *operation–sequence results*, infer the underlying sequence of operations and apply it to each corresponding string, step by step. Below are the operation–sequence results for several example strings:

```
{
    "raw": "ANsqAN2E6b",
    "1": "ANVsqANV2E6b",
    "2": "ANuVsqANV2E6b",
    "3": "ANluVsqANV2E6b",
    "4": "AN7EluVsqANV2E6b",
    "5": "AN9W7EluVsqANV2E6b",
    "6": "ANQY9W7EluVsqANV2E6b",
    "7": "ANzJQY9W7EluVsqANzJV2E6b",
    "8": "ANzZzJQY9W7EluVsqANzJV2E6b",
    "9": "ANG6zZzJQY9W7EluVsqANzJV2E6b",
    "10": "ANlDG6zZzJQY9W7EluVsqANlDzJV2E6b"}

{
    "raw": "M24ycZMANW",
    "1": "M24ycZMANVW",
    "2": "M24ycZMANuVW",
    "3": "M24ycZMANluVW",
    "4": "M24ycZMAN7EluVW",
    "5": "M24ycZMAN9W7EluVW",
    "6": "M24ycZMANQY9W7EluVW",
    "7": "M24ycZMANzJQY9W7EluVW",
    "8": "M24ycZMANzZzJQY9W7EluVW",
    "9": "M24ycZMANG6zZzJQY9W7EluVW",
    "10": "M24ycZMANlDG6zZzJQY9W7EluVW"}
```

Please infer the sequence of operations and apply it to the following string:

```
"kOt2OANnFc"
```

**Question:**

```
{
    'raw': 'kOt2OANnFc',
    '1': '', '2': '', '3': '', '4': '',
    '5': '', '6': '', '7': '', '8': '',
    '9': '', '10': ''
}
```

**Reasoning Process:**

1. Locate every substring ``AN'' in kOt2OANnFc and append V immediately after each occurrence.

2. In kOt2OANVnFc, locate the *first* occurrence of ``AN'' and append u after it.

3. In kOt2OANuVnFc, locate the *first* occurrence of ``AN'' and append l after it.

4. In kOt2OANluVnFc, locate the *first* occurrence of ``AN'' and append 7E after it.

5. In kOt2OAN7EluVnFc, locate the *first* occurrence of ``AN'' and append 9W after it.

6. In kOt2OAN9W7EluVnFc, locate the *first* occurrence of ``AN'' and append QY after it.

7. In kOt2OANQY9W7EluVnFc, locate *every* occurrence of ``AN'' and append zJ after each one.

8. In `kOt2OANzJQY9W7EluVnFc`, locate the *first* occurrence of `''AN''` and append `zZ` after it.

9. In `kOt2OANzZzJQY9W7EluVnFc`, locate the *first* occurrence of `''AN''` and append `G6` after it.

10. In `kOt2OANG6zZzJQY9W7EluVnFc`, locate *every* occurrence of `''AN''` and append `lD` after each one.

**Answer:**

```
{
    'raw': 'kOt2OANnFc',
    '1': 'kOt2OANVnFc',
    '2': 'kOt2OANuVnFc',
    '3': 'kOt2OANluVnFc',
    '4': 'kOt2OAN7EluVnFc',
    '5': 'kOt2OAN9W7EluVnFc',
    '6': 'kOt2OANQY9W7EluVnFc',
    '7': 'kOt2OANzJQY9W7EluVnFc',
    '8': 'kOt2OANzZzJQY9W7EluVnFc',
    '9': 'kOt2OANG6zZzJQY9W7EluVnFc',
    '10': 'kOt2OANlDG6zZzJQY9W7EluVnFc'
}
```

## E.9 CODE PUZZLE

### E.9.1 BASE ASSUMPTIONS

**Code–Puzzle Overview.** A *code puzzle* is an intellectual task that centres on program semantics, blending logical reasoning with algorithmic analysis. Typically, the puzzle presents a snippet of code (pseudocode, a function, or a flowchart) or merely its observable behaviour via a validator. The solver must deduce the program's behaviour, compute its output, or reverse-engineer possible inputs.

**Basic rules.**

1. **Given program structure**. The puzzle supplies pseudocode, a function, or execution constraints that may involve branching, loops, recursion, memoisation, or side effects.

2. **Reasoning objectives**.
   - Compute the program's final output or key intermediate states;
   - Verify whether the program satisfies a property (e.g., correctness, termination);
   - Reverse-engineer inputs or reconstruct missing code logic.

3. **Efficiency**. Although brute-force simulation can yield answers, an ideal solution should rely on structured logical analysis or algorithmic design (e.g., recursive derivation, dynamic programming, mathematical induction). Some puzzles explicitly require the solver to present the reasoning steps or critical logic.

**Given information.**

1. **Code logic**: the program body, function definitions, pseudocode, or execution constraints;

2. **Input description**: concrete input values, ranges, edge cases, or test instances;

3. **Output specification**: the expected format (numerical value, intermediate variable, Boolean, execution path, etc.);

4. **Extra constraints**: time complexity, resource limits, recursion depth, and so forth.

**Objectives.**    A puzzle may require one or more of the following:

1. **Accurate output derivation**: determine the program's final output for the given input;

2. **Intermediate-state analysis**: reveal key execution paths and intermediate computations;

3. **Property verification**: check whether the program meets a specified property (e.g., sorting, balance, termination);

4. **Input reverse-engineering**: infer possible inputs from a desired output;

5. **Logic completion**: fill in missing parts so that the program meets its requirements;

6. **Clear reasoning**: when required, provide a step-by-step logical derivation.

### E.9.2    EXAMPLE 1

**Concrete Rules:**    A pair of baby rabbits starts reproducing from the 3rd month after their birth. Each month, every rabbit pair that has reached this age gives birth to one new pair. The newborn pairs also follow the same rule, starting to reproduce from the 3rd month after they are born. Assume no rabbits ever die. At the beginning of the 1st month, there is one newborn rabbit pair. The rabbits continue to reproduce according to this pattern.

We use an integer list li of length n to represent the total number of rabbit pairs in each month, where li[i] denotes the number of rabbit pairs in the (i+1)-th month. For example: li = [1, 1, 2] means there are 1, 1, and 2 rabbit pairs in the 1st, 2nd, and 3rd months, respectively.

Now, given n = 10, please simulate the growth of rabbit pairs from month 1 to month 10 and output a list of length 10 representing the number of rabbit pairs each month.

**Question:**

```
["_", "_", "_", "_", "_",
"_", "_", "_", "_", "_"]
```

**Reasoning Process:**    Let $f(n)$ denote the number of rabbit pairs in month $n$.

According to the problem statement, each pair begins to reproduce in its third month of life (the month of birth counts as month 1), i.e. after completing two full months. Consequently, the number of newborn pairs in month $n$ equals the number of pairs that existed two months earlier, $f(n-2)$. This yields the recurrence

$$f(n) = f(n-1) + f(n-2).$$

**Initial conditions**

$$f(1) = 1 \quad \text{(only the original pair)},$$
$$f(2) = 1 \quad \text{(no reproduction yet)}.$$

**Month-by-month values**

$$f(3) = f(2) + f(1) = 1 + 1 = 2,$$
$$f(4) = f(3) + f(2) = 2 + 1 = 3,$$
$$f(5) = f(4) + f(3) = 3 + 2 = 5,$$
$$f(6) = f(5) + f(4) = 5 + 3 = 8,$$
$$f(7) = f(6) + f(5) = 8 + 5 = 13,$$
$$f(8) = f(7) + f(6) = 13 + 8 = 21,$$
$$f(9) = f(8) + f(7) = 21 + 13 = 34,$$
$$f(10) = f(9) + f(8) = 34 + 21 = 55.$$

Hence the number of rabbit pairs from month 1 through month 10 is

$$[1, \ 1, \ 2, \ 3, \ 5, \ 8, \ 13, \ 21, \ 34, \ 55].$$

**Answer:**

```
[1, 1, 2, 3, 5, 8, 13, 21, 34, 55]
```

E.10   TRAIN GAME CARD PUZZLE

E.10.1   BASE ASSUMPTIONS

**Train Pushing (Card Matching) Game.**   *Train Pushing* is a competitive card game based on matching card ranks. It is suitable for two or more players and proceeds by sequentially playing cards and collecting matched piles, with the goal of eventually capturing all cards.

**Basic rules.**

1. **Deck configuration**. A standard 54-card deck is used (including the two jokers). Typically $2-4$ players participate and the deck is dealt out evenly.

2. **Initial state**. Each player keeps their personal pile face-down and may not look at the cards. The first player is chosen by the previous winner, randomly, or by mutual agreement.

3. **Turn order**. In clockwise (or counter-clockwise) order, each player on their turn flips the *top* card of their pile and places it face-up onto a shared discard stack, revealing its *rank only* (suit is ignored).

4. **Match–trigger mechanism**. If the rank of the played card matches the rank of *any* card already in the shared stack (jokers are considered identical), a "train grab" is triggered. The just-played card is the *tail*; the earlier matching card is the *head*. The current player collects *all* cards from head to tail (inclusive) and places them *in order, head first* beneath their personal pile, without shuffling.

5. **Continued play**. After collecting, the same player's next turn simply flips the new top card of their pile. Turn order otherwise proceeds normally; collecting cards does not alter seating order.

6. **Victory**. Play continues until only one player still has cards; that player wins.

**Given information.**

1. Number of players ($2-4$, but more are possible).

2. Deck and dealing: a 54-card deck dealt evenly.

3. Play direction: clockwise or counter-clockwise.

4. Current state: contents of the shared stack, remaining card count and order for each player.

5. Matching rules as specified above.

**Objectives.**   Starting from a given game state, a puzzle may ask the solver to achieve one or more of the following:

1. **Simulate the game**: follow turn order, play cards, and perform train grabs until the game ends.

2. **Determine the winner**: identify which player ultimately wins.

3. **Record key events**: log each train-grab event, noting head/tail ranks and the collected interval of cards.

4. **Reproduce the action sequence**: output statements such as "Player $X$ plays $Y$; train grab triggered; Player $X$ collects cards from $Z$ to $W$."

5. **Strategy evaluation** (for certain variants): analyse whether different collection strategies affect the win rate.

### E.10.2 EXAMPLE 1

**Concrete Rules:** A deck of cards contains 1 big joker and 1 little joker, with 2 suits (hearts and spades), and 2 cards per suit (numbered 1-2). There are 2 players. Player 1's initial deck is: ["heart 1", "spade 1", "little joker"]. Player 2's initial deck is: ["big joker", "spade 2", "heart 2"]. Please provide the sequence of cards played by the players. The play sequence is a list composed of multiple rounds. Each round is represented by a list, where the order of cards indicates the sequence of cards in the shared pile on the table during that round. Every time a train is triggered (i.e., a match occurs and cards are collected), that marks the end of a round. Each element in the list corresponds to the sequence of cards played in that round.

**Question:**

```
[
    ["_", "_", "_"],
    ["_", "_", "_", "_"],
    ["_", "_", "_"],
    ["_", "_"],
    ["_", "_", "_", "_"],
    ["_", "_"]
]
```

**Reasoning Process:** The game starts with two players, each holding three face-down cards. In clockwise order they flip the top card of their personal pile and place it face-up on the central stack. Whenever the newly revealed card matches the rank of *any* card already in the stack (both jokers count as the same rank), the current player immediately "grabs the train": all cards from the *engine* (first matching card) to the *caboose* (just-played card), inclusive, are taken and placed beneath that player's pile in order (engine at the very bottom, caboose on top). The same player then continues to play. Iterating this produces six consecutive play segments:

1. Player 1 plays $\heartsuit 1 \to$ Player 2 Joker$_B$ $\to$ Player 1 $\spadesuit 1$; ranks match, and Player 1 collects all three cards.

2. Player 1 Joker$_S$ $\to$ Player 2 $\spadesuit 2 \to$ Player 1 $\spadesuit 1 \to$ Player 2 $\heartsuit 2$; match, and Player 2 collects the last three cards.

3. Leaving the Joker$_S$ at the bottom, Player 2 $\heartsuit 2 \to$ Player 1 Joker$_B$; match, and Player 1 collects these three cards.

4. Player 1 $\heartsuit 1 \to$ Player 2 $\spadesuit 1$; match, and Player 2 collects both cards.

5. Player 2 $\spadesuit 2 \to$ Player 1 Joker$_B$ $\to$ Player 2 $\spadesuit 1 \to$ Player 1 $\heartsuit 2$; match, and Player 1 collects four cards.

6. Player 1 Joker$_S$ $\to$ Player 2 $\heartsuit 1$; no further match occurs and Player 2 runs out of cards, so the game ends with Player 1 the winner.

Thus the six-round play log described in the puzzle is obtained.

**Answer:**

```
[
    ["heart 1", "big joker",
     "spade 1"],
    ["little joker", "spade 2",
     "spade 1", "heart 2"],
    ["little joker", "heart 2",
     "big joker"],
    ["heart 1", "spade 1"],
    ["spade 2", "big joker",
     "spade 1", "heart 2"],
    ["little joker", "heart 1"]
]
```

# F  ALL LOGICAL REASONING METADATA

In this section, we present a condensed version of over 100 logical reasoning puzzle metadata from ExCLUB, as shown below; the complete dataset is available in the code implementation.

## F.1  0-9

There are 3 logical reasoning puzzles in the category 0-9, as shown in Table 25.

Table 25: Concise metadata for logical reasoning puzzles in category 0-9.

| name | task | dynamic | monotonic | rule_based | dimension | probabilistic | formal | model | method |
|------|------|---------|-----------|------------|-----------|---------------|--------|-------|--------|
| 15 Puzzle | Path Search | dynamic | non-monotonic | rule-based | 2D | deterministic | formal | deduction | heuristic |
| 2048 | Rule-Based Simulation | dynamic | non-monotonic | rule-based | 2D | probabilistic | formal | deduction | constraint |
| 8 Puzzle | Path Search | dynamic | non-monotonic | rule-based | 2D | deterministic | formal | deduction | heuristic |

**15 Puzzle**   : The 15 Puzzle is a sliding tile puzzle where the player must rearrange a scrambled 4x4 grid of 15 numbered tiles into sequential order by moving them one at a time into an adjacent empty space.

**2048**   : Given an initial grid state and a sequence of moves, simulate the 2048 game mechanics—sliding, merging, and adding new tiles—to determine the final grid state.

**8 Puzzle**   : The user must find the shortest sequence of sliding tile moves to rearrange a scrambled 3x3 grid of numbers from a given initial state to a specific goal configuration.

## F.2  A

There are 2 logical reasoning puzzles in the category A, as shown in Table 26.

Table 26: Concise metadata for logical reasoning puzzles in category A.

| name | task | dynamic | monotonic | rule_based | dimension | probabilistic | formal | model | method |
|------|------|---------|-----------|------------|-----------|---------------|--------|-------|--------|
| ARC-AGI | Structural Transformation | static | monotonic | rule-based | 2D | deterministic | formal | induction | constraint |
| Arrow Maze | Grid Constraint | static | non-monotonic | rule-based | 2D | deterministic | formal | deduction | constraint |

**ARC-AGI**   : The core of the puzzle is to identify a single, abstract transformation rule from a few demonstration pairs and apply it to solve a new, unseen test grid.

**Arrow Maze**   : The core of the puzzle is to fill a grid with directional arrows such that for each number clue, its value equals the sum of the lengths of all continuous arrow rays starting from its adjacent cells.

## F.3  B

There are 4 logical reasoning puzzles in the category B, as shown in Table 27.

Table 27: Concise metadata for logical reasoning puzzles in category B.

| name | task | dynamic | monotonic | rule_based | dimension | probabilistic | formal | model | method |
|------|------|---------|-----------|------------|-----------|---------------|--------|-------|--------|
| Big Bench Symbolic | Rule-Based Simulation | static | non-monotonic | rule-based | 1D | deterministic | formal | mixed | causal |
| Binario | Grid Constraint | static | non-monotonic | rule-based | 2D | deterministic | formal | deduction | constraint |
| Boolean Expressions | Rule-Based Simulation | static | monotonic | rule-based | 1D | deterministic | formal | deduction | constraint |
| Buggy Tables | Rule-Based Simulation | static | non-monotonic | rule-based | 2D | deterministic | formal | deduction | constraint |

**Big Bench Symbolic**   : The user must deduce a hidden transformation rule that converts an input list of integers into an output list based on a series of examples, and then apply this rule to a final test case.

**Binario** : The user must fill a square grid with zeros and ones, ensuring that each row and column has an equal number of each value, no more than two identical numbers are adjacent, and all rows and columns are unique.

**Boolean Expressions** : The user must evaluate a series of complex boolean expressions, which uniquely blend arithmetic, logical operators, and real-world factual statements, to determine which ones are true.

**Buggy Tables** : The puzzle requires the solver to first restore a tabular dataset by reversing a single, systematic bug and then execute a specified data analysis query on the corrected data.

## F.4 C

There are 14 logical reasoning puzzles in the category C, as shown in Table 28.

Table 28: Concise metadata for logical reasoning puzzles in category C.

| name | task | dynamic | monotonic | rule_based | dimension | probabilistic | formal | model | method |
|---|---|---|---|---|---|---|---|---|---|
| Calcudoko | Grid Constraint | static | non-monotonic | rule-based | 2D | deterministic | formal | deduction | constraint |
| Campsite | Grid Constraint | static | non-monotonic | rule-based | 2D | deterministic | formal | deduction | constraint |
| Car Painting | Path Search | static | non-monotonic | rule-based | 1D | deterministic | formal | deduction | constraint |
| Character search | Rule-Based Simulation | static | monotonic | rule-based | 1D | deterministic | formal | deduction | constraint |
| Checkmate in One | Game Decision | static | non-monotonic | rule-based | 2D | deterministic | formal | deduction | constraint |
| Cipher | Rule-Based Simulation | static | non-monotonic | rule-based | 2D | deterministic | formal | deduction | constraint |
| code | Rule-Based Simulation | dynamic | non-monotonic | rule-based | XD | deterministic | formal | deduction | constraint |
| Combinatorial calculation | Path Search | static | non-monotonic | rule-based | 1D | deterministic | formal | deduction | constraint |
| Constrained Linear Arrangement | Rule-Based Simulation | static | monotonic | rule-based | 1D | deterministic | formal | deduction | constraint |
| Countdown | Path Search | static | non-monotonic | rule-based | 1D | deterministic | formal | deduction | constraint |
| Cryptanalysis | Rule-Based Simulation | static | monotonic | rule-based | 1D | deterministic | formal | deduction | constraint |
| Cryptarithm | Path Search | static | non-monotonic | rule-based | 1D | deterministic | formal | deduction | constraint |
| Crypto KKA | Rule-Based Simulation | static | non-monotonic | rule-based | 3D | deterministic | formal | deduction | constraint |
| Crypto KPA | Structural Transformation | static | non-monotonic | rule-based | 1D | deterministic | formal | mixed | heuristic |

**Calcudoko** : The puzzle requires filling an N×N grid with numbers 1 to N such that each number appears once per row and column, and the numbers within specified 'cages' combine arithmetically to a given target.

**Campsite** : The core of the Campsite puzzle is to place a tent orthogonally adjacent to each tree on a grid, ensuring no two tents are adjacent and that the number of tents in each row and column matches the specified counts.

**Car Painting** : The puzzle is to reorder a sequence of colored items to minimize the number of adjacent color changes, under the constraint that each item can only be displaced by a limited distance from its original position.

**Character search** : The solver must decrypt an encrypted string by applying the inverse of a given character-shifting cipher rule.

**Checkmate in One** : The solver must identify the single, unique chess move that delivers an immediate checkmate from a given board position.

**Cipher** : The solver must perform a deterministic cryptographic transformation on a given string by strictly following a complete and explicit set of rules and parameters.

**code** : Deduce the output of a described computational process by analyzing its underlying rules and logic.

**Combinatorial calculation** : The user must insert arithmetic operators and parentheses into a fixed sequence of numbers to create an expression that evaluates to a specific target value.

**Constrained Linear Arrangement** : The user must deduce the single correct linear sequence of a set of unique items by logically applying a series of constraints on their positions and relationships.

**Countdown** : Construct an arithmetic expression using a given set of integers, each exactly once, to equal a target number, under the constraint that every intermediate calculation must result in a positive integer.

**Cryptanalysis** : The solver must deduce a secret target value, sequence, or set by logically interpreting a series of clues, where each clue provides feedback on a specific guess.

**Cryptarithm** : The puzzle requires finding the unique one-to-one mapping of letters to digits (0-9) that satisfies a given arithmetic equation.

**Crypto KKA** : The user must decrypt a given ciphertext by applying the rules of a specified classical cipher using a provided key.

**Crypto KPA** : The user must analyze a given plaintext and its corresponding ciphertext to deduce the secret classical encryption algorithm and key, and then apply this to decrypt a new target ciphertext.

## F.5 D

There are 3 logical reasoning puzzles in the category D, as shown in Table 29.

Table 29: Concise metadata for logical reasoning puzzles in category D.

| name | task | dynamic | monotonic | rule_based | dimension | probabilistic | formal | model | method |
|---|---|---|---|---|---|---|---|---|---|
| Dyck Language | Rule-Based Simulation | static | monotonic | rule-based | 1D | deterministic | formal | deduction | constraint |
| Dyck Language Errors | Rule-Based Simulation | dynamic | non-monotonic | rule-based | 2D | deterministic | formal | deduction | constraint |
| Dyck Language Reasoning Errors | Rule-Based Simulation | dynamic | non-monotonic | rule-based | 2D | deterministic | formal | deduction | constraint |

**Dyck Language** : The puzzle requires appending the minimal sequence of closing brackets to an incomplete string to create a well-formed bracket sequence.

**Dyck Language Errors** : The puzzle requires identifying the 1-based index of the first error in a given bracket sequence according to specified matching and nesting rules.

**Dyck Language Reasoning Errors** : The user must audit a provided step-by-step trace of a stack-based validation process for a balanced bracket sequence and identify all steps where the reported stack state is incorrect.

## F.6 E

There are 2 logical reasoning puzzles in the category E, as shown in Table 30.

Table 30: Concise metadata for logical reasoning puzzles in category E.

| name | task | dynamic | monotonic | rule_based | dimension | probabilistic | formal | model | method |
|---|---|---|---|---|---|---|---|---|---|
| Eight Queens puzzle | Grid Constraint | static | non-monotonic | rule-based | 2D | deterministic | formal | deduction | constraint |
| Element operations | Rule-Based Simulation | static | monotonic | rule-based | 2D | deterministic | formal | deduction | constraint |

**Eight Queens puzzle** : The puzzle requires placing a specified number of queens on a grid such that no two queens can attack each other, while also adhering to any initial queen placements or forbidden cells.

**Element operations** : The puzzle requires transforming a character matrix by deleting entire rows, columns, or the elements on diagonals if they form a symmetrical (palindromic) sequence.

## F.7 F

There are 3 logical reasoning puzzles in the category F, as shown in Table 31.

Table 31: Concise metadata for logical reasoning puzzles in category F.

| name | task | dynamic | monotonic | rule_based | dimension | probabilistic | formal | model | method |
|---|---|---|---|---|---|---|---|---|---|
| FOLIO | Rule-Based Simulation | static | monotonic | rule-based | 1D | deterministic | formal | deduction | constraint |
| Full Crosswords | Grid Constraint | static | non-monotonic | rule-based | 2D | deterministic | formal | deduction | constraint |
| Futoshiki | Grid Constraint | static | non-monotonic | rule-based | 2D | deterministic | formal | deduction | constraint |

**FOLIO** : The user must evaluate the logical validity of a conclusion (True, False, or Unknown) based strictly on a given set of premises.

**Full Crosswords** : The user must solve a crossword puzzle where words are formed exclusively by entire rows and columns that are completely free of blocked cells.

**Futoshiki** : The core of Futoshiki is to complete a Latin square grid by placing numbers from 1 to N in each cell, ensuring that all specified greater-than or less-than relationships between adjacent numbers are also satisfied.

### F.8  G

There are 2 logical reasoning puzzles in the category G, as shown in Table 32.

Table 32: Concise metadata for logical reasoning puzzles in category G.

| name | task | dynamic | monotonic | rule_based | dimension | probabilistic | formal | model | method |
|---|---|---|---|---|---|---|---|---|---|
| Game24 | Path Search | static | non-monotonic | rule-based | 1D | deterministic | formal | deduction | constraint |
| Goods Exchange | Rule-Based Simulation | dynamic | non-monotonic | rule-based | 2D | deterministic | formal | deduction | causal |

**Game24** : Using a given set of integers exactly once and the four basic arithmetic operations, the objective is to form an expression that equals 24.

**Goods Exchange** : The solver must meticulously track the ownership of unique items as they are exchanged among a group of participants according to a sequence of actions to determine the final distribution.

### F.9  H

There are 3 logical reasoning puzzles in the category H, as shown in Table 33.

Table 33: Concise metadata for logical reasoning puzzles in category H.

| name | task | dynamic | monotonic | rule_based | dimension | probabilistic | formal | model | method |
|---|---|---|---|---|---|---|---|---|---|
| Hamiltonian Cycle | Path Search | static | non-monotonic | rule-based | XD | deterministic | formal | deduction | constraint |
| Hamiltonian Path | Path Search | static | non-monotonic | rule-based | XD | deterministic | formal | deduction | constraint |
| Hitori | Grid Constraint | static | non-monotonic | rule-based | 2D | deterministic | formal | deduction | constraint |

**Hamiltonian Cycle** : The core of the Hamiltonian Cycle puzzle is to determine if a path exists within a given graph that visits every vertex exactly once before returning to the starting vertex.

**Hamiltonian Path** : Find a path in a given undirected graph that visits every vertex exactly once.

**Hitori** : The core of the puzzle is to black out cells in a grid to eliminate duplicate numbers within each row and column, under the constraints that black cells cannot be orthogonally adjacent and all remaining white cells must form a single, connected group.

### F.10  K

There are 3 logical reasoning puzzles in the category K, as shown in Table 34.

Table 34: Concise metadata for logical reasoning puzzles in category K.

| name | task | dynamic | monotonic | rule_based | dimension | probabilistic | formal | model | method |
|---|---|---|---|---|---|---|---|---|---|
| Kakurasu | Grid Constraint | static | monotonic | rule-based | 2D | deterministic | formal | Deduction | constraint |
| knights_and_knaves | Rule-Based Simulation | static | monotonic | rule-based | 1D | deterministic | formal | deduction | constraint |
| Kukurasu | Grid Constraint | static | non-monotonic | rule-based | 2D | deterministic | formal | deduction | constraint |

**Kakurasu**  : The player must blacken cells in a grid to satisfy given sum constraints for each row and column, where a cell's value is determined by its 1-indexed position within that row or column.

**knights_and_knaves**  : The solver must deduce whether characters are truth-telling Knights or lying Knaves by analyzing their statements, using the core logical principle that a statement is true if and only if the speaker is a Knight.

**Kukurasu**  : Kukurasu is a logic puzzle where players mark cells in a grid to satisfy target sums for each row and column, with each marked cell's value weighted by its column index for the row's sum and by its row index for the column's sum.

## F.11   L

There are 6 logical reasoning puzzles in the category L, as shown in Table 35.

Table 35: Concise metadata for logical reasoning puzzles in category L.

| name | task | dynamic | monotonic | rule_based | dimension | probabilistic | formal | model | method |
|---|---|---|---|---|---|---|---|---|---|
| Letter logic diagram | Grid Constraint | static | monotonic | rule-based | 2D | deterministic | formal | deduction | constraint |
| Light Up | Grid Constraint | static | non-monotonic | rule-based | 2D | deterministic | formal | deduction | constraint |
| Lights out | Rule-Based Simulation | dynamic | non-monotonic | rule-based | 3D | deterministic | formal | deduction | causal |
| line_forming_game | Rule-Based Simulation | dynamic | monotonic | rule-based | 2D | deterministic | formal | deduction | constraint |
| Logic puzzle | Grid Constraint | static | non-monotonic | rule-based | 2D | deterministic | formal | deduction | constraint |
| Logical equations | Algebraic Derivation | static | monotonic | rule-based | XD | deterministic | formal | deduction | constraint |

**Letter logic diagram**  : The user fills an N x N grid with a set of N symbols, adhering to a specified combination of logical constraints such as uniqueness or uniformity within rows, columns, and/or diagonals.

**Light Up**  : Strategically place light bulbs in a grid to illuminate every white cell, ensuring that no bulb illuminates another and that any numbered black cells have the specified count of adjacent bulbs.

**Lights out**  : The puzzle requires calculating the final state of a grid by simulating a sequence of operations where pressing specified buttons cumulatively and cyclically alters the states of predefined lights.

**line_forming_game**  : Given the final state of an N-in-a-row game, determine the winner and construct a valid, turn-based move sequence that results in the provided board.

**Logic puzzle**  : Select a specified number of cells from a numerical grid such that the aggregated value (sum or product) of the selected numbers within each active row and column satisfies a single, global constraint.

**Logical equations**  : The solver must deduce the unique numerical value for each symbol from a given multiset by solving a system of mathematical and logical constraints.

## F.12   M

There are 8 logical reasoning puzzles in the category M, as shown in Table 36.

**Magic Square**  : The user must complete a partially filled square grid with distinct integers so that the sum of the numbers in each row, column, and both main diagonals is the same.

Table 36: Concise metadata for logical reasoning puzzles in category M.

| name | task | dynamic | monotonic | rule_based | dimension | probabilistic | formal | model | method |
|------|------|---------|-----------|------------|-----------|---------------|--------|-------|--------|
| Magic Square | Algebraic Derivation | static | non-monotonic | rule-based | 2D | deterministic | formal | deduction | constraint |
| Mahjong-type | Rule-Based Simulation | dynamic | non-monotonic | rule-based | 2D | deterministic | formal | deduction | constraint |
| Math Path | Rule-Based Simulation | static | non-monotonic | rule-based | 1D | deterministic | formal | deduction | constraint |
| Mathador | Path Search | static | non-monotonic | rule-based | 1D | deterministic | formal | Deduction | constraint |
| Matrix transformation | Rule-Based Simulation | static | monotonic | rule-based | 2D | deterministic | formal | deduction | constraint |
| maze | Path Search | static | non-monotonic | rule-based | 2D | deterministic | formal | deduction | constraint |
| Minesweeper | Grid Constraint | static | monotonic | rule-based | 2D | deterministic | formal | deduction | constraint |
| Mutual generation and restriction | Rule-Based Simulation | dynamic | monotonic | rule-based | 2D | deterministic | formal | deduction | constraint |

**Mahjong-type** : The user must simulate a simplified, multi-round Mahjong game by managing a hand of tiles, applying a strict hierarchy of pattern-matching rules after each draw to determine the final round's outcome.

**Math Path** : The user must deduce the missing single-digit numbers in a mathematical expression to make the equation true, while strictly following the order of operations and rules for integer division.

**Mathador** : Construct a mathematical expression equal to a target value using each number from a given list exactly once and a set of allowed arithmetic operators.

**Matrix transformation** : The puzzle requires the solver to apply a specific geometric transformation, either a 90-degree clockwise rotation or a horizontal flip, to a given 2D matrix.

**maze** : The user must find a continuous path of adjacent, passable cells from a designated start point to an end point on a grid, navigating around impassable obstacles.

**Minesweeper** : The solver must deduce the locations of hidden mines on a grid based on numerical clues that count mines in adjacent cells or along cardinal directions.

**Mutual generation and restriction** : Deduce an opponent's unique sequence of single-use piece plays by applying a complete set of interaction rules to a known history of your own moves and their win/loss/draw outcomes.

## F.13 N

There are 6 logical reasoning puzzles in the category N, as shown in Table 37.

Table 37: Concise metadata for logical reasoning puzzles in category N.

| name | task | dynamic | monotonic | rule_based | dimension | probabilistic | formal | model | method |
|------|------|---------|-----------|------------|-----------|---------------|--------|-------|--------|
| New operator calculation | Rule-Based Simulation | static | monotonic | rule-based | 1D | deterministic | formal | deduction | constraint |
| Nine Puzzle | Path Search | dynamic | non-monotonic | rule-based | 2D | deterministic | formal | deduction | heuristic |
| NL Navigation | Path Search | static | monotonic | rule-based | 2D | deterministic | formal | deduction | constraint |
| Norinori | Path Search | static | non-monotonic | rule-based | 2D | deterministic | formal | deduction | constraint |
| Number Wall | Grid Constraint | static | non-monotonic | rule-based | 2D | deterministic | formal | deduction | constraint |
| Numbrix | Grid Constraint | static | non-monotonic | rule-based | 2D | deterministic | formal | deduction | constraint |

**New operator calculation** : The user must calculate the value of a mathematical expression containing new, custom-defined operators by applying the provided rules for those operators.

**Nine Puzzle** : The user must find a sequence of circular row and column shifts to arrange a 3x3 grid of numbers from 1 to 9 in ascending order.

**NL Navigation** : The puzzle requires interpreting a natural language description of landmarks arranged in a binary tree to find the shortest path from the root to the nearest landmark of a specified type.

**Norinori** : Tile a grid partitioned into regions with non-adjacent dominoes, covering exactly two cells in each region.

**Number Wall**  : Place walls on a grid to create distinct, orthogonally connected islands, where each island contains exactly one number clue that dictates its size.

**Numbrix**  : The core of the Numbrix puzzle is to fill a grid with a continuous sequence of numbers, forming a single, unbroken path where consecutive numbers are always orthogonally adjacent.

## F.14  O

There are 3 logical reasoning puzzles in the category O, as shown in Table 38.

Table 38: Concise metadata for logical reasoning puzzles in category O.

| name | task | dynamic | monotonic | rule_based | dimension | probabilistic | formal | model | method |
|---|---|---|---|---|---|---|---|---|---|
| Object Counting | Rule-Based Simulation | static | monotonic | rule-based | 2D | deterministic | formal | deduction | constraint |
| Object Properties | Rule-Based Simulation | dynamic | non-monotonic | rule-based | 1D | probabilistic | formal | deduction | constraint |
| Operation | Rule-Based Simulation | static | non-monotonic | rule-based | 1D | deterministic | formal | deduction | constraint |

**Object Counting**  : The puzzle requires the solver to parse a narrative, identify items belonging exclusively to the protagonist "I", categorize them, and perform a specific arithmetic calculation on their quantities while ignoring distractor items owned by other characters.

**Object Properties**  : The puzzle requires the solver to track the properties of objects in a collection as they are modified by a sequence of given rules, and then to count the number of objects that satisfy a final set of conditions.

**Operation**  : The puzzle requires calculating the value of a mathematical expression by applying rules for newly defined operators and a custom order of operations.

## F.15  P

There are 3 logical reasoning puzzles in the category P, as shown in Table 39.

Table 39: Concise metadata for logical reasoning puzzles in category P.

| name | task | dynamic | monotonic | rule_based | dimension | probabilistic | formal | model | method |
|---|---|---|---|---|---|---|---|---|---|
| Path movement | Rule-Based Simulation | dynamic | non-monotonic | rule-based | 2D | deterministic | formal | deduction | constraint |
| Pattern recognition | Rule-Based Simulation | static | monotonic | rule-based | 2D | deterministic | formal | deduction | constraint |
| Pooling | Rule-Based Simulation | static | non-monotonic | rule-based | 2D | deterministic | formal | deduction | constraint |

**Path movement**  : Determine the final position of an object on a grid by simulating its movement according to a sequence of commands, accounting for wrap-around boundaries and immobilizing traps.

**Pattern recognition**  : The user must locate the unique square sub-grid within a character matrix that conforms to a given pattern rule and report the coordinates of its specified corner.

**Pooling**  : The user must calculate a new, smaller matrix by applying an aggregation operation, such as finding the maximum or integer average, to the values within a sliding window that moves across an input matrix.

## F.16  R

There are 1 logical reasoning puzzles in the category R, as shown in Table 40.

**Reversi**  : Calculate the final state of a Reversi game board by simulating a sequence of moves, where each move involves placing a piece to outflank and flip the opponent's pieces in all eight directions.

Table 40: Concise metadata for logical reasoning puzzles in category R.

| name | task | dynamic | monotonic | rule_based | dimension | probabilistic | formal | model | method |
|---|---|---|---|---|---|---|---|---|---|
| Reversi | Rule-Based Simulation | dynamic | non-monotonic | rule-based | 3D | deterministic | formal | deduction | constraint |

## F.17 S

There are 24 logical reasoning puzzles in the category S, as shown in Table 41.

Table 41: Concise metadata for logical reasoning puzzles in category S.

| name | task | dynamic | monotonic | rule_based | dimension | probabilistic | formal | model | method |
|---|---|---|---|---|---|---|---|---|---|
| Single-choice self-reasoning | Rule-Based Simulation | static | non-monotonic | rule-based | 1D | deterministic | formal | deduction | constraint |
| Sixteen Puzzle | Path Search | dynamic | non-monotonic | rule-based | 2D | deterministic | formal | deduction | heuristic |
| Skyscraper | Grid Constraint | static | non-monotonic | rule-based | 2D | deterministic | formal | deduction | constraint |
| Skyscraper Puzzle | Grid Constraint | static | non-monotonic | rule-based | 2D | deterministic | formal | deduction | constraint |
| Slant | Grid Constraint | static | non-monotonic | rule-based | 2D | deterministic | formal | deduction | constraint |
| Space Reasoning | Rule-Based Simulation | static | non-monotonic | rule-based | 2D | deterministic | formal | deduction | constraint |
| Space Reasoning Tree | Rule-Based Simulation | static | monotonic | rule-based | XD | deterministic | formal | deduction | constraint |
| Stack Permutation | Rule-Based Simulation | dynamic | non-monotonic | rule-based | 2D | deterministic | formal | deduction | constraint |
| Star Battle | Grid Constraint | static | non-monotonic | rule-based | 2D | deterministic | formal | deduction | constraint |
| Star Placement Puzzle | Grid Constraint | static | monotonic | rule-based | 2D | deterministic | formal | deduction | constraint |
| Statistical counting | Rule-Based Simulation | static | non-monotonic | rule-based | 1D | deterministic | formal | deduction | constraint |
| string | Rule-Based Simulation | dynamic | monotonic | rule-based | 2D | deterministic | formal | induction | causal |
| String deletion and modification | Rule-Based Simulation | dynamic | non-monotonic | rule-based | 2D | deterministic | formal | deduction | constraint |
| String insertion | Rule-Based Simulation | dynamic | monotonic | rule-based | 1D | deterministic | formal | deduction | causal |
| String processing | Rule-Based Simulation | dynamic | non-monotonic | rule-based | 2D | deterministic | formal | deduction | causal |
| String rearrangement | Rule-Based Simulation | static | non-monotonic | rule-based | 1D | deterministic | formal | deduction | constraint |
| String splitting | Rule-Based Simulation | static | monotonic | rule-based | 1D | deterministic | formal | deduction | constraint |
| String synthesis | Rule-Based Simulation | dynamic | non-monotonic | rule-based | XD | deterministic | formal | deduction | constraint |
| sudoku | Grid Constraint | static | non-monotonic | rule-based | 2D | deterministic | formal | deduction | constraint |
| Sudoku with arithmetic rules | Grid Constraint | static | monotonic | rule-based | 2D | deterministic | formal | deduction | constraint |
| Sum Skyscraper | Grid Constraint | static | non-monotonic | rule-based | 2D | deterministic | formal | deduction | constraint |
| Survo | Grid Constraint | static | non-monotonic | rule-based | 2D | deterministic | formal | deduction | constraint |
| Symbolic Hard | Structural Transformation | static | non-monotonic | rule-based | 2D | deterministic | formal | induction | heuristic |
| Synthesis and decomposition | Rule-Based Simulation | dynamic | non-monotonic | rule-based | 1D | deterministic | formal | deduction | causal |

**Single-choice self-reasoning** : The solver must find the single unique combination of answers to a set of multiple-choice questions that makes all the self-referential statements asserted by the chosen options true.

**Sixteen Puzzle** : The Sixteen Puzzle requires players to sort a 4x4 grid of numbered tiles into ascending order by cyclically shifting entire rows or columns.

**Skyscraper** : The goal is to place numbers from 1 to N into an N×N grid, adhering to Latin Square rules (no repeats in any row or column), while also satisfying external clues that specify the number of visible 'buildings' from that line of sight.

**Skyscraper Puzzle** : The Skyscraper Puzzle requires placing numbers 1 to N in a grid, with no repeats in any row or column, to satisfy external clues that count the number of visible buildings from each side.

**Slant** : Fill a grid with diagonal lines to satisfy numeric clues at the intersections while ensuring no closed loops are formed.

**Space Reasoning** : The puzzle requires the solver to deduce the layout of a hidden map of item-labeled nodes by interpreting a narrative of relative movements between them, in order to identify the item at a final queried location.

**Space Reasoning Tree** : The puzzle requires the solver to deduce the cousins of a target node within a hidden tree by piecing together essential relationships from a large collection of randomized and distracting clues.

**Stack Permutation** : The puzzle's core is to determine if a target output sequence can be generated from an input sequence using only the push and pop operations of a last-in, first-out (LIFO) stack.

**Star Battle** : The core of the puzzle is to place exactly one star in each row and column of a grid, ensuring no two stars are adjacent (horizontally, vertically, or diagonally) and that stars only occupy designated playable cells.

**Star Placement Puzzle** : The puzzle requires placing a specified number of stars in each row, column, and region of a partitioned grid, with the critical constraint that no two stars can be adjacent in any direction.

**Statistical counting** : The user must calculate a final score by modifying an initial value based on the occurrences of predefined character runs and substrings within a given input string.

**string** : The puzzle's rule is to first insert the string 'V' after the substring 'AN', and then in each subsequent step, prepend a new specific string to the growing string that was inserted in the previous steps.

**String deletion and modification** : The puzzle requires finding the final state of a string by iteratively applying a given set of conditional rules that modify or delete its prefixes and suffixes.

**String insertion** : The puzzle involves determining the final state of a string after sequentially applying an ordered set of conditional insertion rules, where each successful application modifies the string for all subsequent rules.

**String processing** : The solver must determine the final state of a string after sequentially applying a given series of manipulation commands.

**String rearrangement** : The puzzle requires applying a defined set of rules to transform strings that represent numbers in various formats.

**String splitting** : The puzzle requires partitioning a string into an ordered sequence of substrings by identifying all locations that match a given set of rules and then making all the specified cuts simultaneously.

**String synthesis** : The puzzle involves calculating the final quantities of items by applying a set of transformation rules according to a specified protocol until no more transformations can be made.

**sudoku** : Sudoku is a logic-based puzzle where the objective is to fill an N×N grid with a set of N distinct symbols, ensuring that each symbol appears exactly once in every row, column, and any other specified constraint region.

**Sudoku with arithmetic rules** : The core of the puzzle is to fill a grid with numbers, satisfying both Sudoku-like row and column uniqueness constraints and arithmetic rules within specified regions (cages).

**Sum Skyscraper** : Fill an N x N grid with unique numbers from 1 to N in each row and column, where perimeter clues dictate the sum of the heights of visible skyscrapers from that direction.

**Survo** : The core of the puzzle is to fill the empty cells of a grid with a given set of numbers so that each row and column adds up to its specified sum.

**Symbolic Hard** : The transformation rule is conditional, with the specific rule being determined by a global property or a secondary pattern within the input grid.

**Synthesis and decomposition** : The user must simulate a deterministic resource transformation system by repeatedly applying an ordered list of rules in a cycle until the system reaches a stable state where no further changes are possible.

## F.18 T

There are 5 logical reasoning puzzles in the category T, as shown in Table 42.

Table 42: Concise metadata for logical reasoning puzzles in category T.

| name | task | dynamic | monotonic | rule_based | dimension | probabilistic | formal | model | method |
|------|------|---------|-----------|------------|-----------|---------------|--------|-------|--------|
| tetris | Rule-Based Simulation | dynamic | non-monotonic | rule-based | 3D | deterministic | formal | deduction | constraint |
| Tic Tac Toe | Game Decision | dynamic | non-monotonic | rule-based | 2D | deterministic | formal | deduction | heuristic |
| Time Sequence | Rule-Based Simulation | static | non-monotonic | rule-based | 2D | deterministic | formal | deduction | constraint |
| Train Game Card Puzzle | Rule-Based Simulation | dynamic | non-monotonic | rule-based | 2D | deterministic | formal | deduction | constraint |
| Twiddle | Path Search | dynamic | non-monotonic | rule-based | 3D | deterministic | formal | deduction | heuristic |

**tetris** : Calculate the final state of a Tetris board after a series of blocks are placed according to a specified sequence of operations.

**Tic Tac Toe** : The user must determine the single optimal move for the active player on a given N×N Tic-Tac-Toe board, following a strict hierarchy of winning, blocking, or making a strategic play.

**Time Sequence** : The user must determine the longest possible common meeting time for a group by analyzing and merging individual weekly schedules, each potentially modified by a unique set of constraints and rules.

**Train Game Card Puzzle** : The puzzle requires simulating a turn-based card game to determine the sequence of cards played in successive rounds, where each round ends when a player plays a card that matches the rank of a previous card in the play pile, allowing them to collect the intervening cards.

**Twiddle** : The puzzle requires sorting a grid of numbers into ascending, row-major order by repeatedly rotating 2x2 subgrids counterclockwise.

## F.19 W

There are 4 logical reasoning puzzles in the category W, as shown in Table 43.

Table 43: Concise metadata for logical reasoning puzzles in category W.

| name | task | dynamic | monotonic | rule_based | dimension | probabilistic | formal | model | method |
|------|------|---------|-----------|------------|-----------|---------------|--------|-------|--------|
| Web of Lies | Rule-Based Simulation | static | monotonic | rule-based | XD | deterministic | formal | deduction | constraint |
| Word Sorting | Rule-Based Simulation | static | monotonic | rule-based | 1D | deterministic | formal | deduction | constraint |
| Word Sorting Mistake | Rule-Based Simulation | dynamic | monotonic | rule-based | 1D | deterministic | formal | deduction | constraint |
| Wordscapes | Grid Constraint | static | non-monotonic | rule-based | 2D | deterministic | formal | deduction | constraint |

**Web of Lies** : Players must determine whether characters are consistent truth-tellers or liars by logically analyzing a web of interconnected statements whose truth value is contingent on the speaker's own identity.

**Word Sorting** : The user must sort a list of words lexicographically based on a custom alphabet created by moving a specified, ordered set of letters to the beginning of the standard alphabet.

**Word Sorting Mistake** : The user must analyze a simulated, step-by-step alphabetical sorting of words and identify the precise first thought that contains a logical or factual error.

**Wordscapes** : The core of the puzzle is to fit provided horizontal and vertical words into a grid, ensuring that they correctly interlock at every shared letter.

## F.20 Z

There are 1 logical reasoning puzzles in the category Z, as shown in Table 44.

Table 44: Concise metadata for logical reasoning puzzles in category Z.

| name | task | dynamic | monotonic | rule_based | dimension | probabilistic | formal | model | method |
|---|---|---|---|---|---|---|---|---|---|
| zebra | Grid Constraint | static | monotonic | rule-based | 2D | deterministic | formal | deduction | constraint |

**zebra** : The core of the Zebra Puzzle is to use deductive reasoning to assign a unique set of attributes to a series of ordered entities based on a given list of relational clues.

# G EVOLUTION

This section provides a systematic exposition of the evolution mechanism underlying the proposed LogicEvolve framework. We describe the meaning of evolutionary capability, its categories, execution pipeline, and representative examples, thereby illustrating how LogicEvolve enables scalable and continual expansion of structured logical reasoning tasks.

## G.1 EVOLUTIONARY CAPABILITY OF LOGICEVOLVE

To avoid unintended associations with open-ended or unconstrained forms of "self-evolution," we explicitly narrow the scope of this term. Throughout this paper, evolution refers to *automated structural transformations, difficulty scaling, and large-scale instance synthesis conducted under explicit meta-rule constraints*. This process does not rely on implicit human tuning, nor does it imply any form of open-ended creativity. Instead, LogicEvolve performs systematic extensions of existing or newly created task families while maintaining formal verifiability and programmatic correctness.

It is important to note that LogicEvolve does not simulate biological evolution, does not employ genetic operators, and does not perform fitness-based selection. All transformations arise strictly from structured error signals and programmatic verification, rather than evolutionary computation.

## G.2 TYPES OF EVOLUTION

Building on the operational definition above, LogicEvolve currently supports two primary forms of evolution.

(1) Task-family extension. The system systematically explores a predefined metadata space to generate new structural templates, constraint formulations, and hyperparameter configurations. This allows an existing task family (e.g., Sudoku, Zebra puzzles) to be automatically extended into richer and more diverse variants, effectively broadening the structural coverage of the original family.

(2) Structure-preserving transformations. Within a fixed task type, the system applies controlled structural variations—such as modifying the number of operational primitives, altering the topology of the constraint graph, or adjusting the number of involved entities. These changes substantially expand the state-space size and reasoning depth (e.g., extending a string task from 10 to 11 operations, or increasing card suits from 4 to 5). Such transformations lead to measurable increases in structural complexity rather than superficial textual rewrites.

## G.3 EVOLUTIONARY PIPELINE

When tasked with extending an existing family or synthesizing a new one, the system initiates a structured evolutionary pipeline. A metadata agent first specifies the task objects, observable variables, foundational rules, and the corresponding hyperparameter space, ensuring that they remain codable and verifiable. A generator then instantiates candidate tasks from the meta-rules, while solvers and validators examine each instance for solvability, uniqueness, and controlled search complexity—using code execution, symbolic solvers, and CSP/SAT solvers as needed.

Instances that fail these criteria (e.g., multiple solutions, no solution, uncontrolled search) produce failure logs, which are fed back to the metadata agent to refine rules and adjust the hyperparameter space. Once the task rules stabilize, the system further increases reasoning depth and task difficulty by manipulating hyperparameters such as the number of objects, constraint density, and clue sparsity.

## G.4 EVOLUTIONARY EXAMPLES

Beyond extending classical task families, LogicEvolve has also produced several new task families that, to the best of our knowledge, do not have direct counterparts in existing public datasets or puzzle repositories. All such families are generated through the same metadata–generator–solver closed loop and verified via programmatic uniqueness checks to ensure logical consistency.

### G.4.1 Cascading Filter Views

We introduce a novel puzzle family termed Cascading Filter Views, constructed around a set of propositions each with an underlying binary state (visible / hidden). The task defines a collection of filters, each applying one of three operations to every proposition: Pass, Block, Invert.

A view is formed by cascading multiple filters in sequence, and the puzzle provides only the final visibility pattern of several such views. The solver must jointly infer both the base proposition states and the behavior matrix of every filter in the cascade.

This structure is distinct from existing puzzle families such as Zebra, Sudoku, or black-box reasoning puzzles; instead, it centers on the interplay between filter-operation matrices, multi-view composition, and strict uniqueness constraints, forming a new class of logically structured reasoning tasks.

Below is an example instance of this task family:

**Puzzle Introduction**   Cascading Filter Views is a deductive puzzle about unseen transformations: abstract "filters" act in sequence on a set of propositions, partially revealing what survives, what is blocked, and what is inverted. The solver must reconstruct both the hidden starting states and the individual behavior of each filter, using only the final partial outcomes of several filter sequences and a small set of logical clues.

**Basic Rules**   1. Core Objects 1.1. Propositions 1.1.1. Each puzzle has M distinct propositions, named P1, P2, ..., PM. 1.1.2. Each proposition has a base state before any filter is applied: either Visible or Hidden. 1.1.3. Base states are fixed for a given puzzle instance but are unknown to the solver except where explicitly given as clues.

1.2. Filters 1.2.1. Each puzzle has N distinct filters, named F1, F2, ..., FN. 1.2.2. Each filter acts separately on each proposition that passes through it. 1.2.3. For each ordered pair (filter F, proposition P), F has exactly one action type chosen from the set Pass, Block, Invert: a. Pass: the filter leaves the incoming state of that proposition unchanged. b. Block: the filter forces that proposition's state to Hidden, regardless of its incoming state. c. Invert: the filter flips the state: Visible becomes Hidden, Hidden becomes Visible. 1.2.4. The complete behavior of a filter is the list of its action types over all propositions. 1.2.5. Distinctness rule: No two filters are allowed to have exactly the same behavior on all propositions; that is, for any $i \neq j$, there exists at least one proposition Pk on which Fi and Fj have different action types. 1.2.6. Activity rule: Each filter must have at least one non-Pass action type; a filter that would Pass all propositions is not allowed. 1.2.7. Palette limitation rule: For each filter, among the three action types Pass, Block, Invert, at most two distinct types may actually be used on its propositions. For example, a filter may use Pass and Block, or Pass and Invert, or Block and Invert, but not all three at once.

1.3. Views 1.3.1. Each puzzle has K distinct views, named V1, V2, ..., VK. 1.3.2. Each view is defined by a finite, non-empty ordered list of filters, called its filter chain. For example, V3 might be defined by the list (F2, F1, F4), meaning that F2 acts first, then F1, then F4. 1.3.3. The filter chain for each view is fully known to the solver. 1.3.4. A view has, for each proposition, a final state after its entire filter chain has been applied. This final state is either Visible or Hidden. 1.3.5. In the puzzle description, only some of these final states are revealed as clues; others remain unspecified and must be deduced.

2. State Propagation Through a Filter Chain 2.1. Initialisation 2.1.1. Before any filters are applied, each proposition Pi begins in its base state: Visible or Hidden. 2.2. Application of a single filter 2.2.1. When a proposition with current state S passes through filter F: a. If F's action type on that proposition is Pass, the state remains S. b. If the action type is Block, the state becomes Hidden. c. If the action type is Invert, the state becomes the opposite of S. 2.3. Application of a filter chain in a view 2.3.1. For each view Vx, its filter chain is applied in the given order to all propositions in parallel, starting from their base states. 2.3.2. The output of one filter in the chain becomes the input to the next filter in that chain for the same proposition. 2.3.3. After the last filter in the chain, each proposition has a final state in that view (Visible or Hidden). 2.3.4. The revealed final states in the puzzle statement must be exactly those produced by this process for the correct base states and filter behaviors.

3. Logical and Structural Constraints 3.1. Non-degeneracy of views 3.1.1. Each view's filter chain must use at least one filter. 3.1.2. Across all views combined, each filter must appear in at least one view's chain. 3.1.3. At least one view must contain at least two filters in its chain, so that composite effects can be distinguished from single-filter effects. 3.2. Distinguishability of filters via views 3.2.1. For any pair of distinct filters Fi and Fj, there must exist at least one proposition Pk and at least one view Vx whose filter chain contains Fi or Fj (possibly both) such that, when the correct solution is applied, the final state of Pk under Vx depends on the difference between Fi and Fj. 3.2.2. Informally, the collection of views must be rich enough that no two filters could be exchanged or merged without violating at least one revealed final-state clue. 3.3. Distinguishability of base states 3.3.1. The combination of views and clues must be such that, in the unique solution, changing the base state of any single proposition would force a violation of at least one clue or rule. 3.4. Permitted clue types 3.4.1. The puzzle may include any number of the following clue types, all of which must be logically true in the unique solution: a. Base-state clue: "Proposition Pi is base-visible" or "Proposition Pj is base-hidden". b. View-output clue: "In view Vx, proposition Pi is visible" or "In view Vy, proposition Pj is hidden". These refer to final states after the full filter chain of that view. c. Filter-local clue: For specific Fi and Pj, the action type is specified or ruled out. Examples of allowed forms: - "Filter Fi blocks Pj." - "Filter Fi does not invert Pk." d. Filter-comparison clue (same proposition): For filters Fi and Fj on a specific proposition Pk, their behaviors may be constrained. Allowed forms include: - "Fi and Fj act identically on Pk." - "Fi and Fj act differently on Pk." (meaning their action types are not the same). e. Filter-consistency clue (across propositions): For a single filter Fi and two propositions Pj and Pk, their behaviors may be related. Allowed forms include: - "Fi acts identically on Pj and Pk." - "Fi acts differently on Pj and Pk." f. Palette-use clue: For a specific filter Fi, the number of distinct action types it uses may be restricted more tightly than the general palette limitation. Allowed forms include: - "Fi uses exactly one non-Pass action type overall." - "Fi never blocks any proposition." 3.4.2. Clues must not contradict the general rules, must not introduce additional action types beyond Pass, Block, Invert, and must be expressible using the vocabulary above. 3.5. Forbidden operations for the solver 3.5.1. The solver may not assume any arithmetic relationships on counts of visible propositions, except where explicitly given by a clue. 3.5.2. The solver may not introduce additional filters, propositions, or views beyond those specified. 3.5.3. The solver must treat all unspecified base states, filter behaviors, and view outputs as unknowns constrained only by the rules and the given clues.

4. Valid Solution Definition 4.1. A candidate solution consists of: 4.1.1. A chosen base state (Visible or Hidden) for every proposition Pi. 4.1.2. A chosen action type (Pass, Block, or Invert) for every ordered pair (Fi, Pj). 4.2. A candidate solution is valid if and only if all of the following hold: 4.2.1. When all views' filter chains are applied according to the rules, every revealed view-output clue is satisfied exactly. 4.2.2. All base-state clues are satisfied. 4.2.3. All filter-local, filter-comparison, filter-consistency, and palette-use clues are satisfied. 4.2.4. The Distinctness rule (no duplicate filters) holds. 4.2.5. The Activity rule (no filter that is purely Pass) holds. 4.2.6. The Palette limitation rule (at most two action types per filter) holds. 4.2.7. All non-degeneracy and distinguishability constraints specified in Section 3 are satisfied. 4.3. A puzzle instance is considered correctly solved when the solver has determined and can state unambiguously: 4.3.1. The base state (Visible or Hidden) of every proposition. 4.3.2. The action type (Pass, Block, or Invert) of every filter on every proposition.

**Required Given Conditions**    5. Minimal Structural Requirements for a Generated Puzzle 5.1. Size constraints 5.1.1. M (number of propositions) must be at least 3. 5.1.2. N (number of filters) must be at least 2. 5.1.3. K (number of views) must be at least 2. 5.2. Specified objects 5.2.1. The puzzle statement must list all propositions P1, . . . , PM. 5.2.2. The puzzle statement must list all filters F1, . . . , FN. 5.2.3. The puzzle statement must list all views V1, . . . , VK. 5.2.4. For each view Vx, the puzzle statement must specify its filter chain as a finite ordered list of filters drawn from F1, . . . , FN. 5.2.5. At least one base-state clue or one view-output clue must be present, otherwise all base states could be symmetrically swapped without effect. 5.2.6. Enough view-output clues must be given so that the resulting system of constraints can, in principle, distinguish between different assignments of base states and filter behaviors. 5.3. Clue coverage 5.3.1. For at least one view Vx and at least two different propositions, their final states must be specified. 5.3.2. For each filter Fi, there must exist at least one clue (of any allowed type) that refers directly or indirectly to Fi, to avoid completely unconstrained filters. 5.3.3. For each proposition Pj, there must exist at least one clue that refers either to its base state or to its final state in some view.

6. Uniqueness Requirement 6.1. A puzzle instance is valid only if there exists exactly one valid solution in the sense of Section 4. 6.2. From the perspective of a generator program, to certify uniqueness: 6.2.1. Enumerate all possible assignments of base states and filter behaviors consistent with the size parameters M, N, K and the general rules (including Distinctness, Activity, and Palette limitation). 6.2.2. For each assignment, simulate all views' filter chains and test against all given clues. 6.2.3. Count how many assignments satisfy all rules and clues. 6.2.4. The puzzle is acceptable only if this count is exactly one. 6.3. If more than one solution exists, the generator must add or strengthen clues (of the allowed types) or adjust the structure of views until uniqueness is achieved.

**Final Objective** 7. Goal of the Solver 7.1. The solver's objective is to logically deduce the unique configuration of base states and filter behaviors that satisfies all given clues and all global rules. 7.2. A solver is considered to have completed the puzzle when they can: 7.2.1. Specify, for every proposition Pi, whether it is base-visible or base-hidden. 7.2.2. Specify, for every ordered pair (Fi, Pj), whether Fi passes, blocks, or inverts Pj. 7.2.3. Show, if required, that applying all views' filter chains to these base states reproduces every revealed final state and respects every clue, with no contradictions. 7.3. No alternative configuration that differs on any base state or any filter action may also satisfy all constraints; if such an alternative exists, the puzzle instance is invalid under these meta-rules and should not be presented as a Cascading Filter Views puzzle.

**Puzzle:**

```
Cascading\_Filter\_Views { Puzzle Instance:

All general rules of the Cascading\_Filter\_Views type apply.

Propositions:
- P1
- P2
- P3

Filters:
- F1
- F2
- F3

Views (filter chains, applied left to right):
- V1: F1, then F2
- V2: F2, then F3
- V3: F3, then F1

Clues:
1. Base states
   - P1 is base-visible.
   - P2 is base-hidden.
   - P3 is base-visible.

2. Filter actions (each statement refers to the action of that
filter on that proposition):
   - F1 passes P1.
   - F1 inverts P2.
   - F1 passes P3.
   - F2 blocks P1.
   - F2 passes P2.
   - F2 passes P3.
   - F3 inverts P1.
   - F3 blocks P2.
   - F3 inverts P3.
```

```
3. View outputs (final states after the entire filter chain
of that view):
   - In view V1, P1 is hidden.
   - In view V1, P2 is visible.
   - In view V1, P3 is visible.
   - In view V2, P1 is visible.
   - In view V2, P2 is hidden.
   - In view V2, P3 is hidden.
   - In view V3, P1 is hidden.
   - In view V3, P2 is visible.
   - In view V3, P3 is hidden.
```

**Answer:**

```
{
    "base_states": {
        "P1": "Visible",
        "P2": "Hidden",
        "P3": "Visible"
    },
    "filters": { d
        "F1": {
            "P1": "Pass",
            "P2": "Invert",
            "P3": "Pass"
        },
        "F2": {
            "P1": "Block",
            "P2": "Pass",
            "P3": "Pass"
        },
        "F3": {
            "P1": "Invert",
            "P2": "Block",
            "P3": "Invert"
        }
    }
}
```

### G.4.2 KAKURASU SHADOW WHEELS

We propose a novel puzzle family termed Kakurasu Shadow Wheels, built around a set of multi-layer circular discs. Each disc is partitioned into sectors that are either solid or transparent, and every disc can be rotated in discrete steps. Each shadow card specifies a pair of discs (front vs. back) together with a target shadow pattern, where the occlusion rule is deterministic: the front disc takes precedence over the back disc.

The solver must determine a unique global rotation configuration for all discs such that every shadow card's target pattern is simultaneously satisfied. This induces a reasoning structure characterized by multi-layer directional occlusion and global consistency across multiple observations, which is fundamentally different from traditional Kakurasu puzzles, rotation-based puzzles, or Lights Out variants.

These task families are not superficial rewrites of existing puzzles. They are defined by their own meta-rule systems, synthesized by the same metadata–generator–solver loop, and validated by a programmable uniqueness checker. Representative examples are provided in the appendix.

Below is an example instance from this task family:

**Puzzle Introduction**   Kakurasu Shadow Wheels is a rotational logic puzzle where you orient layered circular plates to reproduce a collection of fixed shadow patterns. The challenge is to deduce a unique orientation for every plate using only how their opaque and transparent wedges combine when stacked.

**Basic Rules**   1. Components 1.1 The puzzle has D distinct source wheels, where D is an integer greater than or equal to 2. 1.2 Each wheel is a circle divided into K equal angular sectors, where K is an integer greater than or equal to 3 and the same for all wheels in a single puzzle. 1.3 Every sector on a wheel is either solid (blocking light) or open (letting light through). These are fixed properties of the wheel and do not change during solving. 1.4 Each wheel has a printed reference marker on its rim to indicate orientation. Rotating a wheel means turning it so that different sectors line up with the reference marker position.

2. Orientations 2.1 The orientation of a wheel is defined by which one of its K sectors is currently aligned with the reference marker position; thus each wheel has exactly K possible orientations. 2.2 Wheels may only be rotated in whole-sector steps. Flipping wheels over, mirroring them, or rotating them by partial-sector amounts is not allowed.

3. Shadow stacking 3.1 A shadow clue involves exactly two distinct source wheels: one designated as the upper wheel and the other as the lower wheel. 3.2 In a shadow clue, the upper wheel is imagined to be stacked directly on top of the lower wheel, sharing the same center and the same reference marker position. 3.3 For any chosen orientations of the two wheels, the resulting shadow around the circle is determined sector by sector: a) At each sector position, light is blocked (the shadow is solid) if the corresponding sector of the upper wheel is solid, or if the upper sector is open and the corresponding lower sector is solid. b) Light passes through (the shadow is open) at a sector only if both the upper and the lower sectors at that position are open. 3.4 The shadow pattern produced by a particular stacking is itself a circle of K sectors, each sector either solid or open.

4. Shadow cards (clues) 4.1 For each ordered pair of wheels used as a clue, the puzzle provides exactly one shadow card. 4.2 A shadow card specifies: a) The identity of the upper wheel. b) The identity of the lower wheel. c) A fixed target shadow pattern: a circle of K sectors marked solid or open. 4.3 Shadow cards are printed or presented in a fixed orientation and may not be rotated, mirrored, or otherwise altered by the solver. 4.4 A configuration of wheel orientations is said to satisfy a shadow card if, when the named upper and lower wheels are stacked in that order with their current orientations, the resulting shadow pattern matches the card's target pattern exactly, sector by sector, without any additional rotation.

5. Global configuration 5.1 A candidate solution to the puzzle is an assignment of one orientation (one of the K possibilities) to every source wheel. 5.2 A candidate solution is valid if and only if it satisfies every shadow card in the puzzle simultaneously. 5.3 The wheels themselves are never altered: their own solid or open sector patterns remain fixed; only their orientations may be changed. 5.4 The relative vertical order of wheels in a shadow card is fixed; the solver may not swap "upper" and "lower" for any card.

6. Forbidden actions 6.1 The solver may not introduce any new wheels or shadow cards. 6.2 The solver may not partially satisfy a shadow card by rotating the card instead of the wheels; shadow cards are always considered immobile. 6.3 The solver may not assume any interaction between wheels other than those explicitly defined by the given shadow cards.

**Required Given Conditions**   To construct a valid Kakurasu Shadow Wheels puzzle instance that is solvable and unambiguous, a generator must provide at least the following information:

1. Structural parameters 1.1 An integer D greater than or equal to 2 giving the number of distinct source wheels. 1.2 An integer K greater than or equal to 3 giving the number of sectors on each wheel.

2. Wheel definitions 2.1 For each wheel i from 1 to D, specify a fixed pattern of length K indicating, for each sector position around the circle, whether that sector is solid or open. 2.2 At least one wheel must have both solid and open sectors. 2.3 For any wheel whose solid or open pattern has rotational symmetries (for example, repeating the same block of sectors multiple times around the circle),

the set of shadow cards must still ensure that the wheel's final orientation in the unique solution is determined unambiguously.

3. Shadow cards 3.1 Provide a finite set of shadow cards. Each shadow card must specify: a) An ordered pair of distinct wheels (upper, lower). b) A target shadow pattern of length K marking each sector as solid or open. 3.2 Every wheel that appears in the puzzle must participate in at least one shadow card. 3.3 The collection of shadow cards must be consistent: there must exist at least one assignment of orientations to all wheels that satisfies all cards simultaneously. 3.4 The collection of shadow cards must be complete for uniqueness: there must exist exactly one assignment of orientations to all wheels that satisfies all cards simultaneously. a) A puzzle with more than one valid orientation assignment is invalid and must not be presented. 3.5 No shadow card may be contradictory; the full set of shadow cards must remain jointly satisfiable.

4. Minimum instance size 4.1 The smallest recommended non-trivial instance has D equal to 3 wheels and K equal to 4 sectors. 4.2 Puzzles with D equal to 2 wheels are allowed, but care must be taken to avoid trivial symmetry where rotating both wheels together leaves all shadow cards unchanged.

5. Machine-verifiable specification 5.1 A generator must encode wheel patterns and shadow card patterns in a precise, finite form, for example as ordered lists of "solid" and "open" values indexed by sector position from 0 to K minus 1. 5.2 Using this encoding, a solver program must be able to: a) Enumerate all possible orientation assignments. There are exactly K choices for each of the D wheels, so the total number of assignments is finite. b) For each shadow card and each orientation assignment, compute the resulting shadow pattern according to the stacking rule and check for equality with the card's target pattern. c) Confirm that exactly one orientation assignment satisfies every shadow card.

**Final Objective** 1. The puzzle is solved when the orientation of every source wheel is determined such that all shadow cards are simultaneously satisfied. 2. A completed solution can be represented by specifying, for each wheel, which sector index is aligned with the reference marker in the final configuration. 3. Verification of a proposed solution consists of reapplying the stacking rule for every shadow card with the proposed orientations and confirming that every produced shadow pattern matches its card exactly. 4. Any solution that fails even one shadow card is incorrect; if more than one distinct orientation assignment satisfies all shadow cards, the puzzle instance itself is invalid under these meta-rules.

**Puzzle:**

```
Kakurasu Shadow Wheels { New Instance

Global parameters
- Number of wheels D = 3
- Number of sectors per wheel K = 4
- Sector indices: 0, 1, 2, 3 in clockwise order.
- Each sector is either "solid" (blocks light) or "open"
(lets light through).

Orientation convention
- Each wheel i has a fixed internal pattern base_i[0..3] listed
below for orientation 0.
- The orientation of wheel i is an integer o_i in {0,1,2,3}.
- o_i is defined as: the sector with index o_i is aligned with
the fixed reference marker.
- When evaluating shadows, the sector of wheel i that appears
at board position p (p in {0,1,2,3}) under orientation o_i is:
  base_i[(p - o_i) mod 4].

Shadow stacking rule
- A shadow card specifies an UPPER and a LOWER wheel, and a
target pattern T[0..3].
- When the UPPER wheel pattern U[0..3] is stacked on the LOWER
```

```
pattern L[0..3]:
  * At each position p, the resulting shadow S[p] is "solid"
  if U[p] is solid OR (U[p] is open AND L[p] is solid).
  * Equivalently, S[p] is "open" only if BOTH U[p] and L[p]
  are open.
- A shadow card is satisfied iff the produced S[0..3] matches
its target T[0..3] exactly, index by index, with no additional
rotation.
- A solution to the puzzle is a choice of orientations
(o_1, o_2, o_3) such that all shadow cards are simultaneously
satisfied.

Wheel definitions (orientation 0 patterns)
Each pattern is given as an array of 4 entries [sector 0,
sector 1, sector 2, sector 3].

- Wheel 1 (W1)
  base_1 = ["solid", "open", "solid", "solid"]

- Wheel 2 (W2)
  base_2 = ["solid", "solid", "open", "open"]

- Wheel 3 (W3)
  base_3 = ["solid", "open", "open", "open"]

Shadow cards
Each card is written as:
  (Upper wheel, Lower wheel) -> target pattern [T0, T1, T2, T3]

Card 1
- Upper: Wheel 1
- Lower: Wheel 2
- Target pattern:
  ["solid", "solid", "solid", "solid"]

Card 2
- Upper: Wheel 2
- Lower: Wheel 3
- Target pattern:
  ["open", "solid", "solid", "open"]

Card 3
- Upper: Wheel 3
- Lower: Wheel 1
- Target pattern:
  ["solid", "open", "solid", "solid"]

All wheels must share a single set of orientations (o_1, o_2, o_3)
that satisfies all three cards at once. It can be checked by
exhaustive enumeration over o_i in {0,1,2,3} that there is exactly
one such global assignment.
```

**Answer:**

```
[
    0,
    1,
```

```
        2
]
```

## G.5 COMPARATIVE ANALYSIS BEFORE AND AFTER EVOLUTION

To analyze the substantial structural improvement brought about by evolution, we compare the tasks before and after evolution from both qualitative and quantitative dimensions. The results show that evolution leads to a real increase in structural complexity, not just superficial modifications.

### G.5.1 QUANTITATIVE ANALYSIS

To verify whether LogicEvolve produces structurally distinct tasks under different sources of randomness—both the base model's sampling seed and the sampling seed for task-pool sampling during evolution—we run multiple independent small-scale evolution pipelines with different seeds, using CLUB as the initial task pool. We then extract four categories of structural statistics from each generated task under manual verification (with optional LLM assistance):

1. variables, constraints, and dependency relations; 2. the undirected constraint graph and its average shortest-path diameter; 3. the directed dependency graph and its dependency-chain length distribution; 4. an approximate estimate of the feasible configuration space via solver step counts.

Table 45 presents the structural statistics (before and after evolution) for six task instances generated under different random seeds:

Table 45: Structural differences of tasks evolved under different random seeds (before/after evolution). Feasible configuration counts refer to solver search-space estimates, not the number of valid solutions.

| Task (Different Seeds) | Avg. Graph Diameter | Dependency Chain Length | Feasible Configurations |
|---|---|---|---|
| First Evolution – Task 1 | 1.80/0.67 | 3/1 | 1/1 |
| First Evolution – Task 2 | 1.00/1.00 | 1/4 | 1/1 |
| Second Evolution – Task 3 | 1.00/2.00 | 1/2 | 1/1 |
| Second Evolution – Task 4 | 5.00/1.67 | 8/8 | $1/2 \times 10^{13}$ |
| Third Evolution – Task 5 | 1.80/3.50 | 3/5 | $1/4 \times 10^{18}$ |
| Third Evolution – Task 6 | 1.75/0.83 | 2/4 | $7 \times 10^{21}/6 \times 10^{139}$ |

Differences Before and After Evolution Under the Same Seed:

- **Average constraint-graph diameter**: differences of **0–94%** before vs. after evolution under the same seed, with some tasks exhibiting substantially longer-range global dependencies after evolution.

- **Dependency-chain length distribution**: the number of long chains differs by **1–4×**, indicating that some evolved tasks develop markedly deeper reasoning chains.

- **Feasible configuration size**: for roughly **50%** of evolved tasks, the search-space scale differs by **more than two orders of magnitude** across seeds, suggesting essential changes in the feasible configuration space after evolution.

Patterns of Differences Across Seeds After Evolution:

- **Constraint-graph diameter**: varies by **17–422%** across runs, indicating that some runs emphasize highly local constraints, whereas others induce long-range coupling structures.

- **Dependency-chain length**: differs by **2–8×**, showing that some evolutions produce substantially deeper reasoning chains while others yield flatter rule compositions.

- **Feasible configuration space**: for **50%** of the tasks, search-space estimates differ by **more than two orders of magnitude**, implying fundamentally different search problems even though all final tasks maintain a unique valid solution.

These results demonstrate that LogicEvolve does *not* merely rewrite tasks in a deterministic way. Instead, different seeds lead the system to explore *structurally distinct regions* of the task

space—producing tasks with different constraint-graph geometries, reasoning depths, and effective search-space sizes. This seed-driven diversity provides strong evidence of LogicEvolve's genuine evolutionary capacity.

### G.5.2 QUALITATIVE ANALYSIS

In addition to the quantitative statistics reported above, we further conduct a qualitative comparison of the tasks before and after evolution. Our analysis focuses on two dimensions: (1) the **diversity of logical structures** produced by evolution, and (2) the **variation in reasoning depth** induced by hyperparameter expansion. Representative examples are summarized below.

(1) Structural diversity: from local constraints to global hypergraph structures

LogicEvolve expands the original 10 CLUB task types into the 100-task ExCLUB family (here we only sampled 2–3 lower-difficulty variants for demonstration). The following two cases illustrate how evolution leads to structural transformations that go far beyond textual rewriting:

- **Task 1: Maze → Arrow Maze** In CLUB, the maze task requires computing the shortest path through a grid of passable and blocked cells. In ExCLUB, the arrow maze requires filling every grid cell with a directional arrow such that each numeric clue equals the total length of all continuous arrow rays emitted from its adjacent cells. This constitutes a shift from *local path search* to *global arrow-configuration inference with numeric constraints*. The dependency graph transitions from a local adjacency structure to a long-range coupled hypergraph, and the reasoning procedure shifts from single-path expansion to global constraint propagation and backtracking.
- **Task 2: Sudoku → Skyscraper** Sudoku is governed by row/column/box all-different constraints. The skyscraper puzzle adds directional visibility clues on top of a Latin square. Each clue constrains an entire row/column permutation based on how many "buildings" are visible from a given direction. As a result, the constraint graph evolves from local symmetric constraints to a hypergraph combining all-different relations with directional visibility constraints. The reasoning process moves from local candidate elimination to global pruning over complete row/column permutations.

These examples demonstrate that evolution introduces **substantively new structural motifs**—global coupling, multi-view constraints, and hypergraph dependencies—rather than merely rephrasing existing CLUB tasks.

We further compare model failure cases before and after evolution, and observe that the evolved benchmark indeed exposes new error patterns that are rarely triggered in previous versions. For example, when Sudoku is extended into a variant with additional constraints, grok-4 continues to rely on the traditional row–column–box framework and fails to integrate the new constraints, producing outputs that appear valid under the old rules but immediately fail under the updated conditions. Similarly, when the Maze task is expanded into an Arrow Maze with directional indicators, deepseek-r1 continues to apply the stepwise path-expansion strategy used in CLUB and does not exploit the global consistency implied by the arrow directions. As a result, once a local deviation occurs in the long reasoning chain, the model repeatedly propagates the error and ultimately produces answers that conflict with the global arrow structure.

Overall, these cases demonstrate that CLUB's evolution introduces genuinely new structural variations and reasoning mechanisms, enabling the benchmark to reveal failure modes that are difficult to trigger in the original version.

(2) Reasoning-depth diversity: from linear chains to conditional graphs and high-dimensional state transitions

We next compare CLUB1.0 with CLUB2.0, which is generated through a one-step hyperparameter expansion. The following cases highlight how small surface-level changes can cause large structural shifts:

- **Task 1: String Operations (10 ops) → String Operations (11 ops)** The new operation introduces a conditional trigger, transforming a previously linear operation chain into a conditional operation graph. Inferring the operation sequence thus becomes a multi-level

causal reasoning task rather than simple permutation inference, substantially increasing search-space structural complexity and reasoning depth.

- **Task 2: Card Game (4 suits)** → **Card Game (5 suits)** Adding a new suit expands the dimensionality of the game state. The central pile transitions from a finite-state machine with fixed branching to a higher-dimensional state-transition graph. The player's collection path and pile evolution now involve larger branching factors and multi-path trajectories, transforming the task from linear matching simulation into high-dimensional state analysis.

These examples illustrate how seemingly minor hyperparameter changes can yield *explosive structural effects*, enlarging the search space and deepening the reasoning complexity in a non-linear manner.

We further analyze model failure cases before and after evolution and find that increased reasoning depth indeed introduces new reasoning loads, thereby revealing failure modes that are difficult to trigger in the original version. For example, after CLUB2.0's string task introduces a new "conditional slicing" operator, gpt-5-thinking correctly interprets the operator semantics but frequently exhibits index shifts or type confusion along multi-step dependency chains—often producing a list instead of the required final string—indicating local state drift along deeper conditional paths. Similarly, after the maze and card tasks in CLUB2.0 are scaled to significantly larger sizes, gemini-2.5-pro frequently forgets intermediate states within long reasoning chains, causing subsequent steps to lose consistency with earlier ones. This leads to classic "reasoning-chain breakage" and ultimately incorrect answers.

Overall, such errors are relatively rare in CLUB1.0 but become substantially more frequent in the evolved task settings, confirming that the evolution mechanism introduces genuinely new challenges along both structural and depth dimensions.

(3) Summary

Qualitatively, the evolution process induces rich diversity in both **logical structure** (local → global, graphs → hypergraphs) and **reasoning depth** (linear chains → conditional graphs, finite-state → high-dimensional state transitions). Combined with our quantitative analyses (constraint-graph diameters, dependency-chain distributions, feasible configuration scales), these results show that LogicEvolve produces **genuinely new reasoning tasks** with substantive differences in underlying logical complexity and reasoning patterns, rather than superficial task modifications.

At the same time, the evolution of the benchmark systematically exposes latent failure modes that are difficult to trigger in the original version, thereby providing new evidence for understanding the models' true weaknesses and informing more targeted directions for subsequent improvement.

### G.6 CONVERGENCE AND DIVERGENCE IN EVOLUTION

This section further explains the "convergence–divergence" behavior of LogicEvolve across multiple iterations and independent runs. By design, LogicEvolve exhibits two complementary properties: stable convergence within each individual task, and structural divergence across the family of generated tasks. This two-level behavior enables the framework to support both verifiable task construction and broad task-space exploration.

Under fixed task metadata, the core LogicEvolve loop (metadata verification, instance generation, solver correction) consistently converges to a stable task specification within a finite number of iterations. As shown in Table 46, each task in CLUB and ExCLUB reliably produces solvable instances within limited rounds, regardless of the underlying base model or sampling randomness. The solver returns correct solutions, while metadata fields gradually align and consolidate into a coherent structure. This within-task convergence guarantees the verifiability and reliability of LogicEvolve when the task semantics are fixed.

In contrast, when LogicEvolve is used for task rewriting or for generating entirely new task types, its objective is no longer convergence, but exploration of a broader structural space. Different random seeds, structural-transformation paths, and hyperparameter settings drive the generator and meta-rules into distinct structural regions, producing diverse constraint graphs, dependency chains, and solution spaces. These cross-run structural differences are not noise, but an intended mechanism for expanding the task space and increasing the coverage of logical-reasoning patterns.

Table 46: Statistics of LogicEvolve's convergence iterations under fixed-task settings (counting the total rounds of metadata verification, instance generation, and solver correction for each task). The upper section reports results on CLUB (10 tasks across 5 independent runs), and the lower section reports results on ExCLUB (100 tasks under two base-model settings).

| CLUB (10 tasks × 5 runs) | Minimum | Maximum | Average |
|---|---|---|---|
| 1st run | 0 | 43 | 11.63 |
| 2nd run | 1 | 23 | 8.55 |
| 3rd run | 1 | 12 | 6.44 |
| 4th run | 1 | 11 | 5.67 |
| 5th run | 0 | 19 | 8.22 |
| **ExCLUB (100 tasks × two base models)** | **Minimum** | **Maximum** | **Average** |
| 1st run | 2 | 52 | 22.04 |
| 2nd run | 0 | 108 | 53.02 |

It is therefore necessary to distinguish the level at which iteration operates when discussing convergence. Increasing iteration rounds within the same task accelerates internal stabilization and convergence. However, increasing iterations during task generation or structural transformation leads only to broader structural exploration rather than convergence to a single point. The former resembles solving a fixed task, where convergence is naturally expected; the latter resembles inventing a family of tasks, where convergence to a unique structure is inherently impossible.

Notably, although different runs yield task versions with substantial structural variation, they produce highly consistent model-ranking results when evaluating LLM reasoning ability: the relative performance ordering of models remains stable across versions. Furthermore, structural complementarity across versions helps reveal model-specific strengths and weaknesses under different reasoning structures, enabling more comprehensive evaluation and avoiding overfitting to a single structural pattern. Hence, an evolved benchmark does not need to converge to a unique version—multiple independently evolved task families can serve as valid and complementary evaluation resources.

Overall, through its dual mechanism of within-task convergence and across-task divergence, LogicEvolve ensures task verifiability while continually expanding the structural space of logical tasks, providing a natural path toward multi-perspective, broad-coverage, and high-stability logical-reasoning benchmarks.

### G.7 HANDLING ULTRA-COMPLEX TASKS GENERATED DURING EVOLUTION

Certain evolutionary paths may yield task instances whose complexity exceeds that of typical logical puzzles, occasionally producing cases unsolvable by existing CSP solvers, symbolic solvers, or large language models. This subsection clarifies the scope of applicability of our study and the mechanisms we employ to guarantee task correctness.

First, all quantitative experiments in this paper—including CLUB, ExCLUB, and the evolutionary examples—are strictly restricted to a verifiable complexity regime: the portion of the task space for which correctness can be jointly ensured by programmatic uniqueness verification, multi-solver cross-checking, and stratified human auditing. Every instance used in evaluation has passed this verification pipeline. Tasks whose correctness cannot be validated are excluded from the main experiments and the final benchmark release. The limited human intervention present in the current version is restricted to fixing generator bugs and cleaning samples during benchmark construction, and all such actions occur prior to benchmark freeze.

Within the verifiable regime, LogicEvolve employs a multi-layered mechanism to ensure logical consistency and uniqueness: (i) solvers based on procedural code, enumeration, or CSP/SAT/SMT tools are applied to verify the existence and uniqueness of solutions and detect contradictions; (ii) instances are cross-validated using heterogeneous solver types, and only those where all intermediate states and final solutions agree are accepted; (iii) stratified sampling is performed for manual inspection of meta-rule fidelity and ground-truth consistency, with identified issues fed back into the

system for automatic repair. Together, these processes ensure that all evaluated tasks exhibit verifiable correctness and structural stability.

For tasks that fall outside this verifiable boundary, we deliberately refrain from drawing experimental conclusions and treat such instances as directions for future work. During evolution, the framework actively detects unsolvable cases, multi-solution cases, or search instabilities; upon detection, it automatically backtracks to safer hyperparameter regions or adjusts the relevant meta-rules to keep generated tasks within solvable and verifiable bounds.

Looking ahead, we plan to further strengthen the verification pipeline by increasing the formal precision of meta-rules, incorporating more powerful programmatic verification tools, expanding the cross-solver validation ecosystem, and integrating automatic checks for task stability and reverse consistency. For the rare cases of extremely high structural complexity, targeted manual review will be retained when necessary, but such tasks will not be included in the benchmark scoring protocol.

In summary, all experimental conclusions presented in this paper are derived exclusively from the subset of evolution-generated tasks that remain within the verifiable complexity regime. Tasks exceeding this boundary represent promising avenues for future exploration but fall outside the scope of the current evaluation.

