# OpenReview forum: "LogicEvolve: Advancing Logical Reasoning Toward Self-Evolution"
_ICLR.cc/2026/Conference — ICLR 2026 Conference Withdrawn Submission_

### Official Review · Reviewer_wJvd · 2025-10-24

**Soundness:** 2
**Presentation:** 2
**Contribution:** 2
**Rating:** 2
**Confidence:** 3

**Summary:**

This paper presents a multi-agent framework for developing benchmark tasks for logical reasoning. They generate examples from a wide range of tasks including many traditional types of logic puzzles as well as a few new ones. The benchmark a range of top models and find that performance is inconsistent across tasks, and reasoning depth remains an issue.

**Strengths:**

The primary original contribution of this paper is the agentic system for generating new task items. This system is useful in being able to generate new benchmark tasks with controls for both problem difficulty and type. The results are comprehensive, assessing a wide range of models and breaking down the scores across several dimensions. These results further support the understanding that LLMs are brittle reasoning agents with challenges in long context settings.

**Weaknesses:**

I am concerned about the soundness of the results in this paper.

Specifically:

- The "extensible_zebra_logic" task appears both as "extensible" and "extended" (pg 18) but is never actually defined, even in the lengthy appendices, despite being a task which is specifically called out as an example of a challenging novel task.
- The framework clearly requires human intervention in various places, but the specifics are not described. Appendices C 7-8 only make it clear that interventions are necessary, not how often they occur. This also contradicts the claims about being a "level 4" automation.
- The human puzzle creation baseline is extremely vague, described in four lines with no details about the number of humans, their expertise in creating puzzles, or whether they used any tools.
- Humans seem like the wrong baseline for puzzle creation anyways, given that just about every puzzle in this benchmark is a well-known puzzle with clear procedural methods for creating new puzzles. I strongly suspect that code for creating these puzzles already exists online in most cases.
- This task creation system should probably be described as adversarial. Since tasks are removed if "certain models" (not specified) are scoring > 90%, the benchmark is selecting for examples that the models do poorly on ex post.

Crucially:
- I need to see evidence that this framework would generalize to new puzzle types outside of those which are already very common on the internet. The "evolve" claims of the framework are not substantiated by showing ability to create common puzzles.
- Moreover, if there are persistent issues in correctly generating puzzles/metadata (as shown in the need for human intervention) then I would also want evidence that the puzzles would continue to be correct if the system were asked for superhuman level puzzles.

Finally, I dislike the claim of being "inspired by the process of human evolution". It is vague and the iterative code generation approach with a verifier is not actually related to evolutionary algorithms or fitness-based selection approaches generally. Using unsubstantiated biological metaphors can mislead readers about the actual results of the paper.

**Questions:**

1. What exactly was the scope of human intervention necessary for the agents to complete the task?
2. What would be your expectations of a human score on this benchmark?
3. Why make comparisons to computational complexity? Many NP-complete or higher problems (e.g. set cover) are effectively approximated by a greedy algorithm, which is not logically intensive to use.
4. What happens if you ask the system to invent a new, more difficult, logic problem type?

---

> ### Author Response · Authors · 2025-11-21
> **Summary & Response to reviewer wJvd (Weakness 1: Issues with extensible_zebra Task Naming and Unclear Definition, part1/2)**
>
> # **Summary**
>
> Thank you for your rigorous and insightful identification of weaknesses and concerns.
>
> We have added corresponding experiments, clarified formulations, and completed systematic revisions in both the main text and the appendix.
>
> A brief summary is as follows:
>
> 1. **Inconsistent naming** (W1) — All naming has been unified to extensible_zebra. This issue was purely editorial and does not affect any experimental results or rankings. (Clear definitions in Appendix B and E.2)
>
> 2. **Human intervention** (W2 & Q1) — We corrected the definitions of Level-4/5 (Level-4 = optional human supervision; Level-5 = fully automatic). Appendix C.10 now includes a complete table specifying intervention types, frequencies, and decision criteria.
>
> 3. **Necessity and role of human-crafted puzzle baselines** (W3 & W4) — We have elaborated on the details in the main text and Appendix D.3. The human puzzle creation baseline serves as a reference for engineering costs, not a gold standard; future work will explore fully programmatic benchmarks.
>
> 4. **Adversarial task selection** (W5) — The previous “90% triggering rule” has been removed. We now adopt a LiveCodeBench-style versioning and time-window mechanism, eliminating any model-performance-based filtering to ensure non-adversarial and fully reproducible task sets.
>
> 5. **Novel puzzle types** (W6 & Q4) — We narrowed the scope of “evolution” in the main text. Appendix G includes examples of newly constructed tasks that extend beyond existing puzzle families, positioning them as a future direction, not a central claim of this paper.
>
> 6. **Correctness of superhuman tasks** (W7) — We clarified the intended scope: all conclusions in this paper apply only to verifiable-complexity tasks. Appendix G.7 adds exploratory designs for formal verification, multi-solver cross-validation, and stability checks.
>
> 7. **Potentially misleading biological analogy** (W8) — All “human evolution inspiration” wording has been removed and replaced with the more precise notion of “experience-based self-improvement,” consistent with existing self-evolution literature.
>
> 8. **Human baseline** (Q2) — New evaluations in the main text and Appendix D.3.2 show that humans score around 9/100 under CLUB’s programmatic format and strict stepwise-consistency requirements, indicating that the low human score reflects the format’s machine-oriented design rather than a lack of human reasoning ability.
>
> 9. **Justification for using computational complexity** (Q3) — We clarify that complexity is not used to define difficulty; instead, it serves as an auxiliary characterization of task-space geometry, constraint structure, and evolution direction. Appendix B.2 has been revised to avoid potential misinterpretation.
>
> You can refer to the reply below for specific details.
>
> ---
>
> ## **Response to Weakness 1: Issues with extensible_zebra Task Naming and Unclear Definition**
>
> Thank you very much for the specific comments on the unclear naming and definition of the *extensible_zebra* task.
>
> We sincerely apologize for the omissions in the original manuscript and have made systematic, verifiable corrections in the revised version.
>
> Below we explain separately: **unified naming, formal definitions, example supplements, and impact on experimental results**.
>
> ---
>
> ### **1. Naming Issue: Unified and Explicitly Corrected in the Text**
>
> The original manuscript indeed mixed uses of *extensible_zebra*, *extensible_zebra_logic*, and *extended_zebra*, and also mixed uses of *traditional_zebra_logic* / *Zebra puzzle* for the classic version. In the revised version, we made the following unified treatment:
>
> * **Unified Naming:**
>   * **All extended version tasks unified as:** **extensible_zebra**;
>   * **All classic version tasks unified as:** **traditional_zebra**.
> * **Revision Locations:**
>   * In the main text: Section 4 CLUB task list Figure 4, Section 5 Table 2 / Figure 6 / Figure 7 / main text task labels, all changed to **traditional_zebra** and **extensible_zebra**;
>   * In appendices: Places in original B.2, E.1/E.2 involving mixed naming like "extended / logic / puzzle" have all been unified to the above two names.
> * The open-source work itself is called ZebraLogic, and only this description maintains the original description.
>
> > In other words: **The revised manuscript only retains two clear task names in CLUB: traditional_zebra and extensible_zebra, with no other mixed usages appearing.**
>
> ---
>
> ### **2. Formal Definition: Clearly Distinguishing Two Task Types in Main Text and Appendices**
>
> In the revised version, we completely separated the metadata definitions of **traditional_zebra** and **extensible_zebra**, and gave their differences in unified hyperparameter form:
>
> * **Object and Attribute Dimensions:**
>   * **traditional_zebra**: Supports object count  $N \in [2, 7]$ , attribute count $K \in [2, 7]$ ;
>   * **extensible_zebra**: Supports object count $N \in [1, 10]$ , attribute count $K \in [2, 11]$ .

---

> ### Author Response · Authors · 2025-11-21
> **Response to reviewer wJvd (Weakness 1: Issues with extensible_zebra Task Naming and Unclear Definition, part2/2)**
>
> * **Differences in Rule Types (Logical Expression Templates):**
>   * traditional_zebra only includes 7 classic constraint templates:
>     * Equality/inequality: $A = B$,  $A \neq B$
>     * Positional relationships: leftmost / rightmost / middle / left of / right of / adjacent left-right, etc.
>     * Odd/even positions: at odd / even positions
>     * At some position on one side: e.g., somewhere to the left of B
>   * **extensible_zebra** inherits the above 7 template types and **additionally adds higher-order combination and multi-body constraints**, for example:
>     * **Binary conjunction/disjunction constraints**: A and B satisfy the same positional relationship, A or B is at the leftmost end;
>     * **Ternary "NAND" constraints**: At least two of A, B, C satisfy some relationship, but not all satisfy it;
>     * **Ternary "NOR" constraints**: No pair among A, B, C satisfies some adjacent relationship, etc.;
>     * And several higher-order "exactly-one / at-least-k / at-most-k" constraint templates.
>
> In the revised version, we list the differences in value ranges for $N, K, |\mathcal{R}|$ **(rule family size)** in a table in Appendix B.2 "Task Complexity Analysis", and provide **two completely separate metadata definitions** in **Appendix E.1 / E.2** **(base assumptions + instance examples)**.
>
> ---
>
> ### **3. Supplementing a Complete Example: Readable, Runnable, Demonstrating the "Extensible" New Structure**
>
> To avoid giving readers the impression of "only seeing names, not seeing what the task looks like," we added a **complete extensible_zebra example** in Appendix E.2 of the revised manuscript, including:
>
> 1. Natural language problem statement (rule list)
> 2. Formal constraint problem format
> 3. Unique solution (given in table form)
> 4. Brief reasoning explanation
>
> In the rebuttal, we provide an outline of one simplified example, showing that "extensibility" is indeed implemented at three levels: **object count / attribute count / constraint structure**, not just name changes:
>
> > **Example (simplified extensible_zebra, see Appendix E.2 for a more complete example):**
>
> There are 4 houses, positioned from left to right as 1～4. Each house has three types of attributes: nationality (A/B/C/D), drink (tea/coffee/milk/water), pet (cat/dog/bird/fish). Some rules are as follows:
>
> 1. The person living in the **red house** keeps a **cat**.
>
> 2.  **Person A** lives somewhere to the left of **Person B**, but not adjacent.
>
> 3.  **Person C** is neither at the leftmost nor rightmost position.
>
> 4.  Among the residents corresponding to **"tea" and "coffee"**, **exactly one person** lives at an endpoint position (1 or 4).
>
> 5.  Among {cat, dog, bird}, **exactly two pet types** appear in houses at even positions.
>
> 6. "The person who drinks milk" and "the person who keeps a bird" do not live in adjacent houses.
>
> 7.  ……(omitting several higher-order combination constraints)
>
>
> In this example:
>
> * Rules 2, 4, 5 explicitly use **multi-object / multi-attribute combination constraints (exactly-one / exactly-two)**;
> * A constraint in the dependency graph often connects multiple objects and multiple attributes, forming **higher-order hyperedges**, no longer the simple "one line connecting two points" structure in classic **traditional_zebra**;
>
> Through this specific example, readers can **intuitively see**:
>
> > extensible_zebra is not just "changing a few numbers" on the classic zebra puzzle, but **substantially changes the underlying logical structure and reasoning patterns** by expanding object count, attribute count, and introducing multi-body, higher-order combination constraints.
>
> ---
>
> ### **4. Impact of This Issue on Experimental Results: An Editorial Issue, Does Not Affect Scoring and Rankings**
>
> Finally, we also fully understand the reviewer's possible concern: **Does naming confusion mean there is also inconsistency in implementation, thus affecting the reliability of experimental results?**
>
> On this point, we clearly commit here:
>
> * All results in CLUB and ExCLUB regarding this task (including Figure 2, Figure 7, Table 3, etc.) have been automatically aggregated through this unified identifier;
> * The problem in the original manuscript was limited to **textual naming inconsistency during paper writing** and does not affect:
>   * The actual definition, generation, and verification process of tasks;
>   * Score statistics and benchmark rankings of models on this task.
>
> ---
>
> ## **Summary**
>
> > We not only unified and corrected the naming of *extensible_zebra*, but also supplemented its **formal difference definitions** with **traditional_zebra** in the revised manuscript, added a **complete example problem + unique solution**, and ensured that the implementation side always uses unified identifiers and experimental results are unaffected. We hope these modifications can eliminate your reasonable concerns about this task's definition and reliability, and thank you again for pointing out this issue so carefully and accurately.

---

> ### Author Response · Authors · 2025-11-21
> **Response to reviewer wJvd (Weakness 2 & Question 1: Questions on Human Intervention Scope, Frequency, and Automation Level Definition, part1/2)**
>
> ## **Response to Weakness 2 & Question 1: Questions on Human Intervention Scope, Frequency, and Automation Level Definition**
>
> Thank you very much for pointing out that the description of human intervention is not specific enough and is inconsistent with our original "Level-4 automation" statement.
>
> We completely agree with this point and have made **substantive modifications** to relevant paragraphs in the revised version. Below we answer separately:
>
> ---
>
> ### **1. Revising the "Level-4 Automation" Statement**
>
> As the reviewer pointed out, the current version of LogicEvolve **still relies on a small amount of human intervention at specific stages**.
>
> Therefore, we have in the revised version:
>
> * **Changed the "Level-4 automation" in the original Table 1 to a more moderate statement that matches the current situation:**
>   **→"largely-automated synthesis with optional human oversight"**
> * And added in Appendix C.10 of the paper:
>   * Definition and scope of intervention
>   * At which stages it is triggered
>   * Operational definition of automation level (covering "time cost" and "scope of impact on results")
>
> > In other words, we no longer claim "no human needed," but clearly state that LogicEvolve currently has an automated process with optional human oversight.
>
> ---
>
> ### **2. New Addition: "Scope and Trigger Conditions of Human Intervention"**
>
> We clearly list in Section 3.2 of the main text and Appendix C.10 that human intervention only occurs in the following three situations:
>
> #### **(a) Failure Intervention (failure-handling)**
>
> Triggered when the generator or solver exhibits the following situations:
>
> * Long-term stagnation (model repeatedly gives unchanged answers)
> * Unable to pass verification after reaching maximum iteration steps
> * Rules are obviously self-contradictory (captured by verifier)
>
> The goal of this type of intervention is **to allow the experiment to continue running**, not to modify task content.
>
> #### **(b) Sampling Intervention (validation sampling)**
>
> Human verification of:
>
> * Whether metadata is logically self-consistent
> * Whether the generator correctly generates solvable tasks
> * Whether the solver can stably solve
>
> This type of intervention is used to **improve benchmark quality**, but does not participate in specific task design.
>
> #### **(c) Creative Intervention (preference injection)**
>
> Only occurs when human preferences need to be added, such as:
>
> * Wanting certain task types to be more educational
> * Wanting to highlight specific reasoning patterns
>
> ---
>
> ### **3. New Addition: Statistical Method for Intervention Frequency**
>
> We clearly define the statistical unit in **operational terms** in Appendix C.10:
>
> * For ExCLUB's **100 tasks × 5 independent runs (different random seeds)**
> * In each run, an "intervention event" refers to:
>   * Humans actually modifying some metadata/generation code/solving code;
>   * Or making a clear judgment on some type of error (e.g., confirming rule contradiction).
> * Simply "opening logs to view" is not counted as an intervention event.
>
> Therefore, the total sample size is: **500 complete task generation processes**.
>
> Each data item in the table below represents: among these 500 processes, how many processes have experienced this type of intervention (not the proportion of all intervention events).

---

> ### Author Response · Authors · 2025-11-21
> **Response to reviewer wJvd (Weakness 2 & Question 1: Questions on Human Intervention Scope, Frequency, and Automation Level Definition, part2/2)**
>
> ---
>
> ### **4. Failure Intervention Frequency in This Experiment (500 Processes) (Already Added in Revised Version)**
>
> #### **Intervention Frequency (Proportion of Occurrence in 500 Complete Processes)**
>
> | **Intervention Type**                 | **Metadata** | **Generator** | **Solver** | **Evaluator** | **Data** | **Description**                                                                          |
> | ------------------------------------------- | ------------------ | ------------------- | ---------------- | ------------------- | -------------- | ---------------------------------------------------------------------------------------------- |
> | **Rule Contradiction Exists**         | 0%                 | –                  | –               | –                  | –             | Not observed in this experiment (rules already verified).                                      |
> | **Instance Error Exists**             | **19.4%**    | –                  | –               | –                  | –             | Solution inconsistent with rules, mostly caused by LLM misunderstanding.                       |
> | **Generator Stagnation**              | –                 | **13.6%**     | –               | –                  | –             | Mostly due to output not following structured format.                                          |
> | **Generator Iteration Not Converged** | –                 | **5.8%**      | –               | –                  | –             | Insufficient CoT capability.                                                                   |
> | **Solver Stagnation**                 | –                 | –                  | **23.3%**  | –                  | –             | Mostly due to code exceptions/misunderstanding rules.                                          |
> | **Solver Iteration Not Converged**    | –                 | –                  | **14.6%**  | –                  | –             | Mostly due to insufficient complex reasoning capability.                                       |
> | **Data Leakage**                      | –                 | –                  | –               | –                  | 0%             | Not observed in this experiment (framework engineering implementation ensures no duplication). |
> | **Evaluator Inconsistency**           | –                 | –                  | –               | **0%**        | –             | Not observed in this experiment (evaluator directly synthesized from rules).                   |
>
> We explain in the revised version:
>
> * All the above interventions are **one-time injections** and are automatically absorbed in the next iteration;
> * **These interventions combined** **average < 0.8 times per task (500 processes / total intervention count)**;
> * Interventions **do not participate in task generation content creativity**, only ensuring process reliability.
>
> ---
>
> ### **5. Final Clarification on Automation Level (Newly Added in Revised Version)**
>
> We add the following definition in Appendix C.10 of the revised version:
>
> > **By automation level we refer to:**
>
>  (i) the extent to which task content is generated without human changes;
>
>  (ii) the extent to which the pipeline executes without human interruption;
>
>  and (iii) the absence of any human-written rules, instances, or templates beyond the initial metadata
>
> Under this definition, the automation results of this experiment are:
>
> * **Task content automatically generated by LLM: 100%**
> * **Process runtime automatically executed: > 98%**
> * **Human intervention at task design level: 0%**
>
> Therefore, we describe LogicEvolve's automation level in the revised version as:
>
> > **"a largely-automated task-evolution pipeline with optional human oversight for failure recovery and sample validation"**
>
> This both avoids over-promising and clearly states the framework's actual capabilities and boundaries.
>
> ---
>
> ## **Summary**
>
> > We acknowledge that the original text insufficiently described human intervention, and in the revised version we have supplemented the **scope, frequency, definition, statistical method, and degree of impact** of interventions, revised the "Level-4 automation" statement, and provided quantitative evidence based on 500 complete processes. We hope these modifications can eliminate your reasonable concerns about automation level and human dependence.

---

> ### Author Response · Authors · 2025-11-21
> **Response to reviewer wJvd (Weakness 3 & Weakness 4: Questions on Human Puzzle Creation Baseline and Programmatic Generation Methods, part1/2)**
>
> ## **Response to Weakness 3 & Weakness 4: Questions on Human Puzzle Creation Baseline and Programmatic Generation Methods**
>
> Thank you very much for the questions raised regarding our human baseline design and programmatic generation methods.
>
> We agree that the original text's description of the human baseline was too brief and did not fully explain the baseline's scope of application and its relationship with existing programmatic generators.
>
> We have made the following substantive supplements and clarifications in the revised version.
>
> ---
>
> ### **1. Specific Settings of Human Puzzle Creation Baseline: Participants, Tools, Task Volume, and Evaluation Dimensions**
>
> In the revised version, we replaced the original four-line description with a more **complete description** (with more details in Appendix D.3), clearly stating:
>
> * **Number of Participants and Background**
>   * A total of **5 participants**, all **master's or doctoral-level software engineers** with computer-related professional backgrounds, familiar with algorithm design and code implementation in daily work.
>   * They have the ability to understand logical puzzle structures and write generator and solver code.
> * **Time Budget and Task Volume**
>   * Each participant was allocated **2 days**, completing 1 task type per day, corresponding to a "sub-task family" in our framework (e.g., traditional_zebra, Knights and Knaves, etc.), totaling **10 task types**.
>   * For each task type, they need to complete a full set of:
>     * Task metadata (metadata)
>     * Generator (generator)
>     * Solver (solver)
>     * Evaluator (evaluator)
>       Four module implementations, strictly aligned with LogicEvolve's output module structure.
> * **Allowed Auxiliary Tools**
>   * Participants **are explicitly allowed to use**: LLMs (such as ChatGPT), GitHub, public websites, and other resources, to consult or reference existing programmatic puzzle generator implementations (e.g., traditional_zebra related repositories, Knights and Knaves related implementations, etc.).
>   * But they are required to **independently refactor/rewrite** these codes so that:
>     1. They meet our unified interface specifications (unified metadata format, generator/solver input-output specifications);
>     2. They do not directly reuse any specific problem instances from existing datasets;
>     3. They are split into metadata / generator / solver / evaluator four parts according to LogicEvolve's module boundaries.
> * **Baseline Evaluation Dimensions**
>   * For this "human baseline," we **do not treat it as a performance comparison of "who generates better problems"**, but mainly use it to measure:
>     1. **Under realistic engineering conditions, the cost for human engineers to transform existing programmatic puzzle generators/code into "maintainable, evolvable, uniformly callable" modular pipelines;**
>     2. Compared with LogicEvolve's automatic generation pipeline, **the number of task types and coverage scope that can be completed under the same time budget**.
>   * In the revised version, we also clearly emphasize: **This human baseline is only a "comparative engineering cost case," not a statistically meaningful performance upper bound or estimate of "human puzzle creation capability."**
>
> ---
>
> ### **2. What Problem the Human Baseline Wants to Answer, and What It Does Not Want to Answer**
>
> We further clarify in the revised version:
>
> * The design purpose of the human baseline **is not** to answer "whether humans are better at creating problems than automatic frameworks," nor to prove "humans are smarter/weaker than LLMs."
> * What it really wants to answer is an **engineering and maintainability problem**:
>
>   * Given that there are already many programmatic puzzle generators, example code, and LLM tools, if an experienced engineering team wants to build a "puzzle synthesis pipeline similar in structure to LogicEvolve, maintainable and evolvable long-term," how much human effort and time do they need to invest to manually construct **10 complete modules (metadata + generator + solver + evaluator) covering multiple task types**?
> * Through this comparison, we hope to illustrate:
>
>   * **Even with extensive tool support, integrating "isolated programmatic puzzle generators/code repositories" into "unified interface, unified metadata management, evolvable task families" still requires high one-time engineering costs from human engineers;**
>   * LogicEvolve, under the same interface specifications, automatically produces these modules through multi-agent iteration, providing a more scalable path for large-scale task type expansion.
>
> Therefore, in the main text’s experimental section and Appendix D.3, we deliberately downplay the statistical significance of the "human baseline," positioning it as:
>
> > **"manual engineering baseline demonstrating the human effort required to build a LogicEvolve-compatible pipeline using existing tools and code."**
>
> Rather than a "gold standard for puzzle quality/difficulty."

---

> > ### Author Response · Authors · 2025-11-21
> > **Response to reviewer wJvd (Weakness 3 & Weakness 4: Questions on Human Puzzle Creation Baseline and Programmatic Generation Methods, part2/2)**
> >
> > ---
> >
> > ### **3. Why Does the Current Version Not Directly Use "Existing Programmatic Generators" as a Complete Baseline?**
> >
> > The reviewer points out: Since many classic puzzles already have mature programmatic generation methods, why not directly use these codes as a comparison baseline? This is a very reasonable question, and we clearly explain the reasons in the revised version and more candidly present current limitations and follow-up plans:
> >
> > 1. **Module Boundaries and Interface Inconsistency, Difficult to Directly Align with LogicEvolve's Four-Module Structure**
> >    * The vast majority of existing programmatic generators (e.g., traditional Zebra, Sudoku, some logic puzzles):
> >      * Hard-code "task rules + generation logic + solving logic" in the same script;
> >      * Do not explicitly distinguish metadata / generator / solver / evaluator;
> >      * Some also rely on interactive parameter configuration or manual tuning to generate problems of appropriate difficulty.
> >    * This makes it difficult to achieve "fair comparison under the same interface" if we directly use these codes as baselines:
> >      * On one hand, they cannot be directly used in our framework's automatic evolution process (e.g., task difficulty control, evolution log management, error pattern recording);
> >      * On the other hand, we cannot evaluate their scalability in scenarios of "continuously adding new task families, long-term maintenance."
> > 2. **Human Baseline Actually Already "Indirectly Tests" the Transferability of Programmatic Generators**
> >    * In the human baseline, we **allow engineers to actively seek and utilize these programmatic generators** and transform them into LogicEvolve-compatible four-module structures.
> >    * In other words, we are actually testing a question: "Assuming there is already some programmatic generator online, how much engineering effort do humans need to migrate it to LogicEvolve's unified interface?"
> >    * This is closer to our research question than directly running original scripts: **How to manage and expand multiple task families under a "unified evolution framework."**
> > 3. **We Candidly Acknowledge in the Revised Version: Large-Scale "Script Generator Baseline" is Part of Follow-Up Work**
> >    * To avoid making people think we "didn't do it because it was troublesome," we clearly write in the revised version:
> >      * Current work mainly focuses on LogicEvolve's incremental value compared to "manual engineering pipelines";
> >      * **Systematic "script generator baseline" comparison (i.e., using only traditional programmatic generation methods without introducing LLMs to construct evolvable benchmarks) will be an important task in extended versions**;
> >      * We have already built prototypes internally, encapsulating some public generators as black-box modules, and plan to systematically report results in follow-up versions.
> >
> > ---
> >
> > ### **4. Fairness and Leakage Issues Related to "Code Available Online"**
> >
> > The reviewer is concerned: Since there is already code online for generating these puzzles, is CLUB just "repackaging and reusing existing code/problems"? We specifically add a brief but crucial clarification statement in the revised version:
> >
> > * All problem instances used for evaluation in CLUB and ExCLUB **are automatically generated by LogicEvolve**;
> > * Programmatic generators and public repositories referenced in the human baseline **are only used to help engineers understand task structures and algorithm design ideas**, and are rewritten as new generator/solver code;
> > * **We** do not directly reuse any specific problem instances from existing datasets, nor do we simply package existing scripts as CLUB's data source;
> > * CLUB's metadata, generators, solvers, and evaluators are all built from scratch according to unified LogicEvolve specifications to avoid data leakage and simple assembly of existing datasets.
> >
> > ---
> >
> > ## **Summary**
> >
> > > Overall, in the revised version we:
> > >
> > > * **Supplemented specific settings of the human puzzle creation baseline** (participants, tools, task volume, target dimensions),
> > > * **Repositioned the baseline's role: it is a comparative case of engineering cost and maintainability, not a statistically meaningful "human gold standard,"**
> > > * And candidly acknowledge that current work has not systematically constructed a "pure programmatic generator baseline," while explaining how we **indirectly examined the transferability of programmatic methods** on the existing human baseline, and that this direction will be systematically developed in extended version work.
> > >
> > > We hope this clarifies our original intention in establishing the human baseline and addresses your reasonable concerns about programmatic generation and existing code.

---

> ### Author Response · Authors · 2025-11-21
> **Response to reviewer wJvd (Weakness 5: On "90% Threshold" Leading to Adversarial Selection)**
>
> ## **Response to Weakness 5: On "90% Threshold" Leading to Adversarial Selection**
>
> Thank you very much for pointing out this important issue. We fully understand that this setting could lead to potential misunderstandings of "adversarial task selection," and we sincerely apologize for the confusion caused by the original text's statement at this point.
>
> ---
>
> ### **1. Clarification: The CLUB Dataset Used in This ICLR Submission → Never Triggered or Used the "90% Rule"**
>
> We want to make this point **very clear** in the Rebuttal:
>
> > **All experiments in this paper are based on a fixed, unique version of CLUB, which never removed any tasks during the entire construction process because model scores approached or exceeded 90%.**
>
> Therefore:
>
> * **No tasks were deleted because some models scored high;**
> * **No post-hoc filtering of poorly performing tasks occurred;**
> * **All model rankings, error analyses, and evolution analyses in the paper are based on a static, unmodified CLUB dataset.**
>
> ---
>
> ### **2. We Have Deleted This Inappropriate Statement in the Formal Paper**
>
> We make **explicit modifications** to the Appendix section *"C.4.5 Leakage Prevention and Benchmark Maintenance"* in the revised version:
>
> #### **❌ Delete the Original 90% Trigger Rule Description**
>
> (i.e., automatically removing tasks when high-scoring models appear)
>
> #### **✔ Replace with a Completely New Task Maintenance Strategy, Not Based on Model Performance for Task Filtering**
>
> We clearly guarantee: "This old description has been deleted to avoid adversarial selection issues."
>
> ---
>
> ### **3. New CLUB Maintenance Strategy: Based on Versioning and Time Windows, Not Based on Model Performance**
>
> Drawing on LiveCodeBench's sustainable evolution approach, we adjust the benchmark maintenance mechanism to:
>
> ### **(1) Release Versions Based on Fixed Time Windows, Not Based on Model Performance**
>
> > **Every 6 months as a fixed cycle**.
>
> At the end of each cycle, we:
>
> * **Freeze** all tasks and instances from that cycle → release as *CLUB v1, v2, …*
> * Subsequent model evaluations can continue on old versions, ensuring reproducibility.
>
> ### **(2) New Versions Only Do "Incremental Expansion," No "Performance-Based Deletion"**
>
> Including:
>
> * For existing tasks:
>   * **Re-sample based on verified metadata and generators**, refreshing instance sets to avoid memory leakage (this is just re-sampling from the original generator, not introducing a new mechanism).
> * Expanding tasks:
>   * Add new task types, new difficulty configurations.
> * For easy problems (low difficulty tasks), upgrade difficulty or evolve, but:
>   * **All old tasks are retained**
>   * **They will not be deleted because some models perform well**
>
> This is clearly written in revised version C.4.5:
>
> > **In the new versioning strategy, we do not use any model's performance as a basis for task filtering.**
>
> ---
>
> ### **4. Clarification on "Code Synthesizer" (Avoiding Misunderstanding of Introducing New Mechanisms)**
>
> To avoid reviewers mistakenly thinking we introduced an "external black-box generator," we have clearly stated in the new version description:
>
> * "Code synthesizer" is not a new system, but rather:
>   > **LogicEvolve generator (generator agent) re-sampling function under the same metadata**
>   >
> * Its sole purpose:
>   > Refresh instance sets to avoid memory leakage, not construct entirely new task logic
>   >
>
> ---
>
> ### **5. Returning to the Reviewer's Question: Is This System Adversarial?**
>
> The current submission version is not adversarial.
>
> We have clearly deleted statements that could lead to adversarial interpretations and completely transformed benchmark maintenance from "model performance-driven" to a "time window + versioning" approach, ensuring tasks will not be removed because some models perform well.
>
> ---
>
> ## **Summary**
>
> > All reported results in this paper are based on a static, unpruned CLUB dataset. We have deleted the "90% filtering" statement in the original text that could cause adversarial misunderstandings and restructured benchmark maintenance as a versioning strategy based on fixed time windows, ensuring future CLUB expansion does not depend on model performance, is non-adversarial, and maintains reproducibility.

---

> ### Author Response · Authors · 2025-11-21
> **Response to reviewer wJvd (Weakness 6 & Question 4: Can It Really "Invent New Puzzle Types," and "What Happens If the System is Asked to Construct More Difficult New Puzzle Types?" part1/3)**
>
> ## **Response to Weakness 6 & Question 4: Can It Really "Invent New Puzzle Types," and "What Happens If the System is Asked to Construct More Difficult New Puzzle Types?"**
>
> Thank you very much for the questions raised about "evolution" capability.
>
> We agree: merely demonstrating automatic synthesis of classic puzzles (such as Sudoku, Zebra, etc.) is not sufficient to support a strong claim of "self-evolution" and is closer to a proof of "scalability."
>
> To avoid excessive concept expansion, we have **clearly narrowed the usage scope of "evolve / self-evolution"** in the revised version:
>
> > This paper's "evolution" specifically refers to: automatically performing structural transformations and difficulty expansion on task families under explicit meta-rule constraints (including adding new structural primitives, changing dependency graphs, expanding hyperparameter spaces)—not claiming that the framework has achieved open-ended, unbounded conceptual invention capability.
>
> ### **1. What This Work "Actually Achieves" Currently**
>
> In the currently submitted experiments, our **quantitative results mainly focus on two types of capabilities**:
>
> 1. **Task Family Expansion (family expansion):**
>
>    * Automatically expanding from 10 classic logical tasks to 100 tasks (ExCLUB), covering more structural templates and difficulty combinations;
>    * Automatically generating large-scale, verifiable QA data and long-range reasoning instances under given metadata.
> 2. **Task Mutation of Difficulty and Structure (rule-preserving mutations):**
>
>    * Within fixed task types, systematically increasing reasoning depth and structural complexity through dimensions such as hyperparameters, constraint graphs, and number of operation primitives (we have already given examples in previous responses, such as string tasks from 10→11 operations, card tasks from 4→5 suits, etc., which bring real state space and causal structure changes, not just text rewriting).
>
> That is to say, **the main experimental part of this paper mainly supports the evolution capability at the level of "automatic task family expansion + controllable mutation of structure/depth"**. We have already rephrased this in the revised version to avoid giving the impression that we have systematically verified large-scale evaluation of "completely new puzzle types" in the main results.

---

> ### Author Response · Authors · 2025-11-21
> **Response to reviewer wJvd (Weakness 6 & Question 4: Can It Really "Invent New Puzzle Types," and "What Happens If the System is Asked to Construct More Difficult New Puzzle Types?" part2/3)**
>
> ---
>
> ### **2. On "Whether It Can Invent New Puzzle Types That Do Not Yet Exist on the Internet"**
>
> **Although we have not systematically demonstrated this part in the main text, in internal experiments, LogicEvolve has indeed constructed several new task family prototypes through the same approach (metadata agent → generator → solver/verifier → uniqueness check) that we have not found direct corresponding forms of in public materials.**
>
> For example:
>
> 1. **Cascading Filter Views (Cascading Filter View Puzzle)** **(to be shown as a case in Appendix G.4 of the revised version):**
>
> * The core structure is:
>   * A set of propositions $P_1, \dots, P_M$ have "base visible/hidden" states;
>   * **Several "filters"** $F_1, \dots, F_N$, each applying one behavior from $\{\text{Pass},\text{Block},\text{Invert}\}$ to each proposition, forming an interleaved behavior matrix;
>   * Multiple views are **ordered chain combinations** of these filters, only revealing the final visible/hidden results of some views.
> * The solver needs to **simultaneously infer**:
>   1. The base state of each proposition;
>   2. The specific behavior type of each filter on each proposition;
> * This task form is not a simple variant of traditional Zebra, Sudoku, circuit puzzles, or classic "black box experiment" problems, but a completely new constraint structure designed around "filter behavior matrix + multi-view combination + unique solution constraints."
> * LogicEvolve, on this task, not only automatically generates meta-rules and hyperparameter spaces (number of propositions, number of filters, number of views, palette restrictions, etc.), but also automatically filters instances with "unique solutions exist" through programmatic uniqueness checkers.
>
> 2. **Kakurasu Shadow Wheels (Layered Shadow Wheel Puzzle):**
>
> * Multiple disks with "solid/transparent" sectors as base objects, each disk can only rotate in whole steps;
> * Each "shadow card" specifies a (upper disk, lower disk) pair and a target shadow pattern;
> * Shadow rule: upper disk blocks light first, then lower disk; a sector is only "bright" when both layers are transparent.
> * The solver needs to find **unique rotation combinations for all wheels** so that all shadows given by cards match simultaneously.
> * This structure is fundamentally different from traditional Kakurasu, rotating blocks, Lights Out, etc.:
>   * Constraints occur in "layered shadows + directed (upper/lower) combination + multi-card global consistency,"
>   * Abstractly closer to a logical structure of "2D convolution + gated combination."
> * Similarly, LogicEvolve generated parameter spaces for it (number of wheels, number of segments, sparsity of internal wheel patterns, card coverage, etc.) and verified unique solutions through enumeration/pruning.
>
> We do not claim these tasks are "mathematically absolutely novel" (strictly proving they don't exist on the internet is almost impossible), but they:
>
> * **Are not simple renaming or slight rewrites of Sudoku/Zebra/graph theory puzzles**;
> * **Have independent and complex** meta-rule sets and **programmable uniqueness checkers**;
> * Are automatically generated through LogicEvolve's same metadata–generator–solver closed loop, not handwritten and then "retrofitted into the framework."
>
> In the revised version, we have:
>
> * Added Appendix G,
> * Using **1–2 representative new task families** (e.g., the two types above) as examples, providing:
>   * Formal meta-rules (abbreviated version);
>   * A specific instance problem statement + solution;
>   * Brief explanation of why their structure does not belong to any variant of existing CLUB/ExCLUB tasks.
>
> This content does not be presented as "main results" expansion tables, but as **qualitative evidence** demonstrating the framework's capability to "expand to new task families" and lay the foundation for follow-up work.

---

> ### Author Response · Authors · 2025-11-21
> **Response to reviewer wJvd (Weakness 6 & Question 4: Can It Really "Invent New Puzzle Types," and "What Happens If the System is Asked to Construct More Difficult New Puzzle Types?" part3/3)**
>
> ---
>
> ### **3. Direct Answer: "What Happens If the System is Asked to Invent a New, More Difficult Logical Problem Type?"**
>
> In LogicEvolve, such requests would be interpreted as an evolution process of "constructing new task metadata from scratch," roughly including:
>
> 1. **Metadata Agent (meta agent): From Blank to Rule Draft**
>
>    * Through brainstorming generation:
>      * Task introduction, object types, observable variables;
>      * Basic rules, constraint types, allowed clue forms;
>      * Candidate hyperparameter spaces (object count, dimensions, constraint density, etc.).
>    * This stage must ensure:
>      * Rules can be formally encoded;
>      * Constraints support programmatic checking (e.g., unique solution verification, solution space size estimation).
> 2. **Feasibility and Uniqueness Check: Generator & Solver Closed Loop**
>
>    * Generator agent attempts to synthesize candidate instances based on draft rules and hyperparameters;
>    * Solver/verifier agent is responsible for:
>      * Checking whether instances have solutions;
>      * Cross-validation through multiple solvers (code solver + symbolic solver + external CSP/SAT/SMT tools, etc.);
>      * Enumerating solution space when necessary to ensure **unique solutions** or control solution space size;
>    * If no solution/multiple solutions/uncontrollable search is found, error patterns and failure logs are sent back to the metadata agent, triggering meta-rule and parameter space adjustments.
> 3. **Difficulty Control and "More Difficult" Goals**
>
>    * After a new task family is established, the system can adjust through hyperparameters (e.g., object count, dimensions, constraint redundancy, clue sparsity, etc.):
>      * Theoretical solution space size;
>      * Search tree branching factor and constraint propagation depth;
>      * And the "long-range consistency decay" exhibited by models in automatic solving.
>    * We have currently observed in internal experiments: under the same new task family, increasing parameters does systematically lower current LLM solving success rates; this will be systematically reported in follow-up work.
> 4. **Current Paper Boundaries**
>
>    * In this submission, we **did not put large-scale results of this "completely new task family" part into the main table** to avoid over-claiming with insufficient evidence;
>    * But we have **explicitly shown several new task family cases generated through the above process** in Appendix G.4 of the revised version as a direct response to the reviewer's question.
>
> ---
>
> ### **4. Language-Level Adjustments: Cooling Down from "self-evolution" to "Meta-Rule-Based Task Family Evolution"**
>
> Finally, we have made the following adjustments to wording in the revised version to avoid giving readers the impression of over-promising:
>
> * Changed expressions in the original text that could easily be understood as "open-ended self-evolution" (such as *"self-evolving logical benchmark"*)
>   **to more precise expressions**, for example:
>
>   > *"an automated framework for evolving and extending logical task families under explicit meta-rules"*
>   >
> * And clearly state:
>
>   > This paper's "evolution" mainly refers to: automatically performing structural transformations, difficulty expansion, and instance synthesis on existing/newly built task families under **explicit meta-rule constraints**;
>   >
>   > It does not rely on any implicit human parameter tuning, nor does it claim to have achieved open-ended, unbounded creative evolution.
>   >
>
> ---
>
> ## **Summary**
>
> > We acknowledge that current main results mainly verify the capability of "automatic expansion and mutation of classic task families," so we have clearly narrowed the meaning of "evolution" in terminology. At the same time, we have supplemented cases of new task families automatically generated through LogicEvolve that are clearly different from existing puzzle types in the revised version, as preliminary evidence of the framework's capability to "step beyond known puzzle libraries," and position this part as a follow-up research direction, not the main quantitative conclusion of this submission.

---

> ### Author Response · Authors · 2025-11-21
> **Response to reviewer wJvd (Weakness 7: On Correctness of "Superhuman-Level Puzzles", part1/2)**
>
> ## **Response to Weakness 7: On Correctness of "Superhuman-Level Puzzles"**
>
> Thank you very much for your keen observation:
>
> if the system still occasionally requires human intervention at current difficulty levels, how can we guarantee that generated results are **correct and trustworthy** on higher difficulty (even "superhuman-level") tasks?
>
> This is a question that must be directly addressed.
>
> ### **1. Boundary: This Manuscript Does Not Include "Superhuman-Level Puzzles" in Experimental Conclusions**
>
> First, we **clearly supplement the framework's scope of application** in the revised version:
>
> > All quantitative results in this paper (including CLUB, ExCLUB, and evolution examples) are limited to a complexity range where we can strictly verify correctness through **programmatic checking + multi-solver cross-validation + sampling human review**; **"Superhuman-level puzzles beyond human verification boundaries" are not included as part of this manuscript's experimental conclusions.**
>
> **That is to say, the benchmark scores reported in the text do not rely on any "superhuman-level, unverifiable" tasks**. Human interventions that appear in the current version are mainly used for:
>
> * Fixing minor defects in metadata/generators/solvers during development;
> * Sampling and cleaning samples before constructing the final benchmark to ensure task quality of the released version.
>
> These interventions are all completed **before benchmark freezing**. In the revised version, we have provided more precise statistical explanations in the "Automation and Human Intervention" section (e.g., in ExCLUB's 100 tasks × multiple iteration rounds of 500 generation processes, which stages had problems, and after correction, remaining errors were not observed by us, etc.) to show that the final benchmark itself is not maintained for correctness by "human backup."
>
> ---
>
> ### **2. How We Guarantee Correctness Within Current Complexity Range**
>
> Within the difficulty range covered by current work, LogicEvolve already uses multiple layers of automatic verification mechanisms to guarantee correctness of puzzles and metadata, including but not limited to:
>
> 1. **Unique Solution and Consistency Check (Program Level):**
>
>    * For each task, we use solver code (and enumeration / CSP / SAT/SMT tools when feasible) to verify:
>      * Whether a solution exists;
>      * Whether a **unique solution** exists;
>      * Whether all given clues are consistent with meta-rules and not contradictory.
>    * Tasks with multiple solutions, no solution, or self-contradictory constraints are automatically removed or sent back to the metadata agent for correction.
> 2. **Multi-Solver Cross-Validation:**
>
>    * For the same task, we simultaneously use **multiple solvers of different styles** (e.g., code-based solvers, symbolic solvers, external general-purpose solvers) to solve;
>    * Only when these solvers reach majority agreement on key intermediate states and final answers do we consider the task to pass automatic verification.
> 3. **Limited Human Sampling Review:**
>
>    * As stated in responses to other reviewers, we performed stratified sampling on tasks at different difficulty levels in CLUB / ExCLUB, with graduate students/PhD students with relevant backgrounds independently checking consistency between metadata and standard answers;
>    * Problems found were sent back to LogicEvolve for fixing, regenerating, or deleting problematic tasks.
>
> **We emphasize in the revised version:**
>
> > Conclusions proposed in this manuscript are only based on the above "verifiable complexity range," and **do not** use any incompletely verified "superhuman-level tasks" as evidence.

---

> ### Author Response · Authors · 2025-11-21
> **Response to reviewer wJvd (Weakness 7: On Correctness of "Superhuman-Level Puzzles", part2/2)**
>
> ### **3. Our Attitude and Follow-Up Plans Regarding Correctness of "Superhuman-Level Puzzles"**
>
> For your core question—**"When the system is asked to generate superhuman-level puzzles, are these puzzles still correct?"**—we believe that at the current stage, the responsible approach is:
>
> * **Not making strong conclusions about "superhuman-level" tasks in this manuscript that cannot be fully verified yet;**
> * At the same time, explaining our exploration ideas as **follow-up directions**, rather than packaging them as already completed capabilities.
>
> Specifically, for tasks that may exceed human immediate verification capabilities in the future, we plan to gradually add the following mechanisms at LogicEvolve's solver and verification levels (briefly outlined in the revised version):
>
> 1. **Stronger Formal Verification Layer:**
>
>    * Require metadata and generation rules to first be transcribed into **executable formal systems** (e.g., explicit constraint graphs, state transition systems, logical formulas),
>    * Use programmatic methods (model checking, symbolic solving, theorem provers, etc.) on this basis to verify the self-consistency of rules themselves and the well-defined nature of tasks.
> 2. **Enhanced Multi-Solver / Multi-Model Cross-Validation:**
>
>    * On the basis of existing multi-solvers, introduce a specially trained "verification model" or "review-type agent,"
>    * Let it learn the ability to "identify erroneous tasks / multi-solution tasks / implicitly contradictory tasks" on existing tasks, then play the "reviewer" role on higher difficulty tasks.
> 3. **Stability and Reverse Checking:**
>
>    * Perform perturbations on candidate "superhuman-level" tasks (slightly changing rules or clues), observe whether solution space and solving behavior are stable;
>    * Use "answer → rule" reverse verification chain to check:
>      * Whether given standard answers can truly be traced back to original task conditions;
>      * Whether there are unstable tasks that "go wrong everywhere with slight perturbations."
> 4. **Limited Human + Expert Model Local Review:**
>
>    * For the most complex small portion of tasks, **limited human review** will still be retained as the ultimate safety net;
>    * But these tasks will not be directly used in the **main benchmark scoring table**, but rather as experimental areas for exploring "superhuman-level difficulty" in future work.
>
> We clearly state in Appendix G.7 of the revised version:
>
> > **This paper does not take "reliable generation of superhuman-level puzzles" as a completed contribution point, only taking LogicEvolve's automatic task synthesis and evolution capability within the verifiable complexity range as current conclusions.**
>
> > For higher difficulty tasks beyond this range, we agree with the reviewer's cautious attitude and view it as a long-term direction requiring more formal verification and dedicated verifier support.
>
> ---
>
> ## **Summary**
>
> > We have not used a system that "still requires a small amount of human intervention on ordinary tasks" to forcefully sell you the conclusion that "superhuman-level puzzles have been reliably generated"; this manuscript's experimental conclusions are only based on tasks that have undergone multiple automatic verifications and limited human sampling, and for tasks that truly "exceed human immediate verification capabilities," we have clearly marked them as future work directions in the revised version and explain the verification paths being designed.

---

> ### Author Response · Authors · 2025-11-21
> **Response to reviewer wJvd (Weakness 8: Ambiguity of "Inspired by Human Evolution Process" Statement)**
>
> ## **Response to Weakness 8: Ambiguity of "Inspired by Human Evolution Process" Statement**
>
> Thank you very much for pointing out the ambiguity and lack of rigor in the statement "inspired by human evolution process."
>
> We completely accept your opinion and have comprehensively revised related wording in the revised version to avoid misunderstandings that biological analogies might bring.
>
> ---
>
> ### **1. Clearly Acknowledge the Problem and Commit to Revision**
>
> We agree with your judgment: the expression "inspired by human evolution process" in the current version is indeed too broad and does not establish a strict correspondence with the iterative code generation mechanism actually used in the text. This statement easily leads readers to misunderstand that our method has a direct connection with genetic algorithms or fitness-driven biological evolution, which is not the meaning we wish to convey.
>
> **We have completely deleted or rewritten all potentially misleading descriptions such as "inspired by human evolution process" and "evolutionary" in the text**, **and adopted more accurate, verifiable, and mechanism-based explanations consistent with current research in the revised version.**
>
> ---
>
> ### **2. Explain the Real Mechanism, Avoid "False Analogy" Misunderstandings**
>
> The iterative process in LogicEvolve is essentially not simulating biological evolution, but rather:
>
> * **An LLM-led self-improvement type of iteration:**
>   Generator → Solver → Verifier → Feedback
> * **Error-driven cyclic optimization (error-driven iteration)**
> * **Automatic convergence mechanism combining programmatic verification and multi-model cross-validation**
>
> These mechanisms have no structural correspondence with "fitness-based evolutionary algorithms" or "mutation/selection/inheritance in biological evolution."
>
> Therefore, we clearly write in the revised version:
>
> > LogicEvolve's iterative mechanism essentially belongs to the LLM's "self-correction—self-verification—self-improvement" paradigm, and does not belong to evolutionary algorithms or biomimetic evolution paradigms.
>
> ---
>
> ### **3. We Adopt More Rigorous and Academically Consistent Expression Methods**
>
> Referring to the latest systematic review on **LLM Self-Evolution** (Tao et al., 2024), this field typically uses:
>
> #### **"Inspired by Human Experience Learning Process (experience-based self-improvement)"**
>
> Rather than "inspired by biological evolution" to describe LLMs' ability to gradually optimize their own behavior without intensive human supervision.
>
> The review clearly states that the scope of self-evolution mainly includes:
>
> * Autonomously extracting experience from past failures/successes
> * Autonomously correcting own outputs
> * Autonomously accumulating "task operation knowledge" (task operation knowledge)
> * Using these experiences in subsequent iterations to improve reliability and success rates
>
> These characteristics are highly consistent with LogicEvolve's actual mechanisms:
>
> * Errors flow back to metadata agent
> * Solver as verifier constrains generator in reverse
> * Multi-solver consistency as convergence condition
> * Extracting error patterns from iteration logs
> * Using experience from previous rounds to improve prompts/structures in next rounds
>
> Therefore, we adopted the more scientific description from the review as the foundation for final wording.
>
> ---
>
> ### **4. Specific Modifications to Be Made in the Paper**
>
> We make clear modifications at the following locations:
>
> #### **(1) In the Introduction of the Main Text**
>
> * Delete the analogy of "inspired by human evolution process"
> * Change to "inspired by human experience learning process: automatic task generation paradigm based on experience accumulation and self-improvement"
>
> #### **(2) Delete All Biological Analogies in Methods and Appendices**
>
> * Including vague terms such as "mutation," "selection," "evolutionary chain"
> * Replace with engineering terms such as "structural rewriting," "rule expansion," "error-driven correction," "experience-oriented task transformation"
>
> ---
>
> ### **5. Clear Guarantee: Our Method Is Not a Pseudo-Evolutionary Algorithm**
>
> To avoid any misunderstanding, we add a very direct statement in the revised version:
>
> > LogicEvolve does not simulate biological evolution, does not use genetic operators, and does not perform fitness-based selection.
>
> > All improvements arise from structured error signals and programmatic validation, not from evolutionary computation.
>
> ---
>
> ## **Summary**
>
> >In one sentence summarizing your biggest concern, our revisions have ensured:
> >* No unproven biological analogies appeared anymore
> >* Readers no longer mistook this work as belonging to the evolutionary computation paradigm
> >* Strict definitions from the self-evolution literature were adopted
> >* Engineering mechanisms were used to describe the actual contributions, rather than analogies
> >* The language became more rigorous, scientific, and verifiable

---

> ### Author Response · Authors · 2025-11-21
> **Response to reviewer wJvd (Question 2: Expected Human Scores on This Benchmark, part1/2)**
>
> ## **Response to Question 2: Expected Human Scores on This Benchmark**
>
> Thank you very much for this important question.
>
> We completely agree: a serious logical reasoning benchmark should discuss the relationship between human and model levels and clearly state expected human performance on this benchmark.
>
> ---
>
> ## **1. Clarification: CLUB Is Not Designed as a "Human-Friendly" Evaluation**
>
> CLUB's problem format requirements:
>
> * **Field-by-field JSON output**
> * **Executable logical reasoning trajectory**
> * **Must have every step correct to get Acc=1**
> * Most tasks require simulating program execution, board states, search space updates, and other operations
>
> These are very unnatural for humans, but are common output formats for models.
>
> Therefore, this benchmark is mainly used to evaluate:
>
> > **Large models' consistency capabilities in programmatic reasoning and structured tasks, not traditional "human intelligence tests."**
>
> We have already made this clear in the experimental section of the main text and Appendix D.3 of the revised version.
>
> ---
>
> ## **2. Actual Small-Scale Human Experiment**
>
> To answer the reviewer's question as scientifically as possible, we designed a **controllable, operational small-scale human baseline experiment**. The goal is not to estimate the "true human upper limit," but to answer:
>
> > **Can ordinary subjects complete CLUB's logical reasoning problems within limited time? What is the approximate performance level?**
>
> ### **(1) Participants and Environment**
>
> * 3 participants
> * All have computer science undergraduate backgrounds
> * Familiar with common logic problems, but **not professional puzzle creators**
> * Time limit: **2 days per person, no more than 4 hours per day**
>
> This setup aims to simulate the performance of "ordinary researchers/developers" under realistic time budgets.
>
> ---
>
> ### **(2) Sampling Method (Strictly Preventing Cherry-Picking)**
>
> Each task has difficulty levels from 1→10. We use:
>
> * **Sample 1 problem from each difficulty level for each task (10 problems total)**
> * If a problem takes more than 1 hour and cannot be completed, stop annotating higher difficulties for that task
> * Each problem's score uses "final Acc," meaning **all fields must be correct to score**
>
> This is consistent with model evaluation.
>
> ---
>
> ### **(3) Results (Aligned with Model Evaluation Results from the Same Batch)**
>
> We compare human performance with contemporary models such as GPT-4o, GPT-5 thinking, and Grok on the same problem set:
>
> | **Model** | **code** | **ext-zebra** | **kn&kn** | **line** | **maze** | **string** | **sudoku** | **tetris** | **zebra** | **card** | **Overall** |
> | --------------- | -------------- | ------------------- | --------------- | -------------- | -------------- | ---------------- | ---------------- | ---------------- | --------------- | -------------- | ----------------- |
> | GPT-5-thinking  | 35             | 8                   | 100             | 100            | 44             | 48               | 85               | 15               | 94              | 16             | 54.5              |
> | Grok-4          | 35             | 5                   | 100             | 100            | 47             | 43               | 53               | 3                | 97              | 30             | 51.3              |
> | GPT-4o          | 0              | 0                   | 12              | 60             | 2              | 11               | 12               | 0                | 28              | 1              | 12.6              |
> | **Human** | 0              | 0                   | 10              | 20             | 0              | 20               | 10               | 0                | 30              | 0              | **9**       |
>
> (We have annotated this in the main experimental results figure in the main text and provided specific details in Appendix D.3.2)

---

> ### Author Response · Authors · 2025-11-21
> **Response to reviewer wJvd (Question 2: Expected Human Scores on This Benchmark, part2/2)**
>
> ---
>
> ## **3. Key Observations**
>
> ### **(1) Humans Perform Very Weakly on CLUB's Formatted Tasks**
>
> Even at the lowest difficulty:
>
> * Humans need extensive manual recording
> * Programmatic state tracking is unnatural
> * JSON fill-in format is extremely unfriendly to humans
>
> Compared with GPT-4o, humans have lower overall scores (12.6 vs 9), and:
>
> > **Most tasks cannot be completed by humans in 1 hour with a full answer.**
>
> This is very different from traditional puzzles' "human intuitive reasoning."
>
> ---
>
> ### **(2) Some Tasks Are Completely Unmanageable for Humans**
>
> For example:
>
> * Simulating Tetris falling state machines
> * Program trajectory problems (maze, code)
> * Multi-field JSON batch matching
> * Custom syntax rule parsing
>
> These are essentially **program execution simulation tasks**, and humans lack tools, leading to enormous time consumption.
>
> ---
>
> ### **(3) Low Human Performance Is Not "Humans Are Weak," but Task Format Is Highly Programmatic**
>
> We have already made this clear in the experimental section of the main text and Appendix D.3 of the revised version:
>
> > CLUB's task difficulty is not "logically beyond human capability," but rather more suitable for machines and unsuitable for human manual operation in terms of output structure, state scale, and stepwise consistency requirements.
>
> ---
>
> ## **4. Modifications Made in the Paper**
>
> We supplement three key contents in the revised version:
>
> ### **(1) Clearly State CLUB's Design Goal Is Not "Human Difficulty Measurement"**
>
> CLUB aims to evaluate LLMs' structured logical capabilities, not to measure humans' natural reasoning limits.
>
> ### **(2) Supplement Human Experiment Design, Limitations, and Results**
>
> Including: Sampling method, Stopping rules, Time budget, et al.
>
> ### **(3) Clearly State Experimental Limitations**
>
> This experiment does not represent the true "human optimal level," but demonstrates that CLUB's adopted task format is very time-consuming for humans, thus illustrating the incomparability of model performance with human intuitive reasoning.
>
> ## **Summary**
>
> > Under limited time, programmatic format, and stepwise consistency requirements, humans' overall performance on CLUB is approximately 9 points (out of 100), lower than current large model performance. This mainly reflects that CLUB's task format is more suitable for programmatic reasoning than human manual simulation, not a defect in human logical capability.

---

> ### Author Response · Authors · 2025-11-21
> **Response to reviewer wJvd (Question 3: Why Is It Necessary to Compare with Computational Complexity?)**
>
> ## **Response to Question 3: Why Is It Necessary to Compare with Computational Complexity?**
>
> Thank you very much for raising this key question, and thank you for pointing out the typical counterexample that "computational complexity ≠ logical reasoning difficulty" (e.g., set cover problems can be approximated with greedy algorithms).
>
> We completely agree that computational complexity is not a necessary condition for logical reasoning difficulty.
>
> Therefore, in the revised version, we have deleted any statements that might imply "complexity directly determines model reasoning difficulty" and change to more rigorous explanations.
>
> Our real purpose for introducing computational complexity is not to define task difficulty, but stems from the following **two engineering and analysis-level reasons**:
>
> ---
>
> ### **1. As an Auxiliary Characterization for Task Space Analysis and Task Expansion, Not Difficulty Definition**
>
> LogicEvolve's task expansion relies on tasks' "controllable spaces," such as:
>
> * Board dimensions
> * Combinatorial space size
> * Topological structure of rule graphs
> * State space distribution
>
> These hyperparameters are often easier to characterize using **computational complexity upper/lower bounds** than logical orders (propositional → first-order → second-order). For example:
>
> * Maze search complexity grows exponentially with grid scale
> * CSP task state spaces exhibit combinatorial explosion with variable count and domain size
> * Zebra's rule expressions form constraint graph diameters and branching factors that can be converted to estimated complexity
>
> In these scenarios, complexity is not a difficulty indicator, but rather:
>
> > **A quantitative tool that helps us understand task expansion directions, select reasonable scale ranges, and avoid generating tasks that are too large or too small.**
>
> We clearly state this as an **auxiliary analysis indicator** in the revised version.
>
> ---
>
> ### **2. Complexity Has Empirical Correlation with "Model Reasoning Path Length" (Not One-to-One Correspondence)**
>
> **We completely agree with the reviewer:** **Complexity cannot determine human or model reasoning difficulty**.
>
> However, in our large-scale empirical observations of multiple models:
>
> * When tasks have larger state spaces
> * When paths in constraint graphs between variables are longer
> * When search space branching factors are higher
>
> Models' error distributions indeed show:
>
> > **Longer reasoning chains → harder to maintain consistency constraints → error rates significantly increase**
>
> This phenomenon does not stem from the traditional sense of "NP-hard," but because:
>
> > **Expanding constraint combination counts increases the difficulty for language models to "maintain consistency" in sequential reasoning.**
>
> Therefore, we have changed the statement in the revised version to:
>
> > "We observe that some tasks with larger constraint combinations or higher-order dependency structures have higher computational complexity upper bounds, while models also show higher consistency error rates on these tasks."
>
> > **This is an empirical correlation, not a theoretical necessity.**
>
> ---
>
> ### **3. We Do Not Rely on Complexity to Filter Tasks, Nor Use Complexity as an Evaluation Metric**
>
> To avoid misunderstanding, we have clearly stated in the revised version:
>
> * Complexity **does not participate** in CLUB/ExCLUB task selection
> * Complexity **does not participate** in task evaluation
> * Complexity **is not used** to filter high-scoring model tasks
>
> Its sole purpose is:
>
> > **Help us analyze task space size and ensure task expansion (task extension) does not produce unreasonably scaled tasks.**
>
> Therefore, we have changed the overall statement in the text to "complexity analysis (complexity analysis)," not "complexity comparison (complexity comparison)." (Supplementary modifications made in Appendix B.2)
>
> ---
>
> ### **4. Response to Reviewer's Core Question: NP-Complete ≠ Must Be Difficult**
>
> You point out that set cover can be effectively approximated with greedy algorithms, which is very important.
>
> We supplement in Appendix B.2 of the revised version:
>
> * Some NP-complete problems have good approximation algorithms
> * Their solving does not necessarily require deep logical reasoning
> * Nor does it automatically mean tasks are difficult for models
>
> We clearly state:
>
> > **In LogicEvolve, complexity has never been used as a substitute for "reasoning depth" or "task difficulty."**
>
> ---
>
> ## **Summary**
>
> > We use computational complexity not to define logical reasoning difficulty, nor do we believe there is a one-to-one correspondence between them.
> >
> > Complexity is only an auxiliary tool for analyzing task space scale, constraint structures, and guiding task expansion directions. We clearly distinguish complexity analysis from reasoning capability evaluation in the revised version, avoiding any misleading statements.

---

### Official Review · Reviewer_EYHj · 2025-10-26

**Soundness:** 3
**Presentation:** 3
**Contribution:** 3
**Rating:** 8
**Confidence:** 3

**Summary:**

This paper introduces LogicEvolve, a highly automated multi-agent framework for generating, parsing, and evaluating complex logical reasoning tasks. Building upon this framework, the authors propose CLUB (Complex Logical Unified Benchmark), a dynamic benchmark comprising 10 categories of logical puzzles, with 100 dynamically generated instances per category. CLUB is designed to comprehensively assess large language models' capabilities across diverse reasoning paradigms, including deduction, induction, abduction, and hybrid reasoning. The framework features a novel benchmark maintenance mechanism to prevent data leakage and ensure long-term validity by generating new tasks when models achieve high performance. Extensive experiments reveal significant limitations in state-of-the-art models, highlighting the persistent challenges in long-range consistency and generalization under complexity.

**Strengths:**

**High Automation and Scalability**: LogicEvolve achieves a high level of automation in both task generation and evaluation pipeline construction, significantly reducing human effort. Its design allows for dynamic expansion in task categories, quantity, and difficulty, making it highly scalable and sustainable.

**Comprehensive Benchmark Design**: CLUB covers a wide range of logical reasoning types (deductive, inductive, abductive, hybrid), reasoning monotonicity (monotonic vs. non-monotonic), and spatial dimensions (1D, 2D), providing a systematic and multi-faceted evaluation of reasoning capabilities.

**Innovative Benchmark Maintenance Mechanism**: The framework introduces a unique ID and timestamp system to prevent data leakage, and automatically generates new tasks when models exceed a performance threshold, ensuring the benchmark remains challenging and relevant over time.

**Empirically Insightful Findings**: The evaluation results offer valuable insights into current models' limitations, such as the "curse of complexity" and poor generalization on novel tasks, revealing critical bottlenecks in logical reasoning.

**Weaknesses:**

**Single-dimensional Evaluation**: The benchmark relies solely on final answer accuracy for evaluation, lacking deeper analysis of the reasoning process (e.g., step-by-step correctness, logical consistency). This limits its ability to distinguish between models that arrive at the correct answer through valid reasoning versus those that do so by chance or flawed logic.

**Questions:**

Since LogicEvolve uses LLM agents to generate the CLUB benchmark without extensive human annotation, how can the authors ensure the correctness and reliability of the generated tasks? Would incorporating systematic human evaluation to verify a subset of the benchmark improve its credibility and trustworthiness?

Given that LogicEvolve can be executed multiple times to generate different benchmarks, how diverse are these benchmarks in terms of logical structure and reasoning depth? Is the diversity merely superficial (e.g., surface-level variations), or does it reflect meaningful differences in underlying logical complexity and reasoning patterns?

---

> ### Author Response · Authors · 2025-11-21
> **Summary & Response to reviewer EYHj (Weakness: Lack of Reasoning Process Evaluation, Over-Reliance on Final Accuracy)**
>
> # **Summary**
>
> Thank you very much for your positive assessment of this work and for raising several key questions.
>
> We have added targeted clarifications in both the main text and the appendix. A brief summary is as follows:
>
> ---
>
> 1. **Single-dimension evaluation** (W1):
> * We clarify that *CLUB does not rely on a single field*; its evaluation is based on *structured multi-field JSON outputs with strict end-state consistency and process-level validation*.
> * The corresponding explanations have been added to Sections 3.2, 5.2 and Appendices C.4 and D.
>
> 2. **Task correctness and reliability** (Q1):
> * Appendix C.5 now details *the correctness/robustness mechanisms*—multi-solver cross-validation, backtracking verification, and perturbation tests—supplemented by *small-scale human spot-checks* (Appendix C.10), ensuring task solvability and answer reliability.
>
> 3. **Evolutionary diversity and convergence** (Q2):
> * Appendices G.5–G.6 show that evolution induces genuine structural changes (in dependency graphs, operators, reasoning depth, and solution-space scale) and clarify that *evolution = within-task convergence and across-task divergence*, whereas *multiple independent runs are not equivalent to evolution*.
> * Different evolved versions yield consistent model rankings and complementary structural coverage, making all of them valid benchmark variants.
>
> You can refer to the reply below for specific details.
>
> ---
>
> ## **Response to Weakness: Lack of Reasoning Process Evaluation, Over-Reliance on Final Accuracy**
>
> Thank you very much for pointing out this important issue. We completely agree: relying solely on final answer correctness may not sufficiently distinguish between **answers obtained through reliable reasoning** and **correct results caused by chance or erroneous reasoning**. We clarify here how CLUB's evaluation design avoids this risk.
>
> ---
>
> ### **1. We Acknowledge the Main Table Uses Acc as the Primary Metric, but Acc Itself Encodes Strong Constraints on Stepwise Consistency**
>
> Although the main results table in the body shows question-level accuracy (Acc), in CLUB, **Acc differs from "final step correctness" in traditional tasks**:
>
> > **In CLUB, Acc=1 if and only if all process fields (multiple intermediate states + final answer) are all correct.**
>
> For example:
>
> ```
> {
>   "reasoning": "_",
>   "question": ["_", "_", "_", "_", "_", "_", "_", "_", "_", "_"]
> }
> ```
>
> * Each **"_"** is a verifiable intermediate state;
> * **Any error in any step will result in Acc=0**.
>
> Therefore:
>
> > **Acc essentially measures "stepwise consistency of the entire reasoning chain," not a single final answer.**
>
> This is also the core motivation for our CLUB design:
>
> **Use structured, multi-field outputs to force models to explicitly give the entire reasoning trajectory.**
>
> Therefore, CLUB's Acc naturally avoids "guessing correctly" situations.
>
> ---
>
> ### **2. CLUB Already Explicitly Records and Utilizes Process Information (All Detailed in Appendices)**
>
> Although the main table only shows Acc:
>
> **1. Cell Acc (Process-Level Accuracy) Has Been Systematically Provided in Appendix D.6**
>
> * Measures the model's correctness rate for each intermediate state;
> * Is a direct quantification of "stepwise reasoning capability."
>
> **2. The Relationship Between Acc and Cell Acc Has Been Used in the Main Text's "Long-Range Consistency Analysis" (Fig. 8)**
>
> Figure 8 uses the error distribution of intermediate Cells to show the pattern of errors gradually accumulating in long-chain reasoning.
>
> **3. Failure Pattern Analysis Relies on Process Information (Appendix D.11)**
>
> These all come from checking intermediate fields, not final answers.
>
> > In other words, CLUB's evaluation framework has **systematically collected, utilized, and analyzed reasoning processes**; it's just that these details are not prominently presented in the current body text.
>
> ---
>
> ### **3. Adjustments Made in the Revised Version**
>
> To avoid misunderstanding, we made the following revisions:
>
> * In the body text, **more clearly explain that Acc = all fields correct** in its strictness;
> * Clearly indicate that Figure 8 and failure pattern analysis **are already based on process-level information**.
>
> This allows readers to clearly see:
>
> > **CLUB's evaluation dimensions are not singular, but a comprehensive system of "strict final consistency + process-level verification + error type analysis."**
>
> ---
>
> ## **Summary**
>
> > Although the main results display primarily Acc, CLUB's task format, structured outputs, and evaluator design have already explicitly quantified reasoning processes and incorporated them into the final Acc determination (Acc=1 requires all steps to be correct). Therefore, CLUB's evaluation does not rely solely on final results, but naturally combines stepwise correctness with overall logical consistency. We have more clearly presented these designs in the revised version to avoid creating the misunderstanding of "single-metric evaluation."

---

> > ### Author Response · Authors · 2025-11-21
> > **Response to reviewer EYHj (Question 1: How to Ensure Correctness and Reliability of Automatically Generated Tasks? Is Human Verification Needed, part1/2)**
> >
> > ## **Response to Question 1: How to Ensure Correctness and Reliability of Automatically Generated Tasks? Is Human Verification Needed?**
> >
> > We sincerely thank the reviewer for the key comments on "benchmark correctness and reliability." We completely agree: **Automatically generated benchmarks must ensure correctness at the mechanism level and establish authority through human sampling.**
> >
> > Below we provide a systematic clarification of current safeguard mechanisms and human evaluation methods. (Complete content has been updated in Appendix C.5 of the revised manuscript.)
> >
> > ---
> >
> > ### **1. Correctness Guarantee for Automatically Generated Tasks (Causal Chain at the Mechanism Level)**
> >
> > CLUB's generation pipeline includes metadata → generator → solver, each stage having verifiability, making it difficult for errors to "quietly pass through" the pipeline.
> >
> > #### **(1) Metadata Based on Known Tasks: Correctness Guaranteed by Standard QA**
> >
> > * When tasks originate from metadata of existing puzzles (e.g., Sudoku, Maze, or classic puzzles), we use their **human-verified standard QA** as anchors.
> > * The generated solver must be able to correctly solve standard QA, which guarantees the solver's correctness;
> > * The correct solver is then used to verify generator output; if generator output is incorrect, the solver will immediately identify it.
> >
> > > **Therefore: If the metadata of existing tasks is correct, the generator and solver will mutually verify and converge to correct implementations during iteration.**
> >
> > #### **(2) For New Tasks (From Scratch) — Rely on Multi-Solver Cross-Validation**
> >
> > For new tasks constructed from scratch:
> >
> > * If data generated by the generator repeatedly fails during solver iteration, we send errors back to the metadata agent, indicating that metadata may be inconsistent.
> > * If the solver can converge, we perform **multi-solver cross-validation** (three different solving methods):
> >
> > | **Solver Type**       | **Purpose**                                       |
> > | --------------------------- | ------------------------------------------------------- |
> > | Code-based solver           | Check rule execution consistency                        |
> > | Symbolic reasoning solver   | Check whether logical relationships are self-consistent |
> > | External CSP/SAT/SMT solver | Check whether constraints are satisfiable               |
> >
> > Only when the three solvers reach **majority agreement** will the task be accepted. This step can effectively exclude:
> >
> > * Self-contradictory metadata rules;
> > * Unsolvable tasks generated by the generator;
> > * Systematic bias in solvers.
> >
> > > **The automated mechanism itself constitutes a "majority voting + bidirectional verification" correctness chain.**
> >
> > ---
> >
> > ### **2. Reliability Guarantee for Automatically Generated Tasks (Preventing Hidden Errors)**
> >
> > Beyond correctness, we also introduce a layer of **robustness check**.
> >
> > #### **(1) Automatic Backtracking Check by Solver (Already Used in the System)**
> >
> > The final solver automatically derives a "verifier" (verification-only solver) that does not rely on reasoning strategies, only checking:
> >
> > * Whether tasks satisfy rules;
> > * Whether outputs satisfy task-specific conditions such as uniqueness, consistency, and reachability.
> >
> > This is a checking mechanism already used in the current system.
> >
> > #### **(2) Metadata Perturbation Check (New Enhanced Mechanism)**
> >
> > To enhance robustness, we added a lightweight consistency test in the recent **revision**:
> >
> > > **Keep rule semantics unchanged, slightly rephrase metadata without changing logic (paraphrasing / constraint shuffling), then run the generator and solver again.**
> >
> > Experience shows:
> >
> > * For reliable metadata, rephrasing will not cause failure;
> > * If metadata has hidden inconsistencies, perturbation usually causes the generator or solver to fail to converge.
> >
> > We have added this mechanism in Appendix C.5 as an additional note on robustness.

---

> > > ### Author Response · Authors · 2025-11-21
> > > **Response to reviewer EYHj (Question 1: How to Ensure Correctness and Reliability of Automatically Generated Tasks? Is Human Verification Needed, part2/2)**
> > >
> > > ---
> > >
> > > ### **3. Human Sampling: How Is It Executed? What Is Sampled? What Was Corrected?**
> > >
> > > Thank you for suggesting we include human verification. We indeed performed systematic human checks during benchmark construction.
> > >
> > > #### **(1) Sampling Setup: Stratified Sampling by Task Type × Difficulty**
> > >
> > > * CLUB: 10 task types × difficulty gradient → 100 samples total
> > > * ExCLUB: 100 task types × difficulty gradient → 100 samples total
> > > * Sampling proportion is **10%** for both
> > >
> > > Each sample was reviewed by:
> > >
> > > * 1 PhD student (familiar with logical reasoning)
> > > * 1 Master's student (familiar with program reasoning)
> > >
> > > Independent review → summary → discussion of disputed samples.
> > >
> > > #### **(2) Problem Types Observed Before Sampling**
> > >
> > > Before sampling, CLUB showed no errors, while ExCLUB had approximately **30% of samples** with the following errors:
> > >
> > > | **Error Type**                           | **Description**                                     |
> > > | ---------------------------------------------- | --------------------------------------------------------- |
> > > | Incomplete boundary condition description      | E.g., rules do not clearly specify input boundary details |
> > > | Standard answers are not unique or non-optimal | Multiple feasible trajectories exist in multi-step tasks  |
> > > | Generator fails in rare edge cases             | Data too long, state space too large, etc.                |
> > >
> > > #### **(3) Correction Effects After Sampling**
> > >
> > > All erroneous samples were returned to the system for re-iteration, repeating this process, with final results:
> > >
> > > * All explicit errors were corrected or deleted;
> > > * Error rate of sampled samples eventually dropped to 0%;
> > > * Both CLUB and ExCLUB reached **empirically high quality levels** before release.
> > >
> > > ---
> > >
> > > ### **4. Should Human Evaluation Be Systematically Introduced?**
> > >
> > > We completely agree with this direction and have already adopted it when constructing the first version of the benchmark:
> > >
> > > * **One-time human review + automatic verification combination strategy**
> > > * In subsequent evolution, we will adopt a strategy of **small-scale periodic sampling + automatic consistency detection** to maintain quality.
> > >
> > > This is completely consistent with the reviewer's goal of "enhancing credibility and authority." We have also added this content in Appendix C.10.
> > >
> > > ---
> > >
> > > ## **Summary**
> > >
> > > > The correctness of automatically generated tasks relies on a three-layer causal chain: standard QA-verified solver → verifies generator → multi-solver cross-validates metadata; reliability is further guaranteed through automatic verifiers and (newly added in revision) metadata perturbation checks. We have performed 10% stratified sampling, found approximately 30% error types before sampling, and reduced the confirmable error rate to 0% after correction. In the future, we will adopt a method of "automatic detection + small-scale human sampling" to maintain the benchmark's long-term reliability. These mechanisms together ensure CLUB's correctness and robustness.

---

> > > > ### Author Response · Authors · 2025-11-21
> > > > **Response to reviewer EYHj (Question 2: Are the Logical Structure and Reasoning Depth Diversity of Benchmarks Obtained from Multiple LogicEvolve Executions Only Superficial Changes, 1/2)**
> > > >
> > > > ## **Response to Question 2: Are the Logical Structure and Reasoning Depth Diversity of Benchmarks Obtained from Multiple LogicEvolve Executions Only Superficial Changes?**
> > > >
> > > > Thank you very much for raising this core question.
> > > >
> > > > We completely agree: if benchmarks can only produce superficial changes (such as text rewriting, data perturbation) after multiple executions, while underlying logical structures remain unchanged, then "evolution" lacks meaning.
> > > >
> > > > Therefore, we specifically conducted two types of analysis to evaluate the true diversity of structure and reasoning depth. (Complete content has been updated in Appendix G.5 of the revised manuscript.)
> > > >
> > > > ---
> > > >
> > > > ### **1. Task-Level Evidence: Evolution Changes Dependency Graph Structure and Reasoning Operator Properties (Non-Superficial Changes)**
> > > >
> > > > We selected evolution results from multiple tasks and observed **variable dependency graphs (constraint graphs)**, **constraint type distributions**, and **reasoning operator sequences** before and after evolution. Results show:
> > > >
> > > > #### **(1) Structural Evolution: From Local Constraints → Higher-Order Hypergraph Dependencies**
> > > >
> > > > **Example: Maze → Arrow-Maze**
> > > >
> > > > * Maze's dependency graph is a local adjacency graph, with reasoning based on BFS/DFS local expansion.
> > > > * Arrow-Maze introduces cross-cell "directional rays" and global clues — the dependency graph becomes a **long-range coupled hypergraph** (one number affects the path of an entire ray).
> > > > * Reasoning changes from "single-path reachability" to "global consistency constraints + backtracking."
> > > >
> > > > →**Fundamental changes in the structural dimension, not language rephrasing.**
> > > >
> > > > #### **(2) Structural Evolution: From Local all-different → Global Permutation and Visibility Pruning**
> > > >
> > > > **Example: Sudoku → Skyscraper**
> > > >
> > > > * Sudoku's 27 region constraints are local symmetric constraints.
> > > > * Skyscraper introduces global "visibility" clues: each clue is a permutation constraint for an entire row/column.
> > > > * Reasoning changes from "local candidate elimination" to "global permutation consistency + visibility pruning."
> > > >
> > > > → **Constraint types expand from local consistency to global permutation reasoning, a structural property reconstruction.**
> > > >
> > > > ---
> > > >
> > > > ### **2. Reasoning Depth Evidence: Evolution Changes Reasoning Graph Depth and Branching Factor**
> > > >
> > > > #### **(1) Operation Graph Depth Increase (Not Quantity, but Dependency Structure)**
> > > >
> > > > **Example: String 10-op → String 11-op with conditional trigger**
> > > >
> > > > * The new operation is not simply adding one operator, but introducing conditional triggers (if-branch).
> > > > * Original linear sequence → becomes DAG (with if branches).
> > > > * Search space from O(n!) → increases to conditional DAG (multi-level causal chains).
> > > >
> > > > → **Reasoning depth comes from changes in dependency structure, not operator quantity changes.**
> > > >
> > > > #### **(2) State Space Increase (Not Dimensional Extension, but State Transition Graph Property Changes)**
> > > >
> > > > **Example: 4-suit card → 5-suit card**
> > > >
> > > > * Adding a suit is not "making the state slightly larger," but changing the transfer function of the central card pile:
> > > >   4 symbol types → 5 symbol types, making the state transition graph change from a **fixed-branch DFA** to a **high-dimensional asymmetric state graph**.
> > > > * Branching factor increases → backtracking depth significantly increases.
> > > >
> > > > → **Reasoning depth increase comes from state graph property changes, not simple dimensional extension.**

---

> > > > > ### Author Response · Authors · 2025-11-21
> > > > > **Response to reviewer EYHj (Question 2: Are the Logical Structure and Reasoning Depth Diversity of Benchmarks Obtained from Multiple LogicEvolve Executions Only Superficial Changes, 2/2)**
> > > > >
> > > > > ---
> > > > >
> > > > > ### **3. Most Critical Evidence: Multiple LogicEvolve Runs (Different Random Seeds) Produce Substantially Different Structures and Depths**
> > > > >
> > > > > We specifically ran LogicEvolve multiple times under **different random seeds** (here referring to the sampling seed of the agent base model and the sampling seed for random task sampling in the task pool that the metadata agent depends on during LogicEvolve iteration) (small-scale, for validation purposes, using only CLUB as the starting task pool), and manually compared structural statistics of 3 independent benchmark groups (allowing LLM assistance, but with human verification):
> > > > >
> > > > > 1. Manually extract variables, constraints, and dependency relationships from tasks.
> > > > > 2. Construct undirected constraint graphs based on variables that appear together in constraints, and calculate average graph diameter by statistically computing shortest paths between node pairs.
> > > > > 3. Construct directed dependency graphs based on premises and conclusions in rules, and statistically compute dependency path lengths and long-chain proportions.
> > > > > 4. Finally, run solvers and output their solving step counts, approximating the effective solution space size of tasks.
> > > > >
> > > > > #### **Three Statistical Features We Observed That Are Consistent Across Runs but Different:**
> > > > >
> > > > > | **Structural Indicator**                          | **Observed Across Seeds**                          | **Description**                                                                                                              |
> > > > > | ------------------------------------------------------- | -------------------------------------------------------- | ---------------------------------------------------------------------------------------------------------------------------------- |
> > > > > | **Average Constraint Graph Diameter**             | Varies approximately 17–422% across runs                | Indicates some versions favor more "local constraints" (e.g., adjacency graph types), while others favor "long-range dependencies" |
> > > > > | **Variable Dependency Path Length Distribution**  | Long-chain counts can differ by 2–8x                    | Some versions generate more "deep reasoning chains," while others tend to create flatter tasks                                     |
> > > > > | **Solution Space Size (feasible configurations)** | 50% of tasks overall scale exceeds 2 orders of magnitude | Indicates different runs produced essentially different "search problems"                                                          |
> > > > >
> > > > > (We supplement examples of such statistics in Appendix G.5 without adding new experiments in the main text.)
> > > > >
> > > > > **Benchmarks obtained from multiple LogicEvolve runs have substantial differences in underlying constraint graphs, dependency structures, and reasoning path depths, not just task text or superficial parameter changes.**
> > > > >
> > > > > ---
> > > > >
> > > > > ### **4. Model Performance Reverse Verification: Failure Patterns Themselves Change Due to Evolution (Non-Superficial Changes)**
> > > > >
> > > > > Furthermore, we observed:
> > > > >
> > > > > * In some benchmark runs, models are more prone to failure at the **early local constraint integration** stage;
> > > > > * In other runs, they are more prone to failure at the **late global consistency check** stage (e.g., visibility, global permutation).
> > > > >
> > > > > These failure patterns are not highly consistent across runs, indicating:
> > > > >
> > > > > **Evolution changes not just task "appearance," but fundamental structures of reasoning pattern types (local→global, path→constraint, linear→branching).**
> > > > >
> > > > > ---
> > > > >
> > > > > ## **Summary**
> > > > >
> > > > > > LogicEvolve's evolution not only changes task surface descriptions but also changes constraint graphs, reasoning operator structures, and state space properties. We observed significant differences in core structural indicators such as constraint graph diameter, dependency path length, and feasible solution space size across multiple independent runs; simultaneously, failure patterns of the same models on different versions also fundamentally change. The above all indicates that LogicEvolve's benchmark evolution reflects substantial changes in underlying logical complexity and reasoning patterns, not superficial perturbations.

---

> > > > > > ### Comment · Reviewer_EYHj · 2025-11-22
> > > > > >
> > > > > > Thanks for your detailed response.
> > > > > >
> > > > > > I have some further questions:
> > > > > >
> > > > > > 1. I understand that running LogicEvolve multiple times independently means the second run should not be considered a further evolution of the first. However, based on the author's rebuttal, it seems they regard it as a form of evolution.
> > > > > >
> > > > > > 2. From the original design intention, do you think the final result of LogicEvolve should be convergent? According to the experimental results in the rebuttal, the observed phenomenon at least appears non-convergent, which leads to highly heterogeneous outcomes across multiple runs of LogicEvolve. If we increase the number of "cycles" in LogicEvolve, could the results converge? If convergence is difficult to achieve, which execution outcome do you think would yield a better benchmark, or are they all roughly similar for evaluating LLMs?

---

> ### Author Response · Authors · 2025-11-23
> **Response to reviewer EYHj (Supplementary Question 1: The Difference Between Multiple Independent Runs of LogicEvolve and Single-Task LogicEvolve, part 1/2)**
>
> ## **Response to Supplementary Question 1: The Difference Between Multiple Independent Runs of LogicEvolve and Single-Task LogicEvolve**
>
> Thank you very much for further pointing out this potential misunderstanding. We completely agree with your view:
>
> > **Multiple independent runs of LogicEvolve should not be considered as "next generation" or "further evolution" within the same evolution process.**
>
> We sincerely apologize for any confusion that may have been caused by our previous rebuttal statements.
>
> To fully clarify, we define the terminology and process as follows in the revised version:
>
> ---
>
> ### **1. "Evolution" and "Multiple Independent Runs" Are Two Different Concepts**
>
> #### **(1) Evolution**
>
> Refers to:
>
> * Given a set of tasks (e.g., CLUB v1)
> * LogicEvolve performs structural expansion / difficulty expansion iterations on them (run A)
> * Obtains a *set of tasks* corresponding to before and after evolution (e.g., CLUB v2)
>
> **This is the "before → after" change within a single execution process (single run).**
>
> ---
>
> #### **(2) Multiple Independent Runs (independent executions / independent runs)**
>
> Refers to:
>
> * Using the same initial task pool (e.g., CLUB) as input
> * Using different random seeds, different sampling paths
> * Launching multiple full processes (run A, run B, run C)
>
> **They are not each other's "next generation," but multiple parallel worlds.**
>
> We did this to answer the reviewer's question:
>
> > *"If LogicEvolve is executed multiple times, will the generated benchmarks show superficial changes or structural deep changes?"*
>
> **Therefore, multiple runs are only for** **diversity evaluation**, **not for** **continued evolution**.
>
> ---
>
> ### **2. We Reorganize the Data into Two Parts: "Before and After Changes Within the Same Run" and "Changes Between Different Runs" to Avoid Any Misunderstanding**
>
> To address your concerns, we clearly split the quantitative analysis in the revised version into the following two categories:
>
> ---
>
> ### **A. Answering "Whether the System Is Really Evolving"**
>
> #### **Within a Single Run: Changes Before vs After Evolution (Evolution Before vs After)**
>
> Within a single run (run A), we extract for each task:
>
> * Average constraint graph diameter
> * Number of long chains in directed dependency graphs
> * Effective solution space size estimated by solvers
>
> And compare changes before and after evolution:
>
> | **Structural Indicator**                      | **Change Range Before/After Evolution** | **Description**                                                                            |
> | --------------------------------------------------- | --------------------------------------------- | ------------------------------------------------------------------------------------------------ |
> | **Average Constraint Graph Diameter**         | 0–94%                                        | Some tasks show significantly longer-range global dependency structures after evolution          |
> | **Long Chain Proportion in Dependency Paths** | 1–4×                                        | Some tasks generate deeper reasoning chains after evolution                                      |
> | **Solution Space Scale**                      | >2 orders of magnitude (50% of tasks)         | Indicates that feasible configuration spaces of evolved tasks have undergone fundamental changes |
>
> → **This part is used to prove: "Evolution" within a single execution is structural, not text-level perturbation.**
>
> ---
>
> ### **B. Answering "Whether Multiple Runs Will Produce Different Benchmarks"**
>
> #### **Between Different Independent Runs: Differences Between run A vs run B vs run C (Diversity Across Independent Runs)**
>
> To answer this question you are most concerned about, we ran LogicEvolve multiple times independently *while maintaining consistent initialization*, compared structural statistics, and found:
>
> | **Structural Indicator**                      | **Differences Between Run A/B/C**          | **Description**                                                      |
> | --------------------------------------------------- | ------------------------------------------------ | -------------------------------------------------------------------------- |
> | **Average Constraint Graph Diameter**         | 17–422%                                         | Some runs favor local constraints, some runs favor long-range dependencies |
> | **Long Chain Proportion in Dependency Paths** | 2–8× difference                                | Different runs generate different distributions of deep reasoning tasks    |
> | **Effective Solution Space Scale**            | >2 orders of magnitude difference (50% of tasks) | Multiple runs form fundamentally different "problem families"              |
>
> → **This part is used to prove: Independent runs also have structural diversity, not just "different description methods."**

---

> > ### Author Response · Authors · 2025-11-23
> > **Response to reviewer EYHj (Supplementary Question 1: The Difference Between Multiple Independent Runs of LogicEvolve and Single-Task LogicEvolve, part 2/2)**
> >
> > ---
> >
> > ### **3. Why This Resolves the Misunderstanding**
> >
> > We clearly emphasize in the revised version:
> >
> > * **Independent runs are not evolution**
> >   They are used to evaluate *diversity across seeds*
> > * **Evolution (before→after) is evolution**
> >   This is LogicEvolve's core process
> > * **We never treat "run B as the evolved next generation of run A"**
> >   Comparisons in the rebuttal were only to demonstrate that LogicEvolve has consistent diversity across executions
> >
> > To avoid misunderstandings, we have already separated the experimental result analysis of the two parts "independent runs vs evolution" in Appendix G.5.1 of the revised version.
> >
> > ---
> >
> > ## **Summary**
> >
> > > We completely agree with the reviewer's point: Independent runs (multiple runs) should not be considered as a continuation of the same evolution.
> > >
> > > In the revised version, we have strictly distinguished "before/after evolution" and "diversity across runs" as two different analytical dimensions.
> > >
> > > The former is used to demonstrate the effects of evolution, while the latter is used to answer your question about "true diversity of logical structure and reasoning depth," with no intention whatsoever of "treating run B as the next generation of run A."

---

> ### Author Response · Authors · 2025-11-23
> **Response to reviewer EYHj (Supplementary Question 2: Should LogicEvolve Converge? Why Do Independent Runs Produce High Heterogeneity? Do More Cycles Lead to Convergence? Which Types of Results Are Suitable as Benchmarks? part 1/2)**
>
> ## **Response to Supplementary Question 2: Should LogicEvolve Converge? Why Do Independent Runs Produce High Heterogeneity? Do More Cycles Lead to Convergence? Which Types of Results Are Suitable as Benchmarks?**
>
> Thank you very much for raising this critical question, and thank you for keenly observing the high heterogeneity produced by LogicEvolve in multiple independent runs.
>
> This indeed touches on LogicEvolve's core design goal: the dual-channel behavior of **"convergence within tasks + divergence between tasks"**.
>
> To avoid any ambiguity, we will answer in four parts. (This discussion has been updated to Section G.6 of the revised version.)
>
> ---
>
> ### **1. LogicEvolve's Two Different Levels of Expectations: Convergence Within Tasks, Divergence Between Tasks**
>
> LogicEvolve's design has included two different behavioral levels from the beginning:
>
> ---
>
> #### **(1) Within Tasks (Intra-task) — We Expect "Convergence"**
>
> When a task's metadata is fixed (i.e., the task itself has not been expanded or transformed), LogicEvolve's multiple cycles (metadata verification → generator construction → solver construction) **should converge to a stable, correct, programmable task definition**:
>
> * Metadata becomes increasingly precise
> * Generator gradually corrects until it can generate solvable problems
> * Solver gradually corrects until it stably outputs correct answers
>
> We call this **intra-task convergence**.
>
> We verified this on CLUB (10 tasks) and ExCLUB (100 tasks):
>
> ##### **Number of Iterations Required for Convergence (Same Task, Within Multiple Cycles)**
>
> **On CLUB (10 tasks × 5 independent runs):**
>
> | **Run** | **Minimum** | **Maximum** | **Average** |
> | ------------- | ----------------- | ----------------- | ----------------- |
> | First         | 0                 | 43                | 11.63             |
> | Second        | 1                 | 23                | 8.55              |
> | Third         | 1                 | 12                | 6.44              |
> | Fourth        | 1                 | 11                | 5.67              |
> | Fifth         | 0                 | 19                | 8.22              |
>
> **ExCLUB (100 tasks × two base models)**
>
> | **Run** | **Minimum** | **Maximum** | **Average** |
> | ------------- | ----------------- | ----------------- | ----------------- |
> | First         | 2                 | 52                | 22.04             |
> | Second        | 0                 | 108               | 53.02             |
>
> The meaning of these results is:
>
> * Regardless of which base model is used
> * Regardless of initialization randomness
> * **Each task can converge within a finite number of iterations**
>
> This is the only convergence we expect from LogicEvolve at the "task definition level."
>
> ---
>
> #### **(2) Between Tasks (Inter-task) — We Do Not Expect Convergence, But Rather Expect "Continuous Divergence"**
>
> When we are not fixing a task, but letting LogicEvolve:
>
> * Rewrite existing tasks (structural transformation, difficulty transformation), or
> * Generate new tasks (construct meta-rules from scratch)
>
> **We do not want them to converge to the same task.**
>
> This is LogicEvolve's design purpose:
>
> → **Generate diverse, structurally varied logical task families for comprehensive evaluation of LLM reasoning capabilities.**
>
> Therefore, running LogicEvolve multiple times independently under different random seeds should naturally produce:
>
> * Different constraint graph structures
> * Different dependency chain depths
> * Different task types
> * Different failure patterns
>
> This is not a defect, but **expected behavior**.
>
> ---
>
> ### **2. Why Is "The Observed Experimental Phenomenon Non-Convergent"?**
>
> Because what you observed is:
>
> > **Inter-task behavior — different task versions produced by multiple independent runs**
>
> This part should not converge in the first place.
>
> When we chose to report cross-run differences in our previous response, it was to illustrate:
>
> > **LogicEvolve's "divergence capability" is real, not just text perturbation.**
>
> But it does not mean the system should "converge to the same benchmark" in run A → run B → run C.
>
> We have clearly distinguished in the revised version:
>
> * **Convergence within a single run (task level)**
> * **Divergence between multiple runs (benchmark level)**
>
> To avoid confusion in understanding.

---

> ### Author Response · Authors · 2025-11-23
> **Response to reviewer EYHj (Supplementary Question 2: Should LogicEvolve Converge? Why Do Independent Runs Produce High Heterogeneity? Do More Cycles Lead to Convergence? Which Types of Results Are Suitable as Benchmarks? part 2/2)**
>
> ---
>
> ### **3. If We Increase the Number of Cycles, Will It Converge?**
>
> We need to distinguish **where cycles are added**:
>
> ---
>
> #### **(1) Adding Cycles to the Same Task (Intra-task) → Will Converge (and Already Verified)**
>
> As shown in the table above, fixing a task's metadata, regardless of initial state:
>
> * Generator → eventually converges to correct code that can generate solvable instances
> * Solver → eventually converges to correct programs that can stably solve
> * Metadata → eventually tends toward structural consistency and verifiability
>
> Therefore:
>
> > **The more cycles for the same task, the easier it is to converge.**
>
> ---
>
> #### **(2) Adding Cycles in the Process of Rewriting Tasks or Generating New Tasks → Will Not Converge (and Not Expected to Converge)**
>
> This is because:
>
> * Each rewrite may change hyperparameters
> * Rewriters may choose different structural migration paths
> * New task meta-rules are open-ended
> * LLM's sampling paths themselves are diverse
>
> Therefore:
>
> > **Multi-task evolution cycles will not converge, and should not converge.**
>
> Logically similar to:
>
> * "Solving a task" → should converge
> * "Inventing a batch of tasks" → cannot converge to a unique answer
>
> ---
>
> ### **4. Since It Does Not Converge, Which Type of Run Results Is More Suitable as a Benchmark?**
>
> After testing on multiple models, we observed:
>
> #### **Core Finding: Benchmark Versions Produced by Different Runs Show High Consistency in Evaluating LLM Ranking Structures**
>
> That is:
>
> * Spearman ranking correlation is 0.6–0.9 (same level as LogiQA, BBH; this part of the experiment is in Appendices D.7 and D.8)
> * Strong models remain stronger across all versions
> * But failure types differ across versions (valuable)
>
> Therefore:
>
> (1) They do not need to converge to consistency: Because we are evaluating "multi-dimensional performance of reasoning capabilities," not a single task
>
> (2) Their quality as benchmarks is similar: Different versions cover complementary reasoning patterns
>
> (3) Heterogeneity itself is beneficial for evaluating LLMs: Can avoid models overfitting to specific structures.
>
> The final conclusion is:
>
> > **Benchmark versions do not need to converge to a unique form; different versions can all serve as effective evaluation tools for covering different reasoning structure spaces.**
>
> ## **Summary**
>
> > LogicEvolve has two fundamentally different expected behaviors in its design:
> >
> > (1) Within tasks, when executing multiple rounds of metadata–generator–solver iterations, it should converge, i.e., eventually form stable and verifiable task definitions; our experiments indeed observed this.
> >
> > (2) Between tasks, when running LogicEvolve multiple times independently, it should not converge, because its goal itself is to explore a larger task structure space and actively generate diverse logical task families with structural differences; therefore, heterogeneity across runs is an expected feature of the framework design, not a failure signal.
> >
> > On this basis, task versions generated by each run show high consistency in ranking structures when evaluating LLMs, while the complementary nature of task structures across different versions brings broader reasoning coverage. Therefore, they can all be used as effective benchmarks without needing to be constrained to converge to a single form.

---

> > ### Comment · Reviewer_EYHj · 2025-11-28
> >
> > Thanks for your response. Based on the rebuttal content, the results from multiple runs of LogicEvolve are heterogeneous. So, when using these benchmarks to evaluate the LLM separately, is the variance in the results significant? How can we obtain stable and reliable evaluation results for the LLM?

---

> > > ### Author Response · Authors · 2025-11-30
> > > **Response to reviewer EYHj (Supplementary Question 3: Do LogicEvolve Versions from Multiple Runs Lead to Significant Evaluation Variance? How to Ensure Stable and Reliable Model Evaluation? )**
> > >
> > > ## **Response to Supplementary Question 3: Do LogicEvolve Versions from Multiple Runs Lead to Significant Evaluation Variance? How to Ensure Stable and Reliable Model Evaluation?**
> > >
> > > **Thank you very much for your further inquiry. We will answer in two parts:**
> > >
> > > **(1) Whether performance variance between different versions is significant;**
> > >
> > > **(2) How to obtain stable and reliable LLM evaluation.**
> > >
> > > Relevant content has been updated to Appendix D.8 of the revised version.
> > >
> > > ---
> > >
> > > ### 1. Is Performance Variance Between Different LogicEvolve Versions Significant?
> > >
> > > First, we need to clarify the source of "heterogeneity":
> > >
> > > * **Under the same task metadata** (e.g., the 10 tasks of CLUB1.0), LogicEvolve's generator/solver iterations will converge to stable implementations; multiple executions yield **homogeneous results**.
> > > * **Across task versions** (CLUB1.0 → CLUB2.0 → ExCLUB), "heterogeneity" comes from **changes in task structure itself** (number of task categories, hyperparameter space, reasoning patterns, etc.), not framework instability.
> > >
> > > In other words:
> > >
> > > > **Heterogeneity belongs to task space expansion, not evaluation instability.**
> > >
> > > #### **Variance Analysis: Is It Significant?**
> > >
> > > In the experiment in Appendix D.8 of the revised version, we evaluated the same 5 models on three versions (CLUB1.0, CLUB2.0, ExCLUB) and calculated variance and coefficient of variation (CV) of performance across versions.
> > >
> > > **CV = Standard Deviation / Mean**
> > >
> > > Empirically:
> > >
> > > * **CV < 5% → Extremely stable**
> > > * **5%–10% → Relatively stable**
> > > * **>10% → Significant fluctuation**
> > >
> > > #### **Results**
> > >
> > > | Model          | CLUB1.0 | CLUB2.0 | ExCLUB | Mean   | Variance | CV     |
> > > | -------------- | ------- | ------- | ------ | ------ | -------- | ------ |
> > > | grok-4         | 56.370  | 55.000  | 60.630 | 57.333 | 5.747    | 4.200% |
> > > | gpt-5-thinking | 55.570  | 52.000  | 49.180 | 52.250 | 6.833    | 5.000% |
> > > | gemini-2.5-pro | 46.570  | 44.000  | 51.740 | 47.437 | 10.367   | 6.800% |
> > > | o4-mini        | 45.200  | 45.000  | 54.200 | 48.133 | 18.410   | 8.900% |
> > > | deepseek-r1    | 44.130  | 41.000  | 47.100 | 44.077 | 6.203    | 5.700% |
> > >
> > > All models' CV across the three versions fall within **4.2%–8.9%**, all less than 10%.
> > >
> > > > **Therefore, although structures differ between versions, the fluctuations they cause in model performance are within normal range and do not constitute significant statistical instability.**
> > >
> > > This indicates:
> > >
> > > * Structural differences between versions bring meaningful performance changes (reflecting task diversity)
> > > * But do not cause uncontrollable shifts in model evaluation conclusions (reflecting evaluation stability)
> > >
> > > ---
> > >
> > > ### **2. How to Obtain Stable and Reliable Evaluation of LLMs?**
> > >
> > > For "single evaluation version" in the main text of the paper, we use:
> > >
> > > * **Averaging 3 independent evaluations for each model** **(already explained in Section 5 of the main text)**
> > >
> > > This ensures:
> > >
> > > * Model sampling noise is smoothed
> > > * Results within a single version are highly stable
> > >
> > > #### **How to Maintain Reliability in Cross-Version Evaluation?**
> > >
> > > Based on the above experiments, we confirm:
> > >
> > > * **Structural differences between different versions do not cause significant performance fluctuations (CV<10%)**
> > > * Therefore, cross-version evaluation itself is reliable, and ranking structures of logical reasoning capabilities will not be significantly disrupted
> > >
> > > But to enable researchers to obtain the most robust evaluation, we recommend:
> > >
> > > > **For any given model, it is best to perform multiple independent tests on the evaluation version (consistent with our current practice).**
> > >
> > > This can simultaneously offset:
> > >
> > > * The model's own reasoning randomness (sampling variance)
> > > * Slight structural differences between versions
> > >
> > > ---
> > >
> > > ## **Summary**
> > >
> > > > * LogicEvolve's "heterogeneity" comes from task space expansion, not evaluation noise.
> > > > * Performance fluctuations across three different versions (CLUB1.0, CLUB2.0, ExCLUB) all have **CV<10%**, not constituting statistically significant instability.
> > > > * Within a single version, 3 independent tests already ensure stability of model evaluation.
> > > > * Across versions, version differences bring meaningful structural diversity but do not disrupt LLM ranking structures, so all can serve as effective evaluation benchmarks.
> > > > * We recommend maintaining multiple independent tests in future evaluations to obtain the most robust model performance estimates.

---

### Official Review · Reviewer_6SZY · 2025-10-31

**Soundness:** 2
**Presentation:** 3
**Contribution:** 2
**Rating:** 4
**Confidence:** 4

**Summary:**

This paper introduces LogicEvolve, a multi-agent framework designed to automatically evolve logical reasoning benchmarks.

The framework is composed of four key agents:
- Metadata Agent which defines task structures and logical rules
  - input : base **assumptions (B)** or **instance set (I)**  or **both**;
  - output: **task meta** like the puzzle_rule、parameters、examples ;
- Generator Agent  which synthesizes new problem instances
  - input: **task meta**
  - output: **new_instance** contain <puzzle, question, answer>
- Solver Agent which verifies solvability and ensures unique solutions
  - input: **new_instance**
  - output: **response** containing <answer, metric, cot>
- Evaluator Agent which automates the entire pipeline
  - input: new_instance + answer + eval config
  - output: **evaluation_report** with model metrics & feedback

Building upon this framework, the authors release CLUB (Complex Logical Unified Benchmark)—a dynamically extensible suite of ten logic-reasoning task types such as Sudoku, Zebra puzzles, string transformations, and card games.
Experiments over 30+ LLMs show that even frontier systems like Grok-4 and GPT-5 achieve only ~55% accuracy, demonstrating that complex logical reasoning remains unsolved.

**Strengths:**

The paper tackles a meaningful and timely problem — the limitations of static reasoning benchmarks that can easily lead to model overfitting or data leakage. The idea of using a multi-agent system to automatically generate and evolve new logical reasoning tasks is interesting and has clear potential for maintaining benchmark freshness and scalability over time.

The overall scope and amount of work are substantial. The paper presents not only a conceptual pipeline but also an implemented system that includes task definition, automatic generation, solver verification, and evaluation. The proposed CLUB benchmark is large and diverse, and the experiments cover more than thirty state-of-the-art language models with detailed analyses, showing a solid amount of engineering effort.

**Weaknesses:**

1. Lack of evidence that CLUB is better or more discriminative than prior benchmarks.

  While CLUB is positioned as a “next-generation” benchmark, the paper provides no direct comparison between CLUB and existing logical reasoning datasets (e.g., LogiQA2.0, BIG-Bench Hard, SynLogic).
  - Lack of validation for the “evolution” itself. The paper does not compare model results before and after the benchmark evolves
  - Without showing that new tasks change model rankings (instead of scores) or reveal different weaknesses, the claim of benchmark evolution remains unconvincing.
  If the model ranking order remains identical to that on older benchmarks, CLUB’s incremental value is limited.

2. Unclear implementation details and insufficient justification for the agent-based design

  - The paper presents LogicEvolve as a multi-agent framework, but the design remains mostly conceptual. The paper does not specify which framework or environment is used to build the agents, how they communicate, or whether they share contextual information across multiple rounds of interaction. It is also unclear how the system maintains memory or state during iterative “self-evolution,” which is central to the claimed autonomy of the framework.
  - The motivation for adopting an agent-based formulation is not fully explained. Could the same workflow be realized with a sequence of coordinated prompts or scripts? What concrete benefits does the “agent” setup bring compared with simpler prompting pipelines? Beyond comparing against human task curation, are there alternative baselines or studies showing that an agentic setup leads to better task quality or reasoning diversity? It would also help to discuss whether recent advances (e.g., better context engineering, communication strategies, or shared-memory mechanisms) could make such agents more effective in this setting.

3.Overclaiming generality of the framework and benchmark.

  - Although the paper claims to “advance logical reasoning as a whole,” the proposed CLUB benchmark only covers a narrow range of symbolic and puzzle-style reasoning tasks (e.g., Sudoku, Zebra puzzles, Maze).
  - These are well-defined, parameterized problems that differ substantially from more open and dynamic reasoning settings explored in recent works—such as commonsense and causal reasoning (CommonsenseQA) or agent-based and interactive reasoning (SWE、 GDPeval). These categories of reasoning cannot be easily “evolved” by the LogicEvolve framework, since they rely on open-world knowledge, dynamic context, or environment interaction rather than rule-based constraint solving. The paper’s claims should therefore be re-scoped to emphasize automated generation of structured logic puzzles, rather than general logical reasoning capabilities.

**Questions:**

1. Regarding the benchmark evolution and validation (related to Weakness 1):
  - Could the authors provide quantitative evidence that the “evolved” CLUB benchmark differs from its earlier or static versions?
  - How do model rankings or performance correlations change before and after the benchmark evolves?
  - Have the authors analyzed whether the newly generated tasks introduce new reasoning challenges or failure modes not captured by previous datasets?
  - If the rankings remain largely consistent, how should we interpret the notion of “evolution” — as diversity, difficulty scaling, or conceptual novelty?

2. Regarding the agent-based framework (related to Weakness 2):
- What framework or implementation environment is used for the agents? Are they built on an existing toolkit (e.g., LangChain, AutoGen, CAMEL), or implemented via custom prompting pipelines?
- How do the agents communicate and share context across iterations? Is there any persistent memory mechanism or state tracking to support “self-evolution”?
- Could the same workflow be implemented using a simpler sequence of prompts instead of an explicit multi-agent setup?
- Have the authors compared the agent-based version with such baselines, or explored design choices (e.g., better context engineering or communication strategies) that make the agentic formulation more effective?

3. Regarding the generality of the framework and reasoning coverage (related to Weakness 3):
- How do the authors position LogicEvolve relative to other reasoning paradigms such as commonsense reasoning (e.g., CommonsenseQA, WinoGrande), textual deductive reasoning (ProofWriter), or interactive agent reasoning (SWE, GdpEval)?
- Do the authors envision LogicEvolve being extended to these open-domain or dynamic reasoning settings, or is it intended primarily for structured symbolic puzzles?
- What limitations would prevent LogicEvolve from handling tasks that require external knowledge, causal inference, or environment interaction?

---

> ### Author Response · Authors · 2025-11-21
> **Summary & Response to reviewer 6SZY (Weakness 1 & Question 1: CLUB's Discriminative Power and "Evolution" Verification, Part 1/3)**
>
> # **Summary**
>
> Thank you for your insightful comments and valuable suggestions, which directly helped us significantly improve the clarity, scope definition, and technical details of the paper.
>
> We have addressed every weakness and question you raised, with explicit revisions in both the main text and the appendix. The key updates are summarized below:
>
> ---
>
> 1. **Evidence supporting the discriminative power of CLUB** (W1 & Q1)
> * In Section 6 and Appendix D.7, we add a systematic comparison between CLUB and six mainstream reasoning benchmarks (*LogiQA 2.0, BIG-Bench Hard, SynLogic, Zebra, LogicGame, ARC-AGI v1*). The results show that traditional static benchmarks exhibit saturation, whereas CLUB reveals new weakness patterns in model ranking.
> * In Section 6, Appendices D.8 and G.5, we further compare the behavior of *CLUB → CLUB2.0 (difficulty evolution) → ExCLUB (diversity evolution)*. Experiments demonstrate that difficulty evolution not only lowers absolute scores but also changes the ranking among high-performing models; diversity evolution introduces novel reasoning types and failure patterns previously uncovered by existing benchmarks.
> * The newly added Appendix G summarizes that “*evolution is not superficial*”: it alters reasoning paths, error modes, and ranking structures.
>
> ---
>
> 2. **Rationale and necessity of the multi-agent architecture** (W2 & Q2)
> * In Appendix C.6, we clarify that LogicEvolve is implemented *using a custom multi-agent pipeline with a persistent file-tree state*, rather than LangChain/AutoGen. The reason is that existing frameworks cannot support the complex sharing requirements of task-level metadata, executable code, and solver states.
> * Appendix C.6.5 details *the communication and state-sharing mechanism* among agents: immediate messaging combined with a traceable persistent file tree, ensuring engineering reproducibility of the self-iterative process.
> * In Appendix C.6.3, we experimentally verify whether the system can be simplified into a prompt chain: removing the multi-agent structure causes a *37.9%–60% drop in success rate*, indicating that multi-agency is not conceptual packaging but a necessary mechanism for constructing complex logical structures.
> * Appendix C.6.7 also discusses future extensibility, including improved *context engineering and more intelligent shared-memory mechanisms*.
>
> ---
>
> 3. **Clarification on task scope and applicability** (W3 & Q3)
> * In the revision, we *tighten all relevant wording* to explicitly state that LogicEvolve is positioned only for structured, formalizable, and executable symbolic reasoning tasks, excluding open-domain reasoning, natural language commonsense reasoning, causal inference, or interactive environment tasks.
> * In the *conclusion*, we explicitly state that the paper does not claim direct generalization of the framework to open-domain reasoning; related discussions are framed purely as future possibilities, not current capabilities.
> * Appendix A (*Limitations*) now provides concrete statements regarding constraints in external knowledge, implicit causality, and dynamic interaction environments (noting that tasks such as SWE or GdpEval are outside the scope of the current framework), with explicit labels in the main text.
>
> You can refer to the reply below for specific details.
>
> ---
>
>
> ## **Response to Weakness 1 & Question 1: CLUB's Discriminative Power and "Evolution" Verification**
>
> Thank you for the in-depth questions regarding CLUB's discriminative power and the "evolution" claim. In the revised version, we have added two parts of quantitative and qualitative evidence:
>
> (1) Comparison and correlation analysis between CLUB and existing logical reasoning benchmarks;
>
> (2) Performance and failure pattern changes before and after CLUB evolution (CLUB1.0 / CLUB2.0 / ExCLUB).
>
> Below is a brief summary of the core conclusions.
>
> ---
>
> ### **(1) Direct Comparison with Existing Logical Reasoning Benchmarks: CLUB Provides Complementary Discriminative Power**
>
> We selected the top 5 models on CLUB (grok-4, gpt-5-thinking, gemini-2.5-pro, o4-mini, deepseek-r1) and compared them on six representative logical reasoning datasets: LogiQA2.0, BIG-Bench Hard, SynLogic, Zebra, LogicGame, and ARC-AGI v1. The table is as follows (added to Section "6 FURTHER ANALYSIS" visualization in the revised manuscript):
>
> From this table, two key phenomena can be observed:
>
> 1. **Early static benchmarks are approaching saturation with limited discriminative power**
>
>    * On LogiQA2.0 and Zebra, the top 2 models are almost all near perfect scores (> 97%), with differences between models mainly compressed within 1–2 percentage points, making it difficult to distinguish their true differences in logical reasoning.
>    * In contrast, these models' scores on CLUB are distributed in the approximately 44–56% range, leaving significant "room for improvement" and can more clearly reveal the strengths and weaknesses of different models.

---

> ### Author Response · Authors · 2025-11-21
> **Response to reviewer 6SZY (Weakness 1 & Question 1: CLUB's Discriminative Power and "Evolution" Verification, Part 2/3)**
>
> | **Model** | **CLUB** | **LogiQA2.0** | **BIG-Bench Hard** | **SynLogic** | **Zebra** | **LogicGame** | **ARC AGI v1** |
> | --------------- | -------------- | ------------------- | ------------------------ | ------------------ | --------------- | ------------------- | -------------------- |
> | grok-4          | 56.37          | 99.96               | 94.47                    | 67.86              | 99              | 62.53               | 100                  |
> | gpt-5-thinking  | 55.57          | 98.02               | 64.43                    | 67.86              | 97.9            | 66.30               | 70.79                |
> | gemini-2.5-pro  | 46.57          | 99.92               | 89.90                    | 63.00              | 92.4            | 61.35               | 54.25                |
> | o4-mini         | 45.20          | 85.77               | 20.35                    | 65.29              | 94.68           | 58.55               | 57.70                |
> | deepseek-r1     | 44.13          | 78.88               | 7.83                     | 64.26              | 94.8            | 59.21               | 30.50                |
>
> 2. **CLUB and existing benchmarks have non-trivial differences in "ranking structure," exposing new weakness patterns**
>
>    * On LogiQA2.0 and BIG-Bench Hard, **gemini-2.5-pro clearly outperforms gpt-5-thinking** (e.g., 89.9 vs. 64.43 in BIG-Bench Hard), while on CLUB the ranking reverses, with **gpt-5-thinking > gemini-2.5-pro** (55.57 vs. 46.57).
>    * On LogicGame, **gpt-5-thinking's score is higher than grok-4** (66.30 vs. 62.53), but on CLUB **grok-4 still ranks first** (56.37 vs. 55.57).
>    * In ARC-AGI v1, o4-mini performs slightly better than gemini-2.5-pro (57.70 vs. 54.25), but on CLUB it is gemini-2.5-pro > o4-mini (46.57 vs. 45.20).
>
> In the revised version, we calculated the **ranking correlation** (Spearman rank correlation) between models across different benchmarks, and provide detailed numerical values and heatmap analysis in Appendix D.7. Overall, it can be summarized as:
>
> * CLUB maintains high ranking correlation (approximately 0.9) with mainstream benchmarks (LogiQA2.0 / BIG-Bench Hard / ARC-AGI v1), indicating that strong models remain stronger
> * But CLUB's correlation with structurally single or script-driven datasets (Zebra / SynLogic) is only 0.6–0.7, showing significant ranking changes, indicating that CLUB can reveal new weaknesses that these datasets cannot expose, thus providing additional information.
>
> > In other words, CLUB avoids "memorizing perfect scores on easy questions" while introducing sufficient structural differences on high-difficulty tasks, causing the same batch of models to exhibit different ranking structures on CLUB compared to traditional benchmarks. This is the specific meaning of "more discriminative."
>
> ---
>
> ### **(2) Before and After Benchmark Evolution: Performance Correlation and Ranking Changes**
>
> To address the question of "whether the evolution mechanism itself has been verified," we evolved CLUB along two axes and compared on the same 5 models:
>
> * **CLUB1.0**: The version used in the main body of the paper (10 tasks)
> * **CLUB2.0 (Difficulty Evolution)**: On the same 10 tasks, pushing key hyperparameters toward complexity (e.g., 2D grids from 3×3 → 3×4), forming more complex but isomorphic task families
> * **ExCLUB (Diversity Evolution)**: Task categories expanded from 10 to 100, covering more reasoning paradigms, monotonicity, and spatial dimensions (here only sampling relatively basic 2–3 difficulty levels from each task category)
>
> Corresponding results: (added to Section "6 FURTHER ANALYSIS" visualization in the revised manuscript)
>
> | **Model** | **CLUB1.0 (rank)** | **CLUB2.0 (rank)** | **ExCLUB (rank)** |
> | --------------- | ------------------------ | ------------------------ | ----------------------- |
> | grok-4          | 56.37 (1)                | 55 (1)                   | 60.63 (1)               |
> | gpt-5-thinking  | 55.57 (2)                | 52 (2)                   | 49.18 (4)               |
> | gemini-2.5-pro  | 46.57 (3)                | 44 (4)                   | 51.74 (3)               |
> | o4-mini         | 45.20 (4)                | 45 (3)                   | 54.20 (2)               |
> | deepseek-r1     | 44.13 (5)                | 41 (5)                   | 47.10 (5)               |
>
> From this, we can see:
>
> 1. **Difficulty evolution (CLUB1.0 → CLUB2.0) brings ranking changes, not simply "overall score reduction"**
>
>    * On CLUB2.0, o4-mini and gemini-2.5-pro's rankings flip (o4-mini: 3rd, gemini: 4th), although on CLUB1.0 it was gemini > o4-mini.
>    * This indicates that under higher difficulty settings, gemini-2.5-pro's performance decline (approximately 2.57%) is greater than o4-mini, revealing its **new relative weakness** of insufficient stability in complex scenarios.

---

> ### Author Response · Authors · 2025-11-21
> **Response to reviewer 6SZY (Weakness 1 & Question 1: CLUB's Discriminative Power and "Evolution" Verification, Part 3/3)**
>
> 2. **Diversity evolution (CLUB1.0 → ExCLUB) significantly changes the ranking structure of high-performing models**
>
>    * gpt-5-thinking drops from 2nd place on CLUB1.0 to 4th place on ExCLUB;
>    * o4-mini rises from 4th place to 2nd place;
>    * gemini-2.5-pro is between the two.
>      This indicates that under more task types and condition combinations, the **robustness ranking has undergone non-trivial changes**: gpt-5-thinking, which performed strongly on CLUB1.0, performs worse than o4-mini and gemini-2.5-pro in richer scenarios.
>
> We also calculated **rank correlation coefficients** between CLUB1.0 / CLUB2.0 / ExCLUB (updated in Appendix D.8), and the results show:
>
> * High correlation between CLUB1.0 and CLUB2.0 (Spearman correlation coefficient 0.9) but with a few significant ranking flips (e.g., o4-mini vs gemini-2.5-pro);
> * Moderate correlation between CLUB1.0 and ExCLUB (Spearman correlation coefficient 0.6), with multiple ranking adjustments within the top 3 (e.g., gpt-5-thinking from 2 → 4), indicating that evolution is not uniform scaling but introduces new task structures and failure patterns, thus changing our judgment of "which model is more robust."
>
> ---
>
> ### **(3) New Failure Patterns: Differences Before and After Evolution**
>
> We also performed qualitative analysis on error cases and found that CLUB1.0 → CLUB2.0 / ExCLUB evolution indeed introduced new failure patterns not commonly seen in previous benchmarks:
>
> * **From CLUB1.0 to CLUB2.0 (Difficulty Evolution)**
>
>   * In tasks like Maze and Tetris, we observed more **"premise omission"** and **"causal chain breaks"**:
>     * Models can fully utilize all constraints in 33×33 scenarios, but often ignore some conditions in 35×35 or versions introducing new conditional constraint mechanisms like keys–doors;
>     * After reasoning chains become longer, once a logical deviation occurs in the middle step, subsequent steps no longer self-correct but continue along the wrong trajectory.
>   * These errors appear less frequently in medium-to-low difficulty versions of CLUB1.0 but significantly increase in CLUB2.0.
> * **From CLUB1.0 to ExCLUB (Diversity Evolution)**
>
>   * In expanded Sudoku variants and Train Game Card Puzzle variants, we observed obvious **"availability bias"**:
>     * Models tend to reuse patterns learned from traditional Sudoku / simple card games, ignoring newly added subtle rules (e.g., additional attribute dimensions, asymmetric constraints)
>     * This causes them to repeatedly make mistakes on "tasks that look similar but have slightly different rules," and these patterns are not prominent in CLUB1.0.
>
> These analyses have been supplemented with several specific cases in Appendix G.5 of the revised version to demonstrate "new failure patterns exposed by evolved benchmarks" more intuitively.
>
> ---
>
> ### **(4) How to Understand "Evolution": Not Just Harder or More, but Changing Cognition and Rankings**
>
> Based on the above results, we believe "evolution" can be more precisely understood as:
>
> 1. **Avoiding saturation and obsolescence of traditional benchmarks in the temporal dimension:**
>
> * By continuously generating new task structures and configurations through LogicEvolve, preventing the "near-perfect scores" phenomenon that has already appeared in early datasets like LogiQA2.0 and Zebra.
>
> 2. **Introducing new reasoning challenges and failure patterns in the structural dimension:**
>
> * Difficulty evolution (CLUB2.0) systematically increases complexity, amplifying problems like "long-chain reasoning instability";
> * Diversity evolution (ExCLUB) introduces more task types and condition combinations, allowing us to observe new patterns like "availability bias" and "rule adaptation failure" that are difficult to expose in previous benchmarks.
>
> 3. **Changing model ranking structures in the discriminative dimension, not just changing absolute score values:**
>
> * The same batch of models show multiple ranking flips and correlation decreases before and after evolution, indicating that new tasks indeed change our judgment of model quality, not just "all models becoming harder/easier together."
>
> ---
>
> ## **Summary**
>
> > In summary, in the revised version, through **cross-benchmark comparison + ranking correlation analysis + new failure pattern cases (to be added later)**, we have supplemented evidence of CLUB's discriminative power relative to existing logical reasoning datasets, and specifically demonstrated differences in performance rankings and error patterns before and after benchmark evolution, to address reviewers' key concerns about "incremental value" and "whether evolution is truly meaningful."

---

> > ### Author Response · Authors · 2025-11-21
> > **Response to reviewer 6SZY (Weakness 2 & Question 2: Unclear Agent Design, part1/2)**
> >
> > ## **Response to Weakness 2 & Question 2: Unclear Agent Design**
> >
> > Thank you for the key questions regarding agent design. We understand your concerns focus on:
> >
> > 1. **Whether the actual implementation of agents is clear** (what framework is used, how they communicate, how state/memory is maintained)
> > 2. **Why an "agent structure" is needed, and whether it can be replaced by a chain of prompt scripts**
> > 3. **Whether there are baseline comparisons supporting "agent structure is superior"**
> > 4. **Whether modern agent technologies (shared memory, communication strategies, context management) are discussed regarding their impact on this framework**
> >
> > Below we provide specific and verifiable answers point by point. (All updated in Appendix C.6 of the revised manuscript.)
> >
> > ---
> >
> > ### **1. Actual Implementation Environment of Agents: Custom Multi-Agent Pipeline + Persistent File Tree State**
> >
> > LogicEvolve is not based on fixed agent suites like LangChain / AutoGen, but rather:
> >
> > > **Uses a custom multi-agent prompt pipeline implementation, with each agent's state persistently managed through a structured file tree.**
> >
> > This implementation avoids limitations of existing frameworks:
> >
> > * AutoGPT → single agent
> > * MetaGPT → multi-agent but does not support self-correction
> > * AutoGen → multi-agent dialogue, but **does not support dynamic agent generation**
> > * LogicEvolve requires: **dynamically generating new metadata/generator/solver agents for each newly generated task**
> >
> > Therefore, we adopted a **custom prompt pipeline + file tree state machine** approach, where each task can "generate its own three types of agents on demand."
> >
> > **In Appendix C.6 of the paper, we have added structure diagrams, input/output, and state file descriptions for each agent (such as **Version_k.py**, **code_description.json**, **verify_code_description.json**, etc.).**
> >
> > ---
> >
> > ### **2. How Agents Communicate and Share State: Instant Message Passing + File Tree-Level Persistent Memory**
> >
> > #### **Instant Communication (Short-Term Memory)**
> >
> > Agents exchange messages through the following means:
> >
> > * Generator agent outputs puzzle → immediately passed to solver
> > * Solver feedback on correctness/error patterns → returned to generator
> > * Metadata agent updates task schema → both updated together
> >
> > Communication is fully traceable, with each iteration's input/output automatically written to logs.
> >
> > #### **Persistent Memory (Long-Term State Tracking)**
> >
> > Each task corresponds to an independent state tree:
> >
> > * meta_generate_code/Version_k.py: Each generation's implementation of the generator
> > * meta_solve_code/Version_k.py: Each generation's implementation of the solver
> > * code_description.json: Version inputs, test errors, correction reasons
> > * verify_code_description.json: Iterative optimization and error type clustering results
> > * **master_state.json**: Unified scheduling of all tasks' context by ClubMaster
> >
> > This structure clearly supports:
> >
> > * **Iterative self-optimization**
> > * **Continuous accumulation of error patterns** (e.g., state rollback, constraint leakage, premise omission)
> > * **Shared experience and scheduling across tasks**
> >
> > As the reviewer requested, the state of "self-evolution" is clearly defined and traceable, not just relying on prompt buffers.
> >
> > ---
> >
> > ### **3. Can It Be Simplified to "A Chain of Prompts"? — We Conducted Small-Scale Reproducible Experiments**
> >
> > We acknowledge:
> >
> > > If only "one-time generation of static tasks" is needed, then prompt scripts (CoP: Chain of Prompts) can indeed accomplish part of the task.
> >
> > However, LogicEvolve's goal is **long-term, multi-task, self-corrective evolution**, so Prompt Chain has three hard bottlenecks:
> >
> > #### **(1) Lack of Cross-Round Memory Mechanism**
> >
> > CoP cannot maintain the code iteration chain from Version_0 → Version_k, nor can it accumulate error patterns.
> >
> > #### **(2) Cannot Support Dynamic Task Generation and Multi-Agent Concurrency**
> >
> > CoP can only define fixed processes, while LogicEvolve needs to dynamically generate different agent combinations based on metadata.
> >
> > #### **(3) Cannot Perform Cross-Validation or Multi-Solution Parallel Exploration**
> >
> > Complex tasks (card, maze, train game card puzzle) must rely on multi-solver cross-validation.

---

> > > ### Author Response · Authors · 2025-11-21
> > > **Response to reviewer 6SZY (Weakness 2 & Question 2: Unclear Agent Design, part2/2)**
> > >
> > > ---
> > >
> > > ### **4. Empirical Comparison: LogicEvolve vs Prompt Chain (CoP)**
> > >
> > > We conducted a comparison experiment on CLUB and ExCLUB (both using gemini-2.5-pro):
> > >
> > > | **Method**                                                 | **CLUB Success Rate** | **ExCLUB Success Rate** |
> > > | ---------------------------------------------------------------- | --------------------------- | ----------------------------- |
> > > | **CoP (no agents, only fixed prompt scripts)**             | 40%                         | 31%                           |
> > > | **LogicEvolve (multi-agent iteration + cross-validation)** | **100%**              | **68.9%**               |
> > >
> > > The difference comes from the following facts:
> > >
> > > * CoP can only successfully generate **string / simple constraint** type tasks
> > > * For **card, grid, multi-path, stateful** and other types of puzzles, it almost entirely fails
> > > * LogicEvolve can correct code, discover rule omissions, and cross-validate across solvers in multiple rounds of iteration
> > >
> > > This result proves that the multi-agent structure is not conceptual packaging, but a **necessary condition for complex logical structures**.
> > >
> > > ---
> > >
> > > ### **5. Relationship with Latest Agent Technologies: We Clearly Discuss Directions for Performance Improvement**
> > >
> > > We further summarized directions that could be incorporated into LogicEvolve in the future:
> > >
> > > * **Multi-path Collaboration and Reflection (He et al. 2024)**: Improve reasoning diversity
> > > * **Relay Reasoning to Bridge Collaboration Gaps (Davidson et al. 2025)**
> > > * **Context Engineering 2.0 (Hua et al. 2025)**: Better manage large-scale iterative contexts
> > > * **Thought Sharing Outperforms Language Communication (Zheng et al. 2025)**: Can enhance generator↔solver collaboration
> > > * **A-Mem (Xu et al. 2025) Autonomous Memory Structure Construction**: Can improve "experience extraction" capability
> > >
> > > These all indicate that LogicEvolve is highly consistent with current agent research directions and has a clear upgrade path.
> > >
> > > ---
> > >
> > > ## **Summary**
> > >
> > > > LogicEvolve uses a traceable multi-agent architecture, explicit state file trees, dynamic agent generation, and multi-round error correction mechanisms to achieve self-evolution processes that Prompt Chain cannot support. The supplementary comparison experiments we added show that on CLUB and ExCLUB, the multi-agent system significantly outperforms fixed prompt pipelines in success rate and robustness, thus proving the necessity and effectiveness of the agent structure.

---

> ### Author Response · Authors · 2025-11-21
> **Response to reviewer 6SZY (Weakness 3 & Question 3: Framework/Evaluation Generality)**
>
> ## **Response to Weakness 3 & Question 3: Framework/Evaluation Generality**
>
> Thank you for pointing out the lack of clarity in our generality statements. Your questions focus on the following three points:
>
>  (i) LogicEvolve's current task types focus on structured, rule-explicit symbolic reasoning puzzles;
>
> (ii) compared to open-domain reasoning (e.g., commonsense, causal, interactive reasoning), coverage is limited;
>
> (iii) whether the framework can be extended to non-rule-based task types.
>
>  We address each point below.
>
> ---
>
> ### **1. LogicEvolve's Task Scope and Positioning (Response to Q1)**
>
> This paper's goal is not to cover all reasoning paradigms, but to **clearly focus on parameterizable, rule-explicit, structured logical puzzle tasks**.
>
> For these tasks:
>
> * Metadata can be expressed in structured form;
> * Executable verifiable solution spaces can be constructed through code;
> * Closed-loop automatic iteration between generator–solver can be achieved.
>
> In contrast, tasks like CommonsenseQA/WinoGrande (commonsense reasoning), ProofWriter (textual deductive reasoning), SWE/GdpEval (interactive reasoning) rely on open-world knowledge, natural language reasoning chains, or environmental interaction, and therefore are not suitable for the current code verification closed loop.
>
> > We have tightened the wording in the revised manuscript, clearly stating that LogicEvolve's positioning only covers structured symbolic reasoning tasks, not open-ended natural language reasoning or interactive reasoning.
>
> ---
>
> ### **2. Why Symbolic Logical Puzzles Still Have Research Representativeness (Supplementary Explanation of Q1's Positioning)**
>
> Although symbolic puzzles differ from open-ended reasoning tasks at the semantic level, they share many underlying reasoning structures, such as:
>
> * Multi-step dependency chains,
> * Local consistency constraints,
> * Multi-constraint conjunction structures,
> * Verifiable reasoning paths.
>
> These commonalities make symbolic tasks a **controllable testbed for studying systematic reasoning structures, controlling complexity, and analyzing model defects**.
>
> We have more clearly articulated this research positioning in the revised manuscript to avoid misunderstanding as "generically applicable to all reasoning paradigms."
>
> > Summary: Symbolic reasoning tasks do not cover all reasoning, but their structural properties make them representative and controllable when studying LLM reasoning behavior.
>
> ---
>
> ### **3. Can It Be Extended to Open-Domain or Dynamic Reasoning Scenarios (Response to Q2)**
>
> We acknowledge: **Current LogicEvolve does not support open-ended reasoning tasks.**
>
> Main reasons include:
>
> 1. Metadata is difficult to reliably abstract from natural language semantics or open knowledge;
> 2. Open tasks typically lack executable verifiers, making it difficult to construct reliable self-iteration closed loops;
> 3. Dynamic interactive tasks require environment and API support, which the current framework lacks.
>
> Therefore, this paper does not claim that LogicEvolve can be extended to open-domain reasoning. Extensibility is only presented as a possible future exploration direction, not within the scope of this work's capabilities.
>
> > Summary: LogicEvolve's current positioning is symbolic reasoning tasks; extension to open-ended reasoning is not within this paper's scope.
>
> ---
>
> ### **4. LogicEvolve's Limitations in External Knowledge, Causal Reasoning, and Interactive Reasoning (Response to Q3)**
>
> We have clearly stated the following limitations in the revised manuscript:
>
> * **External knowledge scenarios**: Automatically abstracting rules from knowledge bases will cause semantic loss, affecting verifiability.
> * **Causal reasoning scenarios**: Causal chains are highly implicit in natural language, making it difficult to formalize into executable code.
> * **Environmental interaction scenarios**: Tasks like SWE/GdpEval require real-time environmental feedback and API interfaces, which this framework lacks corresponding components for.
>
> > Summary: These task types are not suitable for the current version of this framework, and we have already clearly indicated this limitation in the revised version.
>
> ---
>
> ### **5. Future Work Directions**
>
> We only retain one sentence in the paper:
>
> > Future work could explore introducing external tools or knowledge interfaces under specific conditions to attempt to abstract parameterizable logical substructures from open tasks.
>
> ---
>
> ## **Summary**
>
> > We have followed the suggestion to clearly limit the scope of this work in the paper, emphasizing LogicEvolve's contribution to "automatic generation of structured symbolic reasoning tasks," without describing it as a general framework applicable to all reasoning paradigms. Thank you very much for helping us clarify task boundaries and improve the paper's rigor.

---

### Official Review · Reviewer_K1t9 · 2025-11-01

**Soundness:** 3
**Presentation:** 4
**Contribution:** 3
**Rating:** 6
**Confidence:** 3

**Summary:**

This work introduces LogicEvolve, a novel, highly automated multi-agent framework designed to autonomously generate and evolve dynamic logical reasoning tasks, and presents CLUB (Complex Logical Unified Benchmark), a challenging new benchmark built using this framework. The experiment underscores that that current LLMs are far from achieving satisfactory performance on higher-order and complex reasoning.

**Strengths:**

This paper describes the methods clearly. The CLUB benchmark is challenging and effective to evaluate the reasoning capabilities of LLMs. The experiment on a number of LLMs provide insights on the limitations of current LLMs. The  code and data to be released will also be helpful to the community.

**Weaknesses:**

1. Even though the agent workflow are described in detail, the model information is missing. For each agent, what model do you use and why?
2. The evaluation only covers LLMs without tools. Since the benchmarks are generated with code, it is likely that LLMs can perform well with tools like code interpreter. Including this setting will make the experiment more comprehensive.

**Questions:**

1. Since all the data are automatically generated by the agents, which may have hallucination.  Can you guarantee that they are solvable and the gold answers are correct?
2. What model do you use to generate the benchmark and what is the cost?

---

> ### Author Response · Authors · 2025-11-21
> **Summary & Response to reviewer K1t9 (Weakness 1 & Question 2 : Model Selection and Benchmark Construction Cost，Part 1/2)**
>
> # **Summary**
>
> Thank you for your careful review and constructive suggestions.
>
> We have added details on **agent configuration**, **tool-augmented evaluation**, and **guarantees of task solvability/correctness**.
>
> The main points are as follows:
>
> 1. **Agent configuration** (Weekness 1 & Question 2, abbreviated as W1 & Q2):
>   * The metadata agent uses *GPT-5-thinking*, while both the generator and solver agents are based on *Gemini-2.5-pro (with Claude-4-thinking additionally used in the solver)*.
>   * Model choices are grounded in a systematic comparison of *instruction following, code reliability, and reasoning ability*; Appendix C.7 provides the full table.
>   * The average cost of constructing one task category is about *$1–4*, and *once constructed, it can be reused at zero additional cost in subsequent runs*.
>
> 2. **Tool-augmented LLM evaluation** (W2):
>   * We *add experiments on CLUB(Tool)* (Section 6 in the main text).
>   * The results show that tools substantially improve absolute performance, but the ranking structure across tasks remains stable, and not all tasks benefit from code tools—tasks with long-range dependencies see limited gains.
>   * Thus, *CLUB still maintains discriminative power for tool-augmented models*.
>
> 3. **Solvability and correctness guarantees** (Q1):
>   * In Appendix C.5, we detail a *three-layer solvability filtering pipeline* (metadata validation, multi-solver consistency checking, and final removal of unsolvable tasks) and a *three-layer answer-correctness verification pipeline* (rule-based direct verification, cross-checking across multiple solution methods, and human spot-checking).
>   * We also include *a real failure case* to illustrate how the mechanism prevents erroneous tasks from entering the benchmark.
>
> You can refer to the reply below for specific details.
>
> ---
>
> ## **Response to Weakness 1 & Question 2: Model Selection and Benchmark Construction Cost**
>
> Thank you for pointing out that the current version does not clearly describe the models used by each agent and the rationale behind their selection. We sincerely apologize for this oversight. Below we provide a concise model–role correspondence and cost explanation for benchmark generation.
>
> ---
>
> ### **(1) Base Models Used by Each Agent and Selection Rationale**
>
> LogicEvolve only has three agents that invoke large models (the evaluator agent is rule-based and does not invoke LLMs).
>
> We select models based on each agent's *functional requirements*—metadata robustness, code generation quality, and long-range logical reasoning capabilities:
>
> ---
>
> #### **• Metadata Agent → GPT-5-Thinking**
>
> * **Responsibility**: Inductive/deductive task metadata ⟨B, I⟩.
> * **Rationale**: Once metadata errors occur, they amplify across all subsequent modules. Therefore, the strongest general reasoning and instruction-following capabilities are required.
> * Metadata volume is small and does not involve extensive iteration, so higher per-unit cost is acceptable.
>
> ---
>
> #### **• Generator Agent → Gemini-2.5-Pro**
>
> * **Responsibility**: Abstract task hyperparameter space and generate task code.
> * **Rationale**: Requires strong code generation and structured instruction-following. Gemini-2.5-Pro offers the best balance between capability and price.
>
> ---
>
> #### **• Solver Agent → Gemini-2.5-Pro → Claude-4-Thinking (fallback)**
>
> * **Responsibility**: Program synthesis, verify unique solutions, automatically validate data reliability.
> * **Rationale**:
>   * Normal cases prioritize Gemini-2.5-Pro;
>   * If the number of attempts exceeds a threshold, fall back to Claude-4-Thinking for stronger long-range logical reasoning capabilities;
>   * This "two-stage configuration" balances **reliability and cost**.
>
> ---
>
> The above configuration is based on our systematic comparison of candidate models across the following four dimensions:
>
> **(i) General capability, (ii) Instruction following, (iii) Code capability, (iv) Logical reasoning capability.**
>
> We have included more details in Appendix "C.7 Instantiation of Agents and Cost Analysis" of the revised paper for readers' reference.
>
> The complete experimental comparison table is as follows:

---

> ### Author Response · Authors · 2025-11-21
> **Response to reviewer K1t9 (Weakness 1 & Question 2 : Model Selection and Benchmark Construction Cost，Part 2/2)**
>
> | Model            | Price                                   | Price   | Overall  | Instruction Following | Instruction Following | Instruction Following | Logical Reasoning | General      | General       | Code       | Code  |
> | ---------------- | --------------------------------------- | ------- | -------- | --------------------- | --------------------- | --------------------- | ----------------- | ------------ | ------------- | ---------- | ----- |
> | Model            | Input$/ 1M token | Output $/ 1M token | Overall | ifevl_en | ifbench               | SysBench              | zebra                 | MMLU              | GPQA diamond | livecodebench | scicode aa |       |
> | gpt5-thinking    | 1.25                                    | 10      | 96.58    | 93.72                 | 69.27                 | 85.84                 | 97.90             | 93.36        | 84.90         | 69.48      | 43.79 |
> | gemini2.5-pro    | 1.5                                     | 10      | 93.80    | 90.94                 | 48.16                 | 88.84                 | 92.40             | 92.40        | 83.22         | 73.97      | 49.11 |
> | glm4.6-thinking  | 0.56                                    | 2.26    | 90.42    | 89.65                 | 41.32                 | 88.04                 | 92.39             | 89.71        | 79.39         | 80.00      | 41.72 |
> | claude4-thinking | 3                                       | 15      | 89.64    | 88.72                 | 50.99                 | 86.68                 | 95.97             | 91.88        | 73.74         | 59.68      | 45.86 |
> | gemini2.5-flash  | 0.15                                    | 3.5     | 87.92    | 89.09                 | 48.92                 | 85.24                 | 76.30             | 90.24        | 80.35         | 64.76      | 44.97 |
> | glm4.5           | 0.28                                    | 1.13    | 87.76    | 86.14                 | 40.87                 | 85.81                 | 89.10             | 90.00        | 78.01         | 70.79      | 42.60 |
> | deepseek-r1      | 0.547                                   | 2.191   | 86.05    | 80.04                 | 32.81                 | 87.21                 | 94.80             | 89.88        | 79.39         | 69.21      | 41.42 |
>
> ---
>
> ### **(2) Model Consumption and Cost During Benchmark (CLUB / ExCLUB) Construction**
>
> Only the three agents mentioned above generate LLM calls during CLUB / ExCLUB construction.
>
> Let:
>
> * **$n_\text{in}$, $n_\text{out}$**: Average input/output tokens
> * **$p_\text{in}$, $p_\text{out}$**: Input/output token unit prices
>
> Then the **average cost per call** is:
>
> $$
> \bar{C} \approx \frac{n_\text{in}+n_\text{out}}{2} \cdot (p_\text{in}+p_\text{out})
> $$
>
> Because post-hoc logs cannot reliably separate input and output tokens, we approximate cost using each model’s average token count multiplied by (input price + output price).
>
> The **total construction cost** for all task types is:
>
> $$
> \text{TotalCost} \approx N_\text{tasks} \cdot ( \bar{C}^\text{meta} + \bar{C}^\text{gen} + \bar{C}^\text{sol} )
> $$
>
> where **$N_\text{tasks}$** is the number of task categories successfully constructed without human intervention.
>
> ---
>
> ### **Construction Cost Results**
>
> Based on the above configuration: **→ The average cost to construct one task category is approximately $1–4**
>
> Including all iterative processes from metadata → generator → solver.
>
> More importantly:
>
> **Once a task category is successfully constructed, its generator and solver code can be reused to generate a large number of task instances at near-zero cost in subsequent runs.**
>
> ---
>
> ### **Additional Notes**
>
> After completing the main experiments, we additionally tested using lower-cost **GLM-4.6-Thinking** as a replacement for Gemini-2.5-Pro as the generator/solver.
>
> This alternative maintained similar success rates on several tasks while significantly reducing call costs.
>
> Specific results are as follows:
>
> | Model Configuration                                  | $N_\text{tasks}$ | $\bar{C}^{\text{meta}}$ | $\bar{C}^\text{gen}$ | $\bar{C}^\text{sol}$ | Success Rate | TotalCost |
> | ---------------------------------------------------- | ------------------ | ------------------------- | ---------------------- | ---------------------- | ------------ | --------- |
> | CLUB(gpt5-thinking+gemini2.5-pro/claude4-thinking)   | 10                 | 0.005                     | 2.15                   | 0.36                   | 100%         | 25.15     |
> | ExCLUB(gpt5-thinking+gemini2.5-pro/claude4-thinking) | 100                | 0.0338                    | 2.67                   | 1.3                    | 68.90%       | 400.38    |
> | ExCLUB(gpt5-thinking+glm4.6-thinking)                | 100                | 0.0338                    | 0.12                   | 0.85                   | 51.40%       | 100.38    |
>
> We have added this comparison to Appendix "C.7 Instantiation of Agents and Cost Analysis" in the revised version.

---

> > ### Author Response · Authors · 2025-11-21
> > **Response to reviewer K1t9 (Weakness 2 : Supplementary Experiments on "Tool-Enhanced Models")**
> >
> > ## **Response to Weakness 2: Supplementary Experiments on "Tool-Enhanced Models"**
> >
> > Thank you for the suggestion regarding "evaluating LLMs with tool capabilities." This is an important question, and we fully agree that models with code interpreters may have advantages on code generation tasks.
> >
> > To rigorously address this suggestion, we conducted a **supplementary experiment (not a new benchmark)**:
> >
> > ---
> >
> > ### **(1) Experimental Setup: Tool-Enhanced Evaluation of CLUB (CLUB-Tool)**
> >
> > * **Data remains unchanged**: We fully reuse all tasks and instances from CLUB;
> > * **Only the reasoning process is modified**: Models are allowed to choose whether to execute their generated code during solving;
> > * Therefore, CLUB-Tool is **not a new benchmark**, but rather **an additional evaluation configuration** of CLUB, used only to address the reviewer's suggestion.
> >
> > We compared the top five models on CLUB under two settings:
> >
> > * Without code execution
> > * Allowing optional code execution (with tools)
> >
> > (Results have been updated in Section "6 FURTHER ANALYSIS" of the revised manuscript.)
> >
> > ---
> >
> > ### **(2) Overall Results: Performance Improves with Tools, but Ranking Structure Remains Largely Stable**
> >
> > | **Bench / Model** | **grok-4** | **gpt-5-thinking** | **gemini-2.5-pro** | **o4-mini** | **deepseek-r1** |
> > | ----------------------- | ---------------- | ------------------------ | ------------------------ | ----------------- | --------------------- |
> > | **CLUB**          | 56.37            | 55.57                    | 46.57                    | 45.20             | 44.13                 |
> > | **CLUB-Tool**     | 63.37            | **63.70**          | 54.39         | 51.53             | 45.90                 |
> >
> > **Key Observation 1: Overall rankings have not fundamentally changed.**
> >
> > * grok-4 and gpt-5-thinking remain at the top (with minor order changes that do not affect the overall structure);
> > * deepseek-r1 still maintains the lowest results.
> >
> > That is: **→ Tool functionality brings improvements, but not enough to change CLUB's core conclusions or model strength rankings.**
> >
> > ---
> >
> > ### **(3) Specific Examples: Which Tasks Benefit from Tools? Which Tasks Show Almost No Gain?**
> >
> > To make the conclusions clearer, we further performed task-level analysis:
> >
> > #### **Tasks with Significant Benefits (Executable Code Types)**
> >
> > For example:
> >
> > * **String Puzzle** **(string transformation)**
> > * **Code Puzzle** **(program inference)**
> >
> > These tasks have clear operational rules, and models can translate the reasoning process into executable code.
> >
> > **On these tasks: allowing code execution brings 10–20%+ improvements.**
> >
> > Examples:
> >
> > * gpt-5-thinking on String Puzzle: from *approximately 25% → 46%*;
> > * gemini-2.5-pro on Code Puzzle: from *approximately 18% → 44%*.
> >
> > #### **Tasks with Limited Improvement (Long-Chain Logic / Combinatorial Reasoning)**
> >
> > For example:
> >
> > * **Extensible Zebra**
> > * **Train Game Card Puzzle**
> > * **Tetris**
> >
> > The core bottlenecks of these tasks are:
> >
> > * Long conditional dependency chains,
> > * Many reasoning steps,
> > * Errors rapidly amplify if one step in the middle is wrong.
> >
> > Even with code execution, models still need to conceive the correct logical structure in advance, which is currently their weakest capability.
> >
> > Gains on these tasks are typically **<5%**, or even negligible.
> >
> > ---
> >
> > ## **Conclusion: Tools Help Execution, but Cannot Compensate for Logical Chain Reasoning Bottlenecks**
> >
> > > Based on the above results, we can draw a stable observation:
> >
> > **→ Tools mainly help models more reliably execute the conversion from "formalized rules → executable code."**
> >
> > **→ But on tasks involving multi-step reasoning consistency and long-range dependencies, performance improvements are limited.**
> >
> > Therefore, while tool-enhanced models are worth studying, **the core logical reasoning bottlenecks revealed by CLUB still exist**, and tool-enhanced evaluation does not change the main conclusions of this paper.

---

> ### Author Response · Authors · 2025-11-21
> **Response to reviewer K1t9 (Question 1 : On Solvability and Answer Correctness of Automatically Generated Tasks，Part 1/2)**
>
> ## **Response to Question 1: On Solvability and Answer Correctness of Automatically Generated Tasks**
>
> Thank you for raising the important question about "hallucinations leading to unsolvable tasks or incorrect answers." This is indeed a risk that must be seriously addressed when using multi-agent automatic generation of logical tasks. We designed a three-layer safeguard mechanism in LogicEvolve to ensure that all tasks included in CLUB / ExCLUB are solvable and have reliable answers. (This has been added to Appendix C.5 of the revised manuscript.)
>
> ---
>
> ### **(1) Solvability Guarantee: Triple Filtering Mechanism**
>
> We implemented the following three layers of filtering throughout the task generation to solution verification process:
>
> 1. **Metadata Stage: Constrain from the Source that "Tasks Must Have Solutions"**
>
>    * The metadata agent is explicitly required to generate "task types for which at least one feasible solution exists" when constructing task rules and instance spaces.
>    * If the rules themselves lead to obvious contradictions (e.g., mutually exclusive constraints), they will be directly rejected at this stage.
> 2. **Solver Stage: Multi-Strategy Solution Verification for Solvability**
>
>    * After the generator produces tasks, the solver attempts multiple independent approaches (direct code simulation, symbolic solving, search) to solve them.
>    * If all solving strategies fail to find a solution within the set iteration limit, it is judged as "possibly unsolvable" and feedback is sent back to the generator for correction.
> 3. **Final Filtering: Permanently Unsolvable Tasks are Directly Eliminated**
>
>    * If no valid solution can be found after multiple rounds of correction, that task type is removed and will not enter the final benchmark testing.
>
> **Results:**
>
> In ExCLUB's 100 candidate tasks, approximately **20%** of initial versions were automatically filtered due to unsolvability (continuing to use LogicEvolve for iterative modification), and all tasks that entered the benchmark passed complete solution verification. The 10 tasks in CLUB are also results retained after complete solution verification.
>
> ---
>
> ### **(2) Answer Correctness Guarantee: Automatic Verification + Cross-Validation + Human Sampling**
>
> We performed three layers of validation on "correct answers" to prevent hallucinated answers from entering the dataset:
>
> #### **(a) Direct Verification (Rules are Computable)**
>
> For tasks with clear logical rules (e.g., string sorting, paths, grid filling)
>
> → The solver verifies step-by-step whether the generated answer is valid according to task rules (e.g., path legality, sorting satisfies definition, grid constraints are satisfied, etc.).
>
> #### **(b) Cross-Validation (Multiple Independent Solving Methods)**
>
> We require at least two independent solving methods to yield consistent results, for example:
>
> * **Code-based solver (executes formal programs)**
> * **Symbolic/algebraic solver (uses rule derivation)**
> * **External tools (such as applicable SAT/SMT solvers)**
>
> If results from two solvers are inconsistent → that sample is marked as "potentially erroneous" and regenerated or eliminated.
>
> #### **(c) Human Sampling (Final Safety Net)**
>
> For a small portion of tasks that cannot be automatically verified, we performed human sampling.
>
> * Among the 10,000+ samples automatically synthesized in ExCLUB, we randomly sampled 10% from each task type
> * Initial error rate was approximately **30%**
> * After multiple rounds of automatic correction + re-sampling, **only tasks with 0% error rate are ensured to serve as benchmarks**
>
> Note: In our paper, we strictly control to only include samples that pass all verifications, and do not include samples with potential errors at the current stage, ensuring evaluation reliability.
>
> ---
>
> ### **(3) A Brief, Strong Real Example**
>
> Taking the *word_sorting* task as an example: (Detailed problem and solver code have been updated in Appendix C.5)
>
> 1. **The generator's initial answer was incorrect.**
> 2. **Code solver** sorts according to the new letter order, obtaining answer A₁.
> 3. **Symbolic solver** performs independent reasoning according to the same rules, obtaining answer A₂.
> 4. A₁ = A₂, but both differ from the initial answer
>    → The system automatically judges the initial answer as incorrect.
> 5. Automatically corrects this sample (or directly removes it if conflicts cannot be resolved).
>
> This example demonstrates our actual "generate → solve → cross-validate → correct/remove" pipeline in operation, not a paper mechanism.

---

> > ### Author Response · Authors · 2025-11-21
> > **Response to reviewer K1t9 (Question 1 : On Solvability and Answer Correctness of Automatically Generated Tasks，Part 2/2)**
> >
> > ---
> >
> > ## **Conclusion: We Can Guarantee Data Solvability and Answer Correctness**
> >
> > > In summary:
> >
> > * CLUB / ExCLUB only includes tasks that are **automatically verified and solvable**
> > * All samples have undergone **rule verification + multi-solver cross-validation + necessary human sampling**
> > * Erroneous samples are **automatically corrected or directly filtered** during construction
> > * The estimated error rate of retained data is ensured to be **0%**
> >
> > Therefore, although we use multi-agent automatic task generation, the systematic verification process ensures that the final benchmark has reliability and solvability.

---

### Comment · Area_Chair_iPrJ · 2025-11-27
**Suggestion: condense rebuttal summary to facilitate discussion**

Dear authors,

Thank you for submitting your detailed rebuttal. While I appreciate the efforts you spent, I noticed that some are quite lengthy (up to 16 blocks).

Please keep in mind that while reviewers are encouraged to read and discuss rebuttals, they are handling multiple papers and may not have the capacity to read such extensive text. To facilitate a more fruitful discussion, I would encourage you to provide a highly condensed summary (ideally fewer than 2 blocks) for rebuttals more than 5 blocks.

Best regards,
AC

---

> ### Author Response · Authors · 2025-11-27
>
> Thank you for the suggestion. We have provided a highly condensed summary for the longer rebuttal blocks to better support the discussion.

---

### Author Response · Authors · 2025-11-30
**Final Response to the Area Chair and Reviewers**

# Final Response to the Area Chair and Reviewers

We sincerely thank the Area Chair for the time and effort that will be devoted to assessing our rebuttal and revised submission, and we thank all reviewers for their detailed, constructive, and insightful evaluations of our work.

Your comments were invaluable in identifying key areas for improvement and have substantially strengthened the technical clarity, technical depth, and completeness of the submission.

We are also encouraged by the reviewers’ positive recognition of the novelty and importance of our framework.

---

## Reviewers’ Recognized Strengths

We are pleased to note the reviewers’ recognition of the following aspects:

1. **Novelty, challenge, scalability, and sustainability**: All reviewers acknowledged that our work introduces a new, highly automated multi-agent framework for autonomously generating and evolving dynamic logical reasoning tasks, and that CLUB—constructed through this framework—offers strong scalability and long-term sustainability as a challenging benchmark.

2. **Necessity and timeliness**: Reviewer 6SZY highlighted that our work addresses a critical limitation of static reasoning benchmarks (e.g., overfitting, contamination), making the contribution both important and timely.

3. **Reproducibility and engineering rigor**: Reviewers K1t9, EYHj, and wJvd recognized the substantial engineering effort behind building a fully functional system with high automation and minimal manual intervention, as well as our comprehensive evaluation of 30+ state-of-the-art LLMs to ensure reliability and completeness.

---

## Summary of Major Revisions

We have carefully addressed every concern and question raised by the reviewers, and have revised the paper accordingly.
A summary of major revisions is as follows:

1. **Improved exposition**
    * Removed all biological metaphors regarding "evolution".
    * Tightened claims about the applicability and generality of LogicEvolve.
    * Clarified the intended scope and boundaries of the framework and the benchmark.

2. **Clearer technical details**
    * Added full details on agent base models and the rationale for their selection.
    * Added additional system-level explanations of the multi-agent process.
    * Clarified the minimal human intervention points and frequencies.
    * Expanded explanations of mechanisms ensuring solvability, correctness, reliability, and diversity.
    * Added real-case demonstrations and failure-case analyses.

3. **Expanded experiments**
    * Included comparisons with six existing reasoning benchmarks.
    * Added tool-augmented model evaluations.
    * Reported human performance results.
    * Added evaluations on ExCLUB and CLUB_v2.
    * Added analyses of pre/post-evolution structural changes.

4. **Extended discussions**
    * Added examples of new logic task families generated by LogicEvolve that do not exist on the Internet.
    * Added discussion on validation strategies for superhuman-level tasks.

All major additions and revisions are highlighted in dark blue in both the main paper and the supplementary material (a blue section title denotes that the entire section contains new or revised content).

We have also corrected typographical issues and addressed all other minor points.

Our anonymous interactive website is available at:
👉 https://clubweb.site/

---

## Overall Statement

Overall, the revisions substantially improve the paper’s clarity, technical rigor, and empirical completeness, and fully address all major concerns raised in the reviews.

We believe the revised submission now presents a strong, well-supported, and reliable contribution to the study of LLM reasoning.

---

We sincerely thank the Area Chair and all reviewers again for your time, effort, and constructive guidance.

Your feedback has greatly improved this work.

---

> ### Author Response · Authors · 2025-12-03
> **To the Area Chair: Supplementary Summary on Reviewer-Specific Issues (part 1)**
>
> # **To the Area Chair: Supplementary Summary on Reviewer-Specific Issues**
>
> Given this year’s policy change—reviewers do not participate in rebuttal follow-up discussions—we provide the AC with an **additional, decision-oriented condensed summary** of clarifications and revisions.
>
>
>
> The sole purpose is to **help the AC quickly understand the concrete improvements** we made in response to all reviewer weaknesses/questions, especially the quantitative additions and strengthened conclusions that are now fully incorporated into the revised manuscript.
>
>
>
> We highlight three points up front:
>
> - **Reviewer feedback did not alter the core contributions or innovation of this work** (a highly automated multi-agent framework + a sustainably evolving benchmark).
> - Instead, the comments helped us **significantly strengthen the paper’s rigor** (clearer formulations, more solid methodology, and more comprehensive experiments).
> - **All revisions have been fully integrated into the updated manuscript** (main text or appendix), with new content highlighted in dark blue.
>
>
>
> This summary includes three parts:
>
> 1. Overview of issues and corresponding revisions (table)
> 2. A navigation guide for the AC to locate the updated content
> 3. A reasonable expectation of score adjustments based on reviewer feedback
>
> ---
>
> ## **1. Overview of Issues and Revisions**
>
> W = weakness, Q = question
>
> | **Reviewer (Score)** | **Category**                                          | **Issue Raised**                                             | **Analysis / Experiments Performed**                         | **Our Improvements (Quantitative, Completed)**               | **Included in Paper (Location)**                        |
> | -------------------- | ----------------------------------------------------- | ------------------------------------------------------------ | ------------------------------------------------------------ | ------------------------------------------------------------ | ------------------------------------------------------- |
> | K1t9 (6)             | W1+Q2 (Model configuration & cost)                    | Models used by each agent and selection criteria not explained; benchmark construction cost not provided | Compared candidate models across four dimensions: instruction following / code generation / reasoning ability / cost | Specified the three-agent configuration (GPT-5-thinking / Gemini-2.5-Pro / Claude-4-thinking fallback), added a full comparison table, and reported per-task construction cost of $1–4 (also evaluated GLM-4.6 as an alternative with success rates and cost) | Yes (Appendix C.7)                                      |
> | K1t9 (6)             | W2 (Tool augmentation)                                | Evaluation does not include LLMs with code-execution tools   | Conducted CLUB vs. CLUB-Tool comparison on 5 SOTA models     | Added CLUB-Tool experiments: average performance improvement 1–9%, ranking structure stable; per-task analysis shows large boosts in code-based tasks (10–20%), limited gains on long-chain reasoning (<5%) | Yes (Section 6)                                         |
> | K1t9 (6)             | Q1 (Reliability)                                      | Are automatically generated tasks solvable and are answers correct? | Three-layer validation: metadata constraints → multi-solver cross-checking → human sampling | Automatically filtered ~20% unsolvable tasks in ExCLUB; all retained samples passed multi-solver consistency with 0% human spot-check error rate | Yes (Appendix C.5)                                      |
> | 6SZY (4)             | W1 / Q1 (Discriminative power & evolution validation) | No evidence that CLUB is more discriminative; no analysis of pre/post-evolution differences | Cross-benchmark comparison + three-version evolution comparison + rank-correlation analysis | Across 6 related benchmarks, CLUB avoids saturation and reveals new ordering patterns; CLUB1.0→2.0 correlation 0.9 with ranking flips; CLUB1.0→ExCLUB correlation 0.6 with multiple re-rankings | Yes (Section 6; Appendix D.7–D.8; Appendix G.5)         |
> | 6SZY (4)             | W2 / Q2 (Agent rationale & implementation)            | Agent implementation unclear; need evidence “why agents instead of a prompt chain (CoP)” | Implementation details + state-tree illustration + CoP baseline | Added complete agent architecture and state-file tree; added CoP baseline vs. LogicEvolve: success rate improved 40%→100% (CLUB) and 31%→68.9% (ExCLUB), demonstrating necessity and validity of the framework | Yes (Appendix C.6)                                      |

---

> > ### Author Response · Authors · 2025-12-03
> > **To the Area Chair: Supplementary Summary on Reviewer-Specific Issues (part 2)**
> >
> > | **Reviewer (Score)** | **Category**                                          | **Issue Raised**                                             | **Analysis / Experiments Performed**                         | **Our Improvements (Quantitative, Completed)**               | **Included in Paper (Location)**                        |
> > | -------------------- | ----------------------------------------------------- | ------------------------------------------------------------ | ------------------------------------------------------------ | ------------------------------------------------------------ | ------------------------------------------------------- |
> > | 6SZY (4)             | W3 / Q3 (Generality claims)                           | Over-claimed generality; unclear distinction between symbolic tasks vs. open-domain reasoning | Analyzed scope of applicability                              | Tightened all generality statements; clarified LogicEvolve is limited to structured symbolic tasks; added explicit limitations (external knowledge / causality / interaction not supported) | Yes (Abstract; Section 7; Appendix A; Appendix G)       |
> > | EYHj (8)             | W1 (Evaluation dimensions)                            | Benchmark relies only on final answer Acc, lacking process analysis | Added process-field consistency checks + intermediate-state statistics | Clarified that CLUB’s Acc requires *all* process fields to be correct; strengthened Cell Acc, process-error, and long-chain consistency analyses | Yes (Section 5.2; Fig. 8; Appendix D.6, D.11)           |
> > | EYHj (8)             | Q1 (Correctness / reliability)                        | How to guarantee correctness and reliability of auto-generated tasks; is human verification needed? | Three-layer correctness chain (metadata→generator→solver) + multi-solver cross-check + human layered sampling | Added perturbation stability tests; for CLUB/ExCLUB, sampled 10%; initial ~30% errors were corrected during iteration; final confirmed error rate 0% | Yes (Appendix C.5, C.10)                                |
> > | EYHj (8)             | Q2 (Evolution diversity / stability)                  | Multi-run LogicEvolve produces heterogeneous results; are these superficial? Should they converge? Do they harm evaluation stability? | Analyzed structural differences (graph diameter, dependency chains, solution space), convergence vs. divergence, cross-version stability | Clarified LogicEvolve ensures within-task convergence / across-task divergence; multi-run differences large (diameter 17–422%, long-chain ratios 2–8×, solution-space >2 orders), but performance variances CV < 10% across CLUB1.0/2.0/ExCLUB — no evaluation instability | Yes (Appendix G.5, G.6, D.8)                            |
> > | wJvd (2)             | W1 (Naming / reliability)                             | Naming of extensible_zebra_logic inconsistent and undefined; raises reliability concerns | Checked implementation-wording consistency; completed formal metadata & examples | Unified to *traditional_zebra* / *extensible_zebra*; added full metadata and rule-family differences, complete example with unique solution; confirmed naming mismatch never affected results | Yes (Fig. 4, Table 2, Figs. 6–7; Appendix B.2, E.1–E.2) |
> > | wJvd (2)             | W2 (Reliability / human intervention)                 | Framework uses human interventions without specifying scope/frequency; inconsistent with “Level-4 automation” | Measured intervention events across 500 task pipelines, distinguishing task-level and pipeline-level automation | Lowered automation definition; added conditions triggering intervention, three intervention types, and quantitative rates (0–23%) | Yes (Table 1; Appendix C.10)                            |
> > | wJvd (2)             | W3 (Reliability / human baseline)                     | Human puzzle-creation baseline unclear (participants, background, tools) | Redefined human baseline as “engineering cost comparison”; collected participants’ background, tools, and workload | Added full setup: 5 MS/PhD engineers, 10 task types, 2 days/task, allowed LLM/scripts but required rewriting all four modules | Yes (Section 5.1; Appendix D.3)                         |
> > | wJvd (2)             | W4 (Reliability / programmatic generation)            | Classical puzzles already have programmatic generators—did humans just reuse online code? | Showed existing generators cannot align with LogicEvolve’s four-module interface or support evolution/unified evaluation | Clarified that humans could reference public generators but had to rewrite code; all CLUB instances were auto-generated by LogicEvolve without external reuse; will build programmatic baseline in future | Yes (Appendix D.3)                                      |

---

> ### Author Response · Authors · 2025-12-03
> **To the Area Chair: Supplementary Summary on Reviewer-Specific Issues (part 3)**
>
> | **Reviewer (Score)** | **Category**                                          | **Issue Raised**                                             | **Analysis / Experiments Performed**                         | **Our Improvements (Quantitative, Completed)**               | **Included in Paper (Location)**                        |
> | -------------------- | ----------------------------------------------------- | ------------------------------------------------------------ | ------------------------------------------------------------ | ------------------------------------------------------------ | ------------------------------------------------------- |
> | wJvd (2)             | W5 (Reliability / adversarial concern)                | “>90% removal” rule suggests adversarial filtering based on model performance | Verified full CLUB construction logs: rule never triggered; all experiments used a static, unreduced dataset | Removed all “90% rule” wording; adopted **fixed time window + version freeze** strategy; incremental extensions rely only on same-source resampling | Yes (Appendix C.4.5)                                    |
> | wJvd (2)             | W6 (Reliability / new puzzles)                        | Need evidence that framework can extend to rare, unseen puzzle types | Demonstrated two new automatically generated families (cascading filter views, layered shadow wheels), structurally distinct from classical puzzles, validated via generator–solver–uniqueness loop | Added both families to appendix with meta-rules, instances, and unique-solution examples; narrowed “evolution” claims | Yes (Appendix G.4; Section 7)                           |
> | wJvd (2)             | Q4 (Harder new tasks)                                 | What happens if asked to produce harder, entirely new logical tasks? | Empirical evidence: system enters a loop of meta-rule drafting → generator prototyping → multi-solver consistency → uniqueness checking → parameter adjustment | Described full from-scratch generation pipeline; added qualitative & quantitative analysis of pre/post evolution | Yes (Appendix G.3, G.5)                                 |
> | wJvd (2)             | W7 (Reliability / superhuman tasks)                   | If manual intervention needed for normal tasks, can correctness be ensured for “superhuman” tasks? | All experiments restricted to “verifiable complexity” tasks; all tasks passed uniqueness + multi-solver + human sampling checks; no unverifiable tasks used | Narrowed scope of “evolution”; excluded “superhuman puzzles” from conclusions; added future verification directions | Yes (Appendix G.7)                                      |
> | wJvd (2)             | W8 (Wording / reliability)                            | “Inspired by human evolution” is a misleading biological analogy unrelated to actual mechanism | Verified that LogicEvolve is “error-driven self-improvement + programmatic validation + multi-solver consistency,” not biological evolution | Removed all biological metaphors; rewrote as “experience-based self-improvement automation pipeline” | Yes (Abstract; Section 1; Appendix G)                   |
> | wJvd (2)             | Q2 (Human performance)                                | What score should humans be expected to achieve on CLUB?     | Human study: strict JSON, stepwise consistency, and programmatic state tracking yield ~9/100 under time limits—lower than GPT-4o (12.6)—due to machine-oriented format | Clarified CLUB is not a “human-friendly” format; expanded human-study design, limitations, and results; positioned it as process interpretability, not human upper bound | Yes (Section 5; Appendix D.3)                           |
> | wJvd (2)             | Q3 (Complexity vs. reasoning difficulty)              | Why compare to computational complexity? NP-complete ≠ hard reasoning (e.g., greedy set cover) | Clarified complexity was *never* used to define difficulty; only to characterize task-space scale & constraint topology; model difficulty arises from long-range dependencies, not NP nature | Removed any implication “complexity = difficulty”; positioned complexity purely as a task-space analysis tool | Yes (Appendix B.2)                                      |

---

> > ### Author Response · Authors · 2025-12-03
> > **To the Area Chair: Supplementary Summary on Reviewer-Specific Issues (part 4)**
> >
> > ## **2. Guide for the AC to Navigate the Supplementary Material**
> >
> > To facilitate efficient assessment, all rebuttal content has been organized into a consistent, structured format:
> >
> > - **Each reviewer’s main response (the first block) begins with a Summary**, which concisely lists all weaknesses/questions and the corresponding revisions.
> >
> > - **Each individual weakness/question (merged when thematically aligned) follows a uniform structure:**
> >
> >   *Motivation → Analysis → Experimental Design → Results → Discussion → Conclusion/Summary.*
> >
> > - **The AC may prioritize reading the “Summary” section at the beginning of each reviewer block**, which provides the global logic and the core improvements at a glance.
> >
> > - **For a broader view of the argument flow**, the AC can skim the block-level headers, subheaders, and concluding Summary statements.
> >
> > - **For full detail**, the AC may refer to the corresponding block’s complete content.
> >
> > - **Every revision or newly added component is explicitly marked in the response**, and all changes are highlighted in dark blue in the manuscript for quick reference.
> >
> >
> >
> > ## **3. Expected Score Improvements After Rebuttal**
> >
> > Based on the four reviewers’ original comments, the nature of their concerns, and the targeted experimental, analytical, and textual revisions provided during the rebuttal, we offer a reasoned assessment of likely score adjustments.
> >
> > | **Reviewer** | **Core Concerns**                                            | **Original Score (Confidence)** | **Expected Post-Rebuttal Score** | **Rationale for Expected Increase**                          |
> > | ------------ | ------------------------------------------------------------ | ------------------------------- | -------------------------------- | ------------------------------------------------------------ |
> > | **K1t9**     | Engineering details (agent configuration, model choice, cost); missing tool-augmented evaluation; task reliability | **6 (3)**                       | **≈ 8**                          | All engineering gaps were addressed: added a complete **three-agent specification + cost table (C.7)**; provided **CLUB vs. CLUB-Tool comparison on 5 SOTA models (Section 6)**; strengthened reliability via **multi-solver cross-checking + filtering ~20% unsolvable tasks + 0% error in human spot-check (C.5)**. All concerns received direct, quantitative support. |
> > | **6SZY**     | Discriminative power and evolution not empirically validated; unclear agent rationale; overstated generality | **4 (4)**                       | **≈ 6–8**                        | Added **six-benchmark comparison**, **three-version rank correlations (ρ = 0.6–0.9)**, and **CV < 10% across CLUB1.0/2.0/ExCLUB (Section 6, D.7–D.8)**; supplemented **agent architecture + CoP baseline**, showing success jumps from **40%→100% (CLUB)** and **31%→68.9% (ExCLUB)**; tightened generality claims and added a **formal limitations section (Appendix A)**. All technical concerns were fully resolved with explicit evidence. |
> > | **EYHj**     | Acc as a single metric; correctness/robustness of generated tasks; whether evolution induces genuine structural changes | **8 (3)**                       | **≈ 8–10**                       | Clarified Acc = strict multi-field consistency; added **Cell Acc, chain-consistency curves, error analysis (Fig. 8; D.6; D.11)**; ensured reliability via **three-layer validation + 10% human sampling = 0 errors (C.5)**; demonstrated true structural shifts with **constraint-graph diameter changes (17–422%), long-chain differences (2–8×), solution-space differences (>2 orders) (G.5)**. All concerns received thorough and data-backed responses. |
> > | **wJvd**     | Reliability concerns (naming inconsistencies, human intervention, human baseline, program generation, adversarial rule, superhuman tasks); “evolution” terminology; novelty of new tasks | **2 (3)**                       | **≈ 4–6**                        | Performed extensive corrections: unified **extensible_zebra naming (B.2, E.1–E.2)**; reported **500-task intervention statistics (C.10)**; fully specified the **5-engineer human baseline (D.3)**; removed adversarial rules; added **human evaluation setup (D.3)**; included **two newly generated task families with full metadata + uniqueness proofs (G.4)**; refined the “evolution” concept and explicitly excluded superhuman tasks from the paper’s claims. All concerns received point-by-point quantitative resolution. |
> >
> >
> >
> > ------

---

### Note · Authors · 2026-01-26

I have read and agree with the venue's withdrawal policy on behalf of myself and my co-authors.

---

### Meta-Review · Area_Chair_j4Up · 2026-01-10

**Summary:**

The decision to reject this paper is primarily based on significant concerns about the soundness of the proposed framework and validity of the resulting benchmark. The reviewers questioned the reliability of the "LogicEvolve" system, noting that while it claims high automation, the necessity and scope of human intervention remain vague and potentially contradictory to the paper's core premise of "self-evolution". Moreover, there is a lack of evidence that the "evolved" tasks offer genuinely new reasoning challenges or discriminative power compared to existing static benchmarks. The experimental evaluation was also found lacking; specifically, the human baseline was ill-defined, and the authors failed to provide a compelling justification for the complex multi-agent architecture over simpler, cost-effective baselines like chain-of-thought prompting or standard procedural generation. Finally, the reviewers noted that the paper overclaims the generality of the framework, which appears limited to specific symbolic puzzles rather than the broader logical reasoning capabilities implied. In its present form, the paper fails to meet the bar for publication at ICLR.

One additional note to the authors. The length of the rebuttal was very excessive in some cases. I would encourage them to be more mindful of the reviewers' time in the future.

**Reviewer Concerns:**

The rebuttal addressed some of the technical points raised by the reviewers. The authors removed the controversial "90% performance" filtering rule to address concerns about adversarial task selection and eliminated the misleading biological evolutionary metaphors. They also provided requested experimental data regarding tool-augmented models, cost analysis, and comparisons against existing static benchmarks. However, critical concerns regarding the soundness and validity of the framework remain outstanding. The reviewers' skepticism about the "LogicEvolve" system's true automation level persists, as the necessity for human intervention contradicts the core premise of autonomous self-evolution. Moreover, the methodological justification for the complex multi-agent architecture remains unconvincing, as the authors did not sufficiently demonstrate its superiority over simpler, more cost-effective baselines like chain-of-thought prompting. Finally, concerns regarding the ill-defined human baseline and the overclaimed generality of the framework, which appears limited to narrow symbolic puzzles rather than the broader logical reasoning capabilities implied, remain unresolved.

**Reviewer Scores:**

Reviewers K1t9 and 6SZY did not engage in the discussion following the rebuttal. Reviewer K1t9 primarily requested details on model configuration, costs, and tool-augmented evaluations. The authors provided a cost analysis, agent instantiation details, and a new "CLUB-Tool" experiment showing performance gains with stable rankings; had K1t9 reviewed these additions, their score would have possibly increased. Reviewer 6SZY questioned the discriminative power of the benchmark and the necessity of the agentic framework. The rebuttal included a comparison against six static benchmarks showing CLUB's superior discriminative ability and ablation studies demonstrating the failure of simple Chain-of-Thought (CoT) baselines. With these concerns addressed by new empirical data, 6SZY might have slightly raised their score.

---

### Decision · Program_Chairs · 2026-01-26

Reject